# Quantum Current Algebra in Action: Linearization, Integrability of Classical and Factorization of Quantum Nonlinear Dynamical Systems

Anatolij K. Prykarpatski

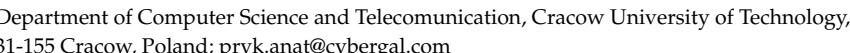

Department of Computer Science and Telecomunication, Cracow University of Technology, 31-155 Cracow, Poland; pryk.anat@cybergal.com

**Abstract:** This review is devoted to the universal algebraic and geometric properties of the non-relativistic quantum current algebra symmetry and to their representations subject to applications in describing geometrical and analytical properties of quantum and classical integrable Hamiltonian systems of theoretical and mathematical physics. The Fock space, the non-relativistic quantum current algebra symmetry and its cyclic representations on separable Hilbert spaces are reviewed and described in detail. The unitary current algebra family of operators and generating functional equations are described. A generating functional method to constructing irreducible current algebra representations is reviewed, and the ergodicity of the corresponding representation Hilbert space measure is mentioned. The algebraic properties of the so called coherent states are also reviewed, generated by cyclic representations of the Heisenberg algebra on Hilbert spaces. Unbelievable and impressive applications of coherent states to the theory of nonlinear dynamical systems on Hilbert spaces are described, along with their linearization and integrability. Moreover, we present a further development of these results within the modern Lie-algebraic approach to nonlinear dynamical systems on Poissonian functional manifolds, which proved to be both unexpected and important for the classification of integrable Hamiltonian flows on Hilbert spaces. The quantum current Lie algebra symmetry properties and their functional representations, interpreted as a universal algebraic structure of symmetries of completely integrable nonlinear dynamical systems of theoretical and mathematical physics on functional manifolds, are analyzed in detail. Based on the current algebra symmetry structure and their functional representations, an effective integrability criterion is formulated for a wide class of completely integrable Hamiltonian systems on functional manifolds. The related algebraic structure of the Poissonian operators and an effective algorithm of their analytical construction are described. The current algebra representations in separable Hilbert spaces and the factorized structure of quantum integrable many-particle Hamiltonian systems are reviewed. The related current algebra-based Hamiltonian reconstruction of the many-particle oscillatory and Calogero–Moser–Sutherland quantum models are reviewed and discussed in detail. The related quasi-classical quantum current algebra density representations and the collective variable approach in equilibrium statistical physics are reviewed. In addition, the classical Wigner type current algebra representation and its application to non-equilibrium classical statistical mechanics are described, and the construction of the Lie–Poisson structure on the phase space of the infinite hierarchy of distribution functions is presented. The related Boltzmann–Bogolubov type kinetic equation for the generating functional of many-particle distribution functions is constructed, and the invariant reduction scheme, compatible with imposed correlation functions constraints, is suggested and analyzed in detail. We also review current algebra functional representations and their geometric structure subject to the analytical description of quasi-stationary hydrodynamic flows and their magneto-hydrodynamic generalizations. A unified geometric description of the ideal idiabatic liquid dynamics is presented, and its Hamiltonian structure is analyzed. A special chapter of the review is devoted to recent results on the description of modified current Lie algebra symmetries on torus and their Lie-algebraic structures, related to integrable so-called heavenly type spatially many-dimensional dynamical systems on functional manifolds.

**Keywords:** diffeomorphism group; current algebra symmetry; current Lie algebra representation; fock space; generating functional; distribution functions; Lie–Poisson structure; coherent states; Lie-Poisson action; Hilbert space linearization; hamiltonian systems; symmetry reduction; integrability; idiabatic states; factorization; heavenly type dynamical systems; integrable dynamical systems; dirac reduction; hydrodynamic flows; entropy; vortex flows

## 1. Introduction

It is an old classical result that the nonrelativistic quantum current algebra realizes [1–3] a representation of the Lie algebra $\mathcal{G}$, related to the semidirect product $G := \mathrm{Diff}(\mathbb{R}^m) \ltimes F(\mathbb{R}^m; \mathbb{R})$ of the topological diffeomorphism group $\mathrm{Diff}(\mathbb{R}^m)$ of the real space $\mathbb{R}^m$ and the space $F(\mathbb{R}^m; \mathbb{R})$ of smooth Schwartz type functions on it. As it was later shown by G. Goldin, with collaborators [4–7], in fact, all nonrelativistic quantum many-particle Hamiltonian systems allow the equivalent representation by means of the current algebra operators and their realization on some specially constructed generalized Hilbert spaces with cyclic vector structure, strongly depending on the groundstate vectors of the corresponding Hamiltonian operators. The detailed analysis of this representation [8–10] made it possible to reveal a deep connection of the specially factorized operator structure of Hamiltonian operators and the quantum complete integrability of the corresponding Heisenberg type operator dynamical systems. Moreover, studying vector field representations of the quantum current algebra, related to the semidirect product $\mathrm{Diff}(\mathbb{S}^1) \ltimes F(\mathbb{S}^1; \mathbb{R})$ of the topological diffeomorphism group $\mathrm{Diff}(\mathbb{S}^1)$ of the circle $\mathbb{S}^1$ and the space $F(\mathbb{S}^2; \mathbb{R})$ of the smooth periodic functions on it, their isomorphism was stated [11–13] with all completely known and up to date Lax type integrable classical dynamical systems on spatially one-dimensional functional manifolds. Closely related algebraic aspects of representation theory of the canonical creation–annihilation operators, both on the Fock space and on related cyclic Hilbert spaces, gave rise to the construction of the effective linearizing scheme of any smooth dynamical system on functional Hilbert space. As the main analytical trick of these schemes is based on the coherent vector representation of the canonical creation–annihilation operators on the Fock space, we describe their unbelievable and impressive applications to the theory of nonlinear dynamical systems on Hilbert spaces, their linearization and integrability, previously initiated in [14,15] and continued in [16]. We briefly review the coherent vector representations of the Bargmann–Segal space $\mathcal{H}_k$ of complex holomorphic functions on $\mathbb{C}^k$, and describe a general approach to constructing coherent states and their applications both to the linearization of nonlinear dynamical systems on Hilbert spaces, and to describing their complete integrability. The latter is developed using the modern Lie-algebraic approach [11,17–19] to nonlinear dynamical systems on Poissonian functional manifolds, and proved to be both unexpected and important for the classification of integrable Hamiltonian flows on Hilbert spaces.

Other very important applications of the current algebra representations are related both to statistical physics, classical and quantum, and to hydrodynamics. The quantum current algebra quasi-classical representations made it possible to analytically describe [20–22] the so-called collective variable approach in equilibrium statistical physics and calculate the main thermodynamical quantities at finite temperatures. The related quantum current algebra quasi-classical Wigner type representations proved to be effective in describing the kinetic theory [23,24] of many-particle systems and calculating both the corresponding evolution equations for the infinite hierarchy of many-particle distribution functions, and developing a new approach to their dynamically compatible splitting, based on the well known Dirac type reduction of Poissonian systems on functional submanifolds.

A very rich geometric structure of liquid flow in a domain $\Omega \subset \mathbb{R}^3$ and its properties can be deeply described by means of the corresponding diffeomorphism group $\mathrm{Diff}(\Omega)$

and its semi-direct products with different functional spaces on the domain $\Omega \subset \mathbb{R}^3$. It is well known that the same physical system is often described using different sets of variables, related with their different physical interpretation. It was observed [25–33] that the corresponding mathematical structures used for describing the analytical properties of hydrodynamical systems are canonically related to each other. Simultaneously, mathematical properties, against a background of their analytical description, make it possible to study additional important parameters [34–50] of different hydrodynamic and magnetohydrodynamic systems. Amongst these, we will mention integral invariants, describing such internal fluid motion peculiarities as vortices, topological singularities [51] and other different instability states, strongly depending [52,53] on imposed isentropic fluid motion constraints. Being interested in their general properties and mathematical structures which are responsible for their existence and behavior, we present [54] a detailed differential geometrical approach to thermodynamically investigating quasi-stationary isentropic fluid motions, paying more attention to the analytical argumentation of tricks and techniques used during the presentation. Amongst the systems analyzed here, we mention the Hamiltonian analysis and adiabatic magneto-hydrodynamic superfluid motion, as well constructing a modified current Lie algebra and describing magneto-hydrodynamic invariants and their geometry. In particular, we studied a modified current Lie algebra symmetry on torus, its Lie-algebraic structure and related integrable heavenly type dynamical systems, describing the quasi-conformal metrics of Riemannian spaces in general relativity.

## 2. The Fock Space, Non-Relativistic Quantum Current Algebra and Its Cyclic Representations

### 2.1. The Fock Space Representation

Let a Hilbert space $\Phi_F$ possess the standard canonical Fock space structure [5,11,55–60], that is

$$\Phi_F = \oplus_{n \in \mathbb{Z}_+} \Phi_{(s)}^{\otimes n}, \tag{1}$$

where subspaces $\Phi_{(s)}^{\otimes n}$, $n \in \mathbb{Z}_+$, are the symmetrized tensor products of the Hilbert space $H \simeq L_2^{(s)}(\mathbb{R}^m; \mathbb{C}^k)$. If a vector $\varphi := (\varphi_0, \varphi_1, ..., \varphi_n, ...) \in \Phi_F$, its norm

$$\|\varphi\|_\Phi := \left( \sum_{n \in \mathbb{Z}_+} \|\varphi_n\|_n^2 \right)^{1/2}, \tag{2}$$

where $\varphi_n \in \Phi_{(s)}^{\otimes n} \simeq L_2^{(s)}((\mathbb{R}^m)^{\otimes n}; \mathbb{C})$ and $\| ... \|_n$ is the corresponding norm in $\Phi_{(s)}^{\otimes n}$ for all $n \in \mathbb{Z}_+$. Note here that concerning the rigging structure (18), there holds the corresponding rigging for the Hilbert spaces $\Phi_{(s)}^{\otimes n}$, $n \in \mathbb{Z}_+$, that is

$$\mathcal{D}_{(s)}^n \subset \Phi_{(s),+}^{\otimes n} \subset \Phi_{(s)}^{\otimes n} \subset \Phi_{(s),-}^{\otimes n} \tag{3}$$

with some suitably chosen dense and separable topological spaces of symmetric functions $\mathcal{D}_{(s)}^n$, $n \in \mathbb{Z}_+$. Concerning expansion (1), we obtain by means of projective and inductive limits [55,57,61,62] the quasi-nucleus rigging of the Fock space $\Phi$ in the form (18).

Consider now any basis vector $|(\alpha)_n) \in \Phi_{(s)}^{\otimes n}$, $n \in \mathbb{N}$, which can be written [56,57,63–65] in the following canonical Dirac ket-form:

$$|(\alpha)_n) := |\alpha_1, \alpha_2, ..., \alpha_n), \tag{4}$$

where, by definition,

$$|\alpha_1, \alpha_2, ..., \alpha_n) := \frac{1}{\sqrt{n!}} \sum_{\sigma \in S_n} |\alpha_{\sigma(1)}) \otimes |\alpha_{\sigma(2)}) ... |\alpha_{\sigma(n)}) \tag{5}$$

and vectors $|\alpha_j) \in H_+, \Phi_{(s)}^{\otimes 1} \simeq H, j, k \in \mathbb{N}$, are bi-orthogonal to each other, that is $(\alpha_k|\alpha_j)_H = \delta_{k,j}$ for any $k, j \in \mathbb{N}$. The corresponding scalar product of base vectors as (5) is given as follows:

$$((\beta)_n|(\alpha)_n) := (\beta_n, \beta_{n-1}, ..., \beta_2, \beta_1|\alpha_1, \alpha_2, ..., \alpha_{n-1}, \alpha_n)$$
$$= \sum_{\sigma \in S_n} (\beta_1|\alpha_{\sigma(1)})_H ... (\beta_n|\alpha_{\sigma(n)})_H := per\{(\beta_i|\alpha_j)_H\}_{i,j=\overline{1,n}}, \quad (6)$$

where "*per*" denotes the permanence of the matrix and $(\cdot|\cdot)$ is the corresponding scalar product in the Hilbert space $H$. Based now on the representation (4), one can define an operator $a^+(\alpha) : \Phi_{(s)}^{\otimes n} \longrightarrow \Phi_{(s)}^{\otimes(n+1)}$ for any $|\alpha) \in H_-$ as follows:

$$a^+(\alpha)|\alpha_1, \alpha_2, ..., \alpha_n) := |\alpha, \alpha_1, \alpha_2, ..., \alpha_n), \quad (7)$$

which is called the "*creation*" operator in the Fock space $\Phi_F$. The adjoint operator $a(\beta) := (a^+(\beta))^* : \Phi_{(s)}^{\otimes(n+1)} \longrightarrow \Phi_{(s)}^{\otimes n}$ with respect to the Fock space $\Phi_F$ (1) for any $|\beta) \in H_-$, called the "*annihilation*" operator, acts as follows:

$$a(\beta)|\alpha_1, \alpha_2, ..., \alpha_{n+1}) := \sum_{\sigma \in S_n} (\beta|\alpha_j)|\alpha_1, \alpha_2, ..., \alpha_{j-1}, \hat{\alpha}_j, \alpha_{j+1}, ..., \alpha_{n+1}), \quad (8)$$

where the hat "$\hat{\cdot}$" over a vector denotes that it should be omitted from the sequence.

It is easy to check that the commutator relationship

$$[a(\alpha), a^+(\beta)] = (\alpha|\beta)_H \quad (9)$$

holds for any vectors $|\alpha) \in H$ and $|\beta) \in H$. Expression (9), owing to the rigging structure (18), can be naturally extended to the general case, when vectors $|\alpha)$ and $|\beta) \in H_-$, conserving its form. In particular, taking $|\alpha) := |\alpha(y)) = \left\{ \frac{1}{\sqrt{2\pi}} e^{i\langle y|x \rangle} \right\}^k \in H_- := L_{2,-}(\mathbb{R}^m; \mathbb{C}^k)$ for any $y \in \mathbb{E}^m$, one easily gets from (9) that

$$[a_i(x), a_j^+(y)] = \delta_{ij}\delta(x - y) \quad (10)$$

for any $i, j = \overline{1, k}$, where we put, by definition, $\langle \cdot|\cdot \rangle$ the usual scalar product in the $m$-dimensional Euclidean space $\mathbb{E}^m := (\mathbb{R}^m; \langle \cdot|\cdot \rangle)$, $a_j^+(y) := a_j^+(y(x))$ and $a_j(y) := a_j(y(x)), j = \overline{1, k}$, for all $x, y \in \mathbb{R}^m$ and denoted by $\delta(\cdot)$ the classical Dirac delta-function.

The construction above makes it possible to observe easily that there exists the unique vacuum vector $|0) \in \Phi_{(s)}^{\otimes 1}$, such that for any $x \in \mathbb{R}^m$

$$a_j(x)|0) = 0 \quad (11)$$

for all $j \in \overline{1, k}$, and the set of vectors

$$\left( \prod_{j=1}^{k} \prod_{i=1}^{n_j} \left( a_j^+ \right)(x_j^{(i)}) \right)|0) \in \Phi_{(s)}^{\otimes n} \quad (12)$$

is total in $\Phi_{(s)}^{\otimes n}$, that is, their linear integral hull over the functional spaces $\Phi_{(s)}^{\otimes n}$ is dense in the Hilbert space $\Phi_{(s)}^{\otimes n}$ for every $n = \sum_{j=1}^{k} n_j \in \mathbb{N}$. This means that for any vector $\varphi \in \Phi_F$, the following canonical representation

$$\varphi = \sum_{n=\sum_{j=1}^{k} n_j \in \mathbb{Z}_+}^{\oplus} \int_{(\mathbb{R}^m)^n} \varphi_{n_1 n_2 \ldots n_s}^{(n)}(x_1^{(1)}, x_1^{(2)}, \ldots, x_1^{(n_1)}; x_2^{(1)}, x_2^{(2)}, \ldots, x_2^{(n_2)}; \ldots \tag{13}$$

$$; x_k^{(1)}, x_k^{(2)}, \ldots, x_k^{(n_m)}) \prod_{j=1}^{k} \frac{1}{\sqrt{n_j!}} \prod_{s=1}^{n_j} a_j^+(x_j^{(s)}) |0\rangle$$

holds with the Fourier type coefficients $\varphi_{n_1 n_2 \ldots n_s}^{(n)} \in \Phi_{(s)}^{\otimes n}$ for all $n = \sum_{j=1}^{k} n_j \in \mathbb{Z}_+$. The latter is naturally endowed with the Gelfand type quasi-nucleus rigging, dual to

$$H_+ \subset H \subset H_-, \tag{14}$$

making it possible to construct a quasi-nucleous rigging of the dual Fock space $\Phi_F := \oplus_{n \in \mathbb{Z}_+} \Phi_{(s)}^{\otimes n}$. Thereby, the chain (14) generates the dual Fock space quasi-nucleolus rigging

$$\mathcal{D} \subset \Phi_{F,+} \subset \Phi_F \subset \Phi_{F,-} \subset \mathcal{D}' \tag{15}$$

with respect to the Fock space $\Phi_F$, easily following from (1) and (14).

Construct now the following self-adjoint operator $\rho(x) : \Phi_F \to \Phi_F$ as

$$\rho(x) := \langle a^+(x) | a(x) \rangle, \tag{16}$$

called the density operator at a point $x \in \mathbb{R}^m$, satisfying the commutation properties:

$$[\rho(x), \rho(y)] = 0,$$
$$[\rho(x), a(y)] = -a(y)\delta(x - y), \tag{17}$$
$$[\rho(x), a^+(y)] = a^+(y)\delta(x - y)$$

for any $x, y \in \mathbb{R}^m$.

Assume now that $\Phi$ is a separable Hilbert space, $F$ is a topological real linear space and $\mathcal{A} := \{A(f) : f \in F\}$ is a family of commuting self-adjoint operators in $\Phi$ (i.e., these operators commute in the sense of their resolutions of the identity) with dense in $\Phi$ domain $\mathrm{Dom} A(f) := D_{A(f)} \subset \Phi, f \in F$. Consider the corresponding Gelfand rigging [57,61,66] of the Hilbert space $\Phi$, i.e., a chain

$$\mathcal{D} \subset \Phi_+ \subset \Phi \subset \Phi_- \subset \mathcal{D}' \tag{18}$$

in which $\Phi_+$ is a Hilbert space, topologically (densely and continuously) and quasi-nucleus (the inclusion operator $i : \Phi_+ \longrightarrow \Phi$ is of the Hilbert–Schmidt type) embedded into $\Phi$, the space $\Phi_-$ is the dual to $\Phi_+$ as the completion of functionals on $\Phi_+$ with respect to the norm $||f||_- := \sup_{||u||_+=1} |(f|u)_\Phi|$, $u \in \Phi$, a linear dense in $\Phi_+$ topological space $\mathcal{D} \subseteq \Phi_+$ is such that $\mathcal{D} \subset D_{A(f)} \subset \Phi$ and the mapping $A(f) : \mathcal{D} \to \Phi_+$ is continuous for any $f \in F$. Then, the following structural theorem [4,5,16,57,61,62,67–69] about the cyclic representations of the family $\mathcal{A} := \{A(f) : f \in F\}$ of commuting self-adjoint operators in the separable Hilbert space $\Phi$ holds.

**Theorem 1.** *Assume that the family of operators $\mathcal{A}$ satisfies the following conditions:*

*(a) for $A(f)$, $f \in F$, the closure of the operator $\overline{A(f)}$ in $\Phi$ coincides with $A(f)$ for any $f \in F$, that is $\overline{A(f)} = A(f)$ on domain $D_{A(f)}$ in $\Phi$;*

*(b) the Range $A(f) \subset \Phi$ for any $f \in F$;*

*(c) for every $\varphi \in \mathcal{D}$ the mapping $F \ni f \longrightarrow A(f)|\varphi) \in \Phi_+$ is linear and continuous;*

*(d) there exists a strong cyclic vector $|\Omega) \in \bigcap_{f \in F} D_{A(f)}$, such that the set of all vectors $|\Omega)$ and $\prod_{j=1}^{n} A(f_j)|\Omega)$, $n \in \mathbb{Z}_+$, is total in $\Phi_+$ (i.e., their linear hull is dense in $\Phi_+$).*

*Then there exists a probability measure $\mu$ on $(F', C_\sigma(F'))$, where $F'$ is the dual of $F$ and $C_\sigma(F')$ is the $\sigma$-algebra generated by cylinder sets in $F'$ such that, for $\mu-$almost every $\eta \in F'$ there is a generalized common eigenvector $\omega(\eta) \in \Phi_-$ of the family $\mathcal{A}$, corresponding to the common eigenvalue $\eta \in F'$, that is for any $\varphi \in \mathcal{D} \subset \Phi_+$ and $A(\mathrm{f}) \in \mathcal{A}$*

$$(\omega(\eta)|A(\mathrm{f})\varphi)_{\Phi_- \times \Phi_+} = \eta(\mathrm{f})(\omega(\eta)|\varphi)_{\Phi_- \times \Phi_+} \tag{19}$$

*with $\eta(\mathrm{f}) \in \mathbb{R}$, denoting here the result of the pairing between $F$ and $F'$.*

*The mapping*

$$\mathcal{D} \ni |\varphi) \longrightarrow (\omega(\eta)|\varphi)_{\Phi_- \times \Phi_+} := \varphi(\eta) \in \mathbb{C} \tag{20}$$

*for any $\eta \in F'$ can be continuously extended to a unitary surjective operator $\mathcal{F}_\eta : \Phi_+ \longrightarrow L_2^{(\mu)}(F'; \mathbb{C})$, where*

$$\mathcal{F}_\eta |\varphi) := \eta(\varphi) \tag{21}$$

*for any $\eta \in F'$ is a generalized Fourier transform, corresponding to the family $\mathcal{A}$. Moreover, the image of the operator $A(\mathrm{f})$, $\mathrm{f} \in F'$, under the $\mathcal{F}_\eta$- mapping is the operator of multiplication by the function $F' \ni \eta \to \eta(\mathrm{f}) \in \mathbb{R}$.*

Now, if to construct the following self-adjoint family $\mathcal{R} := \{\rho(\mathrm{f}) := \int_{\mathbb{R}^m} \rho(x)\mathrm{f}(x)dx : \mathrm{f} \in F\}$ of linear operators in the Hilbert space $\Phi_\mu$, where $F := \mathcal{S}(\mathbb{R}^m; \mathbb{R})$ is the Schwartz functional space dense in $H$, one can derive, making use of Theorem 1, that there exists the generalized Fourier transform (21), such that

$$\Phi_\mu = L_2^{(\mu)}(F'; \mathbb{C}) \simeq \int_{F'}^{\oplus} \Phi_{(\eta)} d\mu(\eta) \tag{22}$$

for some Hilbert space sets $\Phi_{(\eta)}$, $\eta \in F'$, and a suitable measure $\mu$ on $F'$, with respect to which the corresponding joint eigenvector $\omega(\eta) \in \Phi_-$ for any $\eta \in F'$ generates the Fourier transformed family $\{\eta(\mathrm{f}) \in \mathbb{R} : \mathrm{f} \in F\}$. Moreover, if $\dim \Phi_\eta = 1$ for all $\eta \in F'$, the Fourier transformed eigenvector $\omega(\eta) := \Omega(\eta) = 1$ for all $\eta \in F'$.

Now we will consider the family of self-adjoint operators $\rho(\mathrm{f}) : \Phi_\mu \to \Phi_\mu$, $\mathrm{f} \in F$, as generating a unitary family $\mathcal{U} := \{U(\mathrm{f}) : \mathrm{f} \in F\}$, where the operator

$$U(\mathrm{f}) := \exp[i\rho(\mathrm{f})] \tag{23}$$

is unitary, satisfying the abelian commutation condition

$$U(\mathrm{f}_1)U(\mathrm{f}_2) = U(\mathrm{f}_1 + \mathrm{f}_2) \tag{24}$$

for any $\mathrm{f}_1, \mathrm{f}_2 \in F$. Since, in general, the unitary family $\mathcal{U}$ is defined in the Hilbert space $\Phi_\mu$, not coinciding, in general with the canonical Fock type space, the important problem of describing its cyclic unitary representation spaces arises, within which the factorization jointly with relationships (17) hold for any $\mathrm{f} \in F$. This problem can be treated using mathematical tools devised both within the representation theory of $\mathbb{C}^*$-algebras [4,5,57,63] and the Gelfand–Vilenkin [66] approach. Below we will describe the main features of the Gelfand–Vilenkin formalism, being much more suitable for the task, providing a reasonably unified framework of constructing the corresponding representations. The next definitions will be used in our construction.

**Definition 1.** *Let $F$ be a locally convex topological vector space, $F_0 \subset F$ be a finite dimensional subspace of $F$. Let $F^0 \subseteq F'$ be defined by*

$$F^0 := \{\sigma \in F' : \sigma|_{F_0} = 0\}, \tag{25}$$

*and called the annihilator of $F_0$.*

The quotient space $F'^0 := F'/F^0$ may be, evidently, identified with $F'_0 \subset F'$, the adjoint space of $F_0$.

**Definition 2.** *Let $Q \subseteq F'^0$; then the subset*

$$X_{F^0}^{(Q)} := \left\{ \sigma \in F' : \sigma + F^0 \subset Q \right\} \tag{26}$$

*is called the cylinder set with the base $Q$ and the generating subspace $F^0$.*

**Definition 3.** *Let $n = \dim F_0 = \dim F'_0 = \dim F'^0$. One says that a cylinder set $X^{(Q)}$ has Borel base, if $Q$ is a Borel set, when regarded as a subset of $\mathbb{R}^m$.*

The family of cylinder sets with Borel base forms an algebra of sets, which is a key stone for defining measurable sets in and the corresponding measures on $F'$.

**Definition 4.** *The measurable sets in $F'$ are the elements of the $\sigma$-algebra generated by the cylinder sets with Borel base.*

**Definition 5.** *A cylindrical measure in $F'$ is a non-negative $\sigma$-pre-additive function $\mu$ defined on the algebra of cylinder sets with a Borel base and satisfying the conditions $0 \leq \mu(X) \leq 1$ for any $X$, $\mu(F') = 1$ and $\mu\left(\coprod_{j\in\mathbb{N}} X_j\right) = \sum_{j\in\mathbb{N}} \mu(X_j)$, if all sets $X_j \subset F'$, $j \in \mathbb{N}$, have a common generating subspace $F_0 \subset F$.*

**Definition 6.** *A cylindrical measure $\mu$ satisfies the commutativity condition if, and only if, for any bounded continuous function, $\alpha : \mathbb{R}^n \longrightarrow \mathbb{R}$ of $n \in \mathbb{N}$ real variables the function*

$$\alpha[f_1, f_2, ..., f_n] := \int_{F'} \alpha(\eta(f_1), \eta(f_2), ..., \eta(f_n)) d\mu(\eta) \tag{27}$$

*is sequentially continuous in $f_j \in F$, $j = \overline{1, m}$.*

**Remark 1.** *It is known [4,57,66] that in countably normalized spaces, the properties of sequential and ordinary continuity are equivalent.*

**Definition 7.** *A cylindrical measure $\mu$ is countably additive if, and only if, for any cylinder set $X = \coprod_{j\in\mathbb{N}} X_j$, which is the union of countably many mutually disjoints cylinder sets $X_j \subset F'$, $j \in \mathbb{N}$, $\mu(X) = \sum_{j\in\mathbb{N}} \mu(X_j)$.*

The next two standard propositions [4,57,66,70,71], characterizing extensions of the measure $\mu$ on $X = \coprod_{j\in\mathbb{N}} X_j$, hold.

**Proposition 1.** *A countably additive cylindrical measure $\mu$ can be extended to a countably additive measure on the $\sigma$-algebra, generated by the cylinder sets with a Borel base. Such a measure will also be called a cylindrical measure.*

**Proposition 2.** *Let $F$ be a nuclear space. Then, any cylindrical measure $\mu$ on $F'$, satisfying the continuity condition, is countably additive.*

*2.2. Non-Relativistic Quantum Current Algebra and Its Cyclic Representations*

Based on the Fock space $\Phi_F$, defined by (18) and generated by the creation–annihilation operators (7) and (8), the current operator $J(x) : \Phi_F \to \Phi_F^m$, $x \in \mathbb{R}^m$, can be easily constructed as follows:

$$J(x) = \frac{1}{2i}[a^+(x)\,\nabla_x a(x) - \nabla_x a^+(x)\,a(x)], \tag{28}$$

satisfying jointly with the density operator $\rho(x) : \Phi_F \to \Phi_F$, $x \in \mathbb{R}^m$, defined by (16), the following quantum current Lie algebra symmetry [4–7,59,68,72] relationships:

$$[J(g_1), J(g_2)] = iJ([g_1,g_2]), \quad [\rho(f_1), \rho(f_2)] = 0, \qquad (29)$$
$$[J(g_1), \rho(f_1)] = i\rho(\langle g_1|\nabla f_1\rangle),$$

holding for all $f_1, f_1 \in F$ and $g_1, g_2 \in F^m$, where we put, by definition,

$$[g_1,g_2] := \langle g_1|\nabla\rangle g_2 - \langle g_2|\nabla\rangle g_1, \qquad (30)$$

being the usual commutator of vector fields $\langle g_1|\nabla\rangle$ and $\langle g_2|\nabla\rangle$ on the configuration space $\mathbb{R}^m$. It is easy to observe that the current algebra (29) is the Lie algebra $\mathcal{G}$, corresponding to the Banach group $G := \text{Diff}(\mathbb{R}^m) \ltimes F$, the semidirect product of the Banach group of diffeomorphisms $\text{Diff}(\mathbb{R}^m)$ of the $m$-dimensional space $\mathbb{R}^m$ and the Abelian group $F$. As the Lie algebra $\Gamma(\mathbb{R}^m)$ of smooth vector fields on $\mathbb{R}^m$ with the Lie bracket (17) is isomorphic to the Lie algebra $\text{Diff}(\mathbb{R}^m)$ of the Banach diffeomorphism group $\text{Diff}(\mathbb{R}^m)$, it is natural to construct the corresponding unitary operators

$$V(\varphi_t^g) := \exp[iJ(g)], \qquad (31)$$

on the Representation Hilbert space $\Phi_\mu$, where for any $g \in F^m$, there holds $d\varphi_t^g/dt = g(\varphi_t^g)$, $\varphi_t^g(x)|_{t=0} = x \in \mathbb{R}^m$, where $\varphi_t^g \in \text{Diff}(\mathbb{R}^m)$, $t \in \mathbb{R}$. The constructed above exponential currents (23) and (31) constitute together a unitary operator group on the Hilbert space $\Phi$, endowed with the following composition law

$$U(f_1)U(f_2) = U(f_1 + f_2), V(\varphi_1)V(\varphi_2) = V(\varphi_2 \circ \varphi_1), \qquad (32)$$
$$V(\varphi)U(f) = U(f \circ \varphi)V(\varphi)$$

for all $f_1, f_2, f \in F$ and $\varphi, \varphi_2, \varphi \in \text{Diff}(\mathbb{R}^m)$. The operator group (32) is, evidently, isomorphic to the semidirect product group $G$, which is endowed, respectively, with the natural composition law

$$(\varphi_1, f_1) \circ (\varphi_2, f_2) = (\varphi_2 \circ \varphi_1, f_1 + f_2 \circ \varphi_1) \qquad (33)$$

for all $f_1, f_2 \in F$ and $\varphi_1, \varphi_2 \in \text{Diff}(\mathbb{R}^m)$. Concerning a more adequate mathematical description of the Banach diffeomorphism group $\text{Diff}(\mathbb{R}^m)$, it is useful to consider the subgroup $\text{Diff}_0(\mathbb{R}^m)$ of smooth diffeomorphisms of $\mathbb{R}^m$ with compact supports, which is a topological space with the topology given by a counted family of the metrics $||\varphi_1 - \varphi_2||_n := \max_{|k|=\overline{0,n}} \sup_{x \in \mathbb{R}^m} (1 + |x|^2)^n |\varphi_1^{(k)}(x) - \varphi_2^{(k)}|$ for all $n \in \mathbb{Z}_+$ and $\varphi_1, \varphi_2 \in \text{Diff}_0(\mathbb{R}^m)$. So, the diffeomorphism group $\text{Diff}(\mathbb{R}^m)$ can be defined as the completion of the space $\text{Diff}_0(\mathbb{R}^m)$ with respect to the topology introduced above. This way, the constructed group $\text{Diff}(\mathbb{R}^m)$ is topological, locally linear connected and metrizable with a countable topology basis at each of its points. In particular, the group $\text{Diff}(\mathbb{R}^m)$ contains diffeomorphisms with noncompact supports, yet in the limit $|x| \to \infty$, $x \in \mathbb{R}^m$, they can be approximated by the identity mapping in $\text{Diff}(\mathbb{R}^m)$. The latter makes it possible to state that for any $g \in F^m$ the element $\varphi_t^g \in \text{Diff}(\mathbb{R}^m)$ for all $t \in \mathbb{R}$ generates the uniform continuous mapping $F^m \ni g \to \varphi_t^g \in \text{Diff}(\mathbb{R}^m)$.

Proceeding now to the Banach group of currents $G = \text{Diff}(\mathbb{R}^m) \ltimes F$, we have that the separable Hilbert space $\Phi_\mu$ for every irreducible cyclic representation will be unitary equivalent to the Hilbert space (45), which in many physical applications reduces in the case $\dim \Phi_{(\eta)} = 1$ for all $\eta \in F'$ to the following form:

$$\Phi_\mu \simeq L_2^{(\mu)}(F'; \mathbb{C}), \qquad (34)$$

being the space of square integrable functions with respect to the measure $\mu$ on $F'$.

Assume now that an element $\omega \in \Phi_\mu$ is taken arbitrarily and consider the action of the Banach group of currents $G$ on it:

$$U(f)\omega(\eta) = \exp[i(\eta(f)]\omega(\eta), \tag{35}$$

$$V(\varphi)\omega(\eta) = \chi_\varphi(\eta)\omega(\varphi^*\eta)\left[\frac{d\mu(\varphi^*(\eta))}{d\mu(\eta)}\right]^{1/2},$$

where, by definition, $\varphi^*\eta(f) := \eta(f \circ \varphi)$ for all $f \in F$, $\frac{d\mu(\varphi^*\eta)}{d\mu(\eta)}$ is the corresponding Radon–Nikodym derivative [19,73] of the measure $\mu \circ \varphi^*$ with respect to the measure $\mu$ on $F'$ and $\chi_\varphi(\eta)$ is a complex-valued character of the unit norm, satisfying the relationship

$$\chi_{\varphi_2}(\eta)\chi_{\varphi_1}(\varphi_2^*\eta) = \chi_{\varphi_1 \circ \varphi_2}(\eta) \tag{36}$$

for all $\varphi_j \in \text{Diff}(\mathbb{R}^m), j = \overline{1,2}, \eta \in F'$. For the Radon–Nikodym derivative above to exist, the measure $\mu$ on $F'$ should be quasi-invariant with respect to the diffeomorphism group $\text{Diff}(\mathbb{T}^n)$, that is, for any measurable set $Q \subset F'$ the condition $\mu(Q) = 0$ if, and only if, $\mu(\varphi^*Q)$ for arbitrary $\varphi \in \text{Diff}(\mathbb{R}^m)$.

In physics applications, the representation (35) is uniquely determined by the measure $\mu$ on $F'$, which in the general case has a very complicated [20,72] structure, and its analytic construction is nontrivial. One of the fairly effective approaches to this problem is the quantum method of Bogolubov generating functionals developed in [2,6,7,20,74]. Another approach, which is of considerable interest for the theory of dynamical systems, is based on algebraic methods of constructing self-adjoint functional-operator representations of the original current Lie algebra (29). In particular, the representation (35), corresponding to a quantum-mechanical system of $N \in \mathbb{N}$ identical bose-particles localized at points $x_j \in \mathbb{R}^m$, has a measure $\mu$ with supports [20,72] on Dirac delta-functions $\eta := \eta_N \in F'$ of the form:

$$\eta_N(x) = \sum_{j \in \overline{1,N}} \delta(x - x_j) \tag{37}$$

at any $x \in \mathbb{R}^m$ with a measure $\mu$ of the form:

$$d\mu(\eta_N) = \Omega_N^*\Omega_N \prod_{j=\overline{1,N}} dx_j\delta(\eta - \eta_N(x)), \tag{38}$$

where $\Omega_N \in \Phi_N \simeq L_2^{(s)}((\mathbb{R}^m)^{\otimes N};\mathbb{C})$ is the corresponding symmetric ground-state wave function of the related quantum Hamiltonian system, satisfying the conditions (49) and (49), reduced on the invariant subspace $\Phi_N$. Moreover, the following general expressions hold: $\Omega(\eta) = 1$ and for any $\omega \in L_2^{(\mu)}(F';\mathbb{C})$

$$\rho(x)\omega(\eta_N) = \sum_{j \in \overline{1,N}} \delta(x - x_j)\omega(\eta_N), \tag{39}$$

$$J(x)\omega(\eta_N) = \frac{1}{2i}\sum_{j \in \overline{1,N}}\left[\delta(x - x_j) \circ \partial/\partial x_j + \partial/\partial x_j \circ \delta(x - x_j)\right]\omega(\eta_N),$$

where, by definition, $\omega(\eta_N) \in \Phi_N \simeq L_2^{(s)}((\mathbb{R}^m)^{\otimes N};\mathbb{C})$. As a simple consequence of the actions (39), one derives that

$$U(f)\omega(\eta_N) = \exp[i \sum_{j \in \overline{1,N}} f(x_j)]\omega(\eta_N), \tag{40}$$

$$V(\varphi)\omega(\eta_N) = \omega(\varphi^*\eta_N)\left[\left|\det\left(\frac{\partial\varphi(x)}{\partial x}\right)\right|\right]^{1/2},$$

where we put, for brevity, that the character $\chi_\varphi(\eta_N) = 1$ for all $\varphi \in \mathrm{Diff}(\mathbb{R}^m)$.

*2.3. The Generating Functional Equation, Cyclic Current Algebra Representation and Hamiltonian Operator Groundstate*

Concerning the Fourier transform of a cylindrical measure $\mu$ in $F'$, we will use the following natural definitions.

**Definition 8.** *Let $\mu$ be a cylindrical measure in $F'$. The Fourier transform of $\mu$ is the nonlinear functional*

$$\mathcal{L}(f) := \int_{F'} \exp[i\eta(f)]d\mu(\eta), \tag{41}$$

*coinciding with the characteristic functional of the measure $\mu$.*

**Definition 9.** *The nonlinear functional $\mathcal{L} : F \longrightarrow \mathbb{C}$ on $F$, defined by (41), is called positive definite, if, and only if, for all $f_j \in F$ and $\lambda_j \in \mathbb{C}$, $j = \overline{1,n}$, the condition*

$$\sum_{j,k=1}^{n} \bar{\lambda}_j \mathcal{L}(f_k - f_j)\lambda_k \geq 0 \tag{42}$$

*holds for any $n \in \mathbb{N}$.*

The following important proposition, owing to Gelfand and Vilenkin [4,66], Araki [75] and Goldin [1,4], holds.

**Proposition 3.** *The functional $\mathcal{L} : F \longrightarrow \mathbb{C}$ on $F$, defined by (41), is the Fourier transform of a cylindrical measure on $F'$ if, and only if, it is positive definite, sequentially continuous and satisfying the condition $\mathcal{L}(0) = 1$. Suppose now that we have a continuous unitary representation of the unitary family $\mathcal{U}$ in a suitable Hilbert space $\Phi_\mu$ with a cyclic vector $|\Omega) \in \Phi_\mu$. Then we can put*

$$\mathcal{L}(f) := (\Omega|U(f)|\Omega) \tag{43}$$

*for any $f \in F := \mathcal{S}(\mathbb{R}^n; \mathbb{R})$, being the Schwartz space on $\mathbb{R}^m$, and observe that functional (43) is continuous on $F$ owing to the continuity of the representation. Therefore, this functional is the generalized Fourier transform of a cylindrical measure $\mu$ on $F'$:*

$$(\Omega|U(f)|\Omega) = \int_{\mathcal{S}'} \exp[i\eta(f)]d\mu(\eta). \tag{44}$$

*From the spectral point of view, based on Theorem 1, there is an isomorphism between the Hilbert spaces $\Phi_\mu$ and $L_2^{(\mu)}(F; \mathbb{C})$, defined by $|\Omega) \longrightarrow \Omega(\eta) = 1$ and $U(f)|\Omega) \longrightarrow \exp[i\eta(f)]$ and next extended by linearity upon the whole Hilbert space $\Phi$. In the non-cyclic case, there exists a finite or countably infinite family of measures $\{\mu_k : k \in \mathbb{Z}_+\}$ on $F'$, with $\Phi_\mu \simeq \oplus_{k \in \mathbb{Z}_+} L_2^{(\mu_k)}(F'; \mathbb{C})$ and the unitary operator $U(f) : \Phi_\mu \longrightarrow \Phi_\mu$ for any $f \in F$ corresponds in all $L_2^{(\mu_k)}(F'; \mathbb{C})$, $k \in \mathbb{Z}_+$, to a multiplication operator on the exponent function $\exp[i\eta(f)]$. This means that there exists a single cylindrical measure $\mu$ on $F'$ and a $\mu-$ measurable field of Hilbert spaces $\Phi_{(\eta)}$ on $F'$, such that*

$$\Phi_\mu \simeq \int_{F'}^{\oplus} \Phi_{(\eta)} d\mu(\eta), \tag{45}$$

*with* $U(f) : \Phi_\mu \longrightarrow \Phi_\mu$, *corresponding* [66] *to the operator of multiplication by* $\exp[i\eta(f)]$ *for any* $f \in F$ *and* $\eta \in F'$. *Thereby, having constructed the nonlinear functional* (41) *in an exact analytical form, one can retrieve the representation of the unitary family* $\mathcal{U}$ *on the corresponding Hilbert space* $\Phi_\mu$, *as follows:* $\Phi_\mu = \oplus_{n \in \mathbb{Z}_+} \Phi_n$, *where*

$$\Phi_n = \prod_{j=\overline{1,n}} \rho(x_j)|\Omega), \tag{46}$$

*for all* $n \in \mathbb{N}$.

The cyclic vector $|\Omega) \in \Phi_\mu$ can be, in particular, obtained as the ground state vector of some unbounded self-adjoint positive definite Hamiltonian operator $H : \Phi_\mu \longrightarrow \Phi_\mu$, commuting with the self-adjoint non-negative particle number operator

$$N := \int_{\mathbb{R}^m} dx \rho(x), \tag{47}$$

that is $[H, N] = 0$. Moreover, the conditions

$$H|\Omega) = 0 \tag{48}$$

and

$$\inf_{\varphi \in D_H} (\varphi|H|\varphi) = (\Omega|H|\Omega) = 0 \tag{49}$$

hold for the operator $H : \Phi_\mu \to \Phi_\mu$, where $D_H$ denotes its domain of definition, dense in $\Phi_\mu$. To find the functional (43), which is called the generating Bogolubov type functional for moment distribution functions

$$f_n(x_1, x_2, ..., x_n) := (\Omega| : \rho(x_1)\rho(x_2)...\rho(x_n) : |\Omega), \tag{50}$$

where $x_j \in \mathbb{R}^m$, $j = \overline{1, n}$, and the normal ordering operation: $\cdot$ : is defined [4,6,7,55,56,68] as

$$: \rho(x_1)\rho(x_2)...\rho(x_n) := \prod_{j=1}^{n} \left( \rho(x_j) - \sum_{k=1}^{j-1} \delta(x_j - x_k) \right), \tag{51}$$

it is convenient first to choose the Hamilton operator $H : \Phi_F \to \Phi_F$ in the following secondly quantized [4,5,56] representation

$$H := \frac{1}{2} \int_{\mathbb{R}^m} \langle \nabla_x a^+(x) | \nabla_x a(x) \rangle dx + V(\rho), \tag{52}$$

on the related Fock space $\Phi_F$, where the sign "$\nabla_x''$" means the usual gradient operation with respect to $x \in \mathbb{R}^m$ in the Euclidean space $\mathbb{E}^m \simeq (\mathbb{R}^m; \langle \cdot | \cdot \rangle)$. If the energy spectrum density of the Hamiltonian operator (52) on the cyclic representation Hilbert space $\Phi_\mu$ is bounded from below, in works done by Goldin G.A., Grodnik J., Menikov R. Powers R.T. and Sharp D. [4,5,76] it was stated that this Hamiltonian, modulo the ground state energy eigenvalue, can be algebraically represented on a suitably constructed *current algebra symmetry representation Hilbert space* $\Phi_\mu$, as the positive definite gauge type factorized operator

$$H = \frac{1}{2} \int_{\mathbb{R}^m} \left\langle (K^+(x) - A(x; \rho)) | \rho^{-1}(x)(K(x) - A(x; \rho)) \right\rangle dx, \tag{53}$$

satisfying conditions (48) and (49), where $A(x; \rho) : \Phi_\mu \to \Phi_\mu^m$, $x \in \mathbb{R}^n$, is some specially constructed [68,77] linear self-adjoint operator, satisfying the condition

$$K(x)|\Omega) = A(x; \rho)|\Omega) \tag{54}$$

with the ground state $|\Omega\rangle \in \Phi_\mu$, corresponding to chosen potential operators $V(\rho) : \Phi_\mu \to \Phi_\mu$. The singular structure of the operator (53) was previously analyzed in detail in [2] where, in part, its well-posedness was showed.

The "*potential*" operator $V(\rho) : \Phi_\mu \to \Phi_\mu$ is, in general, a polynomial (or analytical) functional of the density operator $\rho(x) : \Phi_\mu \longrightarrow \Phi_\mu$ for any $x \in \mathbb{R}^m$, and the operator $K(x) : \Phi_\mu \to \Phi_\mu^m$ is defined as

$$K(x) := \nabla_x \rho(x)/2 + iJ(x), \tag{55}$$

where the self-adjoint "current" operator $J(x) : \Phi_\mu \to \Phi_\mu^m$ can be naturally defined (but non-uniquely) from the continuity equality

$$\partial \rho/\partial t = i[H, \rho(x)] = -\langle \nabla | J(x) \rangle, \tag{56}$$

holding for all $x \in \mathbb{R}^m$. Such an operator $J(x) : \Phi_\mu \to \Phi_\mu^m$, $x \in \mathbb{R}^m$, can exist owing to the commutation condition $[H, N] = 0$, giving rise to the continuity relationship (56), if, additionally, to take into account that supports supp $\rho$ of the density operator $\rho(x) : \Phi_\mu \to \Phi_\mu$, $x \in \mathbb{R}^m$, can be chosen arbitrarily, owing to the independence of (56) on the potential operator $V(\rho) : \Phi_\mu \to \Phi_\mu$, but its strict dependence on the corresponding representation (45).

**Remark 2.** *The self-adjointness of the operator* $A(g; \rho) : \Phi_\mu \to \Phi_\mu$, $g \in F$, *can be stated following schemes from works [5,68,72] under the additional existence of such a linear anti-unitary mapping* $T : \Phi_\mu \to \Phi_\mu$ *that the following invariance conditions hold:*

$$T\rho(x)T^{-1} = \rho(x), \qquad T\,J(x)\,T^{-1} = -J(x), \qquad T|\Omega\rangle = |\Omega\rangle \tag{57}$$

*for any $x \in \mathbb{R}^m$. Thereby, owing to conditions (57), the following equalities*

$$K(x)|\Omega\rangle = A(x; \rho)|\Omega\rangle \tag{58}$$

*hold for any $x \in \mathbb{R}^m$, giving rise to the self-adjointness of the operator* $A(g; \rho) : \Phi_\mu \longrightarrow \Phi_\mu$, $g \in F^m$.

It is easy to observe that the time-reversal condition (57) imposes the real value relationship for the real valued ground state $\Omega_N = \overline{\Omega}_N \in \Phi_N \simeq L_2^{(s)}(\mathbb{R}^{m \times N}; \mathbb{C})$ of the canonically represented $N$-particle Hamiltonian $H_N : \Phi_N \to \Phi_N$ for arbitrary $N \in \mathbb{N}$. Moreover, taking into account the relationship (58), one can easily observe that on the invariant subspace $\Phi_N \subset \Phi_F$, the operator $K(x) : \Phi_N \longrightarrow \Phi_N$ is representable as

$$K_N(x) = \sum_{j=\overline{1,N}} \delta(x - x_j)\frac{\partial}{\partial x_j}, \tag{59}$$

entailing the following expression for the related operator $A_N(x; \rho) : \Phi_N \to \Phi_N$ on the subspace $\Phi_N \subset \Phi$ :

$$A_N(x; \rho) = \sum_{j=\overline{1,N}} \delta(x - x_j)\nabla_{x_j} \ln |\Omega_N(x_1, x_2, ..., x_N)|. \tag{60}$$

The latter makes it possible to derive its secondly quantized [56,57,78,79] expression as

$$A(x; \rho) = \int_{\mathbb{R}^{m \times N}} dx_2 dx_3 ... dx_N : \rho(x)\rho(x_2)\rho(x_3)...\rho(x_N) : \nabla_x \ln |\Omega_N(x, x_2, ..., x_N)|, \tag{61}$$

which holds for any $x \in \mathbb{R}^m$ and arbitrary $N \in \mathbb{Z}_+$. Being interested in the infinite particle case when $N \to \infty$, the expression (61) can be naturally decomposed [77,79] as

$$A(x; \rho) := \rho(x)\nabla \frac{\delta}{\delta\rho(x)}W(\rho) = \tag{62}$$

$$= \sum_{n \in \mathbb{Z}_+} \frac{1}{n!} \int_{\mathbb{R}^{m \times n}} dy_1 dy_2 ... dy_n : \rho(x)\rho(y_1)\rho(y)\rho(y_3)...\rho(y_n) : \nabla_x W_{n+1}(x; y_1, y_2, ..., y_n),$$

where the corresponding real-valued coefficients $W_n \in H_2^{(1)}(\mathbb{R}^{m \times n}; \mathbb{R})$ should be such functions that the series (62) were convergent in a suitably chosen representation Fock space $\Phi_F$, for which the resulting ground state $\lim_{N \to \infty} \Omega_N \simeq |\Omega) \in \Phi_F$ is necessarily cyclic and normalized.

Based now on the construction above, one easily deduces from expression (55) that the generating Bogolubov type functional (43) obeys for all $x \in \mathbb{R}^m$ the following functional-differential equation:

$$[\nabla_x - i\nabla_x \mathrm{f}]\frac{1}{2i}\frac{\delta \mathcal{L}(\mathrm{f})}{\delta \mathrm{f}(x)} = \mathrm{A}\left(x; \frac{1}{i}\frac{\delta}{\delta \mathrm{f}}\right)\mathcal{L}(\mathrm{f}), \tag{63}$$

whose solutions should satisfy [3,74] the Fourier transform representation (44), and which were, in part, studied in [74]. In particular, a wide class of special so-called Poissonian white noise type solutions to the functional-differential Equation (63) was obtained in [5,61,62,68,71,80] by means of functional-operator methods in the following generalized form:

$$\mathcal{L}(\mathrm{f}) = \exp\left\{2\int_{\mathbb{R}^m}\mathrm{W}\left(\frac{1}{i}\frac{\delta}{\delta \mathrm{f}}\right)dx\right\}\exp\left(\bar{\rho}\int_{\mathbb{R}^m}\{\exp[i\mathrm{f}(x)] - 1\}dx\right), \tag{64}$$

where $\bar{\rho} = (\Omega|\rho|\Omega) \in \mathbb{R}_+$ is a suitable Poisson process parameter and the operator $\mathrm{A}(x; \rho) : \Phi_\mu \to \Phi_\mu^m, x \in \mathbb{R}^m$, resulting from the expression (62) for some scalar operator $\mathrm{W}(\rho) : \Phi_\mu \to \Phi_\mu$.

**Remark 3.** *It is worth remarking here that solutions to Equation (63) realize the suitable physically motivated representations of the abelian Banach subgroup $F$ of the Banach group $G = \mathrm{Diff}(\mathbb{R}^m) \ltimes F$, mentioned above. In the general case of this Banach group $G$ one can also construct [5,6,16,81] a generalized Bogolubov type functional equation, whose solutions give rise to suitable physically motivated representations of the corresponding current Lie algebra $\mathcal{G}$.*

Recalling now the Hamiltonian operator representation (53), one can readily deduce that the following weak representation Hilbert space $\Phi_\mu$ weak relationship

$$\left(\left\langle \mathrm{A}|\rho^{-1}\mathrm{A}\right\rangle - \left\langle \mathrm{K}^*|\rho^{-1}\mathrm{A}\right\rangle - \left\langle \mathrm{A}|\rho^{-1}\mathrm{K}\right\rangle\right)/2 - \mathrm{V}(\rho) = \epsilon_0, \tag{65}$$

where $\epsilon_0 \in \mathbb{R}$ is the corresponding ground state energy density value. Thus, the main analytical problem is now reduced to constructing the expansion (62) corresponding to a suitable cyclic representation Hilbert space $\Phi_\mu$ of the quantum current algebra (29), compatible with the Hamiltonian operator structure (52).

**Remark 4.** *Here we mention that the operator $\mathrm{K}(x) : \Phi_\mu \to \Phi_\mu^m$, $x \in \mathbb{R}^m$, defined by (55), relates to that from the work [4,5,76] via scaling $\mathrm{K}(x) \to \mathrm{K}(x)/2, x \in \mathbb{R}^m$.*

*2.4. The Hamiltonian Operator Reconstruction and the Cyclic Current Algebra Representation*

We will assume that we are given a Banach current group $G = \mathrm{Diff}(\mathbb{R}^m) \ltimes F$ cyclic representation in a Hilbert space $\Phi_\mu$ with respect to $F$ with a cyclic vector $|\Omega) \in \Phi_+ \subset \Phi_\mu$. Based on the well known Araki reconstruction theorem [5,75] for the canonical Weyl commutation relations, we can first readily obtain from (56) that

$$[\mathrm{H}, U(\mathrm{f})] = J(\nabla \mathrm{f})U(\mathrm{f}) - 1/2\rho(\langle \nabla \mathrm{f}_1 | \nabla \mathrm{f}_2\rangle)U(\mathrm{f}), \tag{66}$$

where $U(\mathrm{f}) = \exp[i\rho(\mathrm{f})], \mathrm{f} \in F$, is an element of the unitary family $\mathcal{U}$. The expression (66) makes it possible to calculate the bilinear form

$$\begin{aligned}(U(\mathrm{f}_1)\Omega|\mathrm{H}|U(\mathrm{f}_2)\Omega) = (U(\mathrm{f}_1)\Omega|J(\nabla \mathrm{f}_1)|U(\mathrm{f}_2)\Omega) - \\ -1/2(U(\mathrm{f}_1)\Omega|\rho(\langle \nabla \mathrm{f}_1 | \nabla \mathrm{f}_2\rangle)|U(\mathrm{f}_2)\Omega)\end{aligned} \tag{67}$$

for any $f_1, f_2 \in F$. Taking into account the symmetry properties (57), we finally deduce from (67) that for arbitrary functions $f_1, f_2 \in F$

$$(U(f_1)\Omega|H|U(f_2)|\Omega) = 1/2(U(f_1)\Omega|\rho(\langle \nabla f_1|\nabla f_2\rangle)|U(f_2)\Omega). \tag{68}$$

The standard reasonings make it possible to state that the bilinear symmetric form (68) determines on $\Phi_\mu$ a self-adjoint non-negative definite Hamiltonian operator $H : \Phi_\mu \to \Phi_\mu$, densely defined on the domain $D_H := \underset{f \in F}{span}\{ \exp[i\rho(f)]|\Omega) \in \Phi_\mu\}$. Really, for any set of functions $f_j \in F, j = \overline{1, n}$, the following inequalities

$$\sum_{j,k=\overline{1,n}} \bar{s}_j s_k \langle \nabla f_j|\nabla f_k\rangle \geq 0, \quad \sum_{j,k=\overline{1,n}} \bar{s}_j s_k (U(f_j)\Omega|\rho(x)|U(f_k)\Omega) \geq 0 \tag{69}$$

hold for any complex numbers $s_j \in \mathbb{C}, j = \overline{1, n}$, and arbitrary $n \in \mathbb{N}$. Since, for any non-negative definite complex matrices $A, B \in \text{End } \mathbb{R}^n$, the matrix $C := \{A_{jk}B_{jk} : j, k = \overline{1, n}\} \in \text{End } \mathbb{C}^n$ proves to be non-negative definite [75,82] too, one ensures that the bilinear form (69) is also non-negative definite. Then, as follows from the classical Friedrichs' theorem [69,83–85], there exists a self-adjoint densely defined and non-negative definite operator $H : \Phi_\mu \to \Phi_\mu$.

*2.5. Current Algebra Representations, Generating Functional Method and Ergodicity of the Hilbert Space Representation Measure*

In view of the importance of the current algebra representations of the Banach group $G = \text{Diff } (\mathbb{R}^m) \ltimes F$ for physics applications, we consider their construction by means of the generating functional method [6,7,20,86]. From the very beginning, let us introduce a governing definition in connection with this method.

**Definition 10.** *A generating functional on a group G is a complex-valued function E on G with the following conditions: (1)　$E(1) = 1, 1 \in G$; (2)　$E(a_1 \exp(tA)a_2)$ is a continuous function of the parameter $t \in \mathbb{R}$ for all $A \in \mathcal{G}$ and $a_1, a_2 \in G$; (3) the matrix $\left\| E(a_k^{-1}a_j) \right\|, k, j = \overline{1, N}$, is positive definite for any $N \in \mathbb{N}$;*

The following theorem [75] holds.

**Theorem 2.** *The function E is a generating functional on G if, and only if, there exists a continuous unitary representation $\pi : G \to \text{Aut}(\Phi_\mu)$ on a separable Hilbert space $(\Phi_\mu; (\cdot|\cdot))$ with a cyclic vector $\Omega \in \Phi_\mu$, such that*

$$E(a) = (\Omega|\pi(a)\Omega) \tag{70}$$

*holds for all $a \in G$.*

The vector $\Omega \in \Phi_\mu$ is said to be cyclic with respect to the representation $\pi : G \to Aut(\Phi_\mu)$, if the set $\{\pi(a)\Omega : a \in G\}$ is complete in $\Phi_\mu$, i.e., is dense in $\Phi_\mu$, if taken together with its linear combinations over $\mathbb{C}$. The significance of this theorem is that one can implicitly construct unitary representations of the Banach current group $G = \text{Diff}(\mathbb{R}^m) \ltimes F$ and, thus, the current Lie algebra $\mathcal{G}$ by means of an appropriately defined generating functional on $G$. This is important, since frequently the latter problem is much simpler than the initial problem.

We now consider the representation $\pi : G \to \text{Aut}(\Phi_\mu)$, restricted to the Abelian subgroup $F$ in the group $G = \text{Diff } (\mathbb{R}^m) \ltimes F$ and its corresponding generating functional $\mathcal{L}(f), f \in F$, in the form

$$\mathcal{L}(f) := (\Omega| \exp[i\rho(f)]\Omega) = \int_{F'} d\mu(\eta) \exp[i\eta(f)], \tag{71}$$

where the cyclic vector $\Omega \in \Phi_\mu$ is normalized to unity: $(\Omega|\Omega) = 1$. In many physically interesting cases [6,7,20,86] the expression (71) can be replaced by means of the following equivalent trace-representation:

$$\mathcal{L}(f) = \text{Tr}(P \exp[i\rho(f)]), \qquad (72)$$

where $P : \Phi_\mu \to \Phi_\mu$ is the corresponding so called statistical operator, depending on the Hamiltonian operator $H : \Phi_\mu \to \Phi_\mu$. The constructed above generating functional (71) should possess the following necessary properties: (1) $\mathcal{L}(f) = \overline{\mathcal{L}(-f)}$ for all $f \in F$; (2) $\mathcal{L}(0) = 1$; (3) $|\mathcal{L}(f)| \leq 1$ for all $f \in F$; (4) $\mathcal{L}(f)$ is a positive definite functional on $F$: the inequality $\sum_{j,k=\overline{1,N}} \bar{c}_k \mathcal{L}(f_k - f_j) c_j \geq 0$ holds for all $c_j \in \mathbb{C}, j = \overline{1,N}$, and arbitrary $N \in \mathbb{N}$. As one can show, a generating functional $\mathcal{L} : F \to \mathbb{C}$, satisfying the properties (1)–(4) always defines [66] a measure $\mu$ on $F'$, defining the searched for unitary representation $\pi : G \to \text{Aut}(\Phi_\mu)$ of the Abelian subgroup $F$ of the current Banach group $G = \text{Diff}(\mathbb{R}^m) \ltimes F$. If the measure $\mu$ is in addition quasi-invariant and the factors $\chi_\varphi(\eta)$ in (35) are known for all $\varphi \in \text{Diff}(\mathbb{R}^m), \eta \in F'$, the corresponding representation of the current Lie algebra $\mathcal{G}$ is completely determined. Yet, if being interested only by irreducible representations of the current Banach algebra $\mathcal{G}$, it is well known [66] that the corresponding measure $\mu$ on $F'$ is ergodic for the diffeomorphism subgroup $\text{Diff}(\mathbb{R}^m)$, that is for any measurable and invariant subset $Q \subset F'$ either $\mu(Q) = 0$, or $\mu(F'\backslash Q) = 0$. Moreover, an arbitrary invariant set is in the general case a nondenumerable union of a family of mutually non-intersecting orbits. Assuming that the orbits containing an invariant subset $Q \subset F'$ are measurable, we obtain that there exist only two possibilities for ergodicity of the cylindrical measure $\mu$ on $F'$: either it is concentrated on one orbit, or each orbit has zero measure, and these two possibilities really occur in applications. For instance, the case when the measure is concentrated on functionals of the form (37) leads to irreducibility of the generating functional representation on the Hilbert space $L_2(\mathbb{R}^{mN}; \mathbb{C})$ for any finite $N \in \mathbb{N}$.

### 2.6. The Creation–Annihilation Heisenberg Algebra, Its Coherent State Representations and Linearization of Nonlinear Dynamical Systems on Hilbert Spaces

It is well known [87,88] that the representation theory of the quantum current algebra in a separable Hilbert space $\Phi_\mu$ is very close to the cyclic Hilbert space representations of the canonical creation–annihilation operator Heisenberg algebra $\mathcal{H}$ family $\{a^+(f), a(f) : \Phi_F \to \Phi_F : f \in F\}$, defined on the Fock space $\Phi_F$. The coherent states, being venerable objects in physics, were invented by Schrëdinger [89], as far back as in 1926, in the context of the quantum harmonic oscillator, they seemed to have lapsed into oblivion for some obscure reasons. About thirty-five years later, they were rediscovered, almost simultaneously, by Glauber [90], Klauder [91,92] and Sudarshan [93], in the context of a quantum optical description of coherent light beams emitted by lasers. Since then, coherent states have pervaded nearly all branches of quantum physics—including, of course, quantum optics in the study of lasers, nuclear, atomic and solid state physics, quantum electrodynamics, quantization and dequantization problems and path integrals, to mention just a few. For original references, the reader is referred to the review [88] and reprint volume of Klauder and Skagerstam [94]. In many of these applications, the question naturally poses itself as to whether it might not be possible to find other families of states, sharing some properties of the original or canonical coherent states, emanating from the quantum oscillator and which could possibly be useful to yet other areas of physics.

Already, in 1926, Schrëdinger had tried unsuccessfully to construct coherent states appropriate to the hydrogen atom problem. This was motivated by the quasi-classical character of the canonical coherent states which made them very desirable for studying the quantization of classical dynamical systems, a point which we discuss in some detail below. The key to the generalization of the notion of a coherent state was the observation by Perelomov [95] and independently by Gilmore [96,97], that the construction of the oscillator coherent states could be reformulated as a problem in group representation theory: the canonical coherent states could be obtained by acting on the oscillator ground

state with the operators of a unitary representation of the group generated by the creation and annihilation operators, namely the Weyl–Heisenberg group.

The link between the Schrëdinger and the Perelomov approaches is the uniqueness theorem [98,99] of von Neumann for the quantum mechanics of a system with finitely many degrees of freedom. In addition, a unitary representation $T : G \rightarrow U(G)$ of a compact symmetry group $G$ in a separable Hilbert space $\Phi$, used for building up the system of canonical coherent states, has the property of square integrability with respect to the left (or right) invariant Haar measure on $G$. Furthermore, the physical states, associated with the coherent states, are not indexed by elements of $G$ itself, but by points in the coset space $G/G^c$, where $G^c$ is the Cartan subgroup of $G$ and is isomorphic to the torus.

Since its introduction in 1972, the concept of coherent states was widely exploited [14–16,87,100,101] in many fields of mathematical physics, whose leading idea consisting of considering the translates of a fixed cyclic vector under a group action is as old as the celebrated Gel'fand-Raikov theorem [66] on locally bicompact groups. Their common properties, namely that the related homogeneous space has a complex homogeneous structure, and the corresponding representation Hilbert space can be identified in the coherent state basis with a space of holomorphic functions on the homogeneous space. As it was stated in [87], a homogeneous complex structure is actually present quite generally, and on the basis of the homogeneous complex structure, the related homogeneous manifolds are just the classical phase spaces on which the group acts through canonical transformations. From this point of view, coherent states can be interpreted just as probability wave packets over the classical phase space, that is a well-known result for the harmonic oscillator coherent states. The converse problem, i.e., the construction of irreducible unitary representations of the group, starting from its phase space realization, was considered in [102] and found a definite mathematical setting.

To look at the coherent vector representation problem within the Fock type space, its main idea becomes very transparent and motivative owing to the classical Bargmann–Segal [103] construction. Namely, there is considered the Hilbert space

$$\mathcal{H}_k := \{f \in H(\mathbb{C}^k) : \int_{\mathbb{C}^n} |f(z)|^2 d\mu(z) \tag{73}$$

of holomorphic functions $H(\mathbb{C}^k), k \in \mathbb{N}$, with the scalar product $(f|g) := \int_{\mathbb{C}^k} \overline{f(z)} g(z) d\mu(z)$ for arbitrary $f, g \in \mathcal{H}_k$ with respect to the measure $d\mu(z) = \pi^{-k} \exp(-\langle z|z \rangle) \frac{d\bar{z} \wedge dz}{(2i)^k}$ for $z \in \mathbb{C}^k$. It is easy to observe that the Hilbert space $\mathcal{H}_k$ is the direct sum of the symmetric polynomial subspaces $\mathcal{H}_k^{(s)}, s \in \mathbb{Z}_+$:

$$\mathcal{H}_k = \oplus_{s=0}^\infty \mathcal{H}_k^{(s)}, \tag{74}$$

where, by definition,

$$\mathcal{H}_k^{(s)} := \{ \sum_{s=n_1+n_2+...n_k} c_{n_1 n_2...n_k} z_1^{n_1} z_2^{n_2} ... z_k^{n_k} : c_{n_1 n_2...n_k}, z_j \in \mathbb{C}, j = \overline{1,k} \}. \tag{75}$$

Moreover, it is easy to check that the polynomials

$$e_{n_1 n_2...n_k}(z) = \prod_{j=\overline{1,k}} \frac{z_j^{n_j}}{\sqrt{n_j!}} \tag{76}$$

form a complete and orthogonal base in $\mathcal{H}_k$, that is $(e_{n_1 n_2...n_k} | e_{m_1 m_2...m_k}) = \prod_{j=\overline{1,k}} \delta_{n_j m_j}$. The next important observation, made by V. Bargmann, was the point boundedness of any

function $f \in \mathcal{H}_k : |f(z)| \leq ||f|| \exp(\langle z|z \rangle /2), z \in \mathbb{C}^n$. Really, for any $f \in \mathcal{H}_k$ there holds the expansion

$$f(z) = \sum_{s \in \mathbb{Z}_+} \sum_{s = n_1 + n_2 + \ldots n_k} (e_{n_1 n_2 \ldots n_k} | f) \prod_{j = \overline{1,k}} \frac{z_j^{n_j}}{\sqrt{n_j!}}, \tag{77}$$

from which one easily ensues, owing to the closedness property and Schwartz inequality on $l_2(\mathbb{C})$, that

$$
\begin{aligned}
|f(z)| \leq & \left( \sum_{s \in \mathbb{Z}_+} \sum_{s = n_1 + n_2 + \ldots n_k} |(e_{n_1 n_2 \ldots n_k} | f)|^2 \right)^{1/2} \times \\
& \times \left( \sum_{s \in \mathbb{Z}_+} \sum_{s = n_1 + n_2 + \ldots n_k} \prod_{j = \overline{1,k}} \frac{|z_j|^{2n_j}}{n_j!} \right)^{1/2} == ||f|| \exp(\langle z|z \rangle /2)
\end{aligned} \tag{78}
$$

The latter makes it possible to define for any $u \in \mathbb{C}^n$ the following dual to (78) bounded functional

$$\hat{u}(f) := f(u), \quad ||\hat{u}|| \leq \exp(\langle u|u \rangle /2) \tag{79}$$

on $\mathcal{H}_k$, whose Riesz representation

$$\hat{u}(f) = (h_u | f) \tag{80}$$

defines the unique element $h_u \in \mathcal{H}_k$, or equivalently

$$f(u) = \int_{\mathbb{C}^k} \overline{h_u(\xi)} f(\xi) \exp(-\langle \xi|\xi \rangle) \frac{d\bar{\xi} \wedge d\xi}{(2i)^k}. \tag{81}$$

Taking into account the orthogonality of the base vectors (76) in $\mathcal{H}_k$, it is easy to calculate that the vector $h_u(\xi) = \exp(\langle u|z \rangle) \in \mathcal{H}_k$, whose norm $||h_u|| = \exp(\langle u|u \rangle /2)$ for any $u \in \mathbb{C}^n$ and which is called the "*coherent vector*". It is worth remarking here that the function representation (81) is well known in the operator theory [104,105] and is called the "reproducing kernel" representation with the kernel $h_u \in \mathcal{H}_k, u \in \mathbb{C}^k$.

The Hilbert space $\mathcal{H}_k$, as the direct sum (74) of symmetrical polynomial subspaces, possesses the Fock space structure, allowing the introduction of the creation operators $a_j^+ : \mathcal{H}_k^{(s)} \to \mathcal{H}_k^{(s+1)}$ for any $j = \overline{1,k}$ and all $s \in \mathbb{Z}_+$ as multiplication operators: for any $f \in \mathcal{H}_k^{(s)} \ a_j^+ f(z) := z_j f(z)$ for any $j = \overline{1,k}$ and all $s \in \mathbb{Z}_+$. The corresponding adjoint expressions $\left( a_j^+ \right)^* := a_j : \mathcal{H}_k^{(s)} \to \mathcal{H}_k^{(s-1)}$ act as $a_j f(z) = \partial/\partial z_j f(z)$ on arbitrary $f \in \mathcal{H}_k^{(s)}$ for any $j = \overline{1,k}$ and all $s \in \mathbb{Z}_+$, where, by definition, $\mathcal{H}_k^{(0)} \simeq \mathbb{C}$. Now one can easily check that the coherent vector $h_u = \exp(\langle u|\cdot \rangle) \in \mathcal{H}_k$ for any $u \in \mathbb{C}^k$ is a common eigenvector of the annihilation operators $a_j : \mathcal{H}_k \to \mathcal{H}_k, j = \overline{1,k}$:

$$a_j h_u(z) = u_j h_u(z) \tag{82}$$

with the eigenvalues $u_j \in \mathbb{C}, j = \overline{1,k}$. It is important also to mention here that the creation–annihilation operators defined above satisfy the canonical commutation relationships:

$$[a_j, a_n] = 0 = [a_j^+, a_n^+], \ [a_j, a_n^+] = \delta_{j,n} \tag{83}$$

for all $j, n = \overline{1,k}$.

The coherent vector representation scheme described above can be respectively generalized to arbitrary symmetric Fock space $\Phi$ that will be effectively used in the sections proceeding below. Returning back to the algebraic properties of coherent states, we proceed to describing their unbelievable and impressive applications to theory of nonlinear dynamical systems on Hilbert spaces, their linearization and integrability, previously initiated

in [14,15] and continued in [16]. We briefly reviewed the cyclic Hilbert space representations of the quantum Heisenberg algebra and presented a general approach to constructing the coherent states and their applications both to the linearization of nonlinear dynamical systems on Hilbert spaces, and to describing their complete integrability. The latter is developed using the modern Lie-algebraic approach [11,17–19] to nonlinear dynamical systems on Poissonian functional manifolds, and proved to be both unexpected and important for the classification of integrable Hamiltonian flows on Hilbert spaces.

Jointly with the cyclic Hilbert space representations of the Heisenberg algebra $\mathcal{H}$, we briefly reviewed the closely related cyclic Hilbert space density representations [4,6,87,88] of the canonical quantum current algebra $\mathcal{G}$ on the circle $\mathbb{S}^1$, whose vector field representations on smooth spatially one-dimensional functional manifolds coincide exactly with the related symmetry algebra of completely integrable nonlinear Hamiltonian systems on these manifolds. Based on the current algebra symmetry structure and their functional representations, an effective integrability criterion is formulated for a wide class of completely integrable Hamiltonian systems on smooth spatially one-dimensional functional manifolds. The algebraic structure of the Poissonian operators and an effective algorithm of their analytical construction are also described.

### 2.7. The Canonical Heisenberg Algebra and Its Cyclic Hilbert Space Representations

Let $(\Phi; (\cdot|\cdot))$ be a separable Hilbert space, $F$ be a topological real linear space and $\mathcal{A} := \{A(f) : f \in F\}$ a family of commuting self-adjoint operators in $\Phi$ (i.e., these operators commute in the sense of their resolutions of the identity) with dense in $\Phi$ domain $Dom\ A(f) := D_{A(f)}, f \in F$. Consider the Gelfand rigging [57,61,66] of the Hilbert space $\Phi$, i.e., a chain

$$\mathcal{D} \subset \Phi_+ \subset \Phi \subset \Phi_- \subset \mathcal{D}' \tag{84}$$

in which $\Phi_+$ is a Hilbert space, topologically (densely and continuously) and quasi-nucleus (the inclusion operator $i : \Phi_+ \longrightarrow \Phi$ is of the Hilbert-Schmidt type) embedded into $\Phi$, the space $\Phi_-$ is the dual to $\Phi_+$ as the completion of functionals on $\Phi_+$ with respect to the norm $||f||_- := \sup\limits_{||u||_+=1} |(f|u)_\Phi|$, $u \in \Phi$, a linear dense in $\Phi_+$ topological space $\mathcal{D} \subseteq \Phi_+$ is such that $\mathcal{D} \subset D_{A(f)} \subset \Phi$ and the mapping $A(f) : \mathcal{D} \to \Phi_+$ is continuous for any $f \in F$. Then, owing to the structural theorem (1) there exists a cyclic representation of the canonical creation–annihilation Heisenberg operator algebra $\mathfrak{H}$ family $\{a^+(f), a(f) : \Phi_\mu \to \Phi_\mu : f \in F\}$ on the separable Hilbert space $\Phi_\mu$, whose generalized Fourier transform is given by the expression

$$\Phi_\mu = L_2^{(\mu)}(F'; \mathbb{C}) \simeq \int_{F'}^{\oplus} \Phi_{(\eta)} d\mu(\eta) \tag{85}$$

for some Hilbert space sets $\Phi_\eta$, $\eta \in F'$, and a suitable measure $\mu$ on $F'$, with respect to which the corresponding joint eigenvector $\omega(\eta) \in \Phi_-$ for any $\eta \in F'$ generates the Fourier transformed family $\{\eta(f) \in \mathbb{R} : f \in F\}$. Moreover, if $\dim \Phi_\eta = 1$ for all $\eta \in F'$, the Fourier transformed eigenvector $\omega(\eta) := \Omega(\eta) = 1$ for all $\eta \in F'$.

Next, we will consider the family of self-adjoint operators $\{P(f), Q(g) : \Phi_\eta \to \Phi_\eta : f, g \in F\}$, as generating a unitary Heisenberg group

$$\mathfrak{H} := \{\exp(iP(f)), V(g) = \exp(iQ(g) : \tag{86}$$
$$P := (a^+ + a)/2, Q := i(a - a^+)/2, f, g \in F, \}$$

satisfying the commutation conditions

$$U(f)V(g) = \exp(-i(f|g))V(g)U(f), \tag{87}$$
$$U(f)U(g) = U(f + g), \quad V(f)V(g) = V(f + g),$$

for any $f, g \in F$. Since, in general, the unitary Heisenberg group $\mathfrak{H}$ is defined on a representation Hilbert space $\Phi_\mu$, not coinciding, in general, with the canonical Fock type space

$\Phi_F$, the important problem of describing its cyclic unitary representation spaces arises, within which the factorization (86) jointly with relationships (87) should hold. Below, we will briefly describe only the main features of the Gelfand–Vilenkin formalism, being much more suitable for the task, providing a reasonably unified framework of constructing the corresponding cyclic representations of the family $\mathcal{A} := \{Q(\mathrm{f}) : \mathrm{f} \in F\}$ of commuting self-adjoint operators in a separable Hilbert space $\Phi$.

Proceeding now to the Heisenberg group $\mathfrak{H}$, the separable Hilbert space $\Phi_\mu$ for its every irreducible representation will be unitary equivalent to the Hilbert space (45), which in many physical applications reduces in the case $\dim \Phi_{(\eta)} = 1$ for all $\eta \in F'$ to the following form:

$$\Phi_\mu \simeq L_2^{(\mu)}(F'; \mathbb{C}), \tag{88}$$

being the space of square integrable functions with respect to the measure $\mu$ on $F'$.

Assume now that an element $\omega \in \Phi_\mu$ is taken arbitrarily and consider [75] the action of the Heisenberg group $\mathfrak{H}$ on it:

$$U(\mathrm{f})\omega(\eta) = \exp[i(\eta(\mathrm{f})]\omega(\eta), \tag{89}$$

$$V(\mathrm{g})\omega(\eta) = \chi_\mathrm{g}(\eta)\omega(\eta + \mathrm{g})\left[\frac{d\mu(\eta + \mathrm{g})}{d\mu(\eta)}\right]^{1/2},$$

where, by definition, for any $\mathrm{f} \in F$ the expression $\frac{d\mu(\eta+\mathrm{g})}{d\mu(\eta)}$ at $\eta \in F'$ means the corresponding Radon–Nikodym derivative [19,73] of the measure $\mu(\circ + \mathrm{g})$ with respect to the measure $\mu$ on $F'$ and $\chi_\mathrm{g}(\eta)$ is a complex-valued character of the unit norm, satisfying the relationship

$$\chi_\mathrm{f}(\eta)\chi_\mathrm{g}(\eta + \mathrm{f}) = \chi_{\mathrm{f}+\mathrm{g}}(\eta) \tag{90}$$

for all $\mathrm{f}, \mathrm{g} \in F \subset H$ and arbitrary $\eta \in F'$. For the Radon–Nikodym derivative above to exist, the measure $\mu$ on $F'$ should be quasi-invariant with respect to the shift group elements $\{F' \ni \eta \to \eta + \mathrm{g} \in F'\}$, that is, for any measurable set $Q \subset F'$ the condition $\mu(Q) = 0$ if, and only if, $\mu(Q + \mathrm{g})$ for arbitrary $\mathrm{g} \in F \subset F'$.

**Definition 11.** *A vector $|u) \in \Phi_\mu$ is called a coherent vector state in the representation Hilbert space $\Phi_\mu$ with respect to an element $u \in H \simeq L_2(\mathbb{R}^m; \mathbb{R}^k)$, if it satisfies the eigenfunction condition*

$$a_j(x)|u) = u_j(x)|u) \tag{91}$$

*for each $j = \overline{1, k}$ and all $x \in \mathbb{R}^m$.*

It is easy to check that for any $u \in H$ the coherent ket-vector $|u) \in \Phi_\mu$ exists: really, the following vector expression

$$|u) := \exp[(u|a^+)_H]|\Omega) \tag{92}$$

where $\Omega \in \Phi_+ \subset \Phi_\mu$ is a cyclic vector for the creation–annihilation operator algebra family $\{a^+(\mathrm{f}), a(\mathrm{f}) : \Phi_\mu \to \Phi_\mu : \mathrm{f} \in F\}$ and satisfies the defining condition (91), where the operator $a^+(u) : \Phi_\mu \to \Phi_\mu, u \in H$, action ensues from the determining condition (19): for any $\varphi \in \Phi_\mu$ there exists a unique vector $\omega(\eta_a) \in \Phi_\mu$ for which

$$(\omega(\eta_a)|(a^+(u)\varphi)_\mu = \eta_a(u)\,(\omega(\eta_a)|\varphi)_\mu \tag{93}$$

for all $u \in H$. Moreover, as the Hilbert space $H \subset F'$, the eigenvalue $\eta_a(u) \in \mathbb{R}$ is bounded jointly with the Hilbert space $\Phi_\mu$ norm

$$\|u\| := (u|u)^{1/2} = \exp(\frac{1}{2}\|u\|_H^2) < \infty, \tag{94}$$

since $u \in H$ and its Hilbert space norm $\|u\|_H$ is *a priori* bounded.

Consider now any function $u \in H$ and observe that the Hilbert spaces embedding mapping

$$\xi : H \ni u \longrightarrow |u) \in \Phi_\mu, \tag{95}$$

defined by means of the coherent vector expression (92), realizes a smooth isomorphism between the Hilbert spaces $H$ and the image $\xi(H) \subset \Phi_\mu$. The inverse mapping $\xi^{-1} : \xi(H) \subset \Phi_\mu \longrightarrow H$ is given by the following exact expression:

$$(u|\eta)_H = (\Omega|a(\eta)|u) / (\Omega|u), \tag{96}$$

holding for any $\eta \in H$. Owing to condition (94), one finds from (96) and the classical Riesz type theorem [85,106] that the corresponding function $u \in H$.

Let now define on the Hilbert space $H$ a nonlinear in general dynamical system (which can, in general, be non-autonomous) in partial derivatives

$$du/dt = K[u], \tag{97}$$

where $t \in \mathbb{R}_+$ is the corresponding evolution parameter, $[u] := (x; u, u_x, u_{xx}, ...,) \in J^{(k)}(\mathbb{R}^m; \mathbb{R}^s)$ belongs to the jet-space $J^{(k)}(\mathbb{R}^m, \mathbb{R}^n)$ of the order $k \in \mathbb{Z}_+$, and, in general, a nonlinear mapping $K : H \longrightarrow H$ is Frechet smooth. Assume also that the corresponding Cauchy problem

$$u|_{t=+0} = u_0 \tag{98}$$

for the nonlinear dynamical system (97) is solvable in the Hilbert space $H$ for any $u_0 \in H$ on an interval $[0, T) \subset \mathbb{R}_+^1$ for some $T > 0$. Thus, there is determined a smooth evolution mapping

$$T_t : H \ni u_0 \longrightarrow u(t|u_0) \in H, \tag{99}$$

for all $t \in [0, T)$. Now, it is natural to consider the following commuting diagram:

$$\begin{array}{ccc} H & \xrightarrow{\xi} & \Phi_\mu \\ T_t \downarrow & & \downarrow \mathrm{T}_t \\ H & \xrightarrow{\xi} & \Phi_\mu, \end{array} \tag{100}$$

where the mapping $\mathrm{T}_t : \Phi_\mu \longrightarrow \Phi_\mu, t \in [0, T)$, is defined from the conjugation relationship on the image $\xi(H) \subset \Phi_\mu$ of the mapping (95):

$$\xi \circ T_t = \mathrm{T}_t \circ \xi \tag{101}$$

Now take coherent vector $|u_0) \in \Phi_\mu$, corresponding to the Cauchy data $u_0 \in H$, and construct the vector

$$|u) := \mathrm{T}_t \cdot |u_0) \in \Phi_\mu \tag{102}$$

for all $t \in [0, T)$. Since the vector (102) is, by construction, coherent, that is

$$a_j(x)|u) := u_j(x, t|u_0)|u) \tag{103}$$

for each $j = \overline{1, k}, t \in [0, T)$ and almost all $x \in \mathbb{R}^m$, owing to the smoothness of the mapping $\xi : H \longrightarrow \Phi_\mu$ with respect to the corresponding norms in the Hilbert spaces $H$ and $\Phi_\mu$, we derive that the coherent vector (102) is differentiable with respect to the evolution parameter $t \in [0, T)$. Thus, one can easily find [14,15] that

$$\frac{d}{dt}|u) = \mathrm{K}(a^+, a)|u), \tag{104}$$

where

$$|u)|_{t=+0} = |u_0) \tag{105}$$

and an operator mapping $K(a^+, a) : \Phi_\mu \longrightarrow \Phi_\mu$ is defined by means of the exact analytical expression

$$K(a^+, a) := (a^+ | K[a])_H. \tag{106}$$

As a result of the consideration above we obtain the following theorem.

**Theorem 3.** *Any smooth nonlinear dynamical system* (97) *in Hilbert space H is representable by means of the Hilbert spaces embedding isomorphism* $\xi : H \longrightarrow \Phi_\mu$ *via the completely linear form* (104).

We now make some comments concerning the solution to the linear Equation (104) under the Cauchy condition (105) in the case of the Fock representation space $\Phi_F$. Since any vector $|\omega) \in \Phi_F$ allows the series representation

$$
\begin{aligned}
|\omega) = \bigoplus_{n=\sum_{j=1}^k n_j \in \mathbb{Z}_+} &\int_{(\mathbb{R}^m)^n} \omega^{(n)}_{n_1 n_2 \dots n_s}(x_1^{(1)}, x_1^{(2)}, \dots, x_1^{(n_1)}; \\
&x_2^{(1)}, x_2^{(2)}, \dots, x_2^{(n_2)}; \dots; x_k^{(1)}, x_k^{(2)}, \dots, x_k^{(n_m)}) \prod_{j=1}^k \left( \frac{1}{\sqrt{n_j!}} \prod_{k=1}^{n_j} dx_k^{(j)} a_j^+(x_j^{(k)}) \right) |\Omega),
\end{aligned}
\tag{107}
$$

where for any $n = \sum_{j=1}^k n_j \in \mathbb{N}$ functions

$$\omega^{(n)}_{n_1 n_2 \dots n_k} \in \bigotimes_{j=1}^k L_2^{(s)}((\mathbb{R}^m)^{n_j}; \mathbb{C}) \simeq L_2^{(s)}(\mathbb{R}^{mn_1} \times \mathbb{R}^{mn_2} \times \dots \mathbb{R}^{mn_k}; \mathbb{C}), \tag{108}$$

its Fock space norm is easily calculated as

$$\|\omega\|^2 = \sum_{n=\sum_{j=1}^k n_j \in \mathbb{N}} \|\omega^{(n)}_{n_1 n_2 \dots n_k}\|_2^2. \tag{109}$$

For the case of the coherent vector $|u) \in \Phi_F$ its norm is easily obtained as $\|u\| = \exp(\|u\|_H^2/2)$, coinciding with the result (94). Moreover, substituting (107) into Equation (104), reduces (104) to an infinite recurrent set of linear evolution equations in partial derivatives on coefficient functions (108). The latter can often be solved [14] step by step analytically in exact form, thereby, making it possible to obtain, owing to representation (96), the exact solution $u \in H$ to the Cauchy problem (98) for our nonlinear dynamical system in partial derivatives (97).

Concerning possible applications of nonlinear dynamical systems like (95) in mathematical physics, it is very important to construct their so called conservation laws or smooth invariant functionals $\gamma : H \longrightarrow \mathbb{R}$ on the Hilbert space $H$. Making use of the quantum mathematics technique described above, one can suggest an effective algorithm for constructing these conservation laws in exact form.

Indeed, consider a vector $|\gamma) \in \Phi_\mu$, satisfying the linear equation:

$$\frac{\partial}{\partial t}|\gamma) + K^*(a^+, a)|\gamma) = 0. \tag{110}$$

Then, the following proposition [14,15] holds.

**Proposition 4.** *The functional*

$$\gamma := (u|\gamma) \tag{111}$$

*is a conservation law for dynamical system* (95), *that is*

$$d\gamma/dt|_K = 0 \tag{112}$$

*along all orbits of the evolution mapping* (99).

It is interesting to reanalyze the dynamical system (104) from the Lie-algebraic point of view [11,19] and represent it as a coadjoint canonical Hamiltonian flow on the corresponding adjoint space to the Hilbert space $\Phi_\mu$, considered as a Lie algebra over the field $\mathbb{C}$. To do this, it is necessary to define the related Lie commutator on the Hilbert space $\Phi_\mu$ : for any vectors $|K_\alpha) := \mathrm{K}_\alpha(a^+, a)^*|\omega) \in \Phi_\mu$ and $|K_\beta) := \mathrm{K}_\beta(a^+, a)^*|\omega) \in \Phi_\mu$, where $\mathrm{K}_\alpha(a^+, a)^*$ and $\mathrm{K}_\beta(a^+, a)^* \in \mathrm{End}\,\Phi_\mu$ are smooth mappings and a central vector $|\omega) \in \Phi_\mu$ is chosen to be fixed, their commutator, defined as

$$[|K_\beta), |K_\alpha)] := [\mathrm{K}_\beta(a^+, a)^*, \mathrm{K}_\alpha(a^+, a)^*|\omega), \tag{113}$$

allows the construction of the related co-adjoint action $\Phi_\mu^* \ni (l| \to ad^*_{|K_\alpha}(u| \in \Phi_\mu^*$ of a vector $|K_\alpha) \in \Phi_\mu$ on a fixed element $(l| \in \Phi_\mu^*$, where for any vector $|\eta) \in \Phi_\mu$ there holds the following identity:

$$(ad^*_{|K_\alpha})(l|\eta) = -(l|[K_\alpha), \eta]). \tag{114}$$

The latter makes it possible to define on the adjoint space $\Phi_\mu^*$ the classical Lie–Poisson bracket for any smooth functionals $\alpha := (l|\alpha)$ and $\beta := (l|\beta) \in \mathcal{D}(\Phi_\mu^*)$:

$$\{\alpha, \beta\} := (l|[\mathrm{grad}\,\alpha(l), \mathrm{grad}\,\beta(l)]_\vartheta|\omega) = (l|\,\mathrm{grad}\,\alpha(l)\vartheta(a^+)\,\mathrm{grad}\,\beta(l)|\omega), \tag{115}$$

where, by definition, $\vartheta(a^+) : \Phi_\mu \to \Phi_\mu$ is some skew-symmetric Poisson operator, the element $(l| := (l(u)| \in \Phi_\mu^*$ for any $u \in H$ is interpreted as the corresponding momentum mapping $H \ni u \xrightarrow{l} (l(u)| := (u| \in \Phi_\mu^*$ for the Poissonian action of the Lie algebra $\Phi_\mu$ on the Hilbert space $H$:

$$\Phi_\mu \times H \ni (\mathrm{grad}\,\gamma(u) \times u) \to \ \mathrm{K}_\gamma|u) \in \Phi_\mu \tag{116}$$

with $\mathrm{K}_\gamma := (a^+|\,K_\gamma[a])_H \in \mathrm{End}\,\Phi_\mu$, $K_\gamma[a]^* = -\vartheta(a^+)\,\mathrm{grad}\,\gamma(l) : \Phi_\mu \to \Phi_\mu$ for arbitrary $\gamma \in \mathcal{D}(\Phi_\mu^*)$. The related action

$$(\mathrm{grad}\,\gamma(u) \times u) \ = K_\gamma[u] \tag{117}$$

is a Hamiltonian vector field on $H$, generated by the corresponding Hamiltonian vector field $K_\gamma : H \to H$ on the Hilbert space $H$ commonly with the invariant Hamiltonian function $\gamma = (u|\gamma) \in \mathcal{D}(H)$. Simultaneously, the flow (117) for any $\gamma \in \mathcal{D}(\Phi_\mu^*)$ naturally generates the linear flow on the adjoint space $\Phi_\mu^* \simeq \Phi_\mu$ in the form $^+$

$$ad^*_{\vartheta(a^+)|\gamma)}(u| = (u|\mathrm{K}_\gamma^*. \tag{118}$$

Moreover, one easily checks that the commutator of vector fields $\ \mathrm{K}_\alpha|u)$ and $\mathrm{K}_\beta|u) \in \Phi_\mu$ equals

$$[\mathrm{K}_\alpha|u), \mathrm{K}_\beta|u)] := [\mathrm{K}_\alpha, \mathrm{K}_\beta]|u) \tag{119}$$

for any smooth conservation laws $\alpha = (u|\alpha)$ and $\beta = (u|\beta) \in \mathcal{D}(H)$ of the dynamical system (104), easily following from the evident conditions $\mathrm{K}^*|\alpha) = 0$ and $\mathrm{K}^*\,|\mu) = 0$. Consider now the following representations of the gradient vectors $\ \mathrm{grad}\,\alpha(u) = |\alpha(a^+)|\omega)$ and $\ \mathrm{grad}\,\beta(u) = |\beta(a^+)|\omega)$ for some fixed central element $|\omega) \in \Phi_\mu$. Then, the Poisson bracket (115) for any $\alpha, \beta \in \mathcal{D}(H)$ is representable as

$$\{\alpha, \beta\} = -(u|[\mathrm{K}_\alpha^*|\omega), \mathrm{K}_\beta^*|\omega)]_\vartheta) = (u|[\mathrm{K}_\alpha, \mathrm{K}_\beta]^*|\omega) = \tag{120}$$

$$= (u|\mathrm{K}_{\{\alpha, \beta\}}^*|\omega) = -(u|\vartheta\,\mathrm{grad}\{\alpha, \beta\}),$$

being completely compatible with the Poissonian action of the Lie algebra $\Phi_\mu$ on the Hilbert space $H$. The obtained result can be summarized as the following theorem.

**Theorem 4.** *If the momentum mapping $H \ni u \xrightarrow{l} (l(u)| := (u| \in \Phi_\mu^*$, related with the nonlinear dynamical system ((97) on the Hilbert space H, is Poissonian, then all its symmetries (117), generated by smooth invariants $\gamma \in \mathcal{D}(\Phi_\mu^*)$, are represented as linear Hamiltonian flows (118) on the adjoint Hilbert space $\Phi_\mu^* \simeq \Phi_\mu$ with respect to the canonical Lie–Poisson bracket (120).*

The theorem above plays a decisive role in constructing within the suitably modified Adler–Kostant–Souriau [11,19] scheme integrable Hamiltonian flows on the adjoint space $\Phi_\mu^*$, equivalent to nonlinear integrable Hamiltonian systems on the functional Hilbert space $H$.

*2.8. Conclusions*

Within the scope of this Section we have described the main mathematical preliminaries and properties of the quantum mathematics techniques suitable for analytical studying of the important linearization problem for a wide class of nonlinear dynamical systems in partial derivatives in Hilbert spaces. This problem was analyzed in much detail using the Gelfand–Vilenkin representation theory [66] of infinite dimensional groups and the Goldin–Menikoff–Sharp theory [4,5,74] of generating Bogolubov type functionals, classifying these representations. The related problem of constructing cyclic Hilbert space representations and retrieving their creation–annihilation generating structure still needs a deeper investigation within the approach devised. Here we mention only that some aspects of this problem within the so-called Poissonian White noise analysis which was analyzed in a series of works [55,70,107,108], based on some generalizations of the Delsarte type characters technique. The above-stated theorem about the Hamiltonian structure of symmetries of a nonlinear dynamical system on a Hilbert space and their linearization on a suitably constructed Hilbert space presents, from a practical point of view, a strong interest, if the related results, obtained in [14,15,109,110] and devoted to the application of the Hilbert spaces embedding method to finding conservation laws and the so called recursion operators for the well [17,111] known Korteweg–de Vries type nonlinear dynamical systems, are taken into account. Moreover, a development of these results within the modern Lie-algebraic approach, based on the Adler–Kostant–Symes construction and applied to nonlinear dynamical systems on Poissonian functional manifolds, proves to be both unexpected and important for the classification of integrable Hamiltonian flows on Hilbert spaces, and inspires a hope for new investigations of coherent states and their applications.

**3. Quantum Current Lie Algebra as a Universal Algebraic Structure of Symmetries of Completely Integrable Nonlinear Dynamical Systems**

*3.1. Quantum Lie Algebra of Currents and Its Vector Field Representations*

We consider the non-relativistic quantum Lie algebra $\mathcal{G}$ of currents [4,12,112,113] on the torus $\mathbb{T}^n$, realized by means of the density $\rho(f)$ and current $J(g)$ operators on the separable Hilbert subspace $\Phi_\mu$:

$$[\rho(f_1), \rho(f_2)] = 0, [\rho(f), J(g)] = J(\langle g | \nabla f \rangle), \tag{121}$$
$$[J(g_1), J(g_2)] = iJ([g_2, g_1]),$$

where $\rho(f) = \int_{\mathbb{T}^n} f(x)\rho(x)dx, J(g) = \int_{\mathbb{T}^n} g(x)J(x)dx$ for $f, f_j \in F \simeq C^\infty(\mathbb{T}^n; \mathbb{R}), g, g_j \in F^n, j = \overline{1,2}$. Their representation on the Fock space $\Phi_F$ is given, respectively, by the following operator expressions: $\rho(x) = a^+(x)a(x)$ and $J(x) = \frac{1}{2i}[a^+(x)\nabla a(x) - \nabla a^+(x)a(x)]$, where $a^+(x)$ is the creation and $a(x)$ is the annihilation operators of bose-particle states at point $x \in \mathbb{T}^n$, satisfying the canonical commutation relationships:

$$[a(x), a(y)] = 0, [a^+(x), a^+(y)],$$
$$[a(x), a^+(y)] = \delta(x - y)$$

for all $x, y \in \mathbb{T}^n$. The current Lie algebra (121) is the infinite-dimensional Lie algebra of the semi-direct product $G := \text{Diff}(\mathbb{T}^n) \ltimes F$ of the Banach Lie group of currents $G := \text{Diff}(\mathbb{T}^n)$ and the abelian functional group $F$, where $\text{Diff}(\mathbb{T}^n)$ is the topological group of diffeomorphisms [2,20] of the torus $\mathbb{T}^n$. If, to introduce [2,20,22,112,114,115] a family of unitary operators $U(\mathrm{f})$ and $V(\varphi_t^{\mathrm{g}}) : \Phi_\mu \to \Phi_\mu$, acting on a Hilbert space $\Phi_\mu$ and defined by the formulas

$$U(\mathrm{f}) = \exp[i\rho(\mathrm{f})], V(\varphi_t^{\mathrm{g}}) = \exp[itJ(\mathrm{g})], \tag{122}$$

where $d\varphi_t^{\mathrm{g}}/dt := \mathrm{g}(\varphi_t^{\mathrm{g}}), t \in \mathbb{R}, \varphi_t^{\mathrm{g}} \in \text{Diff}(\mathbb{T}^n)$ and $\varphi_t^{\mathrm{g}}|_{t=0} = x \in \mathbb{T}^n$, then the following relations

$$U(\mathrm{f}_1)U(\mathrm{f}_2) = U(\mathrm{f}_1 + \mathrm{f}_2), V(\varphi)U(\mathrm{f}) = U(\mathrm{f} \circ \varphi)V(\varphi), \tag{123}$$
$$V(\varphi_1)V(\varphi_2) = V(\varphi_2 \circ \varphi_1)$$

hold for all $\mathrm{f}, \mathrm{f}_j \in F$ and $\varphi, \varphi_j \in \text{Diff}(\mathbb{T}^n), j = \overline{1, 2}$. As was argued in [2], the various unitary representations of the current group $G$ describe different physical systems and their states, and the study of the set of cyclic unitarily irreducible representations of the Banach Lie group relationships (123) is an extremely important and topical problem in the quantum theory of dynamical systems.

For every irreducible cyclic representation of the unitary current group $G$ on the separable Hilbert space $\Phi_\mu$ there exists a unitarily equivalent Hilbert space

$$\Phi_\mu \simeq \int_{F'}^{\oplus} d\mu(\eta) \Phi_{(\eta)}, \tag{124}$$

where $\mu$ is the measure on the space $F'$ of continuous real linear functionals on $F$ and $\Phi_{(\eta)}$ are complex linear finite-dimensional spaces labeled by the index $\eta \in F'$. In the case when $\dim \Phi_{(\eta)} = 1, \Phi_\mu \simeq L_2^{(\mu)}(F'; \mathbb{C})$, the space of complex-valued functions on $F'$, integrable with respect to the measure $\mu$ on $F'$. Moreover, if an element $\omega \in \Phi_\mu$, then for the action of the current group $G$ on this element we have the following representations:

$$U(\mathrm{f})\omega(\eta) = exp[i(\eta(\mathrm{f})]\omega(\eta), \tag{125}$$
$$V(\varphi)\omega(\eta) = \chi_\varphi(\eta)\omega(\varphi^*\eta)\left(\frac{d\mu(\varphi^*\eta)}{d\mu(\eta)}\right)^{1/2}$$

where, by definition, $\varphi^*\eta(\mathrm{f}) := \eta(\mathrm{f} \circ \varphi)$ for all $\mathrm{f} \in F$, $\frac{d\mu(\varphi^*\eta)}{d\mu(\eta)}$ is the corresponding Radon–Nikodym derivative of the measure $\mu \circ \varphi^*$ with respect to the measure $\mu$ on $F'$ and $\chi_\varphi(\eta)$ is a complex-valued character of the unit norm, satisfying the relationship

$$\chi_{\varphi_2}(\eta)\chi_{\varphi_1}(\varphi_2^*\eta) = \chi_{\varphi_1 \circ \varphi_2}(\eta) \tag{126}$$

for all $\varphi_j \in \text{Diff}(\mathbb{T}^n), j = \overline{1, 2}, \eta \in F'$. For the Radon–Nikodym derivative above to exist, the measure $\mu$ on $F'$ should be quasi-invariant with respect to the diffeomorphism group $\text{Diff}(\mathbb{T}^n)$, that is for any measurable set $Q \subset F'$ the condition $\mu(Q) = 0$ if, and only if, $\mu(\varphi^*Q)$ for arbitrary $\varphi \in \text{Diff}(\mathbb{T}^n)$.

In physics applications, the representation (125) is uniquely determined by the measure $\mu$ on $F'$, which in the general case has a very complicated [20,72,114] structure, and its analytic construction is nontrivial. One of the fairly effective approaches to this problem is the quantum method of Bogolubov generating functionals developed in [5–7,20]. Another approach, which is of considerable interest for the theory of dynamical systems, is based on algebraic methods of constructing self-adjoint functional-operator representations of the original current Lie algebra (121). We proceed to its description in the case of the current group $G = \text{Diff}(\mathbb{S}^1) \ltimes F$, where $F \simeq C^\infty(\mathbb{S}^1; \mathbb{R})$ on the circle $\mathbb{S}^1$, taking into account results in [12,22,116–118].

We now introduce the following basis operators of the Lie current Lie algebra (121) for $n = 1$:

$$\rho_j := \int_{\mathbb{S}^1} exp(ijx + i\varepsilon x)\rho(x)dx, \qquad J_k := \int_{\mathbb{S}^1} exp(ikx)J(x)dx, \qquad (127)$$

where $j, k \in \mathbb{Z}$ and $\varepsilon \in \mathbb{R}$ is a parameter. Then from (121) and (127), we find that

$$[\rho_j, \rho_k] = 0, \quad [J_k, \rho_j] = (j + \varepsilon)\rho_{j+k}, \quad [J_k, J_j] = (j - k)J_{k+j} \qquad (128)$$

for all $j, k \in \mathbb{Z}$, that is the set $\mathcal{G} := \{ \rho_j, J_k : \Phi_\mu \to \Phi_\mu : j, k \in \mathbb{Z}\}$ of operators (128) on the representation Hilbert space $\Phi_\mu$, is equivalent to the semidirect product $\mathcal{G}\{J\} \ltimes \mathcal{G}\{\rho\}$ of the Lie subalgebra $\mathcal{G}\{J\} := \{ J_k : \Phi_\mu \to \Phi_\mu : k \in \mathbb{Z}\}$ and the Abelian subalgebra $\mathcal{G}\{\rho\} := \{\rho_j : \Phi_\mu \to \Phi_\mu : j \in \mathbb{Z}\}$ and isomorphic to the current Lie algebra (121) for $n = 1$. It is also worth mentioning [119] that in the case of functional-operator representations, the Lie algebra (128) admits the following central extension by means of the Schwinger cocycle:

$$[\rho_j, \rho_k] = \zeta\rho_{j,-k}, \quad [J_k, \rho_j] = (j + \varepsilon)\rho_{j+k}, \quad [J_k, J_j] = (j - k)J_{k+j} + \nu k(k^2 - 1)\delta_{j,-k}, \qquad (129)$$

where $j, k \in \mathbb{Z}$ and $\zeta, \nu \in \mathbb{R}$ are the Schwinger parameters. The current Lie algebra (129) is called the generalized Virasoro current algebra [44] and has many applications in modern theoretical physics.

It is easy to show that the current Lie algebra (128) for $\varepsilon = 0$ admits the standard representation in the ring of operators $\mathbb{C}[\lambda, \lambda^{-1}][\partial/\partial\lambda], \lambda \in \mathbb{C}^N$, regarded as a Lie subalgebra of the Lie algebra of rational vector fields on $\mathbb{C}^N, N \in \mathbb{N}$. Namely, if we set

$$\rho_j = \sum_{n=\overline{1,N}} \lambda_n^j, \quad J_k = \sum_{n=\overline{1,N}} \lambda_n^{k+1}\partial/\partial\lambda_n, \qquad (130)$$

for $j, k \in \mathbb{Z}$, then the current Lie algebra relations (128) are satisfied identically. In this case. if we make the restriction $|\lambda_n| = 1, \lambda_n = exp(i\theta_n), \theta_n \in [0, 2\pi], n = \overline{1, N}$, then for the current algebra operators $\rho(x)$ and $J(x), x \in \mathbb{S}^1$, we obtain the expressions

$$\rho(x) = \sum_{n=\overline{1,N}} \delta(x - \theta_n), \quad J(x) = \frac{1}{2i}\sum_{n=\overline{1,N}} [\delta(x - \theta_n)\partial/\partial\theta_n + \partial/\partial\theta_n\delta(x - \theta_n)]. \qquad (131)$$

It is readily seen that the operators (131) are $N$-particle representations of the current Lie algebra (121) on the circle $\mathbb{S}^1$, and that the support of the measure $\mu$ on $F'$ in the representation (124) is concentrated on functionals $\eta = \sum_{n=\overline{1,N}} \delta(x - \theta_n)$ and the Hilbert space $L_2^{(\mu)}(F'; \mathbb{C}) \simeq L_2^{(s)}(\mathbb{T}^N; \mathbb{C})$, the space symmetric square integrable functions on the torus $\mathbb{T}^N$. In the general case, the current generalized Lie algebra (129) possesses numerous functional-operator representations by means of vector fields on special infinite dimensional manifolds. As will be shown below, these vector fields are defined on these manifold's so-called completely integrable infinite-dimensional Hamiltonian systems, many of which have applications in theoretical and mathematical physics.

On the infinite-dimensional smooth functional manifold $M \subset C^\infty(\mathbb{T}^n; \mathbb{R}^m), n, m \in \mathbb{N}$ are finite, we consider a homogeneous autonomous nonlinear dynamical system

$$u_t = K[u], \qquad (132)$$

where $K : M \to T(M)$ is a Frechet-smooth vector field on $M, [u] \in J(\mathbb{T}^n; \mathbb{R}^m)$ denotes a point of a finite order [59,120] at the jet-manifold $J(\mathbb{T}^n; \mathbb{R}^m)$ and $t \in \mathbb{R}$ is the evolution parameter. We assume that the vector field (132) is Hamiltonian, i.e., there exists a skew-symmetric Poissonian [18,59,120,121] operator $\vartheta : T^*(M) \to T(M)$ such that condition

$$L_K\vartheta = 0 \sim \vartheta_t - \vartheta K'^{,*} - K'\vartheta = 0 \qquad (133)$$

where $L_K$ denotes the Lie derivative [11,26,59,120,122,123] along the vector field $K : M \to T(M)$, "*prime*" denotes the usual Frechet derivative of a mapping and "*\**" denotes the adjoint mapping subject to the standard bilinear convolution form $(\cdot|\cdot)$ on the product $T^*(M) \times T(M)$ of the tangent and cotangent spaces over the functional manifold $M$. If the condition (133) holds, there exists such a smooth Hamiltonian functional $H_\vartheta \in \mathcal{D}(M) \subset C^\infty(M; \mathbb{R})$ that

$$K[u] = -\vartheta \operatorname{grad} H_\vartheta[u] \tag{134}$$

Assume now that the dynamical system (132) possesses one further algebraically independent solution $\eta : T^*(M) \to T(M)$ to the Equation (133), that is $L_K\eta = 0$, which is Poissonian.

**Definition 12.** *A dynamical system (132) possessing a $(\vartheta, \eta)$-pair of Poissonian operators is said [11,13,18,59] to be bi-Hamiltonian, if for any $\lambda \in \mathbb{R}$ the pencil $(\vartheta\lambda + \eta) : T^*(M) \to T(M)$ is also Poissonian. The Poissonian $(\vartheta, \eta)$-pair is called the Magri type compatible.*

**Definition 13.** *If the Poisson operator $\vartheta : T^*(M) \to T(M)$ is invertible, the operator $\Lambda := \vartheta^{-1}\eta : T^*(M) \to T^*(M)$ is said to be gradient-recursive and satisfies the Noether–Lax equation*

$$L_K\Lambda = 0 \sim \Lambda_t - [\Lambda, K'^{,*}] = 0. \tag{135}$$

*Similarly, the operator $\Phi := \eta\vartheta^{-1} : T(M) \to T(M)$ is said to be symmetry-recursive and satisfies the Noether–Lax equation*

$$L_K\Phi = 0 \sim \Phi_t - [K', \Lambda] = 0. \tag{136}$$

*The inverse operator $\vartheta^{-1} : T(M) \to T^*(M)$ is said to be symplectic, and the operator $\vartheta : T^*(M) \to T(M)$ itself is often called cosymplectic.*

Yet, if the inverse operator $\vartheta^{-1} : T(M) \to T^*(M)$ does not exist, the notions of gradient-recursive and symmetry-recursive operators remain the same: $L_K\Lambda = 0$ and $L_K\Phi = 0$, respectively.

**Definition 14.** *The operator $\Phi : T(M) \to T(M)$ is said to be hereditary-recursive if the bilinear operator*

$$[\Phi', \Phi] : T(M) \times T(M) \to T(M) \tag{137}$$

*is symmetric.*

It is easy to check that the operator $\Phi = \eta\vartheta^{-1} : T(M) \to T(M)$ is hereditary-recursive [11,59,121] if the Poissonian pair $(\vartheta, \eta)$-pair is compatible. The Poissonian $(\vartheta, \eta)$-pair is compatible if, and only if, the operator $\eta\vartheta^{-1}\eta : T^*(M) \to T(M)$ is Poissonian too. Moreover, the operators $\vartheta(\vartheta^{-1}\eta)^n : T^*(M) \to T(M)$ for all $n \in \mathbb{Z}_+$ are Poissonian also.

**Definition 15.** *A vector field $\alpha : M \to T(M)$ is called a homogeneous symmetry of the dynamical system (132) if $L_K\alpha = 0 \sim [K, \alpha] = 0$. Respectively, a vector field $\tau : M \to T(M)$ is called an inhomogeneous symmetry of the dynamical system (132), $\partial\tau/\partial t + [K, \tau] = 0$.*

It is easy to observe that subsets of homogeneous and inhomogeneous symmetries, respectively, are Lie subalgebras of the symmetry space $\Gamma(M)$. Suppose now that for a consistent bi-Hamiltonian dynamical system (132) there exist two nontrivial homogeneous symmetry $\alpha_0 \in \Gamma(M)$ and homogeneous symmetry $\tau_0 \in \Gamma(M)$, such that

$$\begin{aligned}
&L_{\tau_0}\alpha_0 = \varepsilon\alpha_0, \; L_{\alpha_0}\vartheta = 0 = L_{\alpha_0}\eta, \; L_{\tau_0}\vartheta = (\xi - 1/2)\vartheta, \\
&L_{\tau_0}\alpha_0 = \varepsilon\alpha_0, L_{\tau_0}\eta = (\xi + 1/2)\eta, \; L_{\tau_0}\Phi = \Phi,
\end{aligned} \tag{138}$$

where $\varepsilon, \xi \in \mathbb{R}$ are certain numerical parameters. Having assumed that the symmetry-recursive operator $\Phi : T(M) \to T(M)$ is invertible, one can construct the following subsets $Q\{\alpha\} \subset \Gamma(M)$ and $Q\{\tau\} \subset \Gamma(M)$, where

$$Q\{\alpha\} := \{\alpha_j : \Phi^j \alpha_0 : j \in \mathbb{Z}\}, \quad Q\{\tau\} := \{\tau_j : \Phi^j \tau_0 : j \in \mathbb{Z}\}. \tag{139}$$

The following proposition holds.

**Proposition 5.** *The semi-direct product $Q := Q\{\tau\} \ltimes Q\{\alpha\}$ is a Lie subalgebra of symmetries of the dynamical system (132) isomorphic to the current Lie algebra (128).*

**Proof.** The proof is a direct consequence of the relations (138) and (139).

$$u_t = u_{xxx} + u u_x := K[u], \tag{140}$$

$\square$

As a simplest example we consider the classical nonlinear Korteweg–de Vries dynamical system on the functional manifold $M \subset C^\infty(\mathbb{S}^1; \mathbb{R})$, possessing two compatible Poissonian operators

$$\vartheta = \partial, \quad \eta = \partial^3 + (u\partial + \partial u)/3, \tag{141}$$

where $\partial := \partial/\partial x, x \in \mathbb{R}$. Its symmetry-recursive operator equals to the expression

$$\Phi = \eta \vartheta^{-1} = \partial^2 + (u + \partial u \partial^{-1})/3, \tag{142}$$

where $\partial^{-1}(\cdot) := 1/2\left[\int_0^x dx(\cdot) - \int_x^{2\pi} dx(\cdot)\right]$ is the operator of inverse differentiation, $\partial \cdot \partial^{-1} = I$. Then, taking into account the homogeneous symmetry, $\alpha_0 = u_x$ and inhomogeneous $\tau_{-1} = 3/2(1 + t u_x)$ generate, respectively, two subalgebras $Q\{\alpha\} := \{\Phi^j \alpha_0 : j \in \mathbb{Z}\}$ and $Q\{\tau\} := \{\Phi^{j+1}\tau_{-1} : j \in \mathbb{Z}\}$, whose semidirect product $Q = Q\{\tau\} \ltimes Q\{\alpha\}$ is isomorphic to the current Lie algebra $\mathcal{G}$ (128).

*3.2. Completely Integrable Hamiltonian Systems and the Current Algebra Symmetry Integrability Criterion*

In analyzing the dynamical system (132) above, we assumed for it the existence of the consistent $(\vartheta, \eta)$-pair of Poissonian operators, with respect to which it is bi-Hamiltonian. However, if the dynamical system (132) is not bi-Hamiltonian but only Hamiltonian and integrable, then obviously the Noether–Lax Equation (133) has only one solution, which is determined up to multiplication by a constant. On the other hand, if the dynamical system (132) is invariant with respect to the universal Banach Lie group symmetry $G = \text{Diff}(\mathbb{T}^n) \ltimes F$, then for the corresponding Lie algebra of symmetries $Q = Q\{\tau\} \ltimes Q\{\alpha\}$, which is isomorphic to the current Lie algebra $\mathcal{G}$, the next conditions should hold:

$$L_\alpha \vartheta = 0, \quad L_\tau \vartheta = 0 \tag{143}$$

for all $\alpha \in Q\{\alpha\}$ and $\tau \in Q\{\tau\}$. In addition, one easily ensues from (128) the following commutation relationships:

$$(j + \varepsilon)\alpha_{j+1} = [\tau_1, \alpha_j], (j + \varepsilon)\alpha_{j-1} = [\tau_{-1}, \alpha_j], (j + \varepsilon)\alpha_j = [\tau_0, \alpha_j], \tag{144}$$
$$(j - 1)\tau_{j+1} = [\tau_1, \tau_j], \quad j\tau_j = [\tau_0, \tau_j], \quad (j + 1)\tau_{j-1} = [\tau_{-1}, \tau_j].$$

Algebraic relationships (144) give rise to the following Lie-algebraic relationships

$$L_{\alpha_j}\vartheta = 0 = L_{\alpha_j}\eta, \, , L_{\tau_j}\Lambda = \Lambda^{j+1}, L_{\alpha_j}\Lambda = 0 = L_{\alpha_j}\Phi, L_{\tau_j}\Phi = \Phi^{j+1}, \tag{145}$$
$$L_{\tau_j}\vartheta = (\xi - j - 1/2)\vartheta \Lambda^j, \quad L_{\tau_j}\eta = (\xi - j + 1/2)\eta \Lambda^j,$$

and show that on the basis of the sl(2) Lie subalgebra $\{\tau_{-1}, \tau_0, \tau_1\}$ jointly with the set of initial homogeneous symmetries $\{\alpha_{-1}, \alpha_1\}$ for $\varepsilon = 0$ or $\{\alpha_0\}$ for $\varepsilon \notin \mathbb{Z}$, as well as the inhomogeneous symmetries $\{\tau_{-2}, \tau_2\}$, one can construct recursively an entire infinite-hierarchy of symmetries $Q\{\tau\} \ltimes Q\{\alpha\}$, which is isomorphic to the current Lie algebra $\mathcal{G}$ (128) by virtue of the construction. In addition, in accordance with the Noether relations (143) there exist two infinite hierarchies of conservation laws to the dynamical system (132), namely, the homogeneous functionals $\gamma_j \in \mathcal{D}(M), j \in \mathbb{Z}$, and the inhomogeneous $\zeta_j \in \mathcal{D}(M), j \in \mathbb{Z}$, satisfying the conditions

$$\tau_j = -\vartheta \operatorname{grad} \zeta_j, \ \alpha_j = -\vartheta \operatorname{grad} \gamma_j, \ \{\gamma_j, \gamma_k\} = 0, \tag{146}$$
$$(j + \varepsilon)\gamma_{j+k} = (\operatorname{grad} \gamma_j | \tau_k) = \{\gamma_j, \tau_k\},$$
$$\partial \tau_k + \{H, \zeta_k\} = 0, \ \{H, \gamma_j\} = 0, \ \{\zeta_j, \zeta_k\} = (j - k)\zeta_{j+k}$$

for any $j, k \in \mathbb{Z}$, where, by definition, $K[u] = -\vartheta \operatorname{grad} H$, and $\{\cdot, , \cdot\} := (\operatorname{grad}(\cdot) | \vartheta \operatorname{grad}(\cdot))$ denotes the Poisson bracket on the space of functionals $\mathcal{D}(M)$ on the functional manifold $M$.

Direct calculations show that the results described above are valid for all the currently known completely integrable nonlinear dynamical systems, including the nonlinear equations of Schrëdinger type [21,59,114,124,125], the Benney—Kaup and Ito equations [114], the Davey—Stewartson and Yajima—Mel'nikov equations [13], and others, defined on infinite-dimensional manifolds, whose symmetry groups are isomorphic to the universal Banach current group $\mathrm{Dif} f(\mathbb{S}^1) \ltimes F$ on the circle $\mathbb{S}^1$. With regard to "*two- dimensionalized*" integrable dynamical systems of the Kadomtsev–Petviashvily type, it can be asserted that they are closely related [11,13,112,117] to special operator-valued nonlinear integrable dynamical systems, generated by suitably defined iso-spectral Lax type problems [111,126] and which are bi-Hamiltonian with respect to the Poissonian operators on these operator-valued manifolds.

The analysis made above of the correspondence between the universal Lie algebra of currents (128) and the functional Lie algebras of symmetries of integrable infinite-dimensional dynamical systems makes it possible to formulate the following working algorithm as an effective criterion of testing integrability of an arbitrary homogeneous nonlinear dynamical system (132) on the infinite-dimensional manifold $M$.

*Algorithm: If for the dynamical system $u_t = K[u]$ on the functional manifold $M$ there exists the nontrivial $sl(2)$ Lie subalgebra $\{\tau_{-1}, \tau_0, \tau_1\}$ together with a subset of "initial" inhomogeneous symmetries $\{\tau_{-2}, \tau_2\}$ and homogeneous $\{\alpha_{-1}, \alpha_1 : \varepsilon = 0\}$ symmetries, satisfying the conditions*

$$[\tau_0, \tau_2] = 2\tau_2, [\tau_0, \tau_{-2}] = -2\tau_{-2}, [\tau_1, \tau_{-2}] = -3\tau_{-1}, \tag{147}$$
$$[\alpha_{-1}, \alpha_1] = 0, \ [\tau_{-1}, \tau_2] = 3\tau_1, \ [\tau_0, \tau_0] = \varepsilon \alpha_0,$$

*then this dynamical system on $M$ possesses an infinite-dimensional Lie algebra of symmetries $Q = Q\{\tau\} \ltimes Q\{\alpha\}$, isomorphic to the current Lie algebra $G$ (128) of the Banach group $Diff(S^1) \ltimes F$ on the circle $S^1$, and if there exists a nontrivial solution of the Noether–Lax equation $L_K \vartheta = 0$, then our dynamical system is an infinite-dimensional completely integrable Hamiltonian flow on the functional manifold $M$. If at the same time the relations (145) are satisfied, then the dynamical system $u_t = K[u]$ on $M$ is bi-Hamiltonian and possesses an hereditary-recursive operator $\Lambda = \vartheta^{-1}\eta$, where $\eta(\xi - 3/2) = L_{\tau_1}\vartheta$, and, by virtue of the gradient-holonomic algorithm [11,13,59,116], a standard Lax type representation.*

*If the conditions (143) are not satisfied, the dynamical system does not possess bi- Hamiltonian structure, but there is an additional infinite-dimensional inhomogeneous hierarchy of conservation laws satisfying the conditions (146).*

### 3.3. Integrable Systems, Their Symmetry Analysis and Structure of the Poissonian Operators

Suppose we are given the homogeneous nonlinear dynamical system (132) on the functional manifold $M$ and pose a question of the existence for this dynamical system of a bi-Hamiltonian structure on $M$ and effective methods of determining it in explicit form.

In accordance with the gradient-holonomic algorithm *[11,13,59,116]* for investigating the integrability of nonlinear dynamical systems, we can successively establish in explicit form the presence for our system (132) of an infinite functionally independent and naturally ordered by means of the parameter $\lambda \in \mathbb{R}$ hierarchy $\quad \gamma_j \in \mathcal{D}(M), j \in \mathbb{Z}_+$, of conservation laws. In addition, by virtue of the homogeneity of the dynamical system (132), it always possesses *a priori* two commuting to each other homogeneous symmetries, which are defined on $M$ by the vector fields $d/dx$ and $d/dt$. We can also consider the equivalent realization of these vector fields $d/dx$ and $d/dt$ on $M$ as Hamiltonian systems [11,17,18,123] on the infinite-dimensional manifold of jets $J(\mathbb{S}^1; \mathbb{R}^m) \simeq M$ with respect to a symplectic structure $\omega^{(2)} \in \Omega^1(J(\mathbb{S}^1; \mathbb{R}^m)$. We denote by $\alpha^{(1)} \in \Omega^1(J(\mathbb{S}^1; \mathbb{R}^m)$ a Liouville type 1-form, for which $d\alpha^{(1)} = \omega^{(2)}$ and take into account that, by definition, there holds the conditions $i_{d/dx}\omega^{(2)}[u] = -d\gamma[u]$ and $L_{d/dx}\,\omega^{(2)}[u] = 0$, where $\gamma[u] \in \Omega^0(J(\mathbb{S}^1; \mathbb{R}^m)$ denotes the density of the corresponding conservation law $\gamma \in \mathcal{D}(M)$ at point $u \in M$. Based now on the Cartan representation [127] of the Lie derivative $L_{d/dx} = i_{d/dx}d + di_{d/dx}$, one easily obtains the following general relationship: $\gamma[u] = \alpha^{(1)}[u](d/dx) = (\psi[u]|u_x)\,\mathrm{mod}(d/dx)$ for some element $\psi \in T^*(M)$ at any point $u \in M$, which should simultaneously satisfy the compatibility condition $d\,L_{d/dt}\psi = 0$ subject to the vector field $d/dt$ on $M$. The latter gives rise to the analytical expression that is useful in applications, $\vartheta^{-1} = \psi'[u] - \psi'^{,*}[u]$, for the corresponding cosymplectic operator on the manifold $M$, whose inverse mapping $\vartheta : T^*(M) \to T(M)$ is our searched for Poissonian operator for the dynamical system (132).

### 3.3.1. Two-Dimensional Korteweg–de Vries Type Hydrodynamic System

We consider an example of a nonlinear bi-Hamiltonian Korteweg–de Vries type hydrodynamic system on *"two-dimensionalized"* smooth functional manifold $M \simeq J(\mathbb{T}^2; \mathbb{R})$ for which the Noether–Lax property (143), mentioned above, is not satisfied. This system [12,21] has the form

$$u_t = u_{xxy} + 2u_x\partial_x^{-1}u_y + 4uu_y := K[u] \tag{148}$$

and possesses two algebraically-independent Poissonian operators

$$\vartheta = \partial_x, \ \ \eta = \partial_x^3 + 2(u\partial_x + \partial_x u) \tag{149}$$

Moreover, it is readily shown that for the dynamical system, (148) allows the representation $K[u] = \Phi u_y$, where $\Phi := \eta\vartheta^{-1} = \partial_x^2 + 2(u + \partial_x u\partial_x^{-1})$ is the corresponding symmetry-recursive operator on $M$. One can check by direct calculations that the set $\{\tau_{-1}^{(x)} = 1/4(1 + tu_x), \tau_{-1}^{(y)} = 1/4(1 + tu_y)\}$ consists of inhomogeneous symmetries of the dynamical system (148) and the set $\{\alpha_0^{(x)} = u_x, \alpha_0^{(y)} = u_x\}$ consists of homogeneous symmetries. From them, one constructs the following hierarchies of symmetries:

$$Q\{\alpha^{(x)}\} := \{\alpha_j^{(x)} = \Phi^j\alpha_0^{(x)} : j \in \mathbb{Z}\}, Q\{\alpha^{(y)}\} := \{\alpha_j^{(y)} = \Phi^j\alpha_0^{(y)} : j \in \mathbb{Z}\}, \tag{150}$$

$$Q\{\tau^{(x)}\} := \{\tau_j^{(x)} = \Phi^{j+1}\tau_{-1}^{(x)} : j \in \mathbb{Z}\}, Q\{\tau^{(y)}\} := \{\tau_j^{(y)} = \Phi^{j+1}\tau_{-1}^{(y)} : j \in \mathbb{Z}\}$$

The resulting Lie subalgebras $Q^{(x)} := Q\{\tau^{(x)}\} \ltimes Q\{\alpha^{(x)}\}$ and $Q^{(y)} := Q\{\tau^{(y)}\} \ltimes Q\{\alpha^{(y)}\}$ have the following commutation relationships:

$$[\tau_j^{(x)}, \alpha_k^{(y)}] = k\alpha_{j+k}^{(y)}, \ \ [\tau_j^{(y)}, \alpha_k^{(x)}] = (k + 1/2)\alpha_{j+k}^{(x)}, \tag{151}$$

$$[\tau_j^{(x)}, \tau_k^{(y)}] = k\tau_{j+k}^{(y)} - j\tau_{j+k}^{(x)}, \ \ [\alpha_j^{(x)}, \alpha_k^{(y)}] = 0$$

for $j, k \in \mathbb{Z}$, and the Lie subalgebras $Q^{(x)}$ and $Q^{(y)}$ are isomorphic to the current Lie algebra $\mathcal{G}$ (128) on the circle $\mathbb{S}^1$. Taking into account this fact and expressions (151), we readily state that the following sets $Q\{\tau^{(+)}\}$ and $Q\{\tau^{(-)}\}$ of symmetries

$$Q\{\tau^{(+)}\} := \{\tau_j^{(+)} = 1/2(\tau_j^{(x)} + \tau_j^{(y)}) : j \in \mathbb{Z}\}, \tag{152}$$

$$Q\{\tau^{(-)}\} := \{\tau_j^{(-)} = 1/2(\tau_j^{(x)} - \tau_j^{(y)}) : j \in \mathbb{Z}\}$$

satisfy for all $j, k \in \mathbb{Z}$ the commutation relationships

$$[\tau_j^{(-)}, \tau_j^{(+)}] = 0, [\tau_j^{(+)}, \tau_k^{(+)}] = (k - j)\tau_{j+k}^{(+)}, [\tau_j^{(+)}, \alpha_k^{(x)}] = (k + 1/2)\alpha_{j+k}^{(x)}, \tag{153}$$

$$[\tau_j^{(-)}, \alpha_k^{(x)}] = 0, [\tau_j^{(-)}, \alpha_k^{(y)}] = 0, [\tau_j^{(+)}, \alpha_k^{(y)}] = k\alpha_{j+k}^{(y)}, [\tau_j^{(+)}, \tau_k^{(-)}] = k\tau_{j+k}^{(-)}.$$

The latter make it possible to deduce the direct sum of Lie algebras of commuting to each of the other Abelian symmetries

$$Q\{\alpha, \tau^{(-)}\} := Q\{\tau^{(-)}\} \oplus Q\{\alpha^{(x)}\} \oplus Q\{\alpha^{(x)}\}, \tag{154}$$

which jointly with the symmetry Lie subalgebra $Q\{\tau^{(+)}\}$ constitutes the Lie algebra $Q$ constructed above of symmetries to the nonlinear dynamical system (148) as the semidirect product

$$Q = Q\{\tau^{(+)}\} \ltimes Q\{\alpha, \tau^{(-)}\}, \tag{155}$$

being fully isomorphic to the current Lie algebra $\mathcal{G}$ (128). The latter states the invariance of the nonlinear dynamical system (148) with respect to the current symmetry group $G = \mathrm{Diff}(\mathbb{S}^1) \ltimes F$ of the circle $\mathbb{S}^1$.

### 3.3.2. Nonlinear Schrëdinger Type Dynamical System

On a smooth functional manifold $M \subset C^2(\mathbb{R}; \mathbb{C}^2)$ a nonlinear Schrëdinger type dynamical system, which was first considered in [128], looks as

$$\left. \begin{array}{l} \psi_t = i\psi_{xx} + (\psi^2 \bar{\psi})_x, \\ \bar{\psi}_t = -i\bar{\psi}_{xx} + (\bar{\psi}^2 \psi)_x, \end{array} \right\} := K[\psi, \bar{\psi}] \tag{156}$$

and is a bi-Hamiltonian flow with respect to the following two compatible Poisson structures:

$$\vartheta = \begin{pmatrix} 0 & \partial_x \\ \partial_x & 0 \end{pmatrix}, \quad \eta = \begin{pmatrix} -\psi\partial_x^{-1}\psi & -i + \psi\partial_x^{-1}\bar{\psi} \\ i + \bar{\psi}\partial_x^{-1}\psi & -\bar{\psi}\partial_x^{-1}\bar{\psi} \end{pmatrix}. \tag{157}$$

It is easy to check that the following flows on $M$

$$\tau_0 = tK + (x\psi_x + \psi/2, x\bar{\psi}_x + \bar{\psi}/2)^\mathsf{T}, \tag{158}$$

$$\tau_1 = t\alpha_3 + xK + (\psi^2\bar{\psi} + i3/2\psi_x, \bar{\psi}^2\psi - i3/2\psi/2)^\mathsf{T},$$

are nonuniform symmetries of the dynamical system (156), that is

$$\partial\tau_j/\partial t + [K, \tau_j] = 0 \tag{159}$$

for $j = \overline{1, 2}$, where $\alpha_3 := \Phi^2(\psi_x, \bar{\psi}_x)^\mathsf{T}$ and $\Phi := \eta\vartheta^{-1} : T(M) \to T(M)$ is the corresponding symmetry-recursive operator. Moreover, the following algebraic relationships hold:

$$L_K\tau_0 = -\vartheta, \quad L_K\tau_1 = -2\eta, \tag{160}$$

where, as before, $L_K$ denotes the Lie derivative with respect to the vector field $K : M \to T(M)$. Put now, by definition, $\alpha_0 := (-i\psi, i\bar{\psi})^\top$, and $\alpha_j := \Phi^j \alpha_0, \tau_j := \Phi^j \tau$ for $j \in \mathbb{Z}$. Then the following proposition holds.

**Proposition 6.** *The nonlinear Schrëdinger type dynamical system* (156) *is a completely integrable bi-Hamiltonian system on the functional manifold $M$, possessing two independent symmetry Lie subalgebras $Q\{\tau\} := \{\tau_j : j \in \mathbb{Z}\}$ and $Q\{\alpha\} := \{\alpha_j : j \in \mathbb{Z}\}$. Moreover, their semidirect product $Q\{\alpha, \tau\} := Q\{\tau\} \ltimes Q\{\alpha\}$ is isomorphic to the quantum Lie algebra $\mathcal{G}$ of currents* (128) *of the Banach group $G = \mathrm{Diff}(\mathbb{S}^1) \ltimes F$ on the circle $\mathbb{S}^1$.*

### 3.3.3. The Benjamin–Ono Nonlinear Dynamical System

This dynamical system is defined on a functional manifold $M \subset C^2(\mathbb{R}; \mathbb{R})$ as

$$u_t = \mathcal{H}u_{xx} + 2uu_x, \tag{161}$$

where $u \in M$ and $\mathcal{H} : T(M) \to T(M)$ is the classical Hilbert transform

$$(\mathcal{H}\alpha)(x) := \frac{1}{\pi} \int_{-\infty}^{\infty} dy \frac{\alpha(y)}{y - x} \tag{162}$$

for any $\alpha \in T(M)$. The Hilbert transform (162) satisfies the following algebraic properties: $\mathcal{H}^2 = -1, \mathcal{H}^* = -\mathcal{H}$ subject to the standard bilinear convolution form on the product $T^*(M) \times T(M)$. It is easy to check that the dynamical system (161) is Hamiltonian [129] with respect to the Poisson operator

$$\vartheta = \partial/\partial x, \tag{163}$$

that is $u_t = -\vartheta \, \mathrm{grad}\, H$, where the Hamiltonian function $H = \int_{-\infty}^{\infty} dx(u^3/3 + u\mathcal{H}u_x)$. Simple calculations make it possible to state [113] that the following functional expressions

$$\tau_{-1} = 1 + tu_x, \tau_0 = xu_x + u + tK, \tag{164}$$
$$\tau_1 = t\alpha_2 + xK + u^2 - 3/2\mathcal{H}u_x,$$
$$\alpha_2 = [2u^3 + 3H(uu_x) + 3uHu_x - 2u_{xx}]_x$$

are symmetries of the Benjamin–Ono nonlinear dynamical system (161). Moreover, since there hold algebraic relationships $L_{\tau_j}\vartheta = 0$ for $j = \overline{-1, 1}$, we can state that this dynamical system is not bi-Hamiltonian on the functional manifold $M$, as owing to the relationships (145) we should have $L_{\tau_{-1}}\vartheta = (\xi + 1/2)\vartheta\Lambda^{-1} = 0, L_{\tau_0}\vartheta = (\xi - 1/2)\vartheta = 0$ and $L_{\tau_1}\vartheta = (\xi - 3/2)\vartheta\Lambda = 0$, whose common solution is $\xi = 1/2$ and $\eta = 0$. The latter means that the Benjamin–Ono nonlinear dynamical system (161) is not bi-Hamiltonian on the functional manifold $M$, albeit it proves to be bi-Hamiltonian [129] on an extended spatially two-dimensional operator manifold $\hat{M}$, being equivalent to a respectively defined Hilbert–Schmidt operator algebra, whose theory was previously developed in [11,20,21,130,131] and applied to other nonlinear dynamical systems such as Devey–Stewartson, Kadomtsev–Petviashvily, etc.

### 3.4. Conclusions

In this Section, we analyzed the algebraic structure of symmetries of nonlinear integrable infinite-dimensional integrable Hamiltonian dynamical systems. It was stated that the Banach group of currents $\mathrm{Diff}(\mathbb{S}^1) \ltimes C^\infty(\mathbb{S}^1; \mathbb{R})$ on the circle $\mathbb{S}^1$ is a universal symmetry group of all completely integrable bi-Hamiltonian systems. Applications of this phenomenon to the problem of constructing effective criteria of integrability of nonlinear dynamical systems of theoretical and mathematical physics are presented.

## 4. The Current Algebra Representations and the Factorized Structure of Quantum Integrable Many-Particle Hamiltonian Systems

*4.1. The Current Algebra Representation and the Hamiltonian Reconstruction of the Calogero–Moser–Sutherland Quantum Model*

The periodic Calogero–Moser–Sutherland quantum bosonic model on the finite interval $[0, l] \simeq \mathbb{R}/\{[0, l]\mathbb{Z}\}$ is governed by the $N$-particle Hamiltonian

$$H_N := -\sum_{j=\overline{1,N}} \frac{\partial^2}{\partial x_j^2} + \sum_{j \neq k = \overline{1,N}} \frac{\pi^2 \beta(\beta - 1)}{l^2 \sin^2[\frac{\pi}{l}(x_j - x_k)]} \tag{165}$$

on the symmetric Hilbert space $L_2^{(s)}([0, l]^N; \mathbb{C})$, where $N \in \mathbb{Z}_+$ and $\beta \in \mathbb{R}$ is an interaction parameter. As it was stated in very interesting and highly speculative works [132,133], there exists linear differential operators

$$\mathcal{D}_j := \frac{\partial}{\partial x_j} - \frac{\pi \beta}{l} \sum_{k = \overline{1,N}, k \neq j} ctg[\frac{\pi}{l}(x_j - x_k)] \tag{166}$$

for $j = \overline{1, N}$, such that the Hamiltonian (165) is factorized as the bounded from below symmetric operator

$$H_N = \sum_{j=\overline{1,N}} \mathcal{D}_j^+ \mathcal{D}_j + E_n, \tag{167}$$

where

$$E_N = \frac{1}{3}\left(\frac{\pi \beta}{l}\right)^2 N(N^2 - 1) \tag{168}$$

is the ground state energy of of the Hamiltonian operator (165), that there exists such a vector $|\Omega_N) \in L_2^{(s)}([0, l]^N; \mathbb{C})$, satisfying for any $N \in \mathbb{Z}_+$ the eigenfunction condition

$$H_N|\Omega_N) = E_N|\Omega_N) \tag{169}$$

and equals

$$|\Omega_N) = \prod_{j < k = \overline{1,N}} \left(\sin[\frac{\pi}{l}(x_j - x_k)]\right)^\beta, \tag{170}$$

coinciding with the corresponding Bethe anzatz representation [134,135] for the groundstate of the quantum Calogero–Moser-Sutherland model (165).

Being additionally interested in proving the quantum integrability of the Calogero–Moser–Sutherland model (165), we will proceed to its second quantized representation [9,10,13,56,57,60,68,135,136] and studying it by means of the density operator representation approach to the current algebra, described above in Section 2 and devised previously in [1,4–7,76,77].

The secondly quantized form of the Calogero–Moser–Sutherland Hamiltonian operator (165) looks as

$$H = \int_0^l dx \psi_x^+(x)\psi_x(x) + \left(\frac{\pi}{l}\right)^2 \beta(\beta - 1) \int_0^l dx \int_0^l dy \frac{\psi^+(x)\psi^+(y)\psi(y)\psi(x)}{\sin^2[\frac{\pi}{l}(x - y)]}, \tag{171}$$

acting on the corresponding Fock space $\Phi_F := \oplus_{n \in \mathbb{Z}_+} \Phi_n^{(s)}$, $\Phi_n^{(s)} \simeq L_2^{(s)}([0, l]^n; \mathbb{C})$, $n \in \mathbb{Z}_+$. To proceed to the current algebra representation of the Hamiltonian operator (171), it would

useful to recall the factorized representation (167) and construct preliminarily the following singular Dunkl type [132,133,137,138] symmetrized differential operator

$$D_N(x) := \sum_{j=\overline{1,N}} \delta(x - x_j)\frac{\partial}{\partial x_j} -$$

$$- \frac{\pi\beta}{2l} \sum_{j\neq k=\overline{1,N}} \left(\delta(x - x_j)ctg[\frac{\pi}{l}(x_j - x_k)] + \delta(x - x_k)ctg[\frac{\pi}{l}(x_k - x_j)]\right)$$

(172)

on the Hilbert space $L_2^{(s)}([0,l]^N; \mathbb{C})$, $N \in \mathbb{Z}_+$, parametrized by a running point $x \in \mathbb{R}/\{[0,l]\mathbb{Z}\}$. The corresponding secondly quantized representation of the operator (172) looks as

$$D(x) = \psi^+(x)\psi_x(x) - \frac{\pi\beta}{l}\int_0^l dy\, ctg[\frac{\pi}{l}(x-y)] : \psi^+(x)\psi^+(y)\psi(y)\psi(x) : \quad (173)$$

for any $x \in \mathbb{R}/[0,l]\mathbb{Z}$, or on the density operator $\rho : \Phi_F \to \Phi_F$ representation form, as

$$D(x) = \nabla_x\rho(x)/2 + iJ(x) -$$

$$- \frac{\pi\beta}{2l}\int_0^l dy\, \left[ctg[\frac{\pi}{l}(x-y)] : \rho(x)\rho(y) : - ctg[\frac{\pi}{l}(y-x)] : \rho(y)\rho(x) :\right],$$

(174)

which is equivalently representable in a suitable current algebra symmetry representation Hilbert space $\Phi$, as

$$D(x) = K(x) -$$

$$- \frac{\pi\beta}{2l}\int_0^l dy\left[ctg[\frac{\pi}{l}(x-y)] : \rho(x)\rho(y) : - ctg[\frac{\pi}{l}(y-x)] : \rho(y)\rho(x) :\right].$$

(175)

Now, based on the operator (174), one can formulate [10] the following proposition.

**Proposition 7.** *The secondly quantized Calogero–Moser–Sutherland Hamiltonian operator (171) in a suitable current algebra symmetry representation Hilbert space $\Phi$ is weakly equivalent to the factorized Hamiltonian operator*

$$\hat{H} = \int_0^l dx D^+(x)\rho(x)^{-1}D(x) \quad (176)$$

*modulo the ground state energy operator $E : \Phi \to \Phi$, where*

$$E = \frac{1}{3}\left(\frac{\pi\beta}{l}\right)^2 : N^3 : + \left(\frac{\pi\beta}{l}\right)^2 : N^2 :, \quad (177)$$

*where, as before,*

$$N := \int_0^l \rho(x)dx \quad (178)$$

*is the particle number operator, and satisfies the determining conditions*

$$(H - E)|\Omega) = 0, \quad D(x)|\Omega) = 0 \quad (179)$$

*on the suitably renormalized groundstate vector $|\Omega) \in \Phi$ for all $x \in \mathbb{R}/[0,l]\mathbb{Z}$. Moreover, for any integer $N \in \mathbb{Z}_+$ the corresponding projected vector $|\Omega_N) := |\Omega)|_{\Phi_N}$ exactly coincides with the*

*related Bethe groundstate vector for the N-particle Calogero–Moser–Sutherland model ([165](#)) and satisfies the following eigenfunction relationships:*

$$\mathrm{N}|\Omega_N) = N|\Omega_N), \quad \mathrm{E}|\Omega_N) = \left( \frac{1}{3} \left( \frac{\pi\beta}{l} \right)^2 :\mathrm{N}^3: + \left( \frac{\pi\beta}{l} \right)^2 :\mathrm{N}^2: \right) |\Omega_N) = \tag{180}$$

$$= \left[ \frac{1}{3} \left( \frac{\pi\beta}{l} \right)^2 (N^3 - 3N^2 + 2N) + \left( \frac{\pi\beta}{l} \right)^2 N(N-1) \right] |\Omega_N) =$$

$$= \left[ \frac{1}{3} \left( \frac{\pi\beta}{l} \right)^2 (N^3 - 3N^2 + 2N + 3N^2 - 3N) \right] |\Omega_N) =$$

$$= \left[ \frac{1}{3} \left( \frac{\pi\beta}{l} \right)^2 N(N^2 - 1) \right] |\Omega_N) := E_N|\Omega_N),$$

*exactly ensuing the result ([168](#)).*

**Remark 5.** *When deriving the expression ([180](#)), we have used the identities*

$$\rho(x)\rho(y) = \; :\rho(x)\rho(y): +\rho(y)\delta(x-y),$$

$$\rho(x)\rho(y)\rho(z) = \; :\rho(x)\rho(y)\rho(z): + :\rho(x)\rho(y):\delta(y-z)+ \tag{181}$$

$$+ :\rho(y)\rho(z):\delta(z-x)+ :\rho(z)\rho(x):\delta(x-y)+ :\rho(x)\delta(y-z)\delta(z-x),$$

*which hold [5,55,56,77] for the density operator $\rho : \Phi \to \Phi$ at any points $x, y, z \in \mathbb{R}/\{[0,l]\mathbb{Z}\}$.*

Observe now that the operator ([173](#)) can be rewritten down in $\Phi$ as

$$\mathrm{D}(x) = \mathrm{K}(x) - \mathrm{A}(x), \tag{182}$$

where, by definition,

$$\mathrm{K}(x) := \nabla_x \rho(x)/2 + iJ(x), \quad \mathrm{A}(x) := \frac{\pi\beta}{l} \int_0^l dy \, ctg[\frac{\pi}{l}(x-y)] : \rho(x)\rho(y) : \tag{183}$$

for all $x \in \mathbb{R}/\{[0,l]\mathbb{Z}\}$. Recalling now the second condition of ([179](#)), one can rewrite it equivalently as

$$\mathrm{K}(x)|\Omega) = \mathrm{A}(x)|\Omega) \tag{184}$$

on the renormalized ground state vector $|\Omega) \in \Phi$ for all $x \in \mathbb{R}/\{[0,l]\mathbb{Z}\}$. On the other hand, owing to the expression ([176](#)), we obtain the searched for current algebra representation

$$\hat{\mathrm{H}} = \int_0^l dx (\mathrm{K}^+(x) - \mathrm{A}(x))\rho(x)^{-1}(\mathrm{K}(x) - \mathrm{A}(x)) \tag{185}$$

of the Calogero–Moser–Sutherland Hamiltonian operator ([165](#)) on the suitably renormalized Hilbert space $\Phi$, as it was already demonstrated in the work [76,77], using the condition ([184](#)) in the form ([61](#)).

### 4.2. The Current Algebra Representation and Integrability of the Calogero–Moser–Sutherland Quantum Model

We now briefly discuss the quantum integrability of the Calogero–Moser–Sutherland model ([165](#)). Owing to the factorized representation ([185](#)), one can easily observe [8–10]

that for any integer $p \in \mathbb{Z}_+$, the suitably symmetrized Hamiltonian operator densities $\mathrm{h}(x) := \mathrm{D}^+(x)\rho(x)^{-1}\mathrm{D}(x) : \Phi \to \Phi, x \in \mathbb{R}/\{[0,l]\mathbb{Z}\}$, commute to each other and with the particle number operator $\mathrm{N} : \Phi \to \Phi$, that is

$$[\mathrm{h}(x),\mathrm{h}(y)] = 0, \quad [\mathrm{h}(x),\mathrm{N}] = 0 \tag{186}$$

for any $x, y \in \mathbb{R}/[0,l]\mathbb{Z}$. As a result of the commutation property (186), one easily obtains that for any integer $p \in \mathbb{Z}_+$ the symmetric operators

$$\hat{\mathrm{H}}^{(p)} := \int_0^l dx \mathrm{h}(x)^p \tag{187}$$

also commute to each other

$$[\hat{\mathrm{H}}^{(p)}, \hat{\mathrm{H}}^{(q)}] = 0 \tag{188}$$

for all integers $p, q \in \mathbb{Z}_+$, and in particular, commute to the Calogero–Moser–Sutherland Hamiltonian operator (176):

$$[\hat{\mathrm{H}}^{(p)}, \hat{\mathrm{H}}] = 0. \tag{189}$$

Concerning the related $N$-particle differential expressions for the operators (187), it is enough to calculate their projections on the $N$-particle Fock subspace $\Phi_N^{(s)} \subset \Phi_F, N \in \mathbb{N}$. Namely, let an arbitrary vector $|\varphi_N) \in \Phi_N^{(s)}$ be representable as

$$|\varphi_N) := \int_{[0,l]^N} \varphi_N(x_1, x_2, ..., x_N) \prod_{j=\overline{1,N}} dx_j \psi^+(x_j)|0\rangle \tag{190}$$

for some coefficient function $\varphi_N \in L_2^{(s)}([0,l]^N; \mathbb{C})$. Then, by definition,

$$\hat{\mathrm{H}}^{(p)}|\varphi_N) := |\varphi_N^{(p)}), \tag{191}$$

where

$$|\varphi_N^{(p)}) = \int_{[0,l]^N} (H_N^{(p)}\varphi_N)(x_1, x_2, ..., x_N) \prod_{j=\overline{1,N}} dx_j \psi^+(x_j)|0\rangle \tag{192}$$

for a given $p \in \mathbb{Z}_+$ any $N \in \mathbb{Z}_+$. In particular, for $p = 2$, when $\hat{\mathrm{H}}^{(2)} + \mathrm{E} = \mathrm{H} : \Phi_F \to \Phi_F$, one easily retrieves the shifted Calogero–Moser–Sutherland Hamiltonian operator (165):

$$H_N^{(2)} = -\sum_{j=\overline{1,N}} \frac{\partial^2}{\partial x_j^2} + \sum_{j \neq k = \overline{1,N}} \frac{\pi^2 \beta(\beta-1)}{l^2 \sin^2[\frac{\pi}{l}(x_j - x_k)]} - \left(\frac{\pi\beta}{l}\right)^2 \frac{N(N^2-1)}{3}. \tag{193}$$

Respectively for higher integers $p > 2$ the resulting $N$-particle differential operator expressions $H_N^{(p)} : L_2^{(s)}([0,l]^N; \mathbb{C}) \to L_2^{(s)}([0,l]^N; \mathbb{C})$, $N \in \mathbb{Z}_+$, can be obtained the described above way by means of simple yet cumbersome calculations, and which will prove to be completely equivalent to those calculated previously in good work [132].

**Remark 6.** *In the thermodynamical limit, when* $\lim_{N \to \infty, l \to \infty} N/\pi l := \bar{\rho} > 0$, *the structural operator* $\mathrm{D}(x) : \Phi \to \Phi, x \in \mathbb{R}/\{[0,l]\mathbb{Z}\}$, *reduces to*

$$\bar{\mathrm{D}}(x) := \lim_{N/l \to \bar{\rho}} \mathrm{D}(x) = \nabla_x \rho(x)/2 + iJ(x) - \beta \int_{\mathbb{R}} dy \frac{: \rho(y)\rho(x) :}{x - y}, \tag{194}$$

*and, respectively, the operator* (165) *reduces to*

$$\bar{H}_N = -\sum_{j=\overline{1,N}} \frac{\partial^2}{\partial x_j^2} + \beta(\beta-1) \sum_{j \neq k = \overline{1,N}} \frac{1}{(x_j - x_k)^2} \tag{195}$$

on the Hilbert space $L_2^{(s)}(\mathbb{R}^N; \mathbb{C})$ for any $N \in \mathbb{Z}_+$, whose density operator representation in a suitable Hilbert space $\Phi$, respectively, equals

$$\bar{H} = \int_{\mathbb{R}} dx \left( \bar{D}^+(x) \rho(x)^{-1} \bar{D}(x) + \epsilon_0 \right),$$ (196)

where $\epsilon_0 := \lim_{N/l \to \bar{\rho}} \frac{E_N}{l} = \bar{\rho}^3/3$ denotes the average energy density of the reduced Calogero–Moser–Sutherland Hamiltonian operator (195) as $N \to \infty$, exactly coinciding with the before obtained results in [77,133].

## 5. The Dual Current Algebra Density Representation and the Factorized Structure of Quantum Integrable Many-Particle Hamiltonian Systems

*5.1. The Current Algebra Density Representation*

We are now interested in constructing a special density functional representation of the local current algebra (29) on the corresponding representation Hilbert space $\Phi_\mu \simeq \oplus_\rho$ with the cyclic vector $|\Omega) = 1 \in \Phi_\rho$. To do this, let us first consider the *creation* $\psi^+(x)$ and annihilation operators $\psi(x), x \in \mathbb{R}^m$, defined via (56) on the canonical Fock space $\Phi_F$, which can be formally represented as

$$\psi^+(x) = \sqrt{\rho(x)} \exp[-i\vartheta(x)], \psi(x) = \exp[i\vartheta(x)]\sqrt{\rho(x)},$$ (197)

where $\rho(x) : \Phi_F \to \Phi_F$ is our density operator and $\vartheta(x) : \Phi_F \to \Phi_F, x \in \mathbb{R}^m$, is some self-adjoint operator. What is important is the operators $\rho(x)$ and $\vartheta(x) : \Phi_F \to \Phi_F$ realize the canonical [55,56,60,63,85] commutation relationships

$$[\rho(x), \rho(y)] = 0 = [\vartheta(x), \vartheta(y)],$$ (198)

$$[\rho(y), \vartheta(x)] = i\delta(x - y)$$

for any $x, y \in \mathbb{R}^m$. Concerning the current operator $J(x) : \Phi_F \to \Phi_F^m, x \in \mathbb{R}^m$, one can easily obtain its equivalent expression

$$J(x) = \rho(x)\nabla\vartheta(x).$$ (199)

Based on the canonical relationships (198) one can easily obtain, following [72,85,139], that

$$\vartheta(x) = \frac{1}{i}\frac{\delta}{\delta\rho(x)} + i\sigma[\rho(x)],$$ (200)

where $\sigma[\rho(x)] : \Phi_\rho \to \Phi_\rho$ acts on the corresponding Hilbert representation space $\Phi_\rho$ and is some function of the density operator $\rho(x) : \Phi_\rho \to \Phi_\rho, x \in \mathbb{R}^m$. Then, respectively, the current operator (199) is representable in $\Phi_\rho$ as

$$J(x) = -i\rho(x)\nabla\frac{\delta}{\delta\rho(x)} + \rho(x)\nabla\sigma[\rho(x)].$$ (201)

The functional-operator expression (201) proves to make sense [5,72,75,139] as operators on the Hilbert space $\Phi_\rho$ of functional valued complex-functions on the manifold $\mathcal{M}$, coordinated by the density functional parameter $\rho : \Phi_\rho \to \Phi_\rho$ and endowed with the scalar product $(a|b)_{\Phi_\rho} := \int_{\mathcal{M}} \overline{a(\rho)}b(\rho)d\mu(\rho)$ subject to some measure $\mu$ on $\mathcal{M}$. To calculate this measure $\mu$ on $\mathcal{M}$, we will present an explicit isomorphism between this Hilbert space $\Phi_\rho$ and the corresponding Fock space $\Phi$ of spinless bosonic particles in $\mathbb{R}^m$. First, we determine the support $supp\,\mu \subset \mathcal{M}$ of the measure $\mu$, having assumed that the manifold

$$\mathcal{M} = \cup_{n \in \mathbb{Z}_+} \mathcal{M}_n,$$ (202)

where $\mathcal{M}_n := \{a(\rho) : \rho(x) := \sum_{j=1}^{n} \delta(x - c_j) : a \in C^\infty(F'; End\ \Phi_\rho)\}$, where $c_j \in \mathbb{R}^m, j = \overline{1,n}, n \in \mathbb{N}$, are arbitrary vector parameters. The restriction $d\mu_n$ of the measure $\mu$ on the submanifold $\mathcal{M}_n$ can be presented [4,5,57,68,71] as

$$d\mu_n = \gamma_n(c_1, c_2, ..., c_n) \prod_{j=\overline{1,n}} dc_j, \tag{203}$$

where functions $\gamma_n : \mathbb{R}^{m \times n} \to \mathbb{R}_+, n \in \mathbb{N}$, should be determined from the condition (201). In accordance with the manifold structure (202), we can decompose the Hilbert space $\Phi_\rho$ as

$$\Phi_\rho = \oplus_{n \in \mathbb{N}} \Phi_n, \tag{204}$$

where the space $\Phi_n$ depends on the mapping $\sigma : \mathcal{M} \to End(\Phi_\rho)$ and consists of functionals that are bounded on $\mathcal{M}_n$, in particular, for any $a(\rho) \in \mathcal{M}$ the restrictions $a(\rho)|_{\Phi_n}, n \in \mathbb{N}$, consist of functions of vectors $(c_1, c_2, ..., c_n) \in \mathbb{R}^{m \times n}, n \in \mathbb{N}$, respectively. The scalar product in $\Phi_n, n \in \mathbb{N}$, is suitably defined by means of the expressions (203). Now we can construct the isomorphism between the Hilbert spaces $\Phi_n, n \in \mathbb{N}$, and the corresponding components $\Phi_n, n \in \mathbb{N}$, of the corresponding Fock space $\Phi$, representing spinless bosonic particles in $\mathbb{R}^m$. In the Hilbert space $\Phi_n := \Phi_n^{(\sigma)}, n \in \mathbb{N}$, one can easily calculate the eigenfunctions $\varphi_{p_1,p_2,...,p_n}^{(\sigma)}(\rho) \in \Phi_n^{(\sigma)}$ of the free Hamiltonian

$$H_0^{(\sigma)} := \frac{1}{2} \int_{\mathbb{R}^m} dx \langle K^+(x) | \rho^{-1}(x) K(x) \rangle \tag{205}$$

with structural

$$K(x) := \frac{1}{2} \nabla \rho(x) + iJ^{(\sigma)}(x), \quad K^+(x) := \frac{1}{2} \nabla \rho(x) - iJ^{(\sigma)}(x) \tag{206}$$

and the momentum

$$P^{(\sigma)} := \int_{\mathbb{R}^m} dx J^{(\sigma)}(x) \tag{207}$$

operators:

$$H_0^{(\sigma)} \varphi_{p_1,p_2,...,p_n}^{(\sigma)}(\rho) = \left( \sum_{j=\overline{1,n}} E_j \right) \varphi_{p_1,p_2,...,p_n}^{(\sigma)}(\rho), \tag{208}$$

$$P^{(\sigma)} \varphi_{p_1,p_2,...,p_n}^{(\sigma)}(\rho) = \left( \sum_{j=\overline{1,n}} p_j \right) \varphi_{p_1,p_2,...,p_n}^{(\sigma)}(\rho),$$

where $p_j \in \mathbb{R}^m, j = \overline{1,n}$, are momentums of bose-particles in $\mathbb{R}^m$, the operator $H_0^{(\sigma)} : \Phi_\rho \to \Phi_\rho$ is given by the expressions (55), (201) and (205) and the operator $P^{(\sigma)} : \Phi_\rho \to \Phi_\rho$ is given by the expressions (201) and (206), respectively, within which the current operator $J^{(\sigma)}(x) : \Phi_\rho \to \Phi_\rho$ is realized under the condition $\nabla \sigma[\rho(x)] := \sigma \rho(x)^{-1} \nabla \rho(x)$ as

$$J^{(\sigma)}(x) = -i\rho(x) \nabla \frac{\delta}{\delta \rho(x)} + i\sigma \nabla \rho(x) \tag{209}$$

where $\sigma \in \mathbb{R}$ is a fixed real-valued parameter. In this case, the eigenfunctions $\varphi_{p_1,p_2,...,p_n}^{(\sigma)}(\rho) \in \Phi_n^{(\sigma)}, n \in \mathbb{N}$, can be expressed [72,139] as

$$\varphi_{p_1,p_2,...,p_n}^{(\sigma)}(\rho) = \frac{1}{n!} \bar{\varphi}_0^{(\sigma)}(\rho) \left( \prod_{j=\overline{1,n}} B_{p_j}(\rho) \cdot 1 \right), \tag{210}$$

where

$$\bar{\varphi}_0^{(\sigma)}(\rho) := \exp\left[(\sigma - 1/2)\int_{\mathbb{R}^m} dx\rho(x)\ln\rho(x)\right], \tag{211}$$

$$B_{p_j}(\rho) := \int_{\mathbb{R}^m} dx \exp(i\langle p|x\rangle)\rho(x)\exp\left(-\frac{\delta}{\delta\rho(x)}\right). $$

The corresponding $n$-particle Fock subspaces $\Phi_n^{(\sigma)}, n \in \mathbb{N}$, can be naturally represented by means of the vectors

$$|\varphi_n^{(\sigma)}) := \frac{1}{\sqrt{n!}}\int_{\mathbb{R}^{m\times n}}\prod_{j=\overline{1,n}} dp_j\, \varphi_n^{(\sigma)}(p_1, p_2, ..., p_n)a^+(p_1)a^+(p_2)...a^+(p_n)|0\rangle \tag{212}$$

with functions $\varphi_n^{(\sigma)} \in L_2^{(s)}(\mathbb{R}^{m\times n}; \mathbb{C}), n \in \mathbb{N}$, where

$$a^+(p) := \frac{1}{(2\pi)^{m/2}}\int_{\mathbb{R}^m} dx \exp(i\langle x|p\rangle)a^+(x) \tag{213}$$

denotes the momentum creation operator for any $p \in \mathbb{R}^m$.

Moreover, any functional $\varphi_n^{(\sigma)}(\rho) \in \Phi_n^{(\sigma)}, n \in \mathbb{N}$, can be uniquely represented as

$$\varphi_n^{(\sigma)}(\rho) := \int_{\mathbb{R}^{m\times n}}\prod_{j=\overline{1,n}} dp_j\tilde{\varphi}_n^{(\sigma)}(p_1, p_2, ..., p_n)\varphi_{p_1,p_2,...,p_n}^{(\sigma)}(\rho) \tag{214}$$

for $\tilde{\varphi}_n^{(\sigma)} \in L_2^{(s)}(\mathbb{R}^{m\times n}; \mathbb{C})$, since the following condition

$$\left.\left(B_{p_{n+1}}(\rho)\prod_{j=\overline{1,n}} B_{p_j}(\rho)\cdot 1\right)\right|_{\rho=a^+(x)a(x)} |\varphi_n^{(\sigma)}) = 0 \tag{215}$$

holds identically for all $p_j \in \mathbb{R}^m, j = \overline{1, n+1}$, and arbitrary state $|\varphi_n^{(\sigma)}) \in \Phi_F, n \in \mathbb{N}$.

**Remark 7.** *The condition (215) jointly with the constraint $\int_{\mathbb{R}^m} \rho(x)dx = n$ in $\Phi_n^{(\sigma)}, n \in \mathbb{N}$, should be, in general, naturally satisfied for any current algebra representation space $\Phi_\rho$, if and only if $\rho(x) = \sum_{j=\overline{1,n}}\delta(x-c_j) \in \mathcal{M}_n$ for arbitrary $n \in \mathbb{N}$.*

As a result of the construction above, we can state that the Hilbert spaces $\Phi_n^{(\sigma)}, n \in \mathbb{N}$, embed, respectively, isomorphically into the related Fock subspaces $\Phi_n^{(\sigma)}, n \in \mathbb{N}$. As a consequence, we derive that the Hilbert space $\Phi_\rho$ allows an isomorphic embedding into the related Fock space $\Phi_F$.

Consider now, following [5,72,139], the action of the current operator (209) on the basic vectors $\varphi_n^{(\sigma)}(\rho) \in \Phi_n^{(\sigma)}, n \in \mathbb{N}$:

$$J^{(\sigma)}(x)\varphi_n^{(\sigma)}(\rho) = \bar{\varphi}_0^{(\sigma)}(\rho)[-i\rho(x)\nabla\frac{\delta}{\delta\rho(x)} + i\sigma\nabla\rho(x)]\varphi_n^{(\sigma)}(\rho), \tag{216}$$

from which one ensues easily at $\sigma = 1/2$ its $n$-particle representation on the functional manifold $\mathcal{M}_n$:

$$\left.J^{(1/2)}(x)\varphi_n^{(1/2)}(\rho)\right|_{\rho(y)=\sum_{j=\overline{1,n}}\delta(y-c_j)} =$$

$$= \sum_{j=\overline{1,n}}\frac{1}{2}[-i\delta(x-c_j)\nabla_{c_j} + i\nabla_{c_j}\circ\delta(x-c_j)]\tilde{f}_n^{(1/2)}(c_1, c_2, ..., c_n), \tag{217}$$

where we took into account that $\bar{\varphi}_0^{(1/2)}(\rho) = 1$ for all densities $\rho : \Phi \to \Phi$ and have put, by definition, the Fourier transform

$$\tilde{f}_n^{(1/2)}(c_1, c_2, ..., c_n) := \int_{\mathbb{R}^{m \times n}} \prod_{j=\overline{1,n}} dp_j f_n^{(1/2)}(p_1, p_2, ..., p_n) \exp(i \sum_{j=\overline{1,n}} \langle p_j | c_j \rangle) \tag{218}$$

for any fixed particle position vectors $c_j \in \mathbb{R}^n$, $j = \overline{1,n}$, and for arbitrary $n \in \mathbb{N}$. The expression (217), in particular, means that the current operator $J^{(1/2)}(x) : \Phi_\rho \to \Phi_\rho$ is symmetric with respect to the measure $d\mu_n^{(1/2)} := \beta_n \prod_{j=\overline{1,n}} dc_j$ on each functional submanifold $\mathcal{M}^n$ for all $n \in \mathbb{N}$, where the constants $\beta_n \in \mathbb{R}_+$, $n \in \mathbb{N}$, can be determined from the normalization condition $||\varphi_n^{(1/2)}(\rho)||_{\Phi_n^{(1/2)}} = (\varphi_n^{(1/2)} | \varphi_n^{(1/2)})_{\Phi_n^{(1/2)}}^{1/2}$, $n \in \mathbb{N}$. The latter gives rise [1,4,57,68,71,72,139] to the following symbolic measure expression

$$d\mu_n^{(1/2)} := \prod_{x \in \mathbb{R}^m} \delta \left( \rho(x) - \sum_{j=\overline{1,n}} \delta(x - c_j) \right) \prod_{j=\overline{1,n}} \frac{dc_j}{(2\pi)^m} \tag{219}$$

for all $c_j \in \mathbb{R}^n$, $j = \overline{1,n}$, and arbitrary $n \in \mathbb{N}$.

**Remark 8.** *As was aptly observed in [72], the choice $\sigma = 1/2$ makes it possible to realize the current algebra representation on the space $\mathcal{M}$ of analytic functions, which will be a priori assumed for further, that is the corresponding measure can be symbolically expressed as*

$$d\mu_n := \prod_{x \in \mathbb{R}^m} \delta \left( \rho(x) - \sum_{j=\overline{1,n}} \delta(x - c_j) \right) \prod_{j=\overline{1,n}} \frac{dc_j}{(2\pi)^m} \tag{220}$$

*on the subspace $\mathcal{M}_n$ for any $n \in \mathbb{N}$.*

*5.2. The Current Algebra Representation and Hamiltonian Reconstruction: A Many-Dimensional Quantum Oscillator Model*

As a classical application of the construction above, one can consider a density current algebra representation of the quantum Hamiltonian operator

$$\mathrm{H}^{(\omega)} = \frac{1}{2} \int_{\mathbb{R}^m} \langle K(x)^+ | \rho(x)^{-1} K(x) \rangle dx + \frac{1}{2} \int_{\mathbb{R}^m} \langle \omega x | \omega x \rangle \rho(x) dx \tag{221}$$

on the corresponding representation Hilbert space $\Phi_\rho$ of the generalized quantum $N$-particle oscillatory Hamiltonian

$$H_N^{(\omega)} = \frac{1}{2} \sum_{j=\overline{1,N}} \left( \langle \nabla_{x_j} | \nabla_{x_j} \rangle + \langle \omega x_j | \omega x_j \rangle \right) \tag{222}$$

for $N \in \mathbb{Z}_+$ bose-particles in the $m$-dimensional space $\mathbb{R}^m$ under the external oscillatory potential, parametrized by the positive definite frequency matrix $\omega \in End\ \mathbb{R}^m$.

Having shifted the representation Hilbert space $\Phi_\rho$ by the functional $\bar{\varphi}_0^{(1/2)}(\rho) := \exp[-\frac{1}{2} \int_{\mathbb{R}^m} \langle x | \omega x \rangle \rho(x) dx] \in \Phi_\rho$, the corresponding current operator (209) becomes

$$J^{(\omega)}(x) = -i\rho(x) \nabla \frac{\delta}{\delta \rho(x)} + \frac{i}{2} \nabla \rho(x) - i\omega x \rho(x), \tag{223}$$

simultaneously entailing the related $K$-operator changing

$$K(x) = \rho(x) \nabla \frac{\delta}{\delta \rho(x)} \to K^{(\omega)}(x) = \rho(x) \nabla \frac{\delta}{\delta \rho(x)} + \omega x \rho(x) \tag{224}$$

for any $x \in \mathbb{R}^m$. The latter gives rise, respectively, to the following equivalent current algebra functional representation of the oscillatory Hamiltonian (221):

$$\mathrm{H}^{(\omega)} = \frac{1}{2} \int_{\mathbb{R}^m} \langle K^{(\omega)}(x)^+ | \rho(x)^{-1} K^{(\omega)}(x) \rangle dx + \frac{1}{2} \mathrm{tr}\omega \int_{\mathbb{R}^m} \rho(x)dx \qquad (225)$$

for any positive defined matrix $\omega \in End\,\mathbb{R}^m$. The shifted current operator (223) makes it possible to construct the suitably deformed free particle measure

$$d\mu_1^{(\omega)}(\rho) := \exp\left( - \int_{\mathbb{R}^m} dx \rho(x) \langle x | \omega x \rangle \right) d\mu_1^{(1/2)}(\rho) \qquad (226)$$

on the one-particle functional manifold $\mathcal{M}_1$, for which the following expression

$$(\Omega | \mathrm{H}^{(\omega)} | U(\mathrm{f}) | \Omega) = \int_{\mathcal{M}} \exp[i\rho(\mathrm{f})] d\mu_1^{(\omega)}(\rho) \qquad (227)$$

holds for any test function $\mathrm{f} \in F$. The latter, jointly with the related ground state condition $|\Omega) = 1 \in \Phi_\rho$, makes it possible to easily calculate the scalar product elements

$$(U(\mathrm{f}_1)\Omega | \mathrm{H}^{(\omega)} | U(\mathrm{f}_2) | \Omega) = \int_{\mathbb{R}^m} \exp[i\mathrm{f}_1(c) + i\mathrm{f}_2(c)] \exp(-\langle c | \omega c \rangle) \frac{dc}{(2\pi)^m} \qquad (228)$$

for any test functions $\mathrm{f}_1, \mathrm{f}_2 \in F$. The expression (228) makes it possible to successfully calculate the matrix elements $(\rho(\mathrm{f}_{p_1})\Omega | \mathrm{H}^{(\omega)} | \rho(\mathrm{f}_{p_2}) | \Omega)$ of the Hamiltonian $\mathrm{H}^{(\omega)} : \Phi_\rho \to \Phi_\rho$ on the corresponding eigenvectors $\rho(\mathrm{f}_p)|\Omega) \in \Phi_\rho$ for arbitrary $p = p_1, p_2 \in \mathbb{N}$ and, therefore, to find its spectrum.

Consider now the operator (55), taking into account the analytical current representation (216) at $\sigma = 1/2$:

$$K(x)\varphi_n^{(1/2)}(\rho) = [\rho(x)\nabla\tfrac{\delta}{\delta\rho(x)} - 1/2\nabla\rho(x)]\varphi_n^{(1/2)}(\rho) +$$
$$+ 1/2\nabla\rho(x)\varphi_n^{(1/2)}(\rho) = \rho(x)\nabla\tfrac{\delta}{\delta\rho(x)}\varphi_n^{(1/2)}(\rho) \qquad (229)$$

for any $n \in \mathbb{N}$. Having substituted instead of $\varphi_n^{(1/2)}(\rho) \in \Phi_\rho^{(n)}, n \in \mathbb{N}$, the ground state eigenfunction $\Omega(\rho) = 1 \in \Phi_\rho$, we can easily retrieve the before derived expression (61). Moreover, based on the representation (224) and the definition (54), one can calculate that

$$K^{(\omega)}(x)\bar{\varphi}_0^{(1/2)}(\rho) = \left[\rho(x)\nabla\frac{\delta}{\delta\rho(x)} + \omega x \rho(x)\right]\bar{\varphi}_0^{(1/2)}(\rho) = 0 = \qquad (230)$$
$$= \mathrm{A}^{(\omega)}(x;\rho)\bar{\varphi}^{(1/2)}(\rho),$$

where $\bar{\varphi}_0^{(1/2)}(\rho) = \exp[-\frac{1}{2}\int_{\mathbb{R}^m}\langle x|\omega x\rangle\rho(x)dx] \in \Phi_\rho^{(1/2)} \simeq \Phi_\rho$. The latter means, in particular, that the corresponding multiplication operator $\mathrm{A}^{(\omega)}(x;\rho) = 0$, or, respectively,

$$K(x)\bar{\varphi}_0^{(1/2)}(\rho) := \mathrm{A}(x;\rho)\bar{\varphi}_0^{(1/2)}(\rho) = -\omega x \rho(x)\bar{\varphi}_0^{(1/2)}(\rho), \qquad (231)$$

where $\bar{\varphi}_0^{(1/2)}(\rho) := |\Omega(\rho)) \in \Phi_\rho$ is the corresponding ground state vector in $\Phi_\rho$ for the oscillatory Hamiltonian operator (222). Making use of the operator (226), based on expression (64), one can present a special solution to the functional Equation (63) in the form

$$\mathcal{L}(\mathrm{f}) = \exp\left( - \int_{\mathbb{R}^m} dx \langle \omega x | x \rangle \frac{1}{2i} \frac{\delta}{\delta\mathrm{f}(x)} \right) \exp\left( \bar{\rho} \int_{\mathbb{R}^m} \{\exp[i\mathrm{f}(x)] - 1\} dx \right), \qquad (232)$$

confirming similar statements from [5,6,77].

*5.3. Conclusions*

In this Section, we have reviewed the development and applications of an effective algebraic scheme of constructing density operator and density functional representations for the local quantum current algebra and its application to quantum Hamiltonian and symmetry operators reconstruction. We analyzed the corresponding factorization structure for quantum Hamiltonian operators, spatially governing many- and one-dimensional integrable dynamical systems. The quantum generalized oscillatory and Calogero–Moser–Sutherland models of spin-less bose-particles were analyzed in detail. The central vector of the density operator current algebra representation proved to be the ground vector state of the corresponding completely integrable factorized quantum Hamiltonian system in the classical Bethe anzatz form. The latter makes it possible to quantum classify completely integrable Hamiltonian systems a priori, allowing the factorized form and those whose groundstate is of the Bethe anzatz from.  These and related aspects of the factorized and completely integrable quantum Hamiltonians systems are planned to be studied in other places.

## 6. The Quantum Current Algebra Quasi-Classical Representations and the Collective Variable Approach in Equilibrium Statistical Physics

*Introductory Notes*

We consider a large system of $N \in \mathbb{N}$ (one-atomic and spinless) bose-particles with a fixed density $\bar{\rho} := N/\Lambda$ in a volume $\Lambda \subset \mathbb{R}^3$, which is specified by a quantum-mechanical Hamiltonian operator $\hat{H} : L_2^{(sym)}(\mathbb{R}^{3N}; \mathbb{C}) \to L_2^{(sym)}(\mathbb{R}^{3N}; \mathbb{C})$ of the form:

$$\hat{H} := -\frac{\hbar^2}{2m} \sum_{j=1}^{N} \nabla_j^2 + \sum_{j<k}^{N} V(x_j - x_k), \tag{233}$$

where $\nabla_j := \partial/\partial x_j$, $j = \overline{1,N}$, $\hbar$—the Planck constant, $m \in \mathbb{R}_+$—a particle mass and $V(x - y) := V(|x - y|)$, $x, y \in \Lambda$,—a two-particle potential energy, allowing a partition $V = V^{(l)} + V^{(s)}$, where $V^{(s)}$—a short range potential of the Lennard–Johns type and $V^{(l)}$—a long range potential of the Coulomb type. Making use of the second quantization representation [13,22,56,59,63,140,141], the Hamiltonian (233) as $\Lambda \to \mathbb{R}^3$ and $N \to \infty$ can be written as a sum $H = H_0 + V$, where

$$H_0 := -\frac{\hbar^2}{2m} \int_{\mathbb{R}^3} d^3 x \psi^+ \nabla_x^2 \psi, \tag{234}$$

$$V := \frac{1}{2} \int_{\mathbb{R}^3} d^3 x \int_{\mathbb{R}^3} d^3 y V(x - y) \psi^+(x) \psi^+(y) \psi(y) \psi(x),$$

and the operator $H : \Phi_F \to \Phi_F$ acts on the corresponding Fock space $\Phi_F$ and $\psi^+(x), \psi(y)$: $\Phi_F \to \Phi_F$ are the creation and annihilation operators  at points $x \in \mathbb{R}^3$ and $y \in \mathbb{R}^3$. Assume now that our particle system is in a thermodynamically equilibrium state at an "*inverse*" temperature $\mathbb{R}_+ \ni \beta \to \infty$. Assume also that this equilibrium state is compatible with the respectively constructed quantum current algebra $\mathcal{G}$ representation in a separable Hilbert space  $\Phi_\mu$ [4,6,56,68,141], whose generating cyclic vector $\Omega \in \Phi_\mu$ realizes the ground state of the Hamiltonian operator $H : \Phi_\mu \to \Phi_\mu$. Then, the corresponding *n*-particles distribution functions can be written down [56,86,142] as

$$f_n(x_1, x_2, ..., x_n) := (\Omega| : \rho(x_1)\rho(x_1)...\rho(x_n) : \Omega), \tag{235}$$

where $n \in \mathbb{N}$, $\rho(x) : \Phi_\mu \to \Phi_\mu$, $x \in \mathbb{R}^3$—the density operator acting on the Hilbert space $\Phi_\mu$ and $: \cdot :$—the related Wick normal ordering, naturally ensued from that defined over the creation and annihilation operators, and $\Omega \in \Phi_\mu$ is the ground state of the Hamiltonian

(234) at the temperature $\beta \to \infty$, normed by the stability condition $(\Omega|\Omega) = 1$. Having introduced the corresponding Bogolubov generating functional

$$\mathcal{L}(\mathrm{f}) := (\Omega|\exp[i\rho(\mathrm{f})]\Omega) \tag{236}$$

for any "test" Schwartz function $\mathrm{f} \in F \simeq \mathcal{S}(\mathbb{R}^3; \mathbb{R})$, where $\rho(\mathrm{f}) := \int_{\mathbb{R}^3} d^3x \mathrm{f}(x)\rho(x)$, then for the *n*-particle distribution functions (235) one can get the expression

$$f_n(x_1, x_2, ..., x_n) =: \frac{1}{i}\frac{\delta}{\delta \mathrm{f}(x_1)}\frac{1}{i}\frac{\delta}{\delta \mathrm{f}(x_2)}...\frac{1}{i}\frac{\delta}{\delta \mathrm{f}(x_n)} : \mathcal{L}(\mathrm{f})|_{\mathrm{f}=0}. \tag{237}$$

Here $x_j \in \mathbb{R}^3$, $j = \overline{1,n}$, $n \in \mathbb{N}$, and the symbol "$: \frac{1}{i}\frac{\delta}{\delta \mathrm{f}(x_1)}\frac{1}{i}\frac{\delta}{\delta \mathrm{f}(x_2)}...\frac{1}{i}\frac{\delta}{\delta \mathrm{f}(x_n)} :$" imitates the normal ordering symbol "$: :$" action on operator expressions $\rho(x_1)\rho(x_1)...\rho(x_n)$, that is

$$: \frac{1}{i}\frac{\delta}{\delta \mathrm{f}(x_1)} := \frac{1}{i}\frac{\delta}{\delta \mathrm{f}(x_1)}, \tag{238}$$
$$: \frac{1}{i}\frac{\delta}{\delta \mathrm{f}(x_1)}\frac{1}{i}\frac{\delta}{\delta \mathrm{f}(x_2)} := \frac{1}{i}\frac{\delta}{\delta \mathrm{f}(x_1)}[\frac{1}{i}\frac{\delta}{\delta \mathrm{f}(x_2)} - \delta(x_1 - x_2)],$$

and so on. Consider now the expression (236) at some $\beta \in \mathbb{R}_+$, making use of the statistical operator $P : \Phi_\mu \to \Phi_\mu$ and the "*shifted*" 'Hamiltonian $\mathrm{H}^{(\lambda)} := \mathrm{H} - \lambda \int_{\mathbb{R}^3} d^3x\rho(x)$ with $\lambda \in \mathbb{R}$ being a suitable "*chemical*" potential:

$$\mathcal{L}(\mathrm{f}) := \mathrm{Tr}(P\exp[i\rho(\mathrm{f})]), \quad P := \frac{\exp(-\beta \mathrm{H}^{(\mu)})}{\mathrm{Tr}\exp(-\beta \mathrm{H}^{(\mu)})}, \tag{239}$$

where "Tr" means the operator trace-operation on the Hilbert space $\Phi_\mu$. Keeping in mind within the task of studying distribution functions (235) in the classical statistical mechanics case, we need to calculate the trace in (239) as $\hbar \to 0$. The latter gives rise to the following expressions:

$$\mathcal{L}(\mathrm{f}) = Z(\mathrm{f})/Z(0), \quad Z(\mathrm{f}) := \exp[-\beta V(\delta)]\mathcal{L}_0(\mathrm{f}), \tag{240}$$
$$\mathcal{L}_0(\mathrm{f}) = \exp(\varsigma \int_{\mathbb{R}^3} d^3x\{\exp[i\mathrm{f}(x)] - 1\}),$$

where $\varsigma := \exp(\beta\lambda)(2\pi\hbar^2\beta m)^{-3/2}$ is the system "*activity*" [56,142], and

$$V(\delta) := \frac{1}{2}\int_{\mathbb{R}^3} d^3x \int_{\mathbb{R}^3} d^3y V(x - y) : \frac{1}{i}\frac{\delta}{\delta \mathrm{f}(x)}\frac{1}{i}\frac{\delta}{\delta \mathrm{f}(y)} : . \tag{241}$$

Based on expressions (240) and (241) we can formulate the following proposition.

**Proposition 8.** *The functional (236) satisfies [20,56,86] the following functional Bogolubov type equation:*

$$[\nabla_x - i\nabla_x\mathrm{f}(x)]\frac{1}{i}\frac{\delta\mathcal{L}(\mathrm{f})}{\delta \mathrm{f}(x)} \tag{242}$$
$$= -\beta \int_{\mathbb{R}^3} d^3y \nabla_x V(x - y) : \frac{1}{i}\frac{\delta}{\delta \mathrm{f}(x)}\frac{1}{i}\frac{\delta}{\delta \mathrm{f}(y)} : \mathcal{L}(\mathrm{f}),$$

*with the expression (240) being its exact functional-analytic solution.*

Below, we will proceed to constructing effective analytic tools allowing the exact functional-analytic solutions to the Bogolubov functional Equation (242) to be found, describing equilibrium many-particle dynamical systems, as well as generalizing the obtained results for the case of non-equilibrium dynamical many particle systems.

### 7. The Bogolubov-Zubarev "Collective" Variables Transform

Taking into account the two-particle potential energy partition $V = V^{(s)} + V^{(l)}$, owing to the representation (240) one can easily write down the following expression for generating functional $Z(f)$, $f \in F$:

$$Z(f) = \exp[-\beta V^{(s)}(\delta)]\mathcal{L}^{(l)}(f), \quad \mathcal{L}^{(l)}(f) := \exp[-\beta V^{(l)}(\delta)]\mathcal{L}_0(f), \tag{243}$$

where we put

$$V^{(l)}(\delta) := \frac{1}{2}\int_{\mathbb{R}^3} d^3x \int_{\mathbb{R}^3} d^3y V^{(l)}(x-y) : \frac{1}{i}\frac{\delta}{\delta f(x)}\frac{1}{i}\frac{\delta}{\delta f(y)} : , \tag{244}$$

$$V^{(s)}(\delta) := \frac{1}{2}\int_{\mathbb{R}^3} d^3x \int_{\mathbb{R}^3} d^3y V^{(s)}(x-y) : \frac{1}{i}\frac{\delta}{\delta f(x)}\frac{1}{i}\frac{\delta}{\delta f(y)} : .$$

Needing to calculate the functional $\mathcal{L}^{(l)}(f)$, $f \in F$, corresponding to the long range part $V^{(l)}$ of the full potential energy $V : \Phi_\mu \to \Phi_\mu$, we will apply the analogue of Bogolubov–Zubarev [143,144] "collective" variables transform within the grand canonical ensemble, suggested before in [20,68,145,146]. Namely, denote by $\mathcal{L}^{(l)}_{(n)}(f)$, $n \in \mathbb{N}$,—a partial solution to the functional Equation (242), possessing exactly $n \in \mathbb{N}$ particles. Then, owing to the results of [86], for $\mathcal{L}^{(l)}_{(n)}(f)$, $n \in \mathbb{N}$, there holds the following exact expression:

$$\mathcal{L}^{(l)}_{(n)}(f) = \int_{\mathbb{R}^3} d^3x_1 \int_{\mathbb{R}^3} d^3x_2 ... \int_{\mathbb{R}^3} d^3x_n \prod_{j=1}^n \exp[if(x_j)]\exp(-\beta V_n^{(l)}), \tag{245}$$

where $V_n^{(l)}$—the long term part potential energy of an $n-$particle group of the system. Then we get that

$$\mathcal{L}^{(l)}(f) := \sum_{n \in \mathbb{Z}_+} \frac{z^n}{n!}\mathcal{L}^{(l)}_{(n)}(f)Q_0^{-1}, \quad Q_0 := (\sum_{n \in \mathbb{Z}_+} \frac{z^n}{n!}\mathcal{L}^{(l)}_{(n)}(0))^{-1}. \tag{246}$$

The sum in (246) can be calculated exactly, taking into account the expression

$$\mathcal{L}^{(l)}_{(n)}(f) = \int \mathcal{D}(\omega)\{z\int_{\mathbb{R}^3} d^3x \exp[if(x)]g(x;\omega)\}^n J(\omega), \tag{247}$$

where $\mathcal{D}(\omega) := \prod_{k \in \mathbb{R}^3} \frac{i}{2}(d\omega_k^* \wedge d\omega_k)$, $\omega_k^* := \omega_{-k} \in \mathbb{C}$, $k \in \mathbb{R}^3$,

$$g(x;\omega) := \exp\left[-2\pi i(\int_{\mathbb{R}^3} d^3k\omega_k \exp(ikx) + \frac{\beta}{2}\int_{\mathbb{R}^3} d^3k\nu(k)\right],$$

$$J(\omega) := \exp\left[-\int_{\mathbb{R}^3} d^3k \frac{2\pi^2}{\beta\nu(k)}\omega_k\omega_{-k} + \int_{\mathbb{R}^3} d^3k \ln\frac{\pi}{\beta\nu(k)}\right] \tag{248}$$

and $\nu(k) := (2\pi)^{-3}\int_{\mathbb{R}^3} d^3x V^{(l)}(x)\exp(-ikx)$, $k \in \mathbb{R}^3$. Now, from (246)–(248) one easily finds that

$$\mathcal{L}^{(l)}(f) = \int \mathcal{D}(\omega)\exp(\bar{z}\int_{\mathbb{R}^3} d^3x\{\exp[if(x)] - 1\}g(x;\omega))J^{(l)}(\omega)Q^{-1}, \tag{249}$$

where $\bar{\zeta} := \varsigma \exp(\frac{\beta}{2} \int_{\mathbb{R}^3} d^3k \nu(k)) = \varsigma \exp[\frac{\beta}{2} V^{(l)}(0)]$ and the function $J^{(l)}(\omega), \omega \in \mathbb{R}^3$, allows the following series expansion:

$$J^{(l)}(\omega) := J(\omega) \exp\left[\int_{\mathbb{R}^3} d^3x g(x; \omega)\right] = J(\omega) \exp\left[-\frac{(2\pi)^2}{2!}(2\pi)^3 \int_{\mathbb{R}^3} d^3k \omega_k \omega_{-k} \right. \tag{250}$$

$$\left. + \sum_{n \neq 2} \frac{(-2\pi i)^n}{n!}(2\pi)^3 \int_{\mathbb{R}^3} d^3k_1 \int_{\mathbb{R}^3} d^3k_2 ... \int_{\mathbb{R}^3} d^3k_n \prod_{j=1}^{n} \omega_{k_j} \delta\left(\sum_{J=1}^{N} k_j\right)\right].$$

The expression (249) can now be represented [22,115,117,118] in the following cluster Ursell form:

$$\mathcal{L}^{(l)}(f) = \exp\left(\sum_{n=1}^{\infty} \frac{\bar{z}^n}{n!} \int_{\mathbb{R}^3} d^3x_1 \int_{\mathbb{R}^3} d^3x_2 ... \int_{\mathbb{R}^3} d^3x_n \prod_{j=1}^{n} \{\exp[if(x)] - 1\} g_n(x_1, x_2, ..., x_n)\right). \tag{251}$$

Here for any $n \in \mathbb{Z}_+$

$$g_n(x_1, x_2, ..., x_n) := \sum_{\sigma[n]} (-1)^{m+1} (m-1)! \prod_{j=1}^{m} R_{\sigma[j]}(x_k \in \sigma[j]),$$

$$R_n(x_1, x_2, ..., x_n) := \sum_{\sigma[n]} \prod_{j=1}^{m} g_{\sigma[j]}(x_k \in \sigma[j]), \tag{252}$$

where $g_n(x_1, x_2, ..., x_n)$, $n \in \mathbb{N}$, are called the $n-$particle Ursell cluster functions, $R_n(x_1, x_2, ..., x_n)$, $n \in \mathbb{N}$, are suitable "correlation" functions [20,56,68] and $\sigma[n]$ denotes a partition of the set $\{1, 2, ..., n\}$ into non-intersecting subsets $\{\sigma[j] : j = \overline{1, m}\}$, that is $\sigma[j] \cap \sigma[k] = \varnothing$ for $j \neq k = \overline{1, m}$, and $\sigma[n] = \cup_{j=1}^{m} \sigma[j]$. Having separated from the function $J^{(l)}(\omega), \omega \in \mathbb{C}^3$, the natural "Gaussian" part $J_0^{(l)}(\omega), \omega \in \mathbb{C}^3$, one can write down that

$$g_1(x_1) = G(\xi_k^{(1)})/G(0), \quad g_2(x_1, x_2) = G(\xi_k^{(2)})/G(0) - g_1(x_1)g_1(x_2), ..., \tag{253}$$

where $\xi_k^{(n)} := -2\pi i \sum_{s=1}^{n} \exp(ikx_s)$, $k \in \mathbb{R}^3$, $n \in \mathbb{N}$,

$$G(\xi_k^{(n)}) := \exp[\mathrm{M}(\xi_k^{(n)})] \int D(\omega) g^{(l)}(\xi_k^{(n)}; \omega) J_0(\omega),$$

$$\mathrm{M}(\xi_k^{(n)}) := \sum_{m \neq 2} \frac{(-2\pi i)^m}{m!} (2\pi)^3 \int_{\mathbb{R}^3} d^3k_1 \int_{\mathbb{R}^3} d^3k_2 ... \int_{\mathbb{R}^3} d^3k_m \delta\left(\sum_{s=1}^{m} k_s\right) \prod_{s=1}^{m} \frac{\delta}{\delta \xi_{k_s}^{(n)}},$$

$$g^{(l)}(\xi_k^{(n)}; \omega) := \prod_{j=1}^{n} g(x_j; \omega). \tag{254}$$

Since the integrals $\int \mathcal{D}(\omega) g^{(l)}(\xi_k^{(n)}; \omega) J^{(l)}(\omega)$, $n \in \mathbb{N}$, one can calculate exactly, the formulae (251) and (253) are sources of the so called "virial" variables for Ursell–Mayer "cluster" correlation functions $g_n(x_1, x_2, ..., x_n)$, $n \in \mathbb{N}$, having important applications. In particular, from the function $J^{(l)}(\omega), \omega \in \mathbb{C}^3$, one gets right away that the cluster expansion for the functions $g_n(x_1, x_2, ...x_n)$, $n \in \mathbb{N}$, are fulfilled by means of the "screened" potential function $\bar{V}^{(l)}(x - y)$, $x, y \in \mathbb{R}^3$, where

$$\bar{V}^{(l)}(x - y) := \int_{\mathbb{R}^3} d^3k \frac{\nu(k) \exp[ik(x - y)]}{1 + \nu(k) \beta \bar{z}(2\pi)^3}. \tag{255}$$

In particular, from (237) and (251) one easily finds that

$$f_1(x_1) = z \int \mathcal{D}(\omega) g(x; \omega) J^{(l)}(\omega) \left[ \int \mathcal{D}(\omega) J^{(l)}(\omega) \right]^{-1} =$$

$$= \bar{\rho} \simeq \bar{z} \exp \left[ \frac{\beta}{2} \int d^3 k \frac{\beta \nu^2(k)(2\pi)^3 \bar{z}}{1 + \nu(k)\beta \bar{z}(2\pi)^3} \right],$$

$$f_2(x_1, x_2) = z^2 \int \mathcal{D}(\omega) g(x_1; \omega) g(x_2; \omega) J^{(l)}(\omega) \left[ \int \mathcal{D}(\omega) J^{(l)}(\omega) \right]^{-1} \simeq$$

$$\simeq \bar{\rho}^2 \exp[-\beta \bar{V}^{(l)}(x_2 - x_1)] \left\{ 1 + \bar{\rho} \int_{\mathbb{R}^3} d^3 x_3 [\exp\left(-\beta \bar{V}^{(l)}(x_1 - x_3)\right) - 1 \right.$$

$$+ \beta \bar{V}^{(l)}(x_1 - x_3)][\exp\left(-\beta \bar{V}^{(l)}(x_2 - x_3)\right) - 1 + \beta \bar{V}^{(l)}(x_2 - x_3]$$

$$+ \bar{\rho} \int_{\mathbb{R}^3} d^3 x_3 [-\beta \bar{V}^{(l)}(x_1 - x_3)][\exp(-\beta \bar{V}^{(l)}(x_2 - x_3)) - 1 + \beta \bar{V}^{(l)}(x_2 - x_3)]$$

$$+ \bar{\rho} \int_{\mathbb{R}^3} d^3 x_3 [-\beta \bar{V}^{(l)}(x_2 - x_3)][\exp(-\beta \bar{V}^{(l)}(x_1 - x_3)) - 1 + \beta V^{(l)}(x_1 - x_3)] + \left. \right\} \dots \tag{256}$$

and so on. The result, presented above, can be obtained by means of slightly formal calculations, based on generalized functions and operator theories [22,115,118,147]. Really, as $\hbar \to 0$ one has that

$$\mathcal{L}^{(l)}(\mathrm{f}) = \exp[-\beta \mathrm{V}^{(l)}(\delta)] \mathcal{L}_0(\mathrm{f}) Q^{-1} = \tag{257}$$

$$= tr \left\{ \exp(-\beta \mathrm{H}_0^{(\mu)}) \exp\left[ -\frac{\beta}{2} \int_{\mathbb{R}^3} d^3 k \nu(k) : \rho_k \rho_{-k} : \right] \exp[i\rho(\mathrm{f})] \right\}$$

$$= tr \left\{ \exp(-\beta \mathrm{H}_0^{(\mu)}) \exp\left[ \frac{\beta}{2} \int_{\mathbb{R}^3} d^3 k \nu(k) \int_{\mathbb{R}^3} d^3 x \rho(x) \right] \right.$$

$$\times \int \mathcal{D}(\omega) \exp\left[ -\int_{\mathbb{R}^3} d^3 k \frac{2\pi^2}{\beta \nu(k)} \omega_k \omega_{-k} - \int_{\mathbb{R}^3} d^3 k 2\pi i \omega_k \rho_k \right] \exp[i\rho(\mathrm{f})] \right\} Q^{-1}$$

$$= \int \mathcal{D}(\omega) J(\omega) tr \left\{ \exp(-\beta \mathrm{H}_0^{(\mu)}) \exp\left[ i \left( \rho, \mathrm{f} - 2\pi \int_{\mathbb{R}^3} d^3 k \omega_k \exp(ikx) - \frac{i\beta}{2} \int_{\mathbb{R}^3} d^3 k \nu(k) \right) \right] \right\} Q^{-1}$$

$$= \int \mathcal{D}(\omega) J(\omega) \mathcal{L}_0(\mathrm{f} - 2\pi \int_{\mathbb{R}^3} d^3 k \omega_k \exp(ikx) - \frac{i\beta}{2} \int_{\mathbb{R}^3} d^3 k \nu(k)) Q^{-1}$$

$$= \int \mathcal{D}(\omega) J^{(l)}(\omega) \exp\left( \int_{\mathbb{R}^3} d^3 k \{\exp[i\mathrm{f}(x)] - 1\} g(x; \omega) \right),$$

where $\mathrm{H}_0^{(\mu)} := \mathrm{H}_0 - \lambda \int_{\mathbb{R}^3} d^3 x \rho(x)$, $\rho_k := \int_{\mathbb{R}^3} d^3 x \rho(x) \exp(ikx)$, $k \in \mathbb{R}^3$. The expression (257) coincides exactly with that of (251), thereby proving the validity of our expressions (240) and (243) for the N.N. Bogolubov type generating functional $\mathcal{L}(\mathrm{f})$, $\mathrm{f} \in F$, satisfying the functional Equation (242) of Proposition (8).

## 8. The Functional-Analytic Solution and Its Ursell–Mayer Type Diagram Expansion

Having considered (243) and (249) as starting expressions with just known functions $g_n(x_1, x_2, ...x_n)$, $n \in \mathbb{N}$, for the functional $\mathcal{L}(\mathrm{f})$, $\mathrm{f} \in F$, one can obtain the following expansion:

$$\mathcal{L}(f) = Z(f)/Z(0), \ \ Z(f) = \exp[-\beta V^{(s)}(\delta)]\mathcal{L}^{(l)}(f)$$

$$= \exp[-\beta V^{(s)}(\delta)] \exp\left[\sum_{n=1}^{\infty} \frac{z^n}{n!} \int_{\mathbb{R}^3} d^3x_1 \int_{\mathbb{R}^3} d^3x_2 ... \int_{\mathbb{R}^3} d^3x_n\right.$$

$$\left. \times \prod_{j=1}^{n} \{\exp[if(x_j)] - 1\} g_n(x_1, x_2, ..., x_n)\right] \tag{258}$$

$$= \exp\left[\sum_{N=1}^{\infty} \frac{1}{N!} W(G_N^{(c)})\right],$$

where functionals $W(G_N^{(c)})$, $N \in \mathbb{N}$, are calculated via the following rule. Denote by $G_N^{(c)}$, $N \in \mathbb{N}$, such a connected graph that: it consists of exactly $N$ generalized vertices of $[\gamma(n_j)]$ type, $j = \overline{1, N}$, and $\sum_{j=1}^{N} n_j$ ordinary vertices of $[\alpha]$ type. Moreover, each vertex $[y(n)]$ is necessarily connected with $n$ vertices of type $[\alpha]$ by means of dashed lines each to other, and $[\alpha]$ vertices can be connected arbitrarily by means of uniform lines. If now, to attribute to each generalized $[\gamma(n)]-$vertex—the factor $g_n(x_1, x_2, ...x_n)$, to each simple $[\alpha]-$vertex—the factor $\varsigma \int_{\mathbb{R}^3} d^3x \exp[if(x)]$, and to the line connecting them—the factor $\{exp[-\beta V^{(s)}(x_{l_1} - x_{l_2})] - 1\}$, then the obtained resulting expression will be exactly equal to the functional $W(G_N^{(c)})$. The final summing up over all such connected graphs gives rise to the expression (257), where the factor $1/N!$ counts the symmetry order of the graph $G_N^{(c)}$ under the generalized vertices permutations. It is evident that, by representing the factor $\exp[if(x)]$, entering the vertex $[\alpha]$, as $\{\exp[if(x)] - 1\} + 1$, the expression (257) can easily be resumed into Ursell–Mayer type expressions but already with other suitable $g_n$—functions, replacing the former ones, giving rise to expansions similar to (256), based already on the "screened" potential (255).

Thereby, we can formulate, taking into account the results of [20,68], the next proposition, characterizing the Bogolubov type generating functional $\mathcal{L}(f)$, $f \in F$, satisfying the functional Equation (242).

**Proposition 9.** *Let the Bogolubov type generating functional $\mathcal{L}(f)$, $f \in \mathcal{S}(\mathbb{R}^3; \mathbb{R})$, represented analytically as a series (258) of graph-generated functionals, satisfy the following conditions:*

*(i) continuity with respect to the natural topology on $F$, $|\mathcal{L}(f)| \leq 1$, $f \in F$;*

*(ii) positivity: $\sum_{j,k=1}^{n} c_j c_k^* \mathcal{L}(f_j - f_k) \geq 0$ for any $f_j \in F$ and all $c_j \in \mathbb{C}$, $j = \overline{1, n}$, $n \in \mathbb{N}$;*

*iii) symmetry and normalization conditions: $\mathcal{L}^*(f) = \mathcal{L}(-f)$ for all $f \in F$ and $\mathcal{L}(0) = 1$;*

*(iv) translational-invariance: $\mathcal{L}(f) = \mathcal{L}(f_a)$, where $f_a(x) := f(x - a)$, $x, a \in \mathbb{R}^3$, for any $f \in \mathcal{S}(\mathbb{R}^3; \mathbb{R})$;*

*(v) cluster condition or, equivalently, the Bogolubov correlation decay: $\lim_{\lambda \to \infty}[\mathcal{L}(f + g_{\lambda a}) - \mathcal{L}(f_a)\mathcal{L}(g_{\lambda a})] = 0$, $a \in \mathbb{R}^3$, for any $f, g \in F$;*

*(vi) density condition: $\frac{1}{i}\frac{\delta \mathcal{L}(f)}{\delta f(x)}|_{f=0} = \bar{\rho} \in \mathbb{R}_+$.*

*Then the functional (258) solves the Bogolubov type functional equation (242), allowing the positive measure $d\bar{\mu}$, whose Fourier representation on the adjoint tempered generalized functions space $F'$ is exactly*

$$\mathcal{L}(f) = \int_{F'} d\bar{\mu}(\xi) \exp[i\xi(f), \tag{259}$$

*where convolution $\xi(f) := \int_{\mathbb{R}^3} d^3x \xi(x) f(x)$ for $\xi \in F'$ and $f \in F$.*

The obtained result makes it possible to find the many-particle distribution functions (237) and apply them to constructing different thermodynamic functions important [56,65] for applications.

Below, following the Bogolubov method [86], we obtain, based on the expression (245), the important Kirkwood–Saltzbourg–Simansic functional equation for the Bogolubov

generating functional $\mathcal{L}(\mathrm{f})$, $\mathrm{f} \in F$. Namely, making use of the expression (245) we can write down the following relationship:

$$\frac{1}{i}\frac{\delta \mathcal{L}_{(N+1)}(\mathrm{f})}{\delta \mathrm{f}(x)} = \exp[i\mathrm{f}(x)]\frac{(N+1)Z_N}{Z_{N+1}}\mathcal{L}_{(N)}(\mathrm{f}(\cdot) + i\beta V(\cdot - x)) \tag{260}$$

for any $x \in \mathbb{R}^3$, where $Z_N := \int_{\mathbb{R}^{3N}} dx_1 dx_2 ... dx_N \exp(-\beta V_N)$, $N \in \mathbb{N}$.

Since, by definition, $\lim_{N \to \infty} \mathcal{L}_{(N)}(\mathrm{f}) = \mathcal{L}(\mathrm{f})$, $\mathrm{f} \in F$, $\lim_{N \to \infty} \frac{(N+1)Z_N}{Z_{N+1}} := \varsigma \in \mathbb{R}_+$, from (260) one gets right away that

$$\exp[-i\mathrm{f}(x)]\frac{1}{i}\frac{\delta \mathcal{L}(\mathrm{f})}{\delta \mathrm{f}(x)} = \varsigma \mathcal{L}(\mathrm{f}(\cdot) + i\beta V(\cdot - x)), \tag{261}$$

which is called the Kirkwood–Saltzburg–Symanzik functional equation, being very important for proving the Proposition (9) by means of the classical Leray–Schauder fixed point theorem [56,141,148] in some suitably defined Banach space. In particular, at $\mathrm{f} = 0$ from (261) one finds the following important relationship:

$$\bar{\rho} = \varsigma \mathcal{L}(i\beta V(\cdot - x)) \tag{262}$$

for any $x \in \mathbb{R}^3$.

*Conclusions*

In the article, we have showed that the N.N. Bogolubov generating functional method is a very effective tool for studying distribution functions of both equilibrium and non equilibrium states of classical many-particle dynamical systems. In some cases, the N.N. Bogolubov generating functionals can be represented by means of infinite Ursell–Mayer diagram expansions, whose convergence holds under some additional constraints on a statistical system. We also have shown that the Bogolubov idea [56] to use the Wigner density operator transformation to study the non equilibrium distribution functions proved to be very effective, having proposed a new analytic form of non-stationary solutions to the classical N.N. Bogolubov evolution functional equation.

## 9. The Wigner Type Current Algebra Representation and Its Application to Non-Equilibrium Classical Statistical Mechanics

*9.1. Many-Particle Distribution Functions Space and Its Poissonian Structure*

In the case of non-stationary (non-equilibrium) states of the many-particle dynamical systems, the Bogolubov's generating functional (236) does not possess all needed information. To specify this case, we introduce the generating representation functional:

$$\mathcal{L}(\mathrm{f}, \mathrm{g}) = (\Omega| \exp[i\rho(\mathrm{f})] \exp[iJ(\mathrm{g})]\Omega) = \mathrm{Tr}(P \exp[i\rho(\mathrm{f})] \exp[i\,J(\mathrm{g})]), \tag{263}$$

where $\Omega \in \Phi_\mu$ is a cyclic vector of the representation of the current group $G$, satisfying the following additional conditions:

$$\mathrm{T}\rho(\mathrm{f})\mathrm{T}^{-1} = \rho(\mathrm{f}), \quad \mathrm{T}\Omega = \Omega^*, \quad \mathrm{T}J(\mathrm{g})\mathrm{T}^{-1} = -J(\mathrm{g}), \quad \mathrm{T}\mathrm{H}\mathrm{T}^{-1} = \mathrm{H},$$

with the mapping $T : \mathbb{R} \ni t \to -t \in \mathbb{R}$ being the operator of time inversion, and $f \in \mathcal{J}(\mathbb{R}^3; \mathbb{R})$, $\mathrm{g} \in \mathcal{J}(\mathbb{R}^3; \mathbb{R}^3)$ taken arbitrary. In the $N$-particle representation of the current Lie algebra $\mathcal{G}$ (29) for any finite $N \in \mathbb{N}$ the functional $\mathcal{L}(f, \mathrm{g})$ (263) allows the following [5–7] standard finite-particle form:

$$\mathcal{L}(\mathrm{f}, \mathrm{g}) = \int_{\mathbb{R}^3} dx_1 ... \int_{\mathbb{R}^3} dx_N \Omega^*(x_1, ..., x_N) \prod_{j=1}^{N} \exp[i\mathrm{f}(x_j)] \times \tag{264}$$

$$\times \exp[i\xi(x_j, \mathrm{g})]\Omega(x_1, ..., x_N),$$

where $\xi(x,g) = \frac{1}{2i}[g(x)\nabla_x + \nabla_x g(x)]$, $x \in \mathbb{R}^3$, and $\Omega \in L_2(\mathbb{R}^{3N};\mathbb{C})$ is a cyclic state. The operator $\exp[i\xi(x,g)]$ acts on any function $\omega_N \in L_2(\mathbb{R}^{3N};\mathbb{C})$ by the rule:

$$\exp[i\xi(x,g)]\omega_N(x_1,...,x_N) = (\phi^*\omega_N)(x_1,...,x_N)\left(\det\left\|\frac{\partial\phi(x)}{\partial x}\right\|\right)^{1/2}$$

where $\phi \in \mathrm{Di}ff(\mathbb{R}^3)$ is a diffeomorphism of $\mathbb{R}^3$, corresponding to the vector field $g \in \mathcal{J}(\mathbb{R}^3;\mathbb{R}^3)$, that is $\phi(x) = \phi_t^g$, where $\frac{d}{dt}\phi_t^g = g(\phi_t^g(x))$, $x \in \mathbb{R}^3$. For $N \to \infty$ the expression (264) becomes

$$\mathcal{L}(f,g) = \sum_{n\in\mathbb{Z}_+}\frac{1}{n!}\int_{\mathbb{R}^3}dx_1...\int_{\mathbb{R}^3}dx_n\int_{\mathbb{R}^3}dy_1...\int_{\mathbb{R}^3}dy_n\prod_{j=1}^{n}\left[\delta(x_j - y_j)\times\right. \tag{265}$$
$$\left.\times\left\{\exp[if(x_j)]\exp[i\xi(x_j,g)] - 1\right\}f_n(y_1,...,y_n;x_1,...,x_n)\right]$$

where for all $n \in \mathbb{N}$ Bogolubov's quantum distribution functions [56] are

$$f_n(y_1,...,y_n;x_1,...,x_n) = (\Omega|\psi^+(y_n)...\psi^+(y_1)\psi(x_1)...\psi(x_n)\Omega) \tag{266}$$

and satisfy the compatibility conditions

$$f_n(x_1,...,x_n) := f_n(x_1,...,x_n;x_1,...,x_n), \tag{267}$$

where $x_j = \mathbb{R}^3$, $j = \overline{1,n}$, $n \in \mathbb{N}$.

To proceed with the further study of the classical distribution functions of the many-particle dynamical system, when the inverse temperature $\beta \to 0$, and the Planck constant $\hbar \to 0$. Let us introduce [21,22,149,150] the following quantized selfadjoint Wigner operator $w(x,p) : \Phi_W \to \Phi_W$, $(x,p) \in T^*(\mathbb{R}^3)$

$$w(x,p) = \frac{1}{(2\pi)^n}\int_{\mathbb{R}^3}d\alpha\,\exp(i\langle\alpha|p\rangle)\psi^+\left(x + \frac{\hbar\alpha}{2}\right)\psi\left(x - \frac{\hbar\alpha}{2}\right), \tag{268}$$

where, by definition, $\Phi_W := \lim_{\beta\to\infty}\Phi_\mu$ is the corresponding Hilbert space for the constructed Wigner type current algebra representation with the generating cyclic vector $\Omega \in \Phi_W$. Performing transformation (268) in the expression (263), we can find that

$$\mathcal{L}(f,g) \to \mathcal{L}(\tilde{f}) = \sum_{n\in\mathbb{Z}_+}\frac{1}{n!}\int_{\mathbb{R}^3\times\mathbb{R}^3}dx_1dp_1...\times$$
$$\times\int_{\mathbb{R}^3\times\mathbb{R}^3}dx_ndp_n\prod_{j=1}^{n}\left\{\exp(i\tilde{f}(x_j,p_j)) - 1\right\}f_n(x_1,p_1;...;x_n,p_n), \tag{269}$$

for some functions $\tilde{f} \in \mathcal{J}(\mathbb{R}^3 \times \mathbb{R}^3;\mathbb{R}^3)$. From the expression (269) it also follows that

$$\mathcal{L}(f) = (\Omega|\exp[iw(f)]\Omega) = \mathrm{Tr}(P\exp[iw(f)]) \tag{270}$$

where $w(f) = \int_{T^*(\mathbb{R}^3)}dxdp\,w(x,p)f(x,p)$, $\tilde{f} \in \mathcal{J}(\mathbb{R}^3 \times \mathbb{R}^3;\mathbb{R}^3)$, $P : \Phi_W \to \Phi_W$ is the Gibbs statistical operator and $\mathrm{Tr}: \mathrm{End}(\Phi_W) \to \mathbb{C}$ is the corresponding trace-operator, defined on the space $\mathcal{B}(\Phi_W)$ of the nuclear operators on the corresponding Hilbert space representation $\Phi_W$. The corresponding quantum current Lie algebra $\mathcal{G}$ suitably transforms [55,56] into the Abelian Lie algebra $\mathcal{G}_W$ of the operator functionals $\{w(f) \in \mathcal{G} : f \in \mathcal{J}(\mathbb{R}^3 \times \mathbb{R};\mathbb{R})\}$.

Consider now a quantum dynamical system of many identical particles with the average nonvanishing density $\bar{\rho} = \lim_{\Lambda\nearrow\mathbb{R}^3}(N/A) \in \mathbb{R}_+^1\setminus\{0\}$ as $N \to \infty$ and $\Lambda \nearrow \mathbb{R}^3$ in the

Van Hove's sense [150,151]. Then, according to [1,4,59], the Hamiltonian operator (234) in the Wigner representation (268) looks as

$$\mathrm{H} = \int\limits_{T^*(\mathbb{R}^3)} dz \frac{p^2}{2m} w(z) + \int\limits_{T^*(\mathbb{R}^3)} dz \int\limits_{T^*(\mathbb{R}^3)} dz' V(x - y') : w(z)w(z') :, \tag{271}$$

where $z = (x, p) \in T^*(\mathbb{R}^3), z' = (y, q) \in T^*(\mathbb{R}^3)$ and $dz = dxdp$, $dz' = dydq$ are the standard phase space measures in $T^*(\mathbb{R}^3)$ and the ordering : : operation is naturally inherited from (238). According to the Heisenberg's principle [56], the evolution equation with respect to temporal variable $t \in \mathbb{R}_+$ for an arbitrary observable operator quantity $\mathrm{A} : \Phi_W \to \Phi_W$ in the Wigner type representation space $\Phi_W$ is

$$d\mathrm{A}/dt = \frac{i}{\hbar}[\mathrm{H}, \mathrm{A}], \tag{272}$$

where $[\cdot, \cdot]$ is a usual operator commutator, naturally ensued from that on the Hilbert space $\Phi_\mu$. Following [20,56,152–154], one can state, that for $\hbar \to 0$ in the weak sense the following theorem is true.

**Theorem 5.** *Let us denote $\mathcal{M}$ as an algebra of the self-adjoint operators with $A(\mathcal{G})$ in the Wigner representation. Then, the operator bracket $[\cdot, \cdot]_0 = \lim\limits_{\hbar \to 0}[\cdot, \cdot]$ on the algebra $\mathcal{M}$ in the weak sense is equivalent to*

$$[a_j, a_n]_0 = \sum_{k=1}^{\min\{j,n\}} \int\limits_{T^*(\mathbb{R}^3)} dz_1 ... \int\limits_{T^*(\mathbb{R}^3)} dz_k : w(z_1)...w(z_k) \times \tag{273}$$

$$\times \left\{ \frac{\delta^k a_j}{\delta w(z_1)...\delta w(z_k)}, \frac{\delta^k a_n}{\delta w(z_1)...\delta w(z_k)} \right\}^{(k)} :,$$

*where $\{\cdot, \cdot\}^{(k)}$ is a standard canonical Poisson bracket on the phase space of $k \in \mathbb{N}$ particles.*

The statement (273) could be proved by means of the next general Bohr–Dirac correspondence principle in the quasi-classical approach:

$$\lim_{\hbar \to 0} \frac{i}{\hbar}[a, b] = \{a, b\}^{(N)}, \tag{274}$$

where $N \in \mathbb{N}$ is a maximal number of the particle in the system and $a, b \in A(\mathcal{G})$ are operators in $N$-particle Hilbert space representation $\Phi_N = L_2(\mathbb{R}^{3N}; \mathbb{C})$, $F = \sum\limits_{j=1}^{N} \delta(x - x_j)$.

Here, it is worth making the following corollary.

*Corollary.* Algebra of the operators of the observable quantities $A(\mathcal{G})$ for $\hbar \to 0$ allows "*hierarchical*" representation

$$A(\mathcal{G}) = \sum_{j \in \mathbb{Z}_+} A_j(\mathcal{G}) \Rightarrow \mathcal{M} = \bigoplus_{j = \mathbb{Z}_+} A_j(\mathcal{G}) \tag{275}$$

along with Lie bracket $[\![\cdot, \cdot]\!]$, which is inducted by the bracket $[\cdot, \cdot]_0$ (273):

$$[\![a, b]\!] = \bigoplus_{l \in \mathbb{Z}_+} \sum_{j,k \in \mathbb{Z}_+} [a_j, b_k]_0^{(l)}, \tag{276}$$

where $a, b \in \mathcal{M}$ in the Wigner representation and the following expansions hold

$$a = \sum_{j \in \mathbb{Z}_+} a_j, \quad b = \sum_{j \in \mathbb{Z}_+} b_j, \quad [a_j, b_k]_0 = \sum_{l \in \mathbb{Z}_+} [a_j, b_k]_0^{(l)}. \tag{277}$$

Consider now the following linear mapping $\alpha : \mathcal{M} \to A(\mathcal{G})$, where

$$\alpha\left( \underset{j \in \mathbb{Z}_+}{\oplus} a_j \right) = \sum_{j = \mathbb{Z}_+} a_j \in A(\mathcal{G}), \tag{278}$$

and the Lie bracket $[\![\cdot, \cdot]\!]$ is defined in $\mathcal{M}$, and the corresponding Lie bracket $[\cdot, \cdot]_\alpha$ (273) in $A(\mathcal{G})$. Let us consider the dual to (278) mapping $\alpha^* : A(\mathcal{G})^* \to \mathcal{M}^*$, where

$$\mathcal{M}^* = \underset{l \in \mathbb{Z}_+}{\oplus} \mathcal{M}_j^*, \quad \mathcal{M} = \underset{l \in \mathbb{Z}_+}{\oplus} \mathcal{M}_j, \tag{279}$$

$$\mathcal{M}^* = \sum_{j \in \mathbb{Z}_+} \left\{ P \in A_j(\mathcal{G})^* : F(a) = \mathrm{Tr}(Pa), \ a \in A(\mathcal{G}) \right\}.$$

Here $P : \Phi_\mu \to \Phi_\mu$ is statistic operator of the initial dynamical system (271), which satisfy the Heisenberg–Liouville equation

$$dP/dt = \frac{i}{\hbar}[P, \mathrm{H}] \tag{280}$$

for all $t \in \mathbb{R}_+$. The expression (280), according to (274), transforms into the quasi-classical Liouville equation in the Wigner representation.

It is easy to check that for element $F \in A(\mathcal{M})^*$ the expression

$$\alpha^* F = (f_1, ..., f_j, ...) = \mathcal{F} \in \mathcal{M}^* \tag{281}$$

defines the representation on the space $\mathcal{M}^*$ of the distribution functions

$$f_j(z_1, ..., z_j) = \mathrm{Tr}(P : w(z_1)...w(z_j) :), \tag{282}$$

where $z_j \in T^*(\mathbb{R}^3)$, $j \in \mathbb{Z}_+$, and for any $a \in \mathcal{M}$

$$a(\mathcal{F}) = \sum_{j \in \mathbb{Z}_+} \int_{T^*(\mathbb{R}^3)} dz_1 ... \int_{T^*(\mathbb{R}^3)} dz_j f_j(z_1, ..., z_j) a_j(z_1, ..., z_j). \tag{283}$$

Let $b(F), c(F) \in D(A(\mathcal{G})^*)$ be linear functionals on $A(\mathcal{G})^*$, then on $D(A(\mathcal{G})^*)$ the standard [59] Lie-Poisson bracket $\{\cdot, \cdot\}_0$ is defined via the rule

$$\{b(F), c(F)\}_0 = F([b, c]_0), \tag{284}$$

where $b, c \in A(\mathcal{G})$ are such that $F(b) = b(F)$, $F(c) = c(F)$, $F \in A^*(\mathcal{G})$. In the same way, the dual Lie–Poisson bracket $\{\!\{\cdot, \cdot\}\!\}$ is defined on the set of functionals $D^*(\mathcal{M})$ over the adjoint space $\mathcal{M}$ (279)

$$\{\!\{b(\mathcal{F}), c(\mathcal{F})\}\!\} = F([\![b, c]\!]), \tag{285}$$

where $F(b) = b(\mathcal{F})$, $F(c) = c(\mathcal{F})$, $F \in \mathcal{M}^*$.

**Definition 16.** *It is said that mapping of the Lie algebras $\alpha : \mathcal{M} \to A(\mathcal{G})$ is canonical (or Poissonian [59]), if for all $b(\mathcal{F})$ and $c(\mathcal{F})$ the following equality holds*

$$\alpha^* \{b(\mathcal{F}), c(\mathcal{F})\}_0 = \{\!\{\alpha^* b(\mathcal{F}), \alpha^* c(\mathcal{F})\}\!\}, \tag{286}$$

*where $\mathcal{F} = \alpha^* F \in \mathcal{M}^*$.*

From reasonings presented above we can formulated the following proposition.

**Proposition 10.** *Let A and $\mathcal{M}$ be two arbitrary Lie algebras and $\alpha : \mathcal{M} \to A$ be a linear mapping. Then dual mapping $\alpha^* : D(A(\mathcal{G})^*) \to D(\mathcal{M}^*)$ is canonical if $\alpha : \mathcal{M} \to A$ is Lie algebras homomorphism.*

As a consequence of the statement above, one derives the next theorem.

**Theorem 6.** *Dual mapping $\alpha^* : D(A(\mathcal{G})^*) \to D(\mathcal{M}^*)$, which was built by means of the hierarchical Lie algebra of the operators $\mathcal{M}$, is canonical.*

Let us consider the generating functional $\mathcal{L}(f)$, $f \in \mathcal{J}(T^*(\mathbb{R}^3); \mathbb{R})$, defined by expression (270) in Wigner representation, and apply the developed above algebraic technique to the calculation of the following quantity:

$$\frac{d}{dt}\mathcal{L}(f) = \lim_{\hbar \to 0} \frac{i}{\hbar} \mathrm{Tr}(P[\mathrm{H}, : \exp[iw(e^{if}-1)]) :]) \tag{287}$$

for the evolution with respect to the temporal parameter $t \in \mathbb{R}$. From (270) one can easily obtain that

$$\frac{d}{dt}\mathcal{L}(f)(\mathcal{F}) = \mathrm{Tr}(P[\mathrm{H}, : \exp(iw(e^{if}-1)) :]) = \alpha^* \{\mathcal{H}(\mathcal{F}), \mathcal{L}(f)(\mathcal{F})\}_0, \tag{288}$$

where for all $\mathcal{F} \in A(\mathcal{G})^*$ the Hamiltonian functional $\mathcal{H}(\mathcal{F}) \in D(A(\mathcal{G})^*)$ is given as

$$\mathcal{H}(\mathcal{F}) = \mathrm{Tr}(P\mathrm{H}) = \int\limits_{T^*(\mathbb{R}^3)} dz T(p) f_1(z) + \tag{289}$$

$$+ \frac{1}{2} \int\limits_{T^*(\mathbb{R}^3)} dz_1 \int\limits_{T^*(\mathbb{R}^3)} dz_2 V(x_1 - x_2) f_2(z_1, z_2).$$

Based on (288) and Theorem 6, we immediately obtain the Hamiltonian evolution equation

$$\frac{d}{dt}\mathcal{L}(f)(\mathcal{F}) = \{\{\mathcal{L}(f)(\mathcal{F}), \mathcal{H}(\mathcal{F})\}\}, \tag{290}$$

where $t \in \mathbb{R}$, $\mathcal{L}(f)(\mathcal{F}) = \alpha^*\mathcal{L}(f)(F)$, $\mathcal{H}(\mathcal{F}) = \alpha^*H(F)$ and $\mathcal{F} \in \mathcal{M}^*$ is arbitrary. Thus, the following theorem is stated.

**Theorem 7.** *The generating Wigner type representation functional $\mathcal{L}(f)(\mathcal{F})$ (270) on the phase space $D(\mathcal{M})$ satisfies the Hamiltonian dynamical system (290) with respect to the Lie–Poisson bracket (285) and Hamiltonian function (289), taken as a smooth functional on $\mathcal{M}^*$.*

Using Equation (290) and formulae (273), (276), we finally get the following non-equilibrium functional Bogolubov's equation [143]

$$\frac{d}{dt}\mathcal{L}(f) = \int\limits_{T^*(\mathbb{R}^3)} dz \{\frac{1}{i} \frac{\delta \mathcal{L}(f)}{\delta f(z)}, T(p)\}^{(1)} + \tag{291}$$

$$+ \frac{1}{2} \int\limits_{T^*(\mathbb{R}^3)} dz_1 \int\limits_{T^*(\mathbb{R}^3)} dz_2 \{: \frac{1}{i} \frac{\delta}{\delta f(z_1)} \frac{1}{i} \frac{\delta}{\delta f(z_2)} :, V(x_1 - x_2)\}^{(2)} \mathcal{L}(f),$$

where for any $n \in \mathbb{N}$

$$: \frac{1}{i} \frac{\delta}{\delta f(z_1)} ... \frac{1}{i} \frac{\delta}{\delta f(z_n)} := \prod_{j=1}^{n} \left[ \frac{1}{i} \frac{\delta}{\delta f(z_j)} - \sum_{k=1}^{j} \delta(z_j - z_k) \right] \qquad (292)$$

and, by definition, $\{\cdot, \cdot\}^{(j)}$ denotes the standard canonical Poisson bracket on the phase space $T^*(\mathbb{R}^3)^j$ for all $j \in \mathbb{Z}_+$.

Taking into account that for functional $\mathcal{L}(f)$, $f \in \mathcal{J}(T^*(\mathbb{R}^3); \mathbb{R})$, there exists the unlimited expansion (269):

$$\mathcal{L}(f) = \sum_{n \in \mathbb{Z}} \frac{1}{n!} \int_{T^*(\mathbb{R}^3)} dz_1 ... \int_{T^*(\mathbb{R}^3)} dz_n \prod_{i=1}^{n} \{\exp[if(z_j) - 1]\} f_n(z_1, ..., z_n), \qquad (293)$$

from (291), we obtain the kinetic equations for the hierarchy of the Bogolubov distribution functions [143]:

$$\frac{\partial}{\partial t} f_n(z_1, ..., z_n) = \{f_n(z_1, ..., z_n), H_n(z_1, ..., z_n)\}^{(n)} + \qquad (294)$$

$$+ \int_{T^*(\mathbb{R}^3)} dz_1 ... \int_{T^*(\mathbb{R}^3)} dz_{n+1} \{f_n(z_1, ..., z_{n+1}), H_n(z_1, ..., z_{n+1}), \sum_{j=1}^{n} V(x_j - x_{n+1})\}^{(n+1)},$$

where $z_j \in \mathbb{R}^3$, $j = 1, ..., n$, are the coefficients of the $n$-particle cluster in $\mathbb{R}^3$, $H_n(z_1, ..., z_n)$ denotes its corresponding energy:

$$H_n(z_1, ..., z_n) = \sum_{j=1}^{n} \frac{p_j^2}{2m} + \frac{1}{2} \sum_{j \neq k=1}^{n} V(x_j - x_k) \qquad (295)$$

Thus, the problem of the construction of the kinetic theory by Bogolubov is reduced to finding the special solutions of the unlimited hierarchy of the Equations (294), where the selection criterion is based on Bogolubov's fundamental weakening correlation principle:

$$\lim_{\| \langle n \rangle - \langle m \rangle \| \to \infty} |f_{n+m}(z_1, ..., z_{n+m}) - f_n(z_1, ..., z_n) f_m(z_{n+1}, ..., z_{n+m})| \to 0 \qquad (296)$$

where $\| \langle n \rangle - \langle m \rangle \| = dist(\{z_i \in T^*(\mathbb{R}^3) : i = 1, ..., n\}, \{z_{i+n} \in T^*(\mathbb{R}^3) : i = 1, ..., m\})$ is a distance between two clusters with $n \in \mathbb{Z}_+$ and $m \in \mathbb{Z}_+$ numbers of the particles. If a special solution of the hierarchy (289) exists in the functional form

$$f_n(z, ..., z_n; t) = f_n(z_1, ..., z_n; f_1(z; t)) \qquad (297)$$

for all $t \in \mathbb{R}_+$ and $n \in \mathbb{Z}_+$, then the corresponding equation for one-particle distribution function of the system in the external field $V_0 : \mathbb{R}^3 \to$ is the following

$$\frac{\partial}{\partial t} f_1(z; t) + \langle p/m | \nabla_x f_1(z; t) \rangle + \langle \nabla_x V_0(x) | \nabla_p f_1(z; t) \rangle = J(f_1(z; t)), \qquad (298)$$

where $J(f_1(z; t))$ is the so called "collision integral" [56,143,149,150,155], and is called the kinetic Boltzmann equation [56,149,156]. Below, we will focus on the such special solutions of the Bogolubov's hierarchy of the Equations (294), using the above developed algebraic method of Bogolubov's generating functional.

### 9.2. Generating Representation Functional and Its Solution Space Structure

Let us consider Bogolubov's functional Equation (291)

$$\frac{d}{dt}\mathcal{L}(f) = \int_{T^*(\mathbb{R}^3)} dz \left\{ \frac{1}{i}\frac{\delta\mathcal{L}(f)}{\delta f(z)}, T(p) \right\}^{(1)} + \tag{299}$$

$$+ \frac{1}{2}\int_{T^*(\mathbb{R}^3)} dz_1 \int_{T^*(\mathbb{R}^3)} dz_2 \left\{ : \frac{1}{i}\frac{\delta}{\delta f(z_1)}\frac{1}{i}\frac{\delta}{\delta f(z_2)} : \mathcal{L}(f), V(x_1 - x_2) \right\}^{(2)},$$

generated by the statistical operator evolution

$$P(t, t_0) = \exp\left[\frac{i}{\hbar}(t_0 - t)\mathbb{H}\right]\bar{P}\exp\left[\frac{i}{\hbar}(t - t_0)\mathbb{H}\right] \tag{300}$$

for $t, t_0 \in \mathbb{R}$, solving the Heisenberg evolution equation

$$\frac{dP}{dt} = \frac{i}{\hbar}[P, H], \quad P\Big|_{t=t_0} = \bar{P} \tag{301}$$

for the statistical Gibbs operator $P : \Phi_W \to \Phi_W$ with $\mathrm{Tr}\bar{P} = 1$.

When $\hbar \to 0$ in the Wigner representation, the expression (300), as an explicit solution of the (301), allows the following expansion

$$\mathcal{L}(f) = \mathrm{Tr}\left(\exp\left[\frac{i}{\hbar}(t_0 - t)\mathrm{H}\right]\bar{P}\exp\left[\frac{i}{\hbar}(t - t_0)\mathrm{H}\right]\exp[iw(f)]\right) = \tag{302}$$

$$= \mathrm{Tr}\left(\exp\left[\frac{i}{\hbar}(t_0 - t)(\mathrm{H}_0 + \mathrm{V})\right]\bar{P}\exp\left[\frac{i}{\hbar}(t - t_0)(\mathrm{H}_0 + \mathrm{V})\right]\exp[iw(f)]\right)\Big|_{\hbar\to 0} =$$

$$= \mathrm{Tr}\left(\exp\left[\frac{i}{\hbar}(t_0 - t)\mathrm{H}_0\right]\bar{P}\exp\left[\frac{i}{\hbar}(t - t_0)\mathrm{H}_0\right]\exp[\pi(t, t_0)]\exp[iw(f)]\right),$$

where we denoted $\mathrm{H} = \mathrm{H}_0 + \mathrm{V}$,

$$\mathrm{H}_0 = \int_{T^*(\mathbb{R}^3)} dz\frac{p^2}{2m}w(z), \mathrm{V} = \int_{T^*(\mathbb{R}^3)} dz_1 \int_{T^*(\mathbb{R}^3)} dz_2 V(x_1 - x_2) : w(z_1)w(z_2) :, \tag{303}$$

$$\exp[\pi(t, t_0)] = P_0(t_0, t)P(t, t_0), \quad P_0(t_0, t) = \exp\left[\frac{i}{\hbar}(t_0 - t)\mathrm{H}_0\right]\bar{P}\exp\left[\frac{i}{\hbar}(t - t_0)\mathrm{H}_0\right].$$

The operator $\pi(t, t_0)$, $t, t_0 \in \mathbb{R}$, in (303) is called a "cluster operator" and allows the next expansion into the unlimited series:

$$\pi(t_0, t) = \sum_{n\in\mathbb{Z}_+}\frac{1}{n!}\int_{T^*(\mathbb{R}^3)} dz_1...\int_{T^*(\mathbb{R}^3)} dz_n\, \pi_n(z_1, ..., z_n; t, t_0)\times \tag{304}$$

$$\times : w(z_1)...w(z_n) : \stackrel{def}{=} \pi(t, t_0; w),$$

where the functions $\pi_n(z_1, ..., z_n; t, t_0)$, $n \in \mathbb{N}$, can be defined uniquely form the representation (303) under the condition that the Gibbs operator $\bar{P} : \Phi_W \to \Phi_W$ is defined explicitly

in the Wigner representation. Thus, from (302)–(304) we obtain the following expressions for the Bogolubov's generating functional

$$\mathcal{L}(f) = \text{Tr}(P_0 \exp[\pi(t, t_0; w)] \exp(iw))\Big|_{\hbar \to 0} = \tag{305}$$

$$= \exp\left[\pi\left(t, t_0; \frac{1}{i}\frac{\delta}{\delta f}\right)\right] \text{Tr}(P_0 \exp[iw(f)]) = \exp\left[\pi\left(t, t_0; \frac{1}{i}\frac{\delta}{\delta f}\right)\right]\mathcal{L}_0(f),$$

where $\mathcal{L}_0(f)$, $f \in \mathcal{J}(T^*(\mathbb{R}^3); \mathbb{R})$, is a generating functional if the initial dynamical system of the particles under absent of interaction, that is

$$\mathcal{L}_0(f)(t, t_0) = \sum_{n \in \mathbb{Z}_+} \frac{1}{n!} \int_{T^*(\mathbb{R}^3)} dz_1 ... \int_{T^*(\mathbb{R}^3)} dz_n \times$$

$$\times f_n\left(x_1 + \frac{p_1}{m}(t_0 - t), p_1; ...; x_n + \frac{p_n}{m}(t_0 - t), p_1\right) \prod_{j=1}^{n} \{\exp[if(z_j)] - 1\}. \tag{306}$$

Applying to (306) when $t_0 \to -\infty$ Bogolubov's correlation weakening (296), we obtain that for all $t \in \mathbb{R}_+$

$$\mathcal{L}_0(f)(t) = \exp\left[\int_{T^*(\mathbb{R}^3)} dz f_1\left(x - \frac{p}{m}t; p\right)\{\exp[if(z)] - 1\}\right], \tag{307}$$

where $\mathcal{L}_0(f)(t) = \lim_{t_0 \to -\infty} \mathcal{L}_0(f)(t, t_0)$. Now, according to (305) and (307), we find that

$$\mathcal{L}(f)(t) = \exp\left[\pi\left(t, t_0; \frac{1}{i}\frac{\delta}{\delta f}\right)\mathcal{L}_0(f)(t)\right] \tag{308}$$

is a solution of Bogolubov's functional Equation (299), where

$$\pi\left(t; \frac{1}{i}\frac{\delta}{\delta f}\right) = \lim_{t_0 \to -\infty} \pi\left(t, t_0; \frac{1}{i}\frac{\delta}{\delta t}\right) \tag{309}$$

for all $t \in \mathbb{R}_+$. To specify the form of the operators (309), we note that operator $\xi(t, t_0; w) = \exp[\pi(t, t_0; w)]$ for all $t, t_0 \in \mathbb{R}_+$ satisfies under $\hbar \to 0$ the following differential evolution relationship:

$$\frac{d\xi}{dt} = \frac{1}{i\hbar}[\xi, \text{H}]_0 + \lim_{\hbar \to 0} \frac{1}{\hbar}\left(\text{V} - P_0^{-1}\text{V}P_0\right)\xi, \tag{310}$$

where all operators are assumed to be given in the Wigner representation. Expanding the operator $\xi(t, t_0, w)$ into the sum of $n$-particles components, $n \in \mathbb{N}$, we find

$$\xi(t, t_0; w) = \sum_{n \in \mathbb{Z}_+} \frac{1}{n!} \int_{T^*(\mathbb{R}^3)} dz_1 ... \int_{T^*(\mathbb{R}^3)} dz_n \, \xi_n(z_1, ..., z_n; t, t_0) : w(z_1)...w(z_n) :, \tag{311}$$

and there is mutually unambiguous correspondence [56,59] between coefficient functions $\xi_n(z_1, ..., z_n; t, t_0)$ in (311) and coefficient functions in the expansion (304)

$$\pi_n(z_1, ..., z_n) = \sum_{\sigma \in \Sigma_n} (-1)^{n+i}(\sigma - 1) \prod_{j=1}^{\infty} \xi_\sigma(z_{\langle k \rangle} \in \sigma_j), \tag{312}$$

$$\xi_n(z_1, ..., z_n) = \sum_{\sigma \in \Sigma_n} \prod_{j=1}^{\infty} \pi_{\sigma_j}(z_{\langle k \rangle} \in \sigma_j).$$

Here, $\sigma \in \Sigma_n$ is an arbitrary partition of the symmetry group $\Sigma_n$ of all permutations of the set of numbers $\{1, 2, ..., n\}$ on the subsets $\{\sigma_j : j = 1, ..., s\}$, which are not intersect,

that is $\bigcup\limits_{j=1}^{n} \sigma_j = \{1, ..., n\}$ and $\xi_{\sigma_j}$ and $\pi_{\sigma_j}$, $j = 1, ..., s$, are the corresponding to this partition coefficient functions. In particular,

$$\xi_1(z_1) = \pi_1(z_1), \quad \pi_2(z_1, z_2) = \xi_2(z_1, z_2) - \xi_1(z_1)\xi_1(z_2)$$

and so on. Thus, on the base of the defined operator series (304) or (311), the problem of the explicit calculations of the distribution functions become very simple. Below we will analyze these series by means of the language of Bogolubov's generating functional $\mathcal{L}(f)$, $f \in \mathcal{J}(T^*(\mathbb{R}^3); \mathbb{R})$, using Bogolubov's functional hypothesis [56,143,149,150,155].

*9.3. Bogolubov–Boltzmann Kinetic Equation in the Frame of Functional Hypothesis*

The generating functional, as it was stated above, is given by the expression

$$\mathcal{L}(f)(t) = \exp\left[\pi\left(t_0; \frac{1}{i}\frac{\delta}{\delta f}\right)\right]\mathcal{L}_0(f)(t). \tag{313}$$

Here, $\mathcal{L}_0(f)(t)$, $t \in \mathbb{R}_+$, is a generating functional of the system of non-interacting particles, which is equal to the expression (307) when $t_0 \to -\infty$. From (313), it follows that for all $t \in \mathbb{R}_+$ for the *n*-particle distribution function $f_n(z_1, ..., z_n; t)$ the general functional relationship holds

$$f_n(z_1, ..., z_n; t) := f_n(z_1, ..., z_n; f_1(z; t)). \tag{314}$$

Respectively, the generating functional (313) satisfies, according to (290) when $t = 0$, the following dynamic equation:

$$\frac{d}{dt}\mathcal{L}(f) = \int\limits_{T^*(\mathbb{R}^3)} dz \left\{\frac{1}{i}\frac{\delta\mathcal{L}(f)}{\delta f(z)}, T(p)\right\}^{(1)} + \tag{315}$$

$$+ \frac{1}{2} \int\limits_{T^*(\mathbb{R}^3)} dz_1 \int\limits_{T^*(\mathbb{R}^3)} dz_2 \left\{: \frac{1}{i}\frac{\delta}{\delta f(z_1)}\frac{1}{i}\frac{\delta}{\delta f(z_2)} : \mathcal{L}(f), V(x_1 - x_2)\right\}^{(2)},$$

Let us put $f_1(z) \to f_1(z; \tau)$, where $\tau \in \mathbb{R}_-$ and that

$$\frac{\partial f_1(z; \tau)}{\partial \tau} = \{f_1(z; \tau), T(p)\}^{(1)} + \int\limits_{T^*(\mathbb{R}^3)} dz_1\{f_1(z_1)f_1(z), V(x_1 - x)\}^{(2)}. \tag{316}$$

Then from (315), we also obtain that

$$\frac{d}{d\tau}\mathcal{L}(f) = \{\{\mathcal{L}(f), \mathcal{H}(\mathcal{F})\}\} = \int\limits_{T^*(\mathbb{R}^3)} dz \left\{\frac{1}{i}\frac{\delta\mathcal{L}(f)}{\delta f(z)}, T(p) + \int\limits_{T^*(\mathbb{R}^3)} dz_1 f_1(z_1), V(x_1 - x)\right\}^{(1)} \tag{317}$$

for all $\tau \in \mathbb{R}_-$. Then Equation (317) can be rewritten in the following way:

$$\frac{d}{d\tau}\mathcal{L}(f) = \{\{\mathcal{L}(f), \tilde{\mathcal{H}}(\mathcal{F})\}\} = \int\limits_{T^*(\mathbb{R}^3)} dz \left\{\frac{1}{i}\frac{\delta\mathcal{L}(f)}{\delta f(z)}, \tilde{H}(f_1)\right\}^{(1)}, \tag{318}$$

where, by definition, $\tilde{H}(f_1) := \frac{\delta}{\delta f_1}\tilde{\mathcal{H}}(\mathcal{F})$ and

$$\tilde{\mathcal{H}}(\mathcal{F}) = \int\limits_{T^*(\mathbb{R}^3)} dz \frac{p^2}{2m} f_1(z) + \frac{1}{2} \int\limits_{T^*(\mathbb{R}^3)} dz_1 \int\limits_{T^*(\mathbb{R}^3)} dz_2 f_1(z_1)f_1(z_2)V(x_1 - x_2), \tag{319}$$

is the Vlasov-type Hamiltonian of the self-consistent particles interaction. Let us define the following mapping on the phase space of $n \in \mathbb{Z}_+$ particles:

$$S_n(\tau)x_j = x_j(\tau), \quad S_n(\tau)p_j = p_j(\tau), \tag{320}$$

where for all $\tau \in \mathbb{R}_-, j = \overline{1, n}$,

$$\frac{dx(\tau)}{dt} = \{\tilde{H}, x(\tau)\}^{(1)}, \quad \frac{dp(\tau)}{dt} = \{\tilde{H}, p(\tau)\}^{(1)}, \tag{321}$$

$$\tilde{H} = \sum_{j=1}^{n} \frac{p_j^2}{2m} + \frac{1}{2} \sum_{j=k}^{n} V(x_j - x_k).$$

It easy to see that the system of Equations (321) gives the exact solution [157] for the dual Equation (316) in the form of the sum of $\delta$-functions of $n \in \mathbb{Z}_+$ particles:

$$f_1(z) = \sum_{j=1}^{n} \delta(z - z_j), \tag{322}$$

where $z_j \in \mathbb{R}^3, j = 1, ..., n$, are the coordinates of the cluster. Using (320) from (318) we obtain that for all $\tau \in \mathbb{R}$

$$\frac{d}{d\tau}\mathcal{L}(f)(\tau) = \{\{\mathcal{L}(f)(\tau), \tilde{\mathcal{H}}(\mathcal{F})\}\} + \frac{1}{2} \int\limits_{\mathbb{R}^3 \times \mathbb{R}^3} dz_1 \int\limits_{T^*(\mathbb{R}^3)} dz_2 \times \tag{323}$$

$$\times \left\{ : \frac{1}{i} \frac{\delta}{\delta f(z_1)} \frac{1}{i} \frac{\delta}{\delta f(z_2)} : \mathcal{L}(f), V(x_1 - x_2)(\tau) \right\}^{(2)},$$

where we denoted

$$\mathcal{L}(f)(\tau) = S(\tau)\mathcal{L}(f|S(-\tau)f_1), \tag{324}$$
$$V(x_1 - x_2)(\tau) = S(\tau)V(S(-\tau)(x_1 - x_2)),$$
$$f_2(z, z_1)(\tau) = S(\tau)f_2(z_1, z|S(-\tau)f_1)$$

Integrating the Equation (323) in limits $\tau \in (-\infty, 0)$, we obtain that

$$\mathcal{L}(f)|_{\tau=0} = \lim_{\tau \to -\infty} S(\tau)\mathcal{L}(f|S(-\tau)f_1) + \int\limits_{-\infty}^{0} d\tau \left\{ \frac{1}{i} \frac{\delta}{\delta f(z)} \mathcal{L}(f)(\tau), \tilde{\mathcal{H}}(\mathcal{F}(\tau)) \right\}^{(1)} +$$

$$+ \frac{1}{2} \int\limits_{-\infty}^{0} d\tau \int\limits_{\mathbb{R}^3 \times \mathbb{R}^3} dz_1 \int\limits_{T^*(\mathbb{R}^3)} dz_2 \left\{ : \frac{1}{i} \frac{\delta}{\delta f(z_1)} \frac{1}{i} \frac{\delta}{\delta f(z_2)} : \mathcal{L}(f), V(x_1 - x_2)(\tau) \right\}^{(2)} \right] \tag{325}$$

We should also note here, that due to the Bogolubov's principle of correlations weakening (296) and using (307) the first item in (325) can be represented in the form

$$\lim_{\tau \to -\infty} S(\tau)\mathcal{L}(f|S(-\tau)f_1) = \lim_{\tau \to \infty} \exp \left[ \int\limits_{T^*(\mathbb{R}^3)} dz S(\tau)f_1(z)(\tau)\{\exp[if(z)] - 1\} \right] =$$

$$= \exp \left[ \int\limits_{T^*(\mathbb{R}^3)} dz f_1(z)\{\exp[i\,f(z)] - 1\} \right]. \tag{326}$$

Applying to the expression (325) the different variants of the successive approximations method [56,143,149,150,155], we can get the generating functional $\mathcal{L}(f)$ in explicit form and then, using formula

$$f_n(z_1, ..., z_n) =: \frac{1}{i} \frac{\delta}{\delta f(z_1)} ... \frac{1}{i} \frac{\delta}{\delta f(z_n)} : \mathcal{L}(f) \Big|_{f=0} \tag{327}$$

for all $n \in \mathbb{Z}_+$ obtain distribution function for any order of perturbation theory. In particular, choosing expansion by the particle density in container $A \in \mathbb{R}^3$ as a small parameter, it is easy to get the modified kinetic Bogolubov–Boltzmann equation for one-particle distribution function $f_1(z;t)$, $z \in T^*(\mathbb{R}^3)$, $t \in \mathbb{R}_+$:

$$\frac{\partial f_1(z_1;t)}{\partial t} + \langle \frac{p}{m} | \nabla_x f_1(z_1;t) \rangle = \int\limits_{T^*(\mathbb{R}^3)} dz_2 \{ \tilde{f}_2(z_1, z_2; t), V(x_1 - x_2) \}^{(2)}, \tag{328}$$

where the function $\tilde{f}_2(z_2, z_1; t)$ is defined according to (325) and (326) by the following expression:

$$\tilde{f}_2(z_1, z_2; t) = f_1(\tilde{z}_1; t) f_1(\tilde{z}_2; t), \tag{329}$$

$$\tilde{z}_j = \lim_{\tau \to \infty} S_2(\tau) S_1(-\tau) z_j \Rightarrow \begin{cases} \tilde{x}_j = \lim_{\tau \to \infty} S_2(-\tau) x_j + \tau \frac{p_j}{m}, \\ \tilde{p}_j = \lim_{\tau \to \infty} S_2(-\tau) p_j, \end{cases}$$

for $j = \overline{1,2}$. Taking into account that the Poisson bracket $\{\cdot, \cdot\}^{(n)}$ is invariant with respect to the mappings $S_n(\tau)$, $n \in \mathbb{Z}_+$, from (329) it is easy to find that

$$\{ \tilde{f}_2(z_1, z_2; t), V(x_2 - x_1) \}^{(2)} = \frac{|p_2 - p_1|}{m} \frac{\partial}{\partial \xi} (f_1(\tilde{z}_1; t) f_1(\tilde{z}_2; t)) - \tag{330}$$

$$- \langle \frac{(\tilde{p}_2 - p_1)}{m} | \nabla_{x_1} f_1(\tilde{z}_1; t) \rangle f_1(\tilde{z}_2; t) + \langle \frac{(\tilde{p}_2 - p_1)}{m} | \nabla_{x_2} f_1(\tilde{z}_2; t) \rangle f_1(\tilde{z}_1; t),$$

where $\xi \in \mathbb{R}^1$ is a parameter of the axis in a cylindrical coordination system which is directed along the vector $(p_2 - p_1) \in \mathbb{E}^3$ and beginning at the point $x_1 \in \mathbb{R}^3$. After substituting (330) into (328), we can get the kinetic Bogolubov–Boltzmann equation [56,143,149,150,155] in the form of (298) with the explicitly defined collision integral $J(f_1)$, obtained from (330) via integration by $\xi \in \mathbb{R}$. Choosing in (326) other approximations of the generating functional $\mathcal{L}(f)$, $f \in \mathcal{J}(T^*(\mathbb{R}^3); \mathbb{R})$, one can find other forms of Bogolubov–Boltzmann kinetic Equations (298).

We can also make a remark concerning the nature of the operator-functional expression (309) or (304). Namely, it is easy to see that generating functional $\mathcal{L}(f)(t, t_0)$, $f \in \mathcal{J}(T^*(\mathbb{R}^3); \mathbb{R})$, allows the following operator-functional representation for all $t, t_0 \in \mathbb{R}$:

$$\mathcal{L}(f)(t, t_0) = \exp \left[ \frac{1}{2} \int\limits_{T^*(\mathbb{R}^3)} dz_1 \int\limits_{T^*(\mathbb{R}^3)} dz_2 \Big\{ : \frac{1}{i} \frac{\delta}{\delta f(z_1)} \times \right. \tag{331}$$

$$\left. \times \frac{1}{i} \frac{\delta}{\delta f(z_2)} :, V(x_1 - x_2) \Big\}^{(2)} (t - t_0) \right] \mathcal{L}_0(f)(t, t_0).$$

Comparing the expressions (331) and (305), we find that for arbitrary $t, t_0$

$$\pi \left( t, t_0; \frac{1}{i} \frac{\delta}{\delta f} \right) = \frac{1}{2}(t - t_0) \times \tag{332}$$

$$\times \int\limits_{T^*(\mathbb{R}^3)} dz_1 \int\limits_{T^*(\mathbb{R}^3)} dz_2 \Big\{ : \frac{1}{i} \frac{\delta}{\delta f(z_1)} \frac{1}{i} \frac{\delta}{\delta f(z_2)} :, V(x_1 - x_2) \Big\}^{(2)}$$

since the functional $\mathcal{L}_0(f)(t, t_0)$, $f \in \mathcal{J}(T^*(\mathbb{R}^3); \mathbb{R})$, is arbitrary. It is easy to see from (332), that operator $\pi\left(t, t_0; \frac{1}{i}\frac{\delta}{\delta f}\right)$ is not poly-local with respect to the functional derivatives $\frac{1}{i}\frac{\delta}{\delta f}$, which corresponds to the singularity in the operator expansion (304). Thus, using the expression (331), the arbitrariness of the initial state and the classical Bogolubov weakening correlation condition gives a possibility to find many types of the solutions via the method of successive approximations, which follows from (331) and the Bogolubov functional hypothesis subject to the generating representation functional of distribution functions.

Having analyzed the Bogolubov generating functional (331) within the quasi-classical Wigner density operator representation (287), one can obtain an exact functional-operator solution to the evolution Bogolubov functional Equation (323):

$$\mathcal{L}(f) = Z(f)/Z(0), \qquad Z(f) = \exp[\tilde{V}(\delta)]\mathcal{L}_0(f) \tag{333}$$

for $f \in \mathcal{J}(T^*(\mathbb{R}^3); \mathbb{R})$. Here we denoted

$$\tilde{V}(\delta) = \sum_{n \in \mathbb{Z}_+} \frac{1}{n!} \int_{T(\mathbb{R}^3)} dz_1 \int_{T(\mathbb{R}^3)} dz_2 ... \tag{334}$$

$$\times \int_{T(\mathbb{R}^3)} dz_n \Phi_n(z_1, z_2, ..., z_n|t) : \frac{1}{i}\frac{\delta}{\delta f(z_1)} \frac{1}{i}\frac{\delta}{\delta f(z_2)} ... \frac{1}{i}\frac{\delta}{\delta f(z_n)} : ,$$

$$\mathcal{L}_0(f) = \sum_{n \in \mathbb{Z}_+} \frac{1}{n!} \int_{T(\mathbb{R}^3)} z_1 \int_{T(\mathbb{R}^3)} dz_2 ... \int_{T(\mathbb{R}^3)} dz_n$$

$$\times \bar{f}_n(x_1 - \frac{p_1}{m}t, x_2 - \frac{p_2}{m}t, ..., x_n - \frac{p_n}{m}t; p_1, p_2, ..., p_n) \prod_{j=1}^{n} \{\exp[if(z_j)] - 1\},$$

where $\bar{f}_n(z_1, z_2, ..., z_n)$, $n \in \mathbb{N}$, are given $n-$particle distribution functions at $t = 0$, that is, owing to the definition (237),

$$\bar{f}_n(z_1, z_2, ..., z_n) := tr(\bar{P} : w(z_1)wz_2)...w(z_n) :)$$

$$=: \frac{1}{i}\frac{\delta}{\delta f(z_1)} \frac{1}{i}\frac{\delta}{\delta f(z_2)} ... \frac{1}{i}\frac{\delta}{\delta f(z_n)} : \mathcal{L}(f)|_{t=0, f=0}, \tag{335}$$

and $\Phi_n(x_1, x_2, ..., x_n; p_1, p_2, ..., p_n|t)$, $n \in \mathbb{Z}_+$, are so-called cluster potential functions, determined recursively by means of the following functional-operator relationships:

$$\log(P_0^{-1}P) := \sum_{n \in \mathbb{Z}_+} \frac{1}{n!} \int_{T(\mathbb{R}^3)} dz_1 \int_{T(\mathbb{R}^3)} dz_2 ... \int_{T(\mathbb{R}^3)} dz_n$$

$$\times \tilde{V}_n(z_1, z_2, ..., z_n|t) : w(z_1)w(z_2)...w(z_n) : \tag{336}$$

with

$$P_0 = \exp(-\frac{it}{\hbar}H_0)\bar{P}\exp(\frac{it}{\hbar}H_0) \tag{337}$$

being the statistical operator of the non-interacting particle system.

If the initial distribution at $t = 0$ is "chaotic", that is for all $n \in \mathbb{N}$, the following relationships

$$\bar{f}_n(z_1, z_2, ..., z_n) = \prod_{j=1}^{n} \bar{f}_1(z_j) \tag{338}$$

hold, one easily gets from (334) and (338) that

$$\mathcal{L}_0(f) = \exp\left(\int_{T(\mathbb{R}^3)} dz f_1(x - \frac{p}{m}t; p)\{\exp[if(z)] - 1\}\right). \tag{339}$$

If the "chaotic" condition is not fulfilled, we can proceed to the usual cluster Ursell–Mayer type representation [20,22,115,118] for the Bogolubov generating functional (333), where

$$\mathcal{L}_0(f) = \exp\left( \sum_{n \in \mathbb{Z}_+} \frac{1}{n!} \int_{T(\mathbb{R}^3)} dz_1 \int_{T(\mathbb{R}^3)} dz_2 ... \int_{T(\mathbb{R}^3)} dz_n \times \right.$$ (340)

$$\left. \times \bar{g}_n \left( x_1 - \frac{p_1}{m}t, x_2 - \frac{p_2}{m}t, ..., x_n - \frac{p_n}{m}t; p_1, p_2, ..., p_n \right) \prod_{j=1}^{n} \{ \exp[if(z_j)] - 1 \} \right),$$

where "*cluster*" distribution functions $\bar{g}_n(z_1, z_2, ..., z_n)$, $n \in \mathbb{N}$, have the form

$$\bar{g}_n(z_1, z_2, ..., z_n) := \sum_{\sigma[n]} (-1)^{m+1} (m-1)! \prod_{j=1}^{m} \bar{F}_{\sigma[j]}(z_k \in \sigma[j]),$$

$$\bar{f}_n(z_1, z_2, ..., z_n) := \sum_{\sigma[n]} \prod_{j=1}^{m} \bar{g}_{\sigma[j]}(z_k \in \sigma[j]),$$ (341)

and $\sigma[n]$ denotes a partition of the set $\{1, 2, ..., n\}$ into non-intersecting subsets $\{\sigma[j] : j = \overline{1, m}\}$, that is $\sigma[j] \cap \sigma[k] = \varnothing$ for $j \neq k = \overline{1, m}$, and $\sigma[n] = \cup_{j=1}^{m} \sigma[j]$. In particular,

$$\bar{g}_1(z_1) = \bar{f}_1(z_1),$$
$$\bar{g}_2(z_1, z_2) = \bar{f}_2(z_1, z_2) - \bar{f}_1(z_1)\bar{f}_1(z_2), ...,$$ (342)

and so on. The classical Bogolubov generating functional (333), owing to (334) and (340), allows a natural infinite series expansion, whose coefficients can be represented as above, by means of the usual Ursell–Mayer type diagram expressions, which can be effectively used for studying the kinetic properties of our many-particle statistical system.

*9.4. The Kinetic Equations for Many-Particle Distribution Functions, Their Lie-Algebraic Structure and Invariant Reductions*

It is well known that the classical Bogolubov–Boltzmann kinetic equations under the condition of many-particle correlations [56,86,142,149–151,155,157–161] at weak short range interaction potentials describe long waves in a dense gas medium. In general, based on the Liouville equations of a finite number of particles in a fixed volume, it is easy to get for these distribution functions a finite chain of the corresponding kinetic equations, within which one can formally proceed to the statistical mechanics limit and get a chain of equations for the limiting distribution functions. There will be strong difficulties here if we try to mathematically justify the correctness of this limiting transition in a chain of multi-particle kinetic equations. If we do not pay attention to this complex problem, and consider a fairly weak interaction between particles under appropriate initial conditions, one can obtain the related Boltzmann equation, characterizing the process of establishing statistical equilibrium. Many of the problems related to these limiting distribution functions can be omitted if the infinite particle statistical physics ensemble is worked with from the very beginning, making use of the secondly quantized representation [56,76,151,162] of the particle states in the corresponding Fock type space.

Relating to the Boltzmann kinetic equation, the same equation, called the Vlasov equation, as it was shown by N. Bogolubov [157], also describes exact microscopic solutions of the infinite Bogolubov chain [86] for the many-particle distribution functions, which was widely studied, making use of both classical approaches in [20,21,56,76,77,142,161–179], and making use of the generating Bogolubov functional method and the related quantum current algebra representations.

A.A. Vlasov proposed his kinetic equation [180] for electron-ion plasma, based on general physical reasonings that in contrast to the short range interaction forces between neutral gas atoms, interaction forces between charged particles slowly decrease with

distance, and therefore the motion of each such particle is determined not only by its pairwise interaction with either particle, but also by the interaction with the whole ensemble of charged particles. In this case, the Bogolubov equation for distribution functions in a domain $\Lambda \subset \mathbb{R}^3$

$$\frac{\partial f_1(z;t)}{\partial t} + \langle \frac{p}{m} | \nabla_x f_1(z;t) \rangle = \int_{T^*(\Lambda)} dz' \{ f_2(z,z';t), V(x-x') \}^{(2)}, \tag{343}$$

where $z := (x,p) \in T^*(\Lambda), t \in \mathbb{R}_+$ is the temporal evolution parameter, $\{\cdot, \cdot\}^{(m)}$ denotes the canonical Poisson bracket [56,122,181] on the product $T^*(\Lambda)^m, m \in \mathbb{N}$, and $V(x - x'), x, x' \in \Lambda$, is an interparticle interaction potential, - reduces to the Vlasov equation if to put in (343)

$$f_2(z,z';t) = f_1(z;t)f_1(z';t), \tag{344}$$

that is to assume that the two-particle correlation function [142,158,161,179] vanishes:

$$g_2(z,z';t) = f_2(z,z';t) - f_1(z;t)f_1(z';t) = 0 \tag{345}$$

for all $z, z' \in T^*(\Lambda)$ and $t \in \mathbb{R}_+$. Then one easily obtains from (343) that

$$\frac{\partial f_1(z;t)}{\partial t} + \langle \frac{p}{m} | \nabla_x f_1(z;t) \rangle = \langle \frac{\partial f_1(z;t)}{\partial p} | \nabla_x \int_{T^*(\Lambda)} dz' \, V(x-x') f_1(z';t) \rangle \tag{346}$$

for all $z \in T^*(\Lambda)$ and $t \in \mathbb{R}_+$. We remark here that the Equation (346) is reversible under the time reflection $\mathbb{R}_- \ni -t \rightleftarrows t \in \mathbb{R}_+$, thus it is obvious that it can not describe thermodynamically stable limiting states of the particle system in contrast to the classical Bogolubov–Boltzmann kinetic equations [20,56,86,142,149,151,161,166], being *a priori* time non-reversible owing to the choice of boundary conditions in the correlation weakening form. This means that in spite of the Hamiltonicity of the Bogolubov chain for the distribution functions, the Bogolubov–Boltzmann equation *a priori* is not reversible. It is also evident that the condition (345) does not break the Hamiltonicity—the Equation (346) is Hamiltonian with respect to the following Lie–Poisson–Vlasov bracket:

$$\{\{a(f), b(f)\}\} := \int_{T^*(\Lambda)} dz \, f(z) \{ \text{grad} \, a(f)(z), \text{grad} \, b(f)(z) \}^{(1)}, \tag{347}$$

where $\text{grad}(\cdot) := \delta(\cdot)/\delta f, f \in D(T^*(\Lambda)) := M_{f_1}$, respectively $a, b \in D(M_{f_1})$ are smooth functionals on the functional manifold $M_{f_1}$, consisting of functions fast decreasing at the boundary $\partial \Lambda$ of the domain $\Lambda \subset \mathbb{R}^3$. The statement above easily ensues from the following proposition.

**Proposition 11.** *Let $M_{\mathcal{F}}$ denote a set of many-particle distribution functions. Then the classical Bogolubov–Poisson bracket [20,21,86,172] on the functional space $D(M_{\mathcal{F}})$ reduces invariantly on the subspace $D(M_{f_1}) \subset D(M_{\mathcal{F}})$ to the Lie–Poisson–Vlasov bracket (347).*

Concerning the general case when we work with an infinite Bogolubov chain of kinetic equations on the many-particle distribution functions and are forced to break it at some place, numbered by some natural number $N \in \mathbb{N}$, the usual approaches always give rise to the resulting inconsistency [155,158] of the chain and, as a result, to the nonphysical solutions. The most successful approach to obtaining the Boltzmann kinetic equation for the one-particle distribution function was suggested many years ago by N. Bogolubov [86,149], based on the effective application of the so called weak correlation condition. So far, regretfully, this approach, being conjugated with the complex problem of solving functional equations, also gives rise to inconsistency of the higher order kinetic equations. Nonetheless, being inspired by former studies [20,76,162] of these problems, based on the geometrical

interpretation of the Bogolubov kinetic equations chain, we devised a new functional analytic approach [23] to constructing its compatible reduction a priori free of any non-physical consequences. We also succeeded in constructing a reduced set of kinetic equations, based on a suitably devised Dirac type invariant reduction scheme of the corresponding many-particle Lie–Poisson phase space. The approach to solving this problem and its different consequences will be analyzed in more detail in sections to follow below.

*9.5. The Classical Lie-Poisson-Vlasov Bracket and Kinetic Equation For The One-Particle Distribution Function*

The bracket expression (347) allows a slightly different Lie-algebraic interpretation, based on considering the functional space $D(M_{f_1})$ as a Poisson manifold, related with the canonical symplectic structure on the diffeomorphism group $\text{Diff}(\Lambda)$ of the domain $\Lambda \subset \mathbb{R}^3$, first described [31,32] in 1887 by Sophus Lie. Namely, the following classical theorem holds.

**Theorem 8.** *The Lie-Poisson bracket at point* $(\mu; \eta) \in T^*_\eta(\text{Diff}(\Lambda))$ *on the coadjoint space* $T^*_\eta(\text{Diff}(\Lambda)), \eta \in \text{Diff}(\Lambda)$, *is equal to the expression*

$$\{f, g\}(\mu) = (\mu|[\delta g(\mu)/\delta \mu, \delta f(\mu)/\delta \mu])_c \tag{348}$$

*for any smooth right-invariant functionals* $f, g \in C^\infty(T^*_\eta(\text{Diff}(\Lambda)); \mathbb{R})$.

**Proof.** By classical definition [31,32,122,131,181] of the Poisson bracket of smooth functions $(\mu|a)_c, (\mu|b)_c \in C^\infty(T^*_\eta(\text{Diff}(\Lambda)); \mathbb{R}), a, b \in \text{diff}(\Lambda) \simeq T_\eta(\text{Diff}(\Lambda))$ on the symplectic space $T^*_\eta(\text{Diff}(\Lambda))$, it is easy to calculate that

$$\{\mu(a), \mu(b)\} := \delta\alpha(X_a, X_b) = \\ = X_a(\alpha|X_b)_c - X_b(\alpha|X_a)_c - (\alpha|[X_a, X_b])_c, \tag{349}$$

where $X_a := \delta(\mu|a)_c/\delta\mu = a \in \text{diff}(\Lambda), X_b := \delta(\mu|b)_c/\delta\mu = b \in \text{diff}(\Lambda)$. Since the expressions $X_a(\alpha|X_b)_c = 0$ and $X_b(\alpha|X_a)_c = 0$ owing the right-invariance of the vector fields $X_a, X_b \in T_\eta(\text{Diff}(\Lambda))$, the Poisson bracket (349) transforms into

$$\{(\mu|a)_c, (\mu|b)_c\} = -(\alpha|[X_a, X_b])_c = \\ = (\mu|[b, a])_c = (\mu|[\delta(\mu|b)_c/\delta\mu, \delta(\mu|a)_c/\delta\mu])_c \tag{350}$$

for all $(\mu; \eta) \in T^*_\eta(\text{Diff}(\Lambda)) \simeq \text{diff}^*(\Lambda)$, and any $a, b \in \text{diff}(\Lambda)$. The Poisson bracket (350) is easily generalized to

$$\{f, g\}(\mu) = (\mu|[\delta g(\mu)/\delta\mu, \delta f(\mu)/\delta\mu])_c \tag{351}$$

for any smooth functionals $f, g \in C^\infty(\text{diff}^*(\Lambda); \mathbb{R})$, finishing the proof. $\square$

Concerning our special problem of describing evolution equations for one-particle distribution functions, we will consider the one particle cotangent space $T^*(\Lambda)$ over a domain $\Lambda \subset \mathbb{R}^3$ and the canonical Poisson bracket $\{\cdot, \cdot\} := \{\cdot, \cdot\}^{(1)}$ on $T^*(\Lambda)$, for which, by definition, for any $f, g \in M_{f_1}$

$$\{f, g\}(z) := \langle\frac{\partial f}{\partial p}|\frac{\partial g}{\partial x}\rangle - \langle\frac{\partial g}{\partial p}|\frac{\partial f}{\partial x}\rangle, \tag{352}$$

where $z = (x, p) \in T^*(\Lambda)$. We denote now by $\mathcal{G} := (M_{f_1}; \{\cdot, \cdot\})$ the related functional Lie algebra and $\mathcal{G}^*$ its adjoint space with respect to the standard bilinear symmetric form $(\cdot|\cdot): M_{f_1} \times M_{f_1} \to \mathbb{R}$ on the product $M_{f_1} \times M_{f_1}$, where

$$(f|g) := \int_{T^*(\Lambda)} f(z)g(z)dz. \tag{353}$$

The constructed Lie algebra $\mathcal{G}$ with respect to the bilinear symmetric form (353) proves to be metrized, that is $\mathcal{G} \simeq \mathcal{G}^*$ and

$$(\{f,g\}|h) = (f|\{g,h\}) \tag{354}$$

for any $f, g$ and $h \in \mathcal{G}$. If $\gamma \in D(\mathcal{G}^*)$ is a smooth functional on $\mathcal{G}^*$, its gradient $\operatorname{grad} \gamma(f) \in \mathcal{G}$ at point $f \in \mathcal{G}^*$ is naturally defined via the limiting expression

$$(g| \operatorname{grad} \gamma(f)) := \frac{d}{d\varepsilon} \gamma(f + \varepsilon g)\Big|_{\varepsilon=0} \tag{355}$$

for arbitrary element $g \in \mathcal{G}^*$. Now we define the Poisson structure $\{\{\cdot,\cdot\}\} : \mathcal{G}^* \times \mathcal{G}^* \to \mathbb{R}$ by means of the standard Lie–Poisson [31,32,59,122,159,181–184] expression:

$$\{\{\gamma,\mu\}\} := (f|\{\operatorname{grad} \gamma(f), \operatorname{grad} \gamma(f)\}) \tag{356}$$

for arbitrary functionals $\gamma, \mu \in D(\mathcal{G}^*)$. It is evident that the expression (356) identically coincides with the Poisson bracket (347).

Consider a functional $\gamma \in D(\mathcal{G}^*)$ and the related coadjoint action of the element $\operatorname{grad} \gamma(f) \in \mathcal{G}$ at a fixed element $f := f_1 \in \mathcal{G}^*$:

$$\partial f_1 / \partial t := ad^*_{\operatorname{grad} \gamma(f_1)} f_1, \tag{357}$$

where $t \in \mathbb{R}$ is the corresponding evolution parameter. It is easy observe that

$$\partial f_1 / \partial t = \{\{\gamma, f_1\}\} \tag{358}$$

is a Hamiltonian equation with the functional $\gamma \in D(\mathcal{G}^*)$ taken as its Hamiltonian, being simultaneously equivalent to the following canonical Hamiltonian flow:

$$\partial f_1 / \partial t = \{f_1, \operatorname{grad} \gamma(f_1)\}, \tag{359}$$

if to choose as a Hamiltonian the following functional

$$\gamma(f_1) := \int_{T^*(\Lambda)} dz_1 \frac{p_1^2}{2m} f_1(z_1) + \frac{1}{2} \int_{T^*(\Lambda)^2} dz_1 dz_2 V(x_1 - x_2) f_1(z_1) f_1(z_2), \tag{360}$$

where $V(x_1 - x_2)$ is a two-particle interaction potential, $x_1, x_2 \in \Lambda$. It is easy to observe here that the Hamiltonian (360) is obtained from the corresponding classical Hamiltonian expression

$$\mathcal{H}(\mathcal{F}) := \int_{T^*(\Lambda)} dz_1 \frac{p_1^2}{2m} f_1(z_1) + \frac{1}{2} \int_{T^*(\Lambda)^2} dz_1 dz_2 V(x_1 - x_2) f_2(z_1, z_2), \tag{361}$$

where $\mathcal{F} = (f_1, f_2, \dots) \in M_{\mathcal{F}}$ denotes an infinite vector from the space $M_{\mathcal{F}} := \prod_{j \in \mathbb{N}} M_{f_j}$ of multiparticle distribution functions, and if to impose on it the constraint (344). Thus, we have stated the following proposition.

**Proposition 12.** *The Boltzmann–Vlasov kinetic Equation (346) is a Hamiltonian system on the functional manifold $\mathcal{G}^* \simeq \mathcal{G} = (M_f; \{\cdot, \cdot\})$ with respect to the canonical Lie–Poisson structure (356) with Hamiltonian (360). As a consequence, the flow (346) is time reversible.*

*9.6. Boltzmann–Vlasov Kinetic Equations and Microscopic Exact Solutions*

Proposition 11, stated above, claims that the Boltzmann–Vlasov Equation (346) is a suitable reduction of the whole Bogolubov chain upon the invariant functional subspace $M_{f_1} \subset M_{\mathcal{F}}$. Moreover, this invariance in no way should be compatible

*a priori* [20,21,24,157,166,171,173,174] with the other kinetic equations from the Bogolubov chain, and can even be contradictory. Nonetheless, as it was stated [157] by N. Bogolubov, namely owing to this invariance of the subspace $M_{f_1} \subset M_{\mathcal{F}}$ the Boltzmann–Vlasov Equation (346) in the case of the Boltzmann–Enskog hard sphere approximation of the inter-particle potential possesses exact microscopical solutions which are compatible with the whole hierarchy of the Bogolubov kinetic equations. The latter is, obviously, equivalent to its Hamiltonicity on the manifold $M_{f_1}$ with respect to the Lie–Poisson bracket (356). The Boltzmann–Enskog kinetic equation [151,157,158,161,179] equals

$$\frac{\partial f_1(z;t)}{\partial t} + \langle \tfrac{p}{m} | \nabla_x f_1(z;t) \rangle =$$

$$= a^2 \int_{\mathbb{S}^2} dn \int_{\mathbb{E}^3} dp' \; \langle p'|n \rangle \langle \tfrac{\tilde{p}'}{m} | n \rangle [f_2(x, \tilde{p}; x + an, \tilde{p}'; t) - f_2(x, p; x - an, p'; t)]$$

(362)

where $\tilde{p} := p + n\langle p' - p|n \rangle$, $\tilde{p}' := p - n\langle p' - p|n \rangle$, $a > 0-$ a particle diameter, $n \in \mathbb{S}^2 -$ a unit vector, $\langle n|n \rangle = 1$, and, by definition, $f_2(z, z'; t) = 0$ for all $z, z' \in T^*(\Lambda)$, $t \in \mathbb{R}$, satisfying the condition $||z - z'|| < a$. The Equation (362) easily reduces to the Vlasov–Enskog equation

$$\frac{\partial f_1(z;t)}{\partial t} + \langle \tfrac{p}{m} | \nabla_x f_1(z;t) \rangle = J_{V-E}(f),$$

$$J_{V-E}(f) = a^2 \int_{\mathbb{S}^2} dn \int_{\mathbb{E}^3} dp' \; \langle p'|n \rangle \langle \tfrac{\tilde{p}'}{m} | n \rangle \times$$

(363)

$$\times [f_1(x, \tilde{p}; t) f_1(x + an, \tilde{p}'; t) - f_1(x, p; t) f_1(x - an, p'; t)]$$

for all $(z; t) \in T^*(\Lambda) \times \mathbb{R}$ owing to its Hamiltonicity on the space $M_{f_1} \subset M_{\mathcal{F}}$. If, in addition, there exists a nontrivial interparticle potential, the equation above is naturally generalized to the kinetic equation

$$\frac{\partial f_1(z;t)}{\partial t} + \langle \tfrac{p}{m} | \nabla_x f_1(z;t) \rangle = J_{V-E}(f) +$$

$$+ \int_{T^*(\Lambda)} dz' \{f_1(z;t) f_1(z';t), V(x - x')\}^{(2)},$$

(364)

which remains to be Hamiltonian on $M_{f_1}$ and possesses, in particular, the following exact singular solution:

$$f_1(z;t) = \sum_{j=\overline{1,N}} \delta(z - z_j(t)),$$

(365)

where $z_j(t) \in T^*(\Lambda), j = \overline{1, N}$—phase space coordinates in $T^*(\Lambda)^N$ of $N \in \mathbb{N}$ interacting particles in the domain $\Lambda \subset \mathbb{R}^3$. Specified above the Hamiltonicity problem and the existence of exact solutions to the Boltzmann–Vlasov kinetic Equation (364) is deeply related to that of describing correlation functions [142,161,179], suitably breaking the infinite Bogolubov chain [20,76,77,86,142,161] of many-particle distribution functions. Namely, if to introduce many-particle correlation functions [142,161,179] for related Bogolubov distribution functions as

$$g_1(z_1) = 0, g_2(z_1, z_2) = f_2(z_1, z_2) - f_1(z_1) f_1(z_2),$$

$$g_3(z_1, z_2, z_3) = f_3(z_1, z_2, z_3) - f_1(z_1) f_1(z_2) f_1(z_3) - f_1(z_1) g_2(z_2, z_3) -$$

$$- f_1(z_2) g_2(z_3, z_1) - f_1(z_3) g_2(z_1, z_2), ...,$$

(366)

where $z_j \in T^*(\Lambda), j \in N$, then the Vlasov Equation (364) is obtained from the Bogolubov hierarchy at $n = 1$ and $g_2(z_1, z_2) = 0$ for all $z_1, z_2 \in T^*(\Lambda)$.

As it was mentioned above, the constraint imposed on the infinite Bogolubov hierarchy is compatible with its Hamiltonicity. Yet in many practical cases, this closedness procedure by means of imposing the conditions like

$$g_{m+1}(z_1, z_2, \ldots, z_{m+1}) = 0 \tag{367}$$

for all $z_s \in T^*(\Lambda), s = \overline{1, m+1}$ at some fixed $m \geq 2$ gives rise to some serious dynamical problems related to its mathematical correctness. Namely, if to close the infinite Bogolubov chain of kinetic equations on many-particle distribution functions in this way, one easily checks that the imposed constraint (367) does not persist in time subject to the evolution of the distribution functions $f_j(z_1, z_2, \ldots, z_j), z_j \in T^*(\Lambda), j = \overline{1, m}$. This means that these naively reduced kinetic equations are written down somehow incorrectly, as the reduced functional submanifold $M_{\mathcal{F}}^{(m)} := \{\mathcal{F} \in M_{\mathcal{F}} : g_{m+1} = 0\}$ should remain invariant in time. To dissolve this problem, we are forced to consider the whole Bogolubov hierarchy of kinetic equations on multiparticle distribution functions as a Hamiltonian system on the functional manifold $M_{\mathcal{F}}$ and correctly reduce it on the constructed above functional submanifold $M_{\mathcal{F}}^{(m)} \subset M_{\mathcal{F}}$ via the classical Dirac type [11,59,63,122,181] procedure. The kinetic equations obtained this way by means of the reduced Lie–Poisson–Bogolubov structure will evidently differ from those naively obtained by means of the direct substitution of the imposed constraint (367) into the Bogolubov chain of kinetic equations, and in due course will conserve the functional submanifold $M_{\mathcal{F}}^{(m)} \subset M_{\mathcal{F}}$ invariant.

*9.7. The Invariant Reduction of the Bogolubov Distribution Functions Chain*

Consider the constructed before Hamiltonian functional $\mathcal{H}(\mathcal{F}) \in D(M_{\mathcal{F}})$ (361)

$$\mathcal{H}(\mathcal{F}) = \int_{T^*(\Lambda)} dz_1 \frac{p_1^2}{2m} f_1(z_1) + \frac{1}{2} \int_{T^*(\Lambda)^2} dz_1 dz_2 V(x_1 - x_2) f_2(z_1, z_2) \tag{368}$$

and calculate the evolution of the distribution functions vector $\mathcal{F} \in M_{\mathcal{F}}$ under the simplest constraint (367) at $m = 1$, that is

$$g_2(z_1, z_2) = f_2(z_1, z_2) - f_1(z_1)f_1(z_2) = 0 \tag{369}$$

for all $z_1, z_2 \in T^*(\Lambda)$. To perform this reduction on $M_{\mathcal{F}}^{(1)} \subset M_{\mathcal{F}}$, we need [11,59,63] to constraint the $\lambda$-extended Hamiltonian expression

$$\mathcal{H}_\lambda(\mathcal{F}) := \mathcal{H}(\mathcal{F}) + \frac{1}{2} \int_{T^*(\Lambda)^2} dz_1 dz_2 \lambda(z_1, z_2)[f_2(z_1, z_2) - f_1(z_1)f_1(z_2)] \tag{370}$$

for some smooth function $\lambda \in D(T^*(\Lambda)^2)$ and next to determine it from the submanifold $M_{\mathcal{F}}^{(1)}$ invariance condition

$$\frac{\partial g_2(z_1, z_2)}{\partial t} = \{\{\mathcal{H}_\lambda(\mathcal{F}), g_2(z_1, z_2)\}\} = $$
$$= \frac{\partial f_2(z_1, z_2)}{\partial t} - \frac{\partial f_1(z_1)}{\partial t} f_1(z_2) - f_1(z_1) \frac{\partial f_1(z_2)}{\partial t} = 0 \tag{371}$$

for all $z_1, z_2 \in T^*(\Lambda)$ and $t \in \mathbb{R}$. To effectively calculate the condition (371), let us first calculate the evolutions for distribution functions $f_1$ and $f_2 \in M_{\mathcal{F}}$ :

$$\frac{\partial f_1(z_1)}{\partial t} = \{\{\mathcal{H}_\lambda(\mathcal{F}), f_1(z_1)\}\} = \left\{ f_1(z_1), \frac{\delta \mathcal{H}_\lambda(\mathcal{F})}{\delta f_1(z_1)} \right\}^{(1)} + \tag{372}$$
$$+ \int_{T^*(\Lambda)} dz_2 \left\{ f_2(z_1, z_2), \frac{\delta \mathcal{H}_\lambda(\mathcal{F})}{\delta f_2(z_1, z_2)} \right\}^{(1)},$$

and

$$\frac{\partial f_2(z_1,z_2)}{\partial t} = \{\{\mathcal{H}_\lambda(\mathcal{F}), f_2(z_1,z_2)\}\} = \left\{ f_2(z_1,z_2), \frac{\delta \mathcal{H}_\lambda(\mathcal{F})}{\delta f_1(z_1)} + \frac{\delta \mathcal{H}_\lambda(\mathcal{F})}{\delta f_1(z_2)} \right\}^{(2)} +$$

$$+ \left\{ f_2(z_1,z_2), \frac{\delta \mathcal{H}_\lambda(\mathcal{F})}{\delta f_2(z_1,z_2)} \right\}^{(2)} + \int_{T^*(\Lambda)} dz_3 \left\{ f_3(z_1,z_2,z_3), \frac{\delta \mathcal{H}_\lambda(\mathcal{F})}{\delta f_2(z_1,z_3)} + \frac{\delta \mathcal{H}_\lambda(\mathcal{F})}{\delta f_2(z_2,z_3)} \right\}^{(2)}, \tag{373}$$

which can be rewritten equivalently as follows:

$$\frac{\partial f_1(z_1)}{\partial t} = -\langle \frac{\partial f_1(z_1)}{\partial p_1} | \int_{T^*(\Lambda)} dz_2 \frac{\partial \lambda(z_1,z_2)}{\partial x_1} f_1(z_2) - \tag{374}$$

$$- \langle \frac{p_1}{m} - \int_{T^*(\Lambda)} dz_2 \frac{\partial \lambda(z_1,z_2)}{\partial p_1} f_1(z_2) | \frac{\partial f_1(z_1)}{\partial x_1} \rangle +$$

$$+ \frac{1}{2} \int_{T^*(\Lambda)} dz_2 \langle \frac{\partial}{\partial x_1}[V(x_1-x_2) + \lambda(z_1,z_2)] | \frac{\partial f_2(z_1,z_2)}{\partial p_1} \rangle -$$

$$- \frac{1}{2} \int_{T^*(\Lambda)} dz_2 \langle \frac{\partial \lambda(z_1,z_2)}{\partial p_1} | \frac{\partial f_2(z_1,z_2)}{\partial x_1} \rangle$$

and

$$\frac{\partial f_2(z_1,z_2)}{\partial t} = -\langle \frac{\partial f_2(z_1,z_2)}{\partial p_1} | \int_{T^*(\Lambda)} dz_2 \frac{\partial \lambda(z_1,z_2)}{\partial x_1} f_1(z_2) \rangle - \tag{375}$$

$$- \langle \frac{\partial f_2(z_1,z_2)}{\partial p_2} | \int_{T^*(\Lambda)} dz_1 \frac{\partial \lambda(z_1,z_2)}{\partial x_2} f_1(z_1) \rangle -$$

$$- \langle \frac{\partial f_2(z_1,z_2)}{\partial x_1} | \frac{p_1}{m} - \int_{T^*(\Lambda)} dz_2 \frac{\partial \lambda(z_1,z_2)}{\partial p_1} f_1(z_2) \rangle -$$

$$- \langle \frac{\partial f_2(z_1,z_2)}{\partial x_2} | \frac{p_2}{m} - \int_{T^*(\Lambda)} dz_1 \frac{\partial \lambda(z_1,z_2)}{\partial p_2} f_1(z_1) \rangle +$$

$$+ \frac{1}{2} \langle \frac{\partial f_2(z_1,z_2)}{\partial p_1} | \frac{\partial}{\partial x_1}[V(x_1-x_2) + \lambda(z_1,z_2)] \rangle +$$

$$+ \frac{1}{2} \langle \frac{\partial f_2(z_1,z_2)}{\partial p_2} | \frac{\partial}{\partial x_2}[V(x_1-x_2) + \lambda(z_1,z_2)] \rangle -$$

$$- \frac{1}{2} \langle \frac{\partial f_2(z_1,z_2)}{\partial x_1} | \frac{\partial \lambda(z_1,z_2)}{\partial p_1} \rangle - \frac{1}{2} \langle \frac{\partial f_2(z_1,z_2)}{\partial x_2} | \frac{\partial \lambda(z_1,z_2)}{\partial p_2} \rangle +$$

$$+ \frac{1}{2} \langle \int_{T^*(\Lambda)} dz_3 \frac{\partial f_3(z_1,z_2,z_3)}{\partial p_1} | \frac{\partial}{\partial x_1}[V(x_1-x_3) + \lambda(z_1,z_3)] \rangle +$$

$$+ \frac{1}{2} \langle \int_{T^*(\Lambda)} dz_3 \frac{\partial f_3(z_1,z_2,z_3)}{\partial p_2} | \frac{\partial}{\partial x_2}[V(x_2-x_3) + \lambda(z_2,z_3)] \rangle -$$

$$- \frac{1}{2} \langle \int_{T^*(\Lambda)} dz_3 \frac{\partial f_3(z_1,z_2,z_3)}{\partial x_1} | \frac{\partial \lambda(z_1,z_2)}{\partial p_1} \rangle - \frac{1}{2} \langle \int_{T^*(\Lambda)} dz_3 \frac{\partial f_3(z_1,z_2,z_3)}{\partial x_2} | \frac{\partial \lambda(z_1,z_2)}{\partial p_2} \rangle$$

Having now substituted temporal derivatives (374) and (375) into the equality (371) in their explicit form, one obtains the following functional relationship:

$$\frac{1}{2} \langle f_1(z_2) \frac{\partial f_1(z_1)}{\partial p_1} | \frac{\partial}{\partial x_1}(V(x_1-x_2) + \lambda(z_1,z_2) -$$
$$- \int_{T^*(\Lambda)} dz_3 f_1(z_3) [V(x_1-x_3) + \lambda(z_1,z_3)]) \rangle +$$
$$+ \frac{1}{2} \langle f_1(z_1) \frac{\partial f_1(z_2)}{\partial p_2} | \frac{\partial}{\partial x_2}(V(x_2-x_1) + \lambda(z_2,z_1) -$$
$$- \int_{T^*(\Lambda)} dz_3 f_1(z_3) [V(x_2-x_3) + \lambda(z_2,z_3)]) \rangle = 0, \tag{376}$$

which is satisfied if

$$\lambda(z_1,z_2) = -V(x_1-x_2) \tag{377}$$

for all $z_1, z_2 \in T^*(\Lambda)$. Taking into account the result (377), one easily obtains from the Equation (374) the invariantly reduced on the submanifold $M_{\mathcal{F}}^{(1)} \subset M_{\mathcal{F}}$ kinetic equation on the one-particle distribution function:

$$\frac{\partial f_1(z_1)}{\partial t} + \langle p_1/m | \frac{\partial f_1(z_1)}{\partial x_1} \rangle = \langle \frac{\partial f_1(z_1)}{\partial p_1} | \frac{\partial}{\partial x_1} \int_{T^*(\Lambda)} dz_2 f_1(z_2) V(x_1 - x_2) \rangle, \tag{378}$$

which can be rewritten in the following compact form:

$$\frac{\partial f_1(z_1)}{\partial t} = \left\{ f_1(z_1), \frac{\delta \tilde{\mathcal{H}}(\mathcal{F})}{\delta f_1(z_1)} \right\}^{(1)}, \tag{379}$$

where we put, by definition,

$$\tilde{\mathcal{H}}(\mathcal{F}) := \int_{T^*(\Lambda)} dz_1 \frac{p_1^2}{2m} f_1(z_1) + \frac{1}{2} \int_{T^*(\Lambda)^2} dz_1 dz_2 V(x_1 - x_2) f_1(z_1) f_1(z_2). \tag{380}$$

The kinetic Equation (378) naturally coincides exactly with that obtained previously from the naively reduced evolution equation

$$\frac{\partial f_1(z_1)}{\partial t} = \{\{\mathcal{H}(\mathcal{F}), f_1(z_1)\}\}|_{M_{\mathcal{F}}^{(1)}} \tag{381}$$

on the submanifold $M_{\mathcal{F}}^{(1)} \subset M_{\mathcal{F}}$, as it is globally invariant [20,172] with respect to the classical Lie–Poisson–Bogolubov structure on $M_{\mathcal{F}}$.

The obtained result can be formulated as the following proposition.

**Proposition 13.** *The first correlation function Dirac type reduction on the functional submanifold $M_{\mathcal{F}}^{(1)} \subset M_{\mathcal{F}}$, formed by relationships (369), reduces the corresponding Bogolubov chain of many-particle kinetic equations to the well known classical Vlasov kinetic equation.*

**Remark 9.** *It is worth mentioning here that the well known classical Bogolubov approximation of the many-particle distribution functions as $f_n(z_1, z_2, \ldots, z_n) := \varphi_n(z_1, z_2, \ldots, z; f_1), z_j \in T(\Lambda), j = \overline{2, n}$, with mapping $\varphi_n : (\ldots) \times M_{f_1} \to \mathbb{R}$, $n \in \mathbb{N} \backslash \{1\}$, presenting smooth nonlinear functionals, independent of the temporal parameter $t \in \mathbb{R}_+$, define a suitably different functional submanifold $\tilde{M}_{\mathcal{F}}^{(1)} \subset M_{\mathcal{F}}$, upon which the reduced evolution flow*

$$\frac{\partial f_1(z_1)}{\partial t} = \{\{\mathcal{H}(\mathcal{F}), f_1(z_1)\}\}|_{M_{\mathcal{F}}^{(1)}} \tag{382}$$

*gives rise to a new Boltzmann type kinetic equation, being compatible with evolution equations for higher distribution functions, free of evolution inconsistencies and completely different from that derived previously by Bogolubov [86].*

In the same way as above, one can explicitly construct the system of invariantly reduced kinetic equations

$$\frac{\partial f_1(z_1)}{\partial t} = \{\{\mathcal{H}(\mathcal{F}), f_1(z_1)\}\}|_{M_{\mathcal{F}}^{(2)}}, \frac{\partial f_2(z_1, z_2)}{\partial t} = \{\{\mathcal{H}(\mathcal{F}), f_2(z_1, z_2)\}\}|_{M_{\mathcal{F}}^{(2)}} \tag{383}$$

on the submanifold $M_{\mathcal{F}}^{(2)} \subset M_{\mathcal{F}}$, which already is not *a priori* globally invariant with respect to the Hamiltonian evolution flows on $M_{\mathcal{F}}$ and whose detail structure and analysis are postponed to another place.

*9.8. Conclusions and Perspectives*

We presented a review of the Boltzmann type kinetic equations in statistical physics as analytical objects based on the non-relativistic current algebra symmetry approach to constructing the Bogolubov generating functional of many-particle distribution functions. We then applied it to an important classical problem of describing Boltzmann–Bogolubov and Boltzmann–Vlasov type kinetic equations, naturally related with an invariantly reduced canonical Hamiltonian system on the infinite-dimensional space of distribution functions subject to the constraints imposed on suitably chosen many-particle correlation functions. As an interesting introductory example of deriving Boltzman–Vlasov type kinetic equations, we considered a quantum-mechanical model of spinless particles with delta-type interaction, having applications [159,185,186] for describing so called Benney type hydrodynamic particle flows. We also reviewed new results on a special class of dynamical systems of Boltzmann–Bogolubov and Boltzmann–Vlasov type on infinite dimensional functional manifolds modeling kinetic processes in many-particle media. There was demonstrated construction of the classical Bogolubov generating functional method in non-equilibrium statistical mechanics within the classical Wigner quasi-classical representation. We also analyzed and presented the kinetic Boltzmann type equation in non-equilibrium statistical mechanics in the frame of the Bogolubov functional hypothesis. Moreover, the Hamiltonian analysis of the infinite hierarchy of many-particle distribution functions was reviewed, and the algebraic structure of the Boltzmann–Bogolubov kinetic equations and their invariant Poissonian reductions were analyzed in detail, including the derivation of the related Boltzmann–Vlasov kinetic equations. Based on the methods and devised techniques, an approach was proposed to invariant reduction of the chain of Bogolubov distribution functions on suitably chosen correlation function constraints, which allowed the derivation of the related modified Boltzmann–Bogolubov kinetic equations for a finite set of multi-particle distribution functions.

We also elaborated in detail effective enough invariant analytical tools reducing the infinite Boltzmann–Bogolubov hierarchy of kinetic equations upon the two-particle correlation function constraint. Within this aspect of invariant reduction of the infinite Boltzmann–Bogolubov hierarchy of kinetic equations that has very important applications, there stays an interesting problem of analytically presenting this reduction upon the three-particle correlation function constraint and deriving a closed system of the Boltzmann type kinetic equations on the corresponding one- and two-particle distribution functions. The similar, yet much more complicated, analytical problem for the future analysis consists of deriving invariantly reduced kinetic equations under the Bogolubov functional hypothesis and its modified versions.

## 10. The Current Algebra Functional Representations and Geometric Structure of Quasi-Stationary Hydrodynamic Flows

*10.1. Introductory Notes*

This section is devoted to compressible liquid or gas motions in which entropy remains locally constant throughout the flow field, i.e., the flow for which the entropy of a moving element along a streamline remains constant, is called isentropic. This means that along different streamlines, the entropy changes normal to the streamlines. As a typical example, one can mention the flow field behind a curved shock wave, where streamlines, passing through different locations along the curved shock wave, experience different increases in entropy. Hence, downstream from this shock, the entropy can be constant along a given streamline but differs from one streamline to another. This type of flow, with entropy constant along streamlines, is defined as isentropic. Flow with entropy constant everywhere is then called homentropic. Here we need to remark that owing to the second law of thermodynamics, an isentropic flow does not strictly exist. We know from thermodynamics that an isentropic flow is defined to be along streamlines both adiabatic and reversible. Yet, all real flows always experience to some extent the irreversible phenomena of friction, thermal conduction, and diffusion. For instance, any non-equilibrium, chemically reacting

flow is always irreversible, when considered to be a closed system. Nonetheless, there are a large number of liquid and gas dynamic problems with entropy increase negligibly slight, which for the purpose of analysis are assumed to be isentropic. Examples are flows through subsonic and supersonic nozzles, as in wind tunnels and rocket engines, or shock-free flows over a wing, fuselage, or other aerodynamic shapes. For all of them, except for a flow near the thin boundary-layer region, adjacent to the surface where friction and thermal conduction effects can be strong, the outer inviscid flow can be considered isentropic. In contrast, if shock waves exist in the flow, the entropy increase across these shocks destroys the assumption of isentropic flow, although the flow along streamlines between shocks may persist to be isentropic.

As an isentropic flow is governed by thermodynamically reversible processes, being adiabatic along a streamline, it needs to be specified with locally defined [187] thermodynamical parameters, such as the medium density $\rho$, the specific entropy $\sigma$, the local medium absolute temperature $T$, the pressure $p$ and the specific energy $e$. All these quantities are related to each other in some way, which can be retrieved following the classical Gibbs reasonings. We assume from the very beginning that the reversible thermodynamical state of the medium under regard is completely locally described by means of the following first pair: ($p$-local *pressure*, $\rho$-*specific density*) of thermodynamical parameters. Assume now that the same thermodynamical state of this medium can also be simultaneously described by means of the following second pair: ($T$-local absolute temperature, $\sigma$-specific entropy). The latter, in particular, means that a suitable functional transformation from one parameter to another, if smooth, is diffeomorphic, which is the Jacobian $J_{(\sigma,T)}(p,\rho)$ of this transformation $\mathbb{R}^2_+ \ni (\sigma,T) \to (p,\rho) \in \mathbb{R}^2_+$ is not degenerate everywhere, i.e.,

$$J_{(\sigma,T)}(p,\rho) = \frac{\partial(p,\rho)}{\partial(\sigma,T)} := \det \begin{pmatrix} \frac{\partial p}{\partial \sigma} & \frac{\partial p}{\partial T} \\ \frac{\partial \rho}{\partial \sigma} & \frac{\partial \rho}{\partial \sigma} \end{pmatrix} \neq 0 \tag{384}$$

at all points $(\sigma,T) \in \mathbb{R}^2_+$. Taking into account that the local absolute temperature $T$ and the adiabatic $\sigma$ parameters are, in general, defined with some scaling ambiguity, we can always put, by definition, that $J_{(\sigma,T)}(p,\rho) = \rho^2 \neq 0$ everywhere. As a simple consequence of multiplying this expression by the unity Jacobian $J_{(\sigma,\rho)}(\sigma,\rho) = 1$ one easily derives that

$$\begin{aligned} J_{(\sigma,T)}(p,\rho) \times J_{(\sigma,\rho)}(\sigma,\rho) = \\ = \frac{\partial(p,\rho)}{\partial(\sigma,T)}\frac{\partial(\sigma,\rho)}{\partial(\sigma,\rho)} = \frac{\partial(\rho,p)}{\partial(\sigma,\rho)}\frac{\partial(\sigma,\rho)}{\partial(\sigma,T)} = \rho^2, \end{aligned} \tag{385}$$

or, equivalently,

$$\frac{\partial(p,\rho)}{\partial(\sigma,\rho)} = \rho^2 \frac{\partial(\sigma,T)}{\partial(\sigma,\rho)} \Longleftrightarrow \frac{\partial(p/\rho^2)}{\partial \sigma}\bigg|_\rho = \frac{\partial T}{\partial \rho}\bigg|_\sigma \tag{386}$$

at all points $(\sigma,\rho) \in \mathbb{R}^2_+$. The equality of partial derivatives above simply means, owing to the well known Montel–Menchoff–Young theorem [188–190], the existence of such a differentiable thermodynamic state function $\mathbb{R}^2_+ \ni (\rho,\sigma) \to e \in \mathbb{R}$, that its differential satisfies the following equality:

$$\delta e(\rho,\sigma) = T\delta\sigma + p\delta\rho/\rho^2. \tag{387}$$

The latter expression presents exactly the written down second thermodynamic law with respect to the locally defined variables, if the smooth function $\mathbb{R}^2_+ \ni (\rho,\sigma) \to e \in \mathbb{R}$ is interpreted as the specific medium energy of the system at the internal absolute temperature $T = T(\rho,\sigma)$ and pressure $p(\rho,\sigma)$ at suitably fixed state parameters $(\rho,\sigma) \in \mathbb{R}^2_+$. Taking into account that our medium is embedded into some domain $M \subset \mathbb{R}^3$, moving in space-time, our next task is to adequately describe the related motion spatial phase space variables, compatible with the corresponding Euler evolution equations.

*10.2. Diffeomorphism Group Structure and Functional Phase Space Description*

　　It is well known that the same physical system is often described using different sets of variables, related with their different physical interpretation. Simultaneously, this same system is endowed with different mathematical structures deeply depending on the geometric scenario used for its description. In general, these structures prove to be not equivalent but some special way connected to each other. In particular, such double descriptions commonly occur in systems with distributed parameters such as hydrodynamics, magnetohydrodynamics and diverse gauge systems, which are effectively described by means of both symplectic and Poisson structures on suitable phase spaces. In particular, it was observed [25–33] that these structures are canonically related to each other. Mathematical properties, lying in a background of their analytical description, make it possible to study additional important parameters [34–50] of different hydrodynamic and magnetohydrodynamic systems, amongst which we will mention integral invariants, describing such internal fluid motion peculiarities as vortices, topological singularities [51] and other different instability states, strongly depending [52,53] on imposed isentropic fluid motion constraints. Being interested in their general properties and mathematical structures, responsible for their existence and behavior, we present a detailed enough differential geometrical approach to investigating thermodynamically quasi-stationary isentropic fluid motions, paying more attention to analytical argumentation of tricks and techniques used during the presentation.

　　In particular, we consider a compressible liquid filling a compact linearly-connected domain $M \subset \mathbb{R}^3$ with smooth boundary $\partial M$, and moving free of external forces. A configuration of this fluid is called the reference or Lagrangian configuration, its points are called material or Lagrangian points and are denoted by $X \in M$ and are referred to as material, or Lagrangian coordinates. We shall not for now be specific about the correct choices of the related functional spaces to be used and refer to works [191,192], where this is discussed in great detail. The manifold $M \subset \mathbb{R}^3$, thought of as the target space of a configuration $\eta \in \mathrm{Diff}(M)$ of the fluid at a different time, is called the spatial or Eulerian configuration, whose points, called spatial or Eulerian points, will be denoted by small letters $x \in M$. Then a motion of the fluid is a time dependent family [26,29,41,48,122,192–195] of diffeomorphisms written as

$$M \ni x_t = \eta(X, t) := \eta_t(X) \in M \tag{388}$$

for any initial configuration $X \in M$ and some mapping $\eta_t \in \mathrm{Diff}(M), t \in \mathbb{R}$. We also are given the mass density $\rho_0 \in \mathcal{R}(M) \subset C^\infty(M; \mathbb{R}_+)$ and the specific entropy $\sigma_0 \in \Sigma(M) \subset C^\infty(M; \mathbb{R}_+)$ of the fluid in the reference configuration, changing in time in such a way that

$$\rho_0(X) = \rho_t(x_t)J_{\eta_t}(x_t), \sigma_0(X) = \sigma_t(x_t), \tag{389}$$

where $J_{\eta_t}(x_t)$ denotes the standard Jacobian determinant of the motion $\eta_t \in \mathrm{Diff}(M)$ at $x_t \in M$ and $\sigma_t(x_t)$ denotes the specific entropy for any $x_t = \eta_t(X) \in M$ and $t \in \mathbb{R}$. For a motion $x_t = \eta_t(X) \in M$ and arbitrary $X \in M, t \in \mathbb{R}$, one usually defines three velocities:

　　the material or Lagrangian velocity

$$V(X, t) = V_t(X) := \partial \eta_t(X)/\partial t, \tag{390}$$

　　the spatial or Eulerian velocity

$$v(x_t, t) = v_t(x_t) := v_t \circ \eta_t(X) \tag{391}$$

　　and convective or body velocity

$$\mathcal{V}(X, t) = \mathcal{V}_t(X) := -\partial X(x_t, t)/\partial t = -\partial \eta_t^{-1}(x_t)/\partial t, \tag{392}$$

being equivalent to the expression $\mathcal{V}_t = \eta_{t,*}^{-1} v_t$ for all $t \in \mathbb{R}$. Since the velocity $v_t : M \in T(M)$ is tangent to $M$ for all $t \in \mathbb{R}$ at $x_t = \eta_t(X) \in M$, it determines a time dependent vector field on $M$. On the other hand, tangency of $V_t(X)$ and $\eta_t(X), X \in M$, means that the velocity $V_t$ is a vector field over a configuration $\eta_t \in \mathrm{Diff}(M)$ on $M$, that is $V_t : M \to T(M)$ is such a map that $V_t(X)$ is tangent to $M$ not at $X \in M$, but at point $x_t = \eta_t(X) \in M$. Simultaneously, the velocity $\mathcal{V}_t(X)$ is a tangent vector to $M$ at $X \in M$, that is $\mathcal{V}_t$ is also a time dependent vector field on $M$. In what will follow, we will think of the fluid as moving smoothly in the domain $M \subset \mathbb{R}^3$, at any time filling it and producing no shocks and cavitation.

We present in the introductory section a modern differential geometric description of the isentropic fluid motion phase space and featuring diffeomorphism group structure, modelling the related dynamics, as well as its compatibility with the quasi-stationary thermodynamical constraints. The next section is devoted to the Hamiltonian analysis of the adiabatic liquid dynamics, within which, following the general approach of [28,41,194], we explain the nature of the related Poissonian structure on the fluid motion phase space, as a semidirect Banach groups product, and a natural reduction of the canonical symplectic structure on its cotangent space to the classical Lie–Poisson bracket on the adjoint space to the corresponding semidirect Lie algebras product. A modification of the Hamiltonian analysis in case of the isothermal liquid dynamics is presented in the next section. We proceed further to studying the differential-geometric structure of the adiabatic magneto-hydrodynamic superfluid phase space and its related motion within the Hamiltonian analysis and invariant theory. We construct there an infinite hierarchy of different kinds of integral magneto-hydrodynamic invariants, generalizing those previously constructed in [194,196], and analyzing their differential-geometric origins. The last section presents charged fluid dynamics on the phase space invariant with respect to an Abelian gauge group transformation.

*10.3. A Modified Current Algebra, Its Functional Representation And Geometric Description of the Ideal Liquid Dynamics*

It is well known that the motion of an ideal compressible and isentropic fluid is governed by the Euler equations

$$\begin{aligned} \partial v / \partial t + \langle v | \nabla \rangle v + \rho^{-1} \nabla p^{(0)} = 0, \\ \partial \rho / \partial t + \langle \nabla | \rho v \rangle = 0, \partial \sigma / \partial t + \langle v | \nabla \rangle \sigma = 0, \end{aligned} \tag{393}$$

where $p_0 : M \to \mathbb{R}$ is the internal fluid pressure, $\sigma = \sigma(x_t, t) = \sigma_t(x_t)$ is the specific entropy at a spatial point $x_t = \eta_t(X) \in M$ for any $t \in \mathbb{R}$, which is fixed owing to the Euler Equation (393), $\nabla := \partial / \partial x$ is the usual gradient on the space of smooth functions $C^\infty(M; \mathbb{R})$ and $\langle \cdot | \cdot \rangle$ denotes the usual convolution on $T(M) \times T(M)$ subject to the usual metric in $\mathbb{R}^3$, reduced on the submanifold $M$. The evolution (393) is considered to be *a priori* thermodynamic equilibrium and quasi-stationary, meaning that the following *infinitesimal heat convective* and strictly mathematical relationship (387), derived above in the Introduction,

$$\delta e_t(\rho_t(x_t), \sigma_t(x_t)) = T_t(x_t) \delta \sigma_t(x_t) + p_t^{(0)}(x_t) \rho_t^{-2}(x_t) \delta \rho_t(x_t) \tag{394}$$

holds for all $x_t \in M$ and $t \in \mathbb{R}$, where $e_t : \mathcal{R}(M) \times \Sigma(M) \to C^\infty(M \times \mathbb{R}; \mathbb{R})$ denotes the internal specific fluid energy, $T_t : M \to \mathbb{R}_+$ denotes the internal fluid absolute temperature, $p_t^{(0)} : M \to \mathbb{R}$ is the internal liquid pressure and the variation sign "$\delta$" means the change subject to both the temporal variable $t \in \mathbb{R}$ and the spatial variable $x_t \in M$.

Let us now analyze the internal mathematical structure of quantities $(\rho_t, \sigma_t) \in \mathcal{R}(M) \times \Sigma(M)$ as the *physical observables* subject to their evolution (393) with respect to the group

diffeomorphisms $\eta_t \in \text{Diff}(M), t \in \mathbb{R}$, generated by the liquid motion vector field $dx_t/dt = v_t(x_t), x_t := \eta_t(X), t \in \mathbb{R}, X \in M$:

$$
\begin{aligned}
\mathcal{L}_{d/dt}(\rho_t d^3 x_t \langle v_t | dx_t \rangle) &= \rho_t d^3 x_t (-\rho_t^{-1} dp_t^{(0)} + d|v_t|^2/2), \\
\mathcal{L}_{d/dt}(\rho_t d^3 x_t) &= 0, \quad \mathcal{L}_{d/dt} \sigma_t = 0,
\end{aligned}
\tag{395}
$$

where $\mathcal{L}_{d/dt} : \Lambda(M) \to \Lambda(M)$ denotes the corresponding Lie derivative with respect to the vector field $d/dt := \partial/\partial t + \langle v_t | \nabla \rangle \in \Gamma(M \times \mathbb{R}; T(M)), t \in \mathbb{R}$. The relationships (395) here simply mean that at every fixed $t \in \mathbb{R}$ the space of physical observables, being by definition, the adjoint space $\mathcal{G}^* := (\Lambda^1(M) \otimes \Lambda^3(M)) \oplus (\Lambda^3(M) \oplus \Lambda^0(M))$ to the vector space $\mathcal{G} := \Gamma(M; T(M)) \times (\Lambda^0(M) \oplus \Lambda^3(M)) \simeq T_{Id}(G)$, the tangent space at the identity $Id$ to the extended differential-functional current group manifold $G := \text{Diff}(M) \times (\Lambda^0(M) \times \Lambda^3(M)) \simeq \text{Diff}(M) \times (\mathcal{R}(M) \times \Sigma(M))$, where we have naturally identified the Abelian group product $\Lambda^0(M) \times \Lambda^3(M)$ with its direct tangent space sum $T(\Lambda^0(M)) \oplus T(\Lambda^3(M))$.

Consider now the natural action $\text{Diff}(M) \times G \to G$ of the $\text{Diff}(M)$-group on the constructed differential-functional manifold $G$:

$$
\begin{aligned}
(\eta \circ \varphi)(X) &:= \varphi(\eta(X)), (\eta \circ r)(X) := r(\eta(X)), \\
\eta \circ (s(X) d^3 X) &:= \eta^*(s(X) d^3 X)
\end{aligned}
\tag{396}
$$

for $\eta \in \text{Diff}(M), X \in M$ and any $(\varphi; r, s) \in \text{Diff}(M) \times (\mathcal{R}(M) \times \Sigma(M))$. Then, taking into account the suitably extended action (396) on the differential-functional manifold $G$, one can formulate the following easily checkable and crucial for what will follow further proposition.

**Proposition 14.** *The functional manifold $G := \text{Diff}(M) \times (\mathcal{R}(M) \times \Sigma(M))$ in Eulerian coordinates is a smooth symmetry Banach group $G := \text{Diff}(M) \ltimes (\mathcal{R}(M) \times \Sigma(M))$, equal to the semidirect product of the diffeomorphism group $\text{Diff}(M)$ and the direct product $\mathcal{R}(M) \times \Sigma(M)$ of the Abelian functional $\mathcal{R}(M) \simeq \Lambda^0(M)$, and density $\Sigma(M) \simeq \Lambda^3(M)$ group, endowed in Eulerian variables with the following right group multiplication law:*

$$
\begin{aligned}
(\varphi_1; r_1, s_1 d^3 x) \circ (\varphi_2; r_2, s_2 d^3 x) &= \\
= (\varphi_2 \cdot \varphi_1; r_1 + r_2 \cdot \varphi_1, s_1 d^3 x + (s_2 d^3 x) \cdot \varphi_1)
\end{aligned}
\tag{397}
$$

*for arbitrary elements $\varphi_1, \varphi_2 \in \text{Diff}(M), r_1, r_2 \in \Lambda^0(M)$ and $s_1 d^3 x, s_2 d^3 x \in \Lambda^3(M)$.*

This proposition allows a simple enough interpretation, namely, it means that the adiabatic mixing of the $G \ni (\varphi_2; r_2, s_2 d^3 x)$-liquid configuration with the $G \ni (\varphi_1; r_1, s_1 d^3 x)$-liquid configuration amounts to summation of their densities and entropies, simultaneously changing the common specific density owing to the fact that some space of the domain $M$ is already occupied by the first liquid configuration and the second one should be diffeomorphically shifted from this configuration to another free part of the spatial domain $M$, whose volume is assumed to be fixed and bounded.

The second important observation concerns the variational one-form (394), which can be naturally interpreted as some constraint on the group manifold $G$ for any fixed initial extended Lagrangian configuration $(\eta; \rho_0, \sigma_0 d^3 X) \in G$, as it follows from the conditions (389):

$$
J_{\eta_t}(X) \rho_t \circ \eta_t(X) := \rho_0(X), \ \sigma_t \circ \eta_t(X) := \sigma_0(X)
\tag{398}
$$

for all $X \in M, \eta_t \in \text{Diff}(M)$ and $t \in \mathbb{R}$. In addition, if to determine, owing to (394) and the streamline adiabatic constraint $\delta \sigma_t(x_t) = 0$ for all $t \in \mathbb{R}$, the specific energy density

$$
e_t(\rho_t, \sigma_t) := w_t^{(0)}(\rho_t, \sigma_t) + c_t(\sigma_t)
\tag{399}
$$

for some still unknown mapping $c_t : \Sigma(M) \to C^\infty(M \times \mathbb{R}; \mathbb{R})$ and the internal potential energy function $w_t^{(0)} : \mathcal{R}(M) \times \Sigma(M) \to C^\infty(M; \mathbb{R})$ of the liquid under regard, the local energy conservation property

$$\frac{d}{dt} \int_{D_t} e_t(\rho_t, \sigma_t) \rho_t(x_t) d^3 x_t = - \int_{D_t} \langle \nabla | p_t^{(0)}(x_t) v_t(x_t) \rangle d^3 x_t \tag{400}$$

holds for all $t \in \mathbb{R}$ and the domain $D_t := \eta_t(D) \subset M$, where a smooth submanifold $D \subset M$ is chosen arbitrarily and $\eta_t : M \to M$ denotes the corresponding evolution subgroup of the diffeomorphism group $\mathrm{Diff}_0(M)$, generated by the Euler evolution Equation (393), becomes compatible with constraint (394) if there holds the following equality:

$$p_t^{(0)}(x_t) = \rho_t(x_t)^2 \partial w_t^{(0)}(\rho_t, \sigma_t) / \partial \rho_t \tag{401}$$

for all $x_t \in M$ and $t \in \mathbb{R}$. In particular, from (400) and (401) the following global internal energy functional

$$H := \int_M [w_t^{(0)}(\rho_t, \sigma_t) + c_t(\sigma_t)] \, \rho_t(x_t) d^3 x_t \tag{402}$$

is conserved that is $dH/dt = 0$ for all $t \in \mathbb{R}$.

As the extended Lagrangian configuration $(\eta; \rho_0, \sigma_0 d^3 X) \in G$ is fixed for all whiles of time $t \in \mathbb{R}$ and the dynamical variables $\rho_t \in \mathcal{R}(M)$ and $\sigma_t \in \Sigma(M)$ depend only on the evolution diffeomorphisms $\eta_t \in \mathrm{Diff}(M)$, $t \in \mathbb{R}$, it is reasonable to consider the constraint (394) as an element of the cotangent space $T^*_{\eta_t}(\mathrm{Diff}(M))$ to the diffeomorphism group $\mathrm{Diff}(M)$ at the point $\eta_t \in \mathrm{Diff}(M)$ for any $t \in \mathbb{R}$.

Determine first the tangent space $T_\eta(G)$ to the group manifold $G$ at point $(\eta; \rho_0, \sigma_0 d^3 X) \in G$, which will be the direct product of the tangent spaces $T_\eta(\mathrm{Diff}(M)), T_{\rho_0}(\Lambda^0(M))$ and $T_{\sigma_0 d^3 X}(\Lambda^3(M))$. The last two tangent spaces are isomorphic, respectively, to themselves, that is $T_{\rho_0}(\Lambda^0(M)) \simeq \Lambda^0(M)$ and $T_{\sigma_0 d^3 X}(\Lambda^3(M)) \simeq \Lambda^3(M)$ at any $X \in M$. Their adjoint spaces are naturally determined as suitably constructed density and functional spaces on the manifold $M : T^*_{\rho_0}(\Lambda^0(M)) \simeq \Lambda^3(M)$ and $T^*_{\sigma_0 d^3 X}(\Lambda^3(M)) \simeq \Lambda^0(M)$. Concerning the tangent space $T_\eta(\mathrm{Diff}(M))$ at a configuration $\eta \in \mathrm{Diff}(M)$ we will make use of the construction, devised before in [122,181,194]. Namely, let $\eta \in \mathrm{Diff}(M)$ be a Lagrangian configuration and determine the tangent space $T_\eta(\mathrm{Diff}(M))$ at $\eta \in \mathrm{Diff}(M)$ as the collection of left invariant vectors $\xi_\eta := L_{\eta,*}\xi$ at $\eta \in \mathrm{Diff}(M)$, where $L_\eta : \mathrm{Diff}(M) \to \mathrm{Diff}(M)$ is, by definition, the left shift on the diffeomorphism group $\mathrm{Diff}(M)$ and $\xi \in T_{Id}(\mathrm{Diff}(M))$ is a tangent vector at the unity $Id \in \mathrm{Diff}(M)$. It is obvious that for all reference points $X \in M$ and any smooth curve $\mathbb{R} \ni \tau \to \eta_\tau \in \mathrm{Diff}(M)$ of diffeomorphisms of $M$, the set of right invariant vectors $\xi(X) = (\eta^{-1} \circ d\eta_t/d\tau)(X))|_{\tau=0} \in T_X(M)$ at point $X \in M$ defines a smooth vector field $\xi : M \to T(M)$ on the manifold $M$. Since, by definition, the tangent space $T_{Id}(\mathrm{Diff}(M))$ coincides with the Lie algebra $\mathrm{Diff}(M)$ of the diffeomorphism group $\mathrm{Diff}(M)$, strictly isomorphic to the Lie algebra $\Gamma(T(M))$ of right invariant vector fields on $M$, the dual space $T^*_{Id}(\mathrm{Diff}(M))$ can be naturally determined from the geometric point of view as the space $diff^*(M)$, consisting of analytic functions on $diff(M)$ and coinciding with the set of one-form densities on $M$:

$$diff^*(M) \simeq \Lambda^1(M) \otimes |\Lambda^3(M)|. \tag{403}$$

Similarly, the cotangent space $T^*_\eta(\mathrm{Diff}(M))$ consists of all one-form densities on $M$ over $\eta \in \mathrm{Diff}(M)$:

$$T^*_\eta(\mathrm{Diff}(M)) = \{\alpha_\eta : M \to T^*(M) \otimes |\Lambda^3(M)| : \alpha_\eta(X) \in T^*_{\eta(X)}(M) \otimes |\Lambda^3(M)|\} \tag{404}$$

subject to the canonical nondegenerate convolution $(\cdot|\cdot)_c$ on $T_\eta^*(\mathrm{Diff}(M)) \times T_\eta(\mathrm{Diff}(M))$: if $\alpha_\eta \in T_\eta^*(\mathrm{Diff}(M))$, $\xi_\eta \in T_\eta(\mathrm{Diff}(M))$, where $\alpha_\eta|_X = \langle \alpha_\eta(X)|dx \rangle \otimes d^3X$, $\xi_\eta|_X = \langle \xi_\eta(X)|\partial/\partial x \rangle$, then

$$(\alpha_\eta|\xi_\eta)_c := \int_M \langle \alpha_\eta(X)|\xi_\eta(X) \rangle d^3X. \tag{405}$$

The construction above makes it possible to identify the cotangent bundle $T_\eta^*(\mathrm{Diff}(M))$ at the fixed Lagrangian configuration $\eta \in \mathrm{Diff}(M)$ to the tangent space $T_\eta(\mathrm{Diff}(M))$, insomuch as the tangent space $T(M)$ is endowed with the natural internal tangent bundle metric $\langle \cdot | \cdot \rangle_g$ at any point $\eta(X) \in M$, identifying $T(M)$ with $T^*(M)$ via the metric isomorphism $\sharp : T^*(M) \to T(M)$. The latter can also be naturally lifted to $T_\eta^*(\mathrm{Diff}(M))$ at $\eta \in \mathrm{Diff}(M)$, namely: for any elements $\alpha_\eta, \beta_\eta \in T_\eta^*(\mathrm{Diff}(M))$, $\alpha_\eta|_X = \langle \alpha_\eta(X)|dx \rangle \otimes d^3X$ and $\beta_\eta|_X = \langle \beta_\eta(X)|dx \rangle \otimes d^3X \in T_\eta^*(\mathrm{Diff}(M))$ we can define the metric

$$(\alpha_\eta|\beta_\eta)_g := \int_M \rho_0(X)\langle \alpha_\eta^\sharp(X)|\beta_\eta^\sharp(X) \rangle_g d^3X, \tag{406}$$

where, by definition, $\alpha_\eta^\sharp(X) := \sharp(\rho_0(X)^{-1}\langle \alpha_\eta(X)|dx \rangle)$, $\beta_\eta^\sharp(X) := \sharp(\rho_0(X)^{-1}\langle \beta_\eta(X)|dx \rangle)$ $\in T_{\eta(X)}(M)$ for any $X \in M$.

The diffeomorphism group $\mathrm{Diff}(M)$ can be naturally restricted to the factor-group $\mathrm{Diff}_0(M) := \mathrm{Diff}(M)/Diff_{\rho_0,\sigma_0}(M)$ subject to the stationary normal symmetry subgroup $\mathrm{Diff}_{\rho_0,\sigma_0}(M) \subset \mathrm{Diff}(M)$, where

$$Diff_{\rho_0,\sigma_0}(M) := \{\varphi \in \mathrm{Diff}(M) : \rho_0(X) = J_{\varphi(X)}\rho_0(\varphi(X)), \sigma_0(X) = \sigma_0(\varphi(X))\} \tag{407}$$

for any $X \in M$. Based on the construction above, one can proceed to constructing smooth flows and functionals on the specially extended group manifold $G_0 := \mathrm{Diff}_0(M) \ltimes (\Lambda^0(M) \times \Lambda^3(M))$ and consider their coadjoint action on the cotangent bundle $T_{g_\eta}^*(G_0)$, $g_\eta := (\eta; \rho_0, \sigma_0) \in G_0$, and relate them in some way to the evolution with respect to the Euler Equation (393). Moreover, as the cotangent bundle $T_{g_\eta}^*(G_0)$, $g_\eta \in G_0$, is a priori endowed with the canonical Poisson structure, one can study both the Hamiltonian flows on it, related with the Euler Equation (393), and a hidden geometrical meaning of the differential constraints like (394).

*10.4. The Hamiltonian Analysis and Related Adiabatic Liquid Dynamics*

We observed above that the liquid motion is adequately described by means of the symmetry diffeomorphism group $\mathrm{Diff}_0(M)$, acting on the target manifold $M \subset \mathbb{R}^3$, and in this way the modeling liquid motion, generated by suitable vector fields on $\mathrm{Diff}_0(M)$. This also means that the fluid motion strongly depends on the constraint (394) on the cotangent bundle $T_{g_\eta}^*(G_0)$, $g_\eta \in G_0$, and *a priori* possesses the canonical Poisson structure on it. Taking into account that the diffeomorphism group $\mathrm{Diff}_0(M)$ acts on the extended group density manifold $G_0 := \mathrm{Diff}_0(M) \ltimes (\Lambda^0(M) \times \Lambda^3(M))$, fixed by the element $(\eta; \rho_0, \sigma_0 d^3X) \in G$, one can suitably construct the canonical Poisson bracket on the cotangent bundle $T_{g_\eta}^*(G_0)$, $g_\eta \in G_0$, using the canonical coordinate variables on it. Namely, let $(\mu_\eta; \rho_0 d^3X, \sigma_0) \in T_{g_\eta}^*(G_0)$, $g_\eta \in G_0$, be coordinates on $T_{g_\eta}^*(G_0)$, where

$$\mu_\eta(X) = \rho_0(X)[V_\eta^\flat(X)]d^3X|_{x=\eta(X)} = \tag{408}$$
$$= \rho_0(X)v^\flat(\eta(X))J_{\eta^{-1}}(x)d^3x := \rho(x)v(x)d^3x,$$
$$r_\eta(X) = \rho_0(X)d^3X = \rho_0(X)d^3X|_{x=\eta(X)} := \rho(x)d^3x,$$
$$s_\eta(X) = \sigma_0(X) = \sigma(\eta(X))|_{x=\eta(X)} := \sigma(x)$$

and $\flat := \sharp^{-1}$, being suitably represented into the Eulerian spatial variables on $T_{g_\eta}^*(G_0)$ at point $(\eta; \rho, \sigma d^3x) \in G_0$. In particular, the quantities $\mu(x) := \rho(x)v(x)d^3x = (\eta^*\mu_\eta)(X)$,

$r(x) := \rho(x)d^3x = (\eta^* r_\eta)(X)$ and $s(x) := \sigma(x) = (\eta^* s_\eta)(X)$ are called, respectively, the Eulerian momentum density, the Eulerian fluid density and entropy variables at point $x = \eta(X) \in M$. The corresponding metric on $T^*_{g_\eta}(G_0)$ is given by the expression

$$((\alpha_{\eta,1}; r_{\eta,1} s_{\eta,1}) | (\alpha_{\eta,2}; r_{\eta,2} s_{\eta,2})) := (\alpha_{\eta,1} | \alpha_{\eta,2}) +$$
$$+ (r_{\eta,1} | r_{\eta,2}) + (s_{\eta,1} | s_{\eta,2}), \tag{409}$$

where $(\alpha_{\eta,1} | \alpha_{\eta,2})$ for $\alpha_{\eta,1}, \alpha_{\eta,2} \in T^*_\eta(\text{Diff}_0(M))$ is determined by (406) and for any $r_{\eta,1}, r_{\eta,2} \in T^*_\eta(\Lambda^3(M))$ and $s_{\eta,1}, s_{\eta,2} \in T^*_\eta(\Lambda^0(M))$ one determines, respectively, as

$$(r_{\eta,1} | r_{\eta,2}) := \int_M (\rho_1(x)\rho_2(x))d^3x, \quad (s_{\eta,1} | s_{\eta,2}) := \int_M (\sigma_1(x)\sigma_2(x))d^3x. \tag{410}$$

Consider now the cotangent bundle $T^*_{g_\eta}(G_0)$ at point $g_\eta = (\eta; \rho, \sigma d^3x) \in G_0$ as a smooth manifold endowed with the canonical symplectic structure on it, equivalent to the corresponding canonical Poisson bracket on $T^*_{g_\eta}(G_0)$. Taking into account that the manifold $T^*_{g_\eta}(G_0)$, shifted by the right $R_{\eta^{-1}}$-action to the manifold $T^*_{Id}(G_0)$, $Id \in G_0$, becomes diffeomorphic to the adjoint space $\mathcal{G}^*$ to the Lie algebra $\mathcal{G}$ of the group $G_0$, as was stated [30–33,41] by S. Lie in 1887, this canonical Poisson bracket on $T^*_\eta(G_0)$ transforms [26,31,32,41,181,195] into the classical Lie–Poisson bracket on the adjoint space $\mathcal{G}^*$. Moreover, the orbits of the group $G_0$ on $T^*_{g_\eta}(G_0)$, $g_\eta = (\eta; \rho, \sigma d^3x) \in G_0$, transform into the corresponding coadjoint orbits on the adjoint space $\mathcal{G}^*$, generated by elements of the Lie algebra $\mathcal{G}$. To construct this Lie–Poisson bracket, we formulate the following preliminary proposition.

**Proposition 15.** *The Lie algebra $\mathcal{G} \simeq \Gamma(M; T(M)) \ltimes (\Lambda^0(M)) \oplus \Lambda^3(M))$ is determined by the following Lie commutator relationships:*

$$[(a_1; r_1, s_1), (a_2; r_2, s_2)] = ([a_1, a_2];$$
$$\langle a_1 | \nabla r_2 \rangle - \langle a_2 | \nabla r_1 \rangle, \langle \nabla | a_1 s_2 \rangle - \langle \nabla | a_2 s_1 \rangle) \tag{411}$$

*for any vector fields $a_1, a_2 \in diff_0(M) \simeq \Gamma(M; T(M))$ and scalar quantities $r_1, r_2 \in \Lambda^0(M)$ and $s_1, s_2 \in \Lambda^3(M)$ on the manifold $M$.*

**Proof.** Proof of the commutation relationships (411) easily follows from the group multiplication (397), if to take into account that tangent spaces $T(\Lambda^0(M)) \simeq \Lambda^0(M)$ and $T(\Lambda^3(M)) \simeq (\Lambda^3(M))$. $\square$

As an example, we calculate, for brevity, the Poisson bracket on the cotangent space $T^*_\eta(\text{Diff}(\mathbb{T}^n))$ at any $\eta \in \text{Diff}(\mathbb{T}^n)$. Consider the cotangent space $T^*_\eta(\text{Diff}(\mathbb{T}^n)) \simeq diff^*(\mathbb{T}^n)$, the adjoint space to the tangent space $T_\eta(\text{Diff}(\mathbb{T}^n))$ of left invariant vector fields on $\text{Diff}(\mathbb{T}^n)$ at any $\eta \in \text{Diff}(\mathbb{T}^n)$, and take the canonical symplectic structure on $T^*_\eta(\text{Diff}(\mathbb{T}^n))$ in the form $\omega^{(2)}(\mu, \eta) := \delta\alpha(\mu, \eta)$, where the canonical Liouville form $\alpha(\mu, \eta) := (\mu | \delta\eta)_c \in \Lambda^1_{(\mu,\eta)}(T^*_\eta(\text{Diff}(\mathbb{T}^n)))$ at a point $(\mu, \eta) \in T^*_\eta(\text{Diff}(\mathbb{T}^n))$ is defined *a priori* on the tangent space $T_\eta(\text{Diff}(\mathbb{T}^n)) \simeq \Gamma(T(M))$ of right-invariant vector fields on the torus manifold $\mathbb{T}^n$. Having calculated the corresponding Poisson bracket of smooth functions $(\mu | a)_c, (\mu | b)_c \in C^\infty(T^*_\eta(\text{Diff}(\mathbb{T}^n)); \mathbb{R})$ on $T^*_\eta(\text{Diff}(\mathbb{T}^n)) \simeq diff^*(\mathbb{T}^n)$, $\eta \in \text{Diff}(\mathbb{T}^n)$, one can formulate the following proposition.

**Proposition 16.** *The Lie–Poisson bracket on the coadjoint space $T^*_\eta(\text{Diff}(\mathbb{T}^n)) \simeq diff^*(\mathbb{T}^n)$ is equal to the expression*

$$\{f, g\}(\mu) = (\mu | [\delta f(\mu)/\delta\mu, \delta g(\mu)/\delta\mu])_c \tag{412}$$

*for any smooth functionals $f, g \in C^\infty(\mathcal{G}^*; \mathbb{R})$.*

**Proof.** By definition [26,122] of the Poisson bracket of smooth functions $(\mu|a)_c, (\mu|b)_c$ $\in C^\infty(T_\eta^*(\text{Diff}(\mathbb{T}^n)); \mathbb{R})$ on the symplectic space $T_\eta^*(\text{Diff}(\mathbb{T}^n))$, it is easy to calculate that

$$\{\mu(a), \mu(b)\} := -\delta\alpha(X_a, X_b) = \\ = -X_a(\alpha|X_b)_c + X_b(\alpha|X_a)_c + (\alpha|[X_a, X_b])_c, \tag{413}$$

where $X_a := \delta(\mu|a)_c/\delta\mu = a \in diff(\mathbb{T}^n), X_b := \delta(\mu|b)_c/\delta\mu = b \in diff(\mathbb{T}^n)$. Since the expressions $X_a(\alpha|X_b)_c = 0$ and $X_b(\alpha|X_a)_c = 0$ owing the right-invariance of the vector fields $X_a, X_b \in T_\eta(\text{Diff}(\mathbb{T}^n))$, the Poisson bracket (412) transforms into

$$\{(\mu|a)_c, (\mu|b)_c\} = (\alpha|[X_a, X_b])_c = \\ = (\mu|[a, b])_c = (\mu|[\delta(\mu|a)_c/\delta\mu, \delta(\mu|b)_c/\delta\mu])_c \tag{414}$$

for all $(\mu, \eta) \in T_\eta^*(\text{Diff}(\mathbb{T}^n)) \simeq diff^*(\mathbb{T}^n), \eta \in \text{Diff}(\mathbb{T}^n)$ and any $a, b \in diff(\mathbb{T}^n)$. The Poisson bracket (412) is easily generalized to

$$\{f, g\}(\mu) = (\mu|[\delta f(\mu)/\delta\mu, \delta g\mu)/\delta\mu])_c \tag{415}$$

for any smooth functionals $f, g \in C^\infty(\mathcal{G}^*; \mathbb{R})$, finishing the proof. $\square$

Proceed now to the Grassmann algebra $\Lambda(M)$ endowed with Hodge [197] star-isomorphism $* : \Lambda(M) \to \Lambda(M)$ subject to the usual metric on the tangent space $T(M)$ and determine the adjoint space to the Abelian subalgebra $\mathcal{R}(M) \oplus \Sigma(M) \simeq \Lambda^0(M) \oplus \Lambda^3(M)$ as the space $*\Lambda^3(M) \oplus *\Lambda^0(M)$ with respect to the following scalar product on $\Lambda(M)$ :

$$(\alpha^{(n)}|\beta^{(m)}) := \delta_{mn} \int_M (\alpha^{(n)} \wedge *\beta^{(m)}) \tag{416}$$

for any $\alpha^{(n)}, \beta^{(m)} \in \Lambda(M), m, n = \overline{0,3}$. Then the adjoint space $\mathcal{G}^*$, owing to the expressions (409) and (389), is described by means of the Eulerian variables $(\mu; \rho d^3 x, \sigma) \in \mathcal{G}^* \simeq (\Lambda^1(M) \otimes |\Lambda^3(M)|) \ltimes (\Lambda^3(M) \oplus \Lambda^0(M))$. The latter makes it possible to calculate the corresponding Lie–Poisson bracket on the adjoint space $\mathcal{G}^*$ at a point $l := (\mu; \rho d^3 x, \sigma) \in \mathcal{G}^*$, generalizing the Poisson bracket (414):

$$\{f, g\}(l) = (l|[\delta f/\delta l, \delta g/\delta l])_c =$$

$$= \int_M d^3 x \left\langle m \left| \left[ \left\langle \frac{\delta f}{\delta m} | \nabla \right\rangle \frac{\delta g}{\delta m} - \left\langle \frac{\delta g}{\delta m} | \nabla \right\rangle \frac{\delta f}{\delta m} \right] \right\rangle + $$

$$+ \int_M \rho d^3 x \left[ \left\langle \frac{\delta f}{\delta m} \left| \nabla \frac{\delta g}{\delta \rho} \right\rangle - \left\langle \frac{\delta g}{\delta m} \left| \nabla \frac{\delta f}{\delta \rho} \right\rangle \right] + \tag{417}$$

$$+ \int_M \sigma \left[ \left\langle \nabla \left| \frac{\delta f}{\delta m} \frac{\delta g}{\delta \sigma} \right\rangle - \left\langle \nabla \left| \frac{\delta g}{\delta m} \frac{\delta f}{\delta \sigma} \right\rangle \right] d^3 x$$

for any smooth functionals $f, g \in C^\infty(\mathcal{G}^*; \mathbb{R})$, where we put, by definition, $\mu(x) := \langle m(x)|dx \rangle \otimes d^3 x, m(x) = \rho(x)v(x) \in T^*(M)$ for all $x \in M$ and any $t \in \mathbb{R}$.

Return now to the constraint (394) in the following variational form:

$$\delta e_t(\rho_t, \sigma_t)/\delta t = T_t(x_t)\delta\sigma_t(x_t)/\delta t + p_t^{(0)}(x_t)\rho_t^{-2}(x_t)\delta\rho_t(x_t)/\delta t, \tag{418}$$

which should hold at any $x_t \in M$ for all $t \in \mathbb{R}$. Insomuch as, owing to the Euler Equation (393), the full (convective) derivative $\delta\sigma_t(x_t)/\delta t = 0$ at any $x_t \in M$ for all $t \in \mathbb{R}$, one checks once more that the expression (399) holds at any $x_t \in M$ for all $t \in \mathbb{R}$. To determine the energy density function (399), we consider the Euler Equation (393) and

check their Hamiltonian structure subject to the Poisson bracket (417), i.e., the existence of a Hamiltonian functional $H : \mathcal{G}^* \to \mathbb{R}$, for which

$$\frac{\partial}{\partial t}(m; \rho, \sigma)^\mathsf{T} = \{H, (m, \rho, \sigma)^\mathsf{T}\} \tag{419}$$

at any element $l = (m := \rho v; \rho, \sigma)^\mathsf{T} \in \mathcal{G}^*$. By means of easy calculations, one obtains from the system (419) the variational gradient vector

$$\delta H(l)/\delta l = (m\rho^{-1}; -|m|^2/(2\rho^2) + w^{(0)}(\rho, \sigma) + \rho \partial w^{(0)}(\rho, \sigma)/\partial \rho, \rho \partial w^{(0)}(\rho, \sigma)/\partial \sigma), \tag{420}$$

from which one derives [11,59,120] via the Volterra homotopy mapping

$$H = \int_0^1 (\delta H(\lambda l)/\delta l | l)_c d\lambda \tag{421}$$

the exact Hamiltonian expression

$$H = \int_M (|m|^2/(2\rho) + \rho w^{(0)}(\rho, \sigma)] d^3x, \tag{422}$$

coinciding with the expression (402) at $c(\sigma) := |m|^2/(2\rho^2) = |v|^2/2$, as $m := \rho v$ for $v \in T(M)$. Thus, we obtain the internal energy density functional (399) as

$$e_t(\rho_t, \sigma_t) = |v_t|^2/2 + w_t^{(0)}(\rho_t, \sigma_t), \tag{423}$$

for all $\rho := \rho_t \in \mathcal{R}(M), \sigma := \sigma_t \in \Sigma(M)$ and $v_t \in T(M)$, satisfying simultaneously both the constraint (394) and the Euler evolution Equation (393) for all $t \in \mathbb{R}$. Moreover, from the condition (400) one easily finds [194] the following important local differential relationship:

$$\partial[\rho_t(x_t)e_t(\rho_t, \sigma_t)]/\partial t + \langle \nabla | \rho_t(x_t)v_t(x_t)(e_t(\rho_t, \sigma_t) + \\ + \rho_t(x_t)\partial w_t^{(0)}(\rho_t, \sigma_t)/\partial \rho_t )\rangle = 0, \tag{424}$$

satisfied for all $x_t \in M$ and $t \in \mathbb{R}$, also confirming the energy conservation (422).

*10.5. The Hamiltonian Analysis and Related Isothermal Liquid Dynamics*

Consider a liquid motion governed by the Euler equations

$$\partial v/\partial t + \langle v|\nabla \rangle v + \rho^{-1}\nabla p^{(0)} = 0, \\ \partial \rho/\partial t + \langle \nabla | \rho v \rangle = 0, \partial T/\partial t + \langle v|\nabla T \rangle = 0, \tag{425}$$

and describing the ideal compressible and isothermal motion of an ideal compressible and adiabatic fluid in a spatial domain $M \subset \mathbb{R}^3$, as the temperature $T_t(x_t) = T_0(x_t)$ at any evolution point $x_t := \eta_t(X) \in M$ for all $X \in M$ and $t \in \mathbb{R}$. The evolution (425) is considered to be *a priori* thermodynamically quasi-stationary, i.e., the following, *infinitesimal convective* energy relationship

$$\delta \tilde{h}_t(\rho_t, T_t) = -\sigma_t(x_t)\,\delta T_t + p_t^{(0)}(x_t)\,\rho_t^{-2}\delta \rho_t \tag{426}$$

holds for all densities $\rho_t \in \mathcal{R}(M)$, temperature $T_t \in \mathcal{T}(M)$ and specific entropy $\sigma_t \in \Sigma(M)$, where $\tilde{h} : \mathcal{R}(M) \times \mathcal{T}(M) \to \mathbb{R}$ denotes the corresponding internal specific fluid "energy" and the variation sign "$\delta$" means the change subject to both the temporal variable $t \in \mathbb{R}$ and the spatial variable $x_t \in M$. Under the imposed *isothermal condition* $\delta T_t = 0$ the expression (426) transforms into

$$\tilde{h}_t(\rho_t, T_t) = |v_t|^2/2 + \tilde{w}_t^{(0)}(\rho_t, T_t), \tag{427}$$

where $\tilde{w}_t^{(0)}(\rho_t, T_t) := w_t^{(0)}(\rho_t, \sigma_t)|_{\sigma_t := \tilde{\sigma}(\rho_t, T_t)} - T_t\sigma_t(\rho_t, T_t)$, is the specific potential liquid energy for *the isothermal flow*, determined at $\sigma_t := \sigma_t(\rho_t, T_t)$, solving the functional relation $T_t = \partial w_t^{(0)}(\rho_t, \sigma_t)/\partial\sigma_t \in \mathcal{T}(M)$ subject to the entropy argument $\sigma_t \in \Sigma(M)$, if the condition $\partial^2 w_t^{(0)}(\rho_t, \sigma_t)/\partial\sigma_t^2 \neq 0$ holds for all densities $\rho_t \in \mathcal{R}(M)$ and $t \in \mathbb{R}$.

Observe now that the third equation of (425) is exactly equivalent to the internal average fluid kinetic energy conservation integral relationship

$$\frac{d}{dt}\int_{D_t}\rho_t(x_t)T_t(x_t)d^3x_t = 0 \tag{428}$$

over the domain $D_t := \eta_t(D) \subset M$, where a smooth submanifold $D = D_t|_{t=0} \subset M$ is chosen arbitrarily and $\eta_t : M \to M, t \in \mathbb{R}$, denotes the corresponding evolution subgroup of the diffeomorphism group $\mathrm{Diff}_0(M)$, generated by the Euler evolution Equation (425). The relationship (428) simply means that if the density function $\rho_t \in \mathcal{R}(M)$ transforms under diffeomorphism group $\mathrm{Diff}_0(M)$ action as the Abelian functional group $\mathcal{R}(M) \simeq \Lambda^0(M)$, the corresponding transformation of the temperature $T_t \in \mathcal{T}(M)$ is induced by the diffeomorphism group $\mathrm{Diff}_0(M)$ action on the related Abelian group $\mathcal{T}(M) \simeq \Lambda^3(M)$. Concerning the energy density (427) one easily obtains the following differential relationship:

$$\partial[\rho_t(x_t)\tilde{h}_t(\rho_t, T_t)]/\partial t + \langle \nabla | \rho_t(x_t)v_t(x_t)\left[\tilde{h}_t(\rho_t, T_t)\rangle + \rho_t\partial\tilde{w}_t^{(0)}(\rho_t, T_t)/\partial\rho_t\right] \rangle = 0, \tag{429}$$

satisfied for all $t \in \mathbb{R}$. As a simple consequence of the relationship (429), one obtains that the following functional

$$\tilde{H} = \int_{D_t}\rho_t(x_t)\tilde{h}_t(\rho_t, T_t)d^3x_t \tag{430}$$

is conserved over the domain $D_t := \eta_t(D) \subset M, t \in \mathbb{R}$, where a smooth submanifold $D = D_t|_{t=0} \subset M$ is chosen arbitrarily.

Similarly to the reasoning above, one can now construct the differential-functional group space $\mathrm{Diff}(M) \times (\mathcal{R}(M) \times \mathcal{T}(M))$ and formulate the following easily checkable proposition. The differential-functional group functional manifold $\mathrm{Diff}(M) \times (\mathcal{R}(M) \times \mathcal{T}(M))$ in Eulerian coordinates is a smooth Banach group $G := \mathrm{Diff}(M) \ltimes (\mathcal{R}(M) \times \mathcal{T}(M))$, equal to the semidirect product of the diffeomorphism group $\mathrm{Diff}(M)$ and the direct product $\mathcal{R}(M) \times \mathcal{T}(M)$ of Abelian functional $\mathcal{R}(M) \simeq \Lambda^0(M)$ and density $\mathcal{T}(M) \simeq \Lambda^3(M)$ groups, endowed with the following group multiplication law:

$$\begin{aligned}(\varphi_1; r_1, \tau_1 d^3x\ )\circ(\varphi_2; r_2, \tau_2 d^3x) = \\ +(\varphi_2\cdot\varphi_1; r_1 + r_2\cdot\varphi_1, \tau_1 d^3x + (\tau_2 d^3x)\cdot\varphi_1)\end{aligned} \tag{431}$$

for arbitrary elements $\varphi_1, \varphi_2 \in \mathrm{Diff}(M), r_1, r_2 \in \Lambda^0(M)$ and $\tau_1 d^3x, \tau_2 d^3x \in \Lambda^3(M)$.

This proposition allows a simple enough interpretation, namely, it means that the adiabatic mixing of the $G \ni (\varphi_2; r_2, \tau_2 d^3x)$ liquid configuration with the $G \ni (\varphi_1; r_1, \tau_1 d^3x)$ liquid configuration amounts to summation of their spatially shifted densities, simultaneously changing the common specific kinetic energy, proportional [55,198,199] to the liquid temperature, owing to the fact that some space in the domain $M$ is already occupied by the first liquid configuration and the second one should be diffeomorphically shifted from this configuration to another free part of the spatial domain $M$ with fixed and bounded volume. The diffeomorphism group $\mathrm{Diff}(M)$ can be naturally restricted to the factor-group $\mathrm{Diff}_0(M) := \mathrm{Diff}(M)/\mathrm{Diff}_{\rho_0, T_0}(M)$ subject to the stationary normal symmetry subgroup $\mathrm{Diff}_0(M) := \mathrm{Diff}_{\rho_0, T_0}(M) \subset \mathrm{Diff}(M)$, where

$$\mathrm{Diff}_{\rho_0, T_0}(M) := \{\varphi \in \mathrm{Diff}(M) : \rho_0(X) = J_{\varphi(X)}\rho_0(\varphi(X)), T_0(X) = T_0(\varphi(X))\} \tag{432}$$

for any $X \in M$. Based on the construction above one can proceed to studying the extended Banach group $G := \mathrm{Diff}_0(M) \ltimes (\Lambda^0(M) \times \Lambda^3(M))$ action on the cotangent bundle $T^*_{g_\eta}(G)$

at $g_\eta := (\eta; \rho_0, T_0) \in G_0$, generated by the fluid evolution with respect to the Euler Equation (425). The related fluid motion is naturally modelled by means of the coadjoint action of the corresponding Lie algebra $\mathcal{G} \simeq T_{g_\eta}(G_0) \simeq \Gamma(M; T(M)) \ltimes (\Lambda^0(M) \oplus \Lambda^3(M))$ of the group $G_0$, $g_\eta = Id \in G_0$, on its adjoint space $\mathcal{G}^* \simeq (\Lambda^1(M) \otimes \Lambda^3(M)) \ltimes (*\Lambda^0(M) \oplus *\Lambda^3(M)) = (\Lambda^1(M) \otimes \Lambda^3(M)) \ltimes (\Lambda^3(M) \oplus \Lambda^0(M))$.

The related Lie structure on $\mathcal{G}$ easily ensues from the action (431):

$$[(a_1; r_1, \tau_1), (a_2; r_2, \tau_2)] = ([a_1, a_2]; \langle a_1 | \nabla r_2 \rangle - \langle a_2 | \nabla r_1 \rangle, \langle \nabla | a_1 \tau_2 \rangle - \langle \nabla | a_2 \tau_1 \rangle) \tag{433}$$

for any representative elements $(a_1; r_1, \tau_1)$ and $(a_2; r_2, \tau_2) \in \mathcal{G}$. Moreover, as the cotangent bundle $T_{g_\eta}^*(G_0)$ at $g_\eta = Id \in G_0$ is diffeomorphic to the adjoint space $\mathcal{G}^*$ to the Lie algebra $\mathcal{G}$ of the Banach group $G_0$, it is *a priori* endowed with the canonical Lie–Poisson structure

$$\{f, g\}(l) = (l|[\delta g / \delta l, \delta f / \delta l])_c =$$
$$= \int_M d^3x \left\langle m \left| \left[ \left\langle \tfrac{\delta f}{\delta m} | \nabla \right\rangle \tfrac{\delta g}{\delta m} - \left\langle \tfrac{\delta g}{\delta m} | \nabla \right\rangle \tfrac{\delta f}{\delta m} \right] \right\rangle + $$
$$+ \int_M \rho d^3x \left[ \left\langle \tfrac{\delta f}{\delta m} \left| \nabla \tfrac{\delta g}{\delta \rho} \right\rangle - \left\langle \tfrac{\delta g}{\delta m} \left| \nabla \tfrac{\delta f}{\delta \rho} \right\rangle \right] + $$
$$+ \int_M T \left[ \left\langle \nabla \left| \tfrac{\delta f}{\delta m} \tfrac{\delta g}{\delta T} \right\rangle - \left\langle \nabla \left| \tfrac{\delta g}{\delta m} \tfrac{\delta f}{\delta T} \right\rangle \right] d^3x \tag{434}$$

for any smooth functional $f, g \in C^\infty(\mathcal{G}^*; \mathbb{R})$, where we put, by definition, an element $l := (m; \rho, T) \simeq (\mu; \rho d^3x, T) \in \mathcal{G}^*$, $\mu(x) := \langle m(x) | dx \rangle \otimes d^3x$, $m(x) = \rho(x) v(x) \in T^*(M)$ for all $x \in M$ and $t \in \mathbb{R}$, one can easily check that the flow (425) is Hamiltonian:

$$dl/dt = \{\tilde{H}, l\} \tag{435}$$

subject to the adjusted Hamiltonian functional (430):

$$\tilde{H} := \int_M \rho_t h_t(\rho_t, T_t) d^3 x_t = \int_M \rho_t (|m_t|^2 / 2\rho_t^2 + \tilde{w}_t^{(0)}(\rho_t, T_t)) d^3 x. \tag{436}$$

satisfying the conservative condition $d\tilde{H}/dt = 0$ for all $t \in \mathbb{R}$, following simultaneously both from (435) and from the differential relationship (429).

### 10.6. The Hamiltonian Analysis and Adiabatic Magneto-Hydrodynamic Superfluid Motion

We start with considering a quasi-neutral superfluid contained in a domain $M \subset \mathbb{R}^3$ and interacting with a "frozen" sourceless magnetic field $B \in \mathcal{B}(M) \subset C^\infty(M; \mathbb{E}^3)$, satisfying the superconductivity conditions

$$\tilde{E} := E + v \times B = 0, \ \partial E/\partial t = \nabla \times B, \tag{437}$$

where $\tilde{E} : M \to \mathbb{E}^3$ is the internal net superfluid electric field, $E = -\partial A/\partial t : M \to \mathbb{E}^3$ and $B = \nabla \times A : M \to \mathbb{E}^3$ are the internal electric and magnetic fields, respectively, generated by the corresponding magnetic vector field potential $A : M \to \mathbb{E}^3$, $v : M \longrightarrow T(M)$ is the superfluid velocity and "$\times$" denotes the usual vector product in the Euclidean space $\mathbb{E}^3$. The following natural boundary conditions $\langle n | v \rangle |_{\partial M} = 0$ and $\langle n | B \rangle |_{\partial M} = 0$ are imposed on the superfluid flow, where $n \in T^*(M)$ is the vector normal to the boundary $\partial M$, which is considered to be smooth almost everywhere.

Then, in adiabatic magneto-hydrodynamics (MHD), quasi-neutral superconductive superfluid motion is described by the following system of evolution equations:

$$\partial v/\partial t + \langle v | \nabla \rangle v + \rho^{-1} \nabla p - \rho^{-1} (\nabla \times B) \times B = 0,$$
$$\partial \rho/\partial t + \langle \nabla | \rho v \rangle = 0, \partial \sigma/\partial t + \langle u | \nabla \sigma \rangle = 0, \partial B/\partial t = \nabla \times (v \times B), \tag{438}$$

where, as before, $\rho := \rho_t \in \mathcal{R}(M)$ is the superfluid density, $B := B_t : M \longrightarrow \mathbb{E}^3$ is the "frozen" into the superfluid magnetic field, $p := p_t : M \longrightarrow \mathbb{R}$ is the internal liquid pressure

and $\sigma := \sigma_t : M \longrightarrow \mathbb{R}$ is the specific superfluid entropy at time $t \in \mathbb{R}$. The latter is related to the internal MHD superfluid specific energy function $e = e_t(\rho_t, \sigma_t)$ owing to the first thermodynamic law:

$$T_t(\rho_t, \sigma_t) \, \delta\sigma_t = \delta e_t(\rho_t, \sigma_t) - p_t \rho_t^{-2} \delta\rho_t, \tag{439}$$

satisfied for any admissible variations of the phase space parameters $\rho_t \in \mathcal{R}(M)$, $\sigma_t \in \Sigma(M)$, where $T_t = T_t(\rho_t, \sigma_t)$ is the internal absolute temperature in the superfluid for $t \in \mathbb{R}$. The isentropic condition $\delta\sigma_t(x_t) = 0$, where $x_t := \eta_t(X) \in M$ for all $X \in M$ and that related to (438) evolution diffeomorphism $\eta_t \in \mathrm{Diff}(M), t \in \mathbb{R}$, entails the following expression for the specific internal energy

$$e_t(\rho_t, \sigma_t) = w_t^{(0)}(\rho_t, \sigma_t) + c_t(\rho_t, B_t), \tag{440}$$

where $w_t^{(0)} : \mathcal{R}(M) \times \Sigma(M) \to C^\infty(M; \mathbb{R})$ is the corresponding internal potential specific energy density and $c_t : \mathcal{R}(M) \times \mathcal{B}(M) \to C^\infty(M; \mathbb{R})$ is some still unknown function, depending in general on the imposed magnetic field $B_t : M \longrightarrow \mathbb{E}^3, t \in \mathbb{R}$.

Let us now analyze, as before, the mathematical structure of quantities $(\rho_t, \sigma_t, B_t) \in \mathcal{R}(M) \times \Sigma(M) \times \mathcal{B}(M)$ as the *physical observables* subject to their evolution (438) with respect to the group diffeomorphisms $\eta_t \in \mathrm{Diff}(M), t \in \mathbb{R}$, generated by the liquid motion vector field $dx_t/dt = v_t(x_t), x_t := \eta_t(X), t \in \mathbb{R}, X \in M$:

$$\mathcal{L}_{d/dt}(\langle \rho_t v_t | dx_t \rangle d^3 x_t) = [-dp_t^{(0)} + \rho_t^{-1} d|v_t|^2/2 + \langle B_t | \nabla \rangle \langle B_t | dx_t \rangle)] \rho_t d^3 x_t, \\ \mathcal{L}_{d/dt}(\rho_t d^3 x_t) = 0, \quad \mathcal{L}_{d/dt}\sigma_t = 0, \quad \mathcal{L}_{d/dt}(*\langle B_t | dx_t \rangle) = 0, \tag{441}$$

where $\mathcal{L}_{d/dt} : \Lambda(M) \to \Lambda(M)$ denotes the corresponding Lie derivative with respect to the vector field $d/dt := \partial/\partial t + \langle v_t | \nabla \rangle \in \Gamma(M \times \mathbb{R}; T(M)), t \in \mathbb{R}$. The relationships (441) mean that the space of physical observables, being by definition, the adjoint space $\mathcal{G}_{em}^* := \Lambda^1(M) d^3 x \times (\Lambda^3(M) \oplus \Lambda^0(M) \oplus \Lambda^2(M))$ to the extended configuration space is equal to $\mathcal{G}_{em} := diff(M) \times (\Lambda^0(M) \oplus \Lambda^3(M) \oplus \Lambda^1(M)) \simeq T_{Id}(G_{em})$, the tangent space at the identity $Id$ to the extended differential-functional group manifold $G_{em} := \mathrm{Diff}(M) \times \Lambda^0(M) \times \Lambda^3(M) \times \Lambda^1(M) \simeq \mathrm{Diff}(M) \times \mathcal{R}(M) \times \Sigma(M) \times \mathcal{B}(M)$, where we have naturally identified the Abelian group product $\Lambda^0(M) \times \Lambda^3(M) \times \Lambda^1(M)$ with its direct tangent space sum $T(\Lambda^0(M)) \oplus T(\Lambda^3(M)) \oplus T(\Lambda^1(M))$.

Consider now the constructed differential-functional current group manifold $G_{em}$ in Eulerian variables, on which one naturally acts the $\mathrm{Diff}(M)$-group $\mathrm{Diff}(M) \times G_{em} \to G_{em}$ the standard way:

$$(\eta \circ \varphi)(X) := \varphi(\eta(X)), (\eta \circ r)(X) := r(\eta(X)), \\ \eta \circ (s(X) d^3 X) := \eta^*(s(X) d^3 X), \\ \eta \circ \langle b(X) | dX \rangle := \eta^* \langle b(X) | d^3 X \rangle \tag{442}$$

for $\eta \in \mathrm{Diff}(M), X \in M$ and any $(\varphi; r, s, b) \in \mathrm{Diff}(M) \times \mathcal{R}(M) \times \Sigma(M) \times \mathcal{B}(M)$. Then, taking into account the suitably extended action (442) on the differential-functional manifold $G_{em}$,, one can formulate the following easily checkable further proposition that is crucial for what will follow.

**Proposition 17.** *The differential-functional current group manifold $G_{em} := \mathrm{Diff}(M) \times \mathcal{R}(M) \times \Sigma(M) \times \mathcal{B}(M)$ in Eulerian coordinates is a smooth symmetry Banach group $G_{em} := \mathrm{Diff}(M) \ltimes (\mathcal{R}(M) \times \Sigma(M) \times \mathcal{B}(M))$, equal to the semidirect product of the diffeomorphism group $\mathrm{Diff}(M)$ and the direct product $\mathcal{R}(M) \times \Sigma(M) \times \mathcal{B}(M)$ of abelian functional $\mathcal{R}(M) \simeq \Lambda^0(M)$, density $\Sigma(M) \simeq \Lambda^3(M)$ and one-form $\mathcal{B}(M) \simeq \Lambda^1(M)$ groups, endowed with the following group multiplication law in Eulerian variables:*

$$(\varphi_1; r_1, s_1 d^3 x, \langle b_1 | dx \rangle) \circ (\varphi_2; r_2, s_2 d^3 x, \langle b_2 | dx \rangle) = \\ = (\varphi_2 \cdot \varphi_1; r_1 + r_2 \cdot \varphi_1, s_1 d^3 x + (s_2 d^3 x) \cdot \varphi_1, \langle b_1 | dx \rangle + \langle b_2 | dx \rangle \circ \varphi_1) \tag{443}$$

*for arbitrary elements* $\varphi_1, \varphi_2 \in \text{Diff}(M), r_1, r_2 \in \Lambda^0(M)$, $s_1 d^3 x, s_2 d^3 x \in \Lambda^3(M)$ *and* $\langle b_1 | dx \rangle$, $\langle b_2 | dx \rangle \in \Lambda^1(M)$.

Thus, one can proceed to studying the corresponding coadjoint action of the Lie algebra $\mathcal{G}_{em} \simeq T_{Id}(G_{em})$, $Id \in G_{em}$, on the adjoint space $\mathcal{G}_{em}^*$. As the Lagrangian configuration $\eta_0 \in \text{Diff}(M)$ and the entropy $\sigma_0 \in \Sigma(M)$ are assumed to be invariant under the Banach diffeomorphism group action $\text{Diff}(M)$, the resulting group action can be reduced to the factor-group $Diff_0(M) := \text{Diff}(M)/Diff_{\eta_0,\sigma_0}(M)$ action on the semidirect group product $G_{em,0} := Diff_0(M) \ltimes (\mathcal{R}(M) \times \Sigma(M) \times \mathcal{B}(M))$. Based on the multiplication law (443), one easily calculates the following Lie algebra commutation relationships:

$$\begin{aligned}
[(a_1; r_1, s_1, b_1), (a_2; r_2, s_2, b_2)] = ([a_1, a_2]; \langle a_1 | \nabla r_2 \rangle - \\
- \langle a_2 | \nabla r \rangle, \langle \nabla | a_1 b_2 \rangle - \langle \nabla | a_2 s_1 \rangle, \langle a_1 | \nabla \rangle b_2 - \\
- \langle a_2 | \nabla \rangle b_1 + \langle b_2 | \circ \nabla a_1 \rangle - \langle b_1 | \circ \nabla a_2 \rangle)
\end{aligned} \tag{444}$$

for any elements $a_1, a_2 \in diff(M) \simeq T(M), r_1, r_2 \in \mathcal{R}(M) \simeq \Lambda^0(M), s_1, s_2 \in \Sigma(M) \simeq \Lambda^3(M)$ and $b_1, b_2 \in \mathcal{B}(M) \simeq \Lambda^1(M)$.

The adjoint space to the semidirect product Lie algebra $\mathcal{G}_{em,0} = diff(M) \ltimes (\mathcal{R}(M) \oplus \Sigma(M) \oplus \mathcal{B}(M))$ can be, naturally, written symbolically as the space $\mathcal{G}_{em,0}^* = (\Lambda^1(M) \otimes \Lambda^3(M)) \times (*\Lambda^0(M) \oplus *\Lambda^3(M) \oplus *\Lambda^1(M)) = diff^*(M) \times (\Lambda^3(M) \oplus \Lambda^0(M) \oplus \Lambda^2(M))$, whereas before, the mapping $* : \Lambda(M) \to \Lambda(M)$ denotes the Hodge isomorphism. Then, taking into account the adjoint space $\mathcal{G}_{em,0}^*$ to the current Lie algebra $\mathcal{G}_{em,0}$ is endowed with the following [27,28,41,177,194,200] canonical Lie–Poisson bracket

$$\{f, g\} := \int_M \langle m | \langle \frac{\delta f}{\delta m} | \nabla \rangle \frac{\delta g}{\delta m} - \langle \frac{\delta g}{\delta m} | \nabla \rangle \frac{\delta f}{\delta m} \rangle d^3 x +$$

$$+ \int_M \rho \left( \langle \frac{\delta f}{\delta m} | \nabla \frac{\delta g}{\delta \rho} \rangle - \langle \frac{\delta g}{\delta m} | \nabla \frac{\delta f}{\delta \rho} \rangle \right) d^3 x + \int_M \sigma \langle \nabla | (\frac{\delta f}{\delta m} \frac{\delta g}{\delta \sigma} - \frac{\delta g}{\delta m} \frac{\delta f}{\delta \sigma}) \rangle d^3 x + \tag{445}$$

$$+ \int_M \left( \langle B | \langle \frac{\delta f}{\delta m} | \nabla \rangle \frac{\delta g}{\delta B} - \langle \frac{\delta g}{\delta m} | \nabla \rangle \frac{\delta f}{\delta B} \rangle + \langle \frac{\delta f}{\delta B} | \langle B | \nabla \rangle \frac{\delta g}{\delta m} \rangle - \langle \frac{\delta g}{\delta B} | \langle B | \nabla \rangle \frac{\delta f}{\delta m} \rangle \right) d^3 x$$

for any smooth functionals $f, g \in \mathcal{D}(\mathcal{G}_{em,0}^*)$ on the adjoint space $\mathcal{G}^*$, where, as before, we denoted by $m := \rho v \in T^*(M)$ the specific momentum of the superfluid. The bracket (445) naturally ensues from the canonical symplectic structure on the cotangent phase space $T^*(G_{em,0})$, as it was previously demonstrated in the section above.

We now write down the first two equations of the Euler MHD system (438) as the local fluid mass and momentum conservation laws in the integral Ampere–Newton form

$$\frac{d}{dt} \int_{D_t} \rho_t d^3 x_t = 0, \quad \frac{d}{dt} \int_{D_t} \rho_t v_t \, d^3 x_t +$$

$$+ \int_{\partial D_t} p_t^{(0)}(x_t) d^2 S_t - \int_{D_t} \langle B_t(x_t) | \nabla \rangle B_t(x_t) d^3 x_t = 0, \tag{446}$$

which is completely equivalent to the relationships (441) and where $p_t^{(0)} : M \to \mathbb{R}_+$ is the net internal superfluid pressure, $(\nabla \times B_t(x_t)) \times B_t(x_t) : M \to C^\infty(M; \mathbb{E}^3)$ is the spatially distributed Lorentz force on the superfluid, $d^2 S_t$ is the respectively oriented surface measure on the boundary $\partial D_t$ for the domain $D_t := \eta_t(D) \subset M, t \in \mathbb{R}$, and a smooth submanifold $D \subset M$ is chosen arbitrarily. Taking into account that $(\nabla \times B_t(x_t)) \times B_t(x_t) = \langle B_t | \nabla \rangle B_t - \nabla \langle B_t | B_t \rangle / 2$ for any $B_t \in \mathcal{B}(M)$, the second integral relationship (446) becomes equivalent to the following:

$$\partial v_t / \partial t + \langle v_t | \nabla \rangle v_t + \rho_t^{-1} \nabla p_t^{(0)}(\rho_t, \sigma_t) - \rho_t^{-1} \langle B_t | \nabla \rangle B_t = 0, \tag{447}$$

where we have represented the internal superfluid pressure quantity as

$$p_t(x_t) := p_t^{(0)}(\rho_t, \sigma_t) - \langle B_t | B_t \rangle / 2 \tag{448}$$

for some mapping $p_t^{(0)} : \mathcal{R}(M) \times \Sigma(M) \to C^\infty(M; \mathbb{R})$, strictly depending only on the internal liquid configuration $\eta_t \in \text{Diff}(M)$ for all $t \in \mathbb{R}$.

Based on the Poisson bracket expression (445), one can now easily determine the Hamiltonian function $H : M \to \mathbb{R}$, corresponding to the Euler evolution equation (438) on the adjoint space $\mathcal{G}^*$:

$$H = \int_M \rho_t(|m_t|^2/(2\rho_t^2) + w_t^{(0)}(\rho_t, \sigma_t) + \\ + |B_t|^2/(2\rho_t))dx_t^3 := \int_M \rho(x_t)e_t(\rho_t, \sigma_t)d^3x_t, \tag{449}$$

where the quantity

$$e_t(\rho_t, \sigma_t) = |m_t|^2/(2\rho_t^2) + w_t^{(0)}(\rho_t, \sigma_t) + \tag{450}$$
$$+ |B_t|^2/(2\rho_t) := |m_t|^2/(2\rho_t^2) + w_t(\rho_t, \sigma_t)$$

denotes the specific internal superfluid energy, modified by means of the "frozen" magnetic field $B_t \in \mathcal{B}(M), t \in \mathbb{R}$, replacing the previously defined internal specified potential energy $w_t^{(0)}(\rho_t, \sigma_t)$ by the shifted specified potential energy quantity $w_t(\rho_t, \sigma_t) := w_t^{(0)}(\rho_t, \sigma_t) + |B_t|^2/(2\rho_t)$. In particular, the Equation (447) reduces to the equivalent Hamilton expression

$$\partial m_t / \partial t = \{H, m_t\} \tag{451}$$

for $m_t \in T^*(M) \simeq diff^*(M)$ and all $t \in \mathbb{R}$. It is also seen that if $B_t \to 0$ uniformly with respect to time $t \in \mathbb{R}$, the internal energy expression (450) brings about that (423). Recall now that the following quasi-stationary second thermodynamic energy conservation law

$$\delta e_t(\rho_t, \sigma_t) = \rho_t^{-2} p_t(x_t) \delta\rho_t + T_t(x_t)\delta\sigma_t \tag{452}$$

holds for all admitted superfluid variations $\delta\rho_t \in \mathcal{R}(M)$ and $\delta\sigma_t \in \Sigma(M), t \in \mathbb{R}$. As, by isentropic assumption, $\delta\sigma_t = 0$ for all $t \in \mathbb{R}$ along fluid streamlines, for the internal pressure one easily obtains the expression $p_t(x_t) = \rho_t^2 \partial w_t^{(0)}(\rho_t, \sigma_t)/\partial\rho_t - \langle B_t|B_t\rangle/2$, exactly coinciding with that of (448).

The Hamiltonian function (449) evidently satisfies the conservation condition $dH/dt = 0$ for all $t \in \mathbb{R}$. To check this directly, it is enough to observe [194] that the following differential relationship

$$\partial e_t(\rho_t, \sigma_t)/\partial t + \langle \nabla | \rho_t v_t \left[ e_t(\rho_t, \sigma_t) + \rho_t \partial w_0(\rho_t, \sigma_t)/\partial\rho_t - |B_t|^2/2 \right] \rangle = 0 \tag{453}$$

holds for all $t \in \mathbb{R}$ and whose integration over the domain $M \subset \mathbb{R}^3$ easily gives rise to the conservation of the Hamiltonian function (449).

### 10.7. A Modified Current Lie Algebra, Magneto-Hydrodynamic Invariants and Their Geometry

The importance of spatial invariants describing the stability [194] of MHD superfluid motion was previously stated long ago [181,193,194,201]. Based on the modern symplectic theory of differential–geometric structures on manifolds, we devise a unified approach to study MHD invariants of compressible superfluid flow, related with specially constructed symmetry structures and commuting to each other vector fields on the phase space.

We start from a useful differential-geometric observation that the magneto-hydrodynamic Euler equations $\Gamma(M; T(M))$ action on the adjoint space to the Lie algebra $\mathcal{G}$ of the modified Banach current group $G = \text{Diff}(M) \ltimes (\Lambda^0(M) \oplus \Lambda^3(M) \oplus *^1(M))$, generated by the following vector field differential relationship:

$$dx_t/dt = v_t(x_t), \tag{454}$$

where $x_t = \eta_t(X) \in M$, $X \in M$, and $v_t : M \to T(M)$, $t \in \mathbb{R}$, is an acceptable time-dependent vector field on the domain $M$, describing the adiabatic superfluid and super-

conductive motion via the diffeomorphism subgroup mappings $\eta_t \in \mathrm{Diff}(M), \eta_t|_{t=0} = \eta_0 \in \mathrm{Diff}_0(M)$. Taking into account that the initial superfluid configuration $\eta_0 \in \mathrm{Diff}(M)$ is fixed, one can define, following reasonings from [81], a new differential relationship

$$dx_\tau / dt = u_t(x_\tau) \tag{455}$$

on the domain $M$ with respect to the evolution variable $\tau \in \mathbb{R}$, parameterized by the time parameter $t \in \mathbb{R}$, where $u_t : M \to T(M)$, is a $\tau$-independent vector field on $M$, generating the diffeomorphism subgroup $\psi_t \in \mathrm{Diff}(M)$, $x_\tau := \psi_t(\eta_0(X)), X \in M$, commuting to that generated by the vector field (454), i.e., $\eta_t \circ \psi_t = \psi_t \circ \eta_t$ for all $t, \tau \in \mathbb{R}$. The action of the diffeomorphism subgroup $\psi_t \in \mathrm{Diff}(M)$ at any fixed time $t \in \mathbb{R}$ can be naturally interpreted as rearranging the particle configurations in the superfluid, not changing its other dynamic characteristics. If to denote the corresponding Lie derivatives with respect to the vector fields (454) and (455) by differential expressions $\mathcal{L}_{d/dt} := \partial/\partial t + \langle v_t | \nabla \rangle \circ : C^\infty(M; \mathbb{R}) \to C^\infty(M; \mathbb{R})$ and $\mathcal{L}_{u_t} := \langle u_t | \nabla \rangle \circ : C^\infty(M; \mathbb{R}) \to C^\infty(M; \mathbb{R})$, the commutation condition $\eta_t \circ \psi_t = \psi_t \circ \eta_t$ for all $t, \tau \in \mathbb{R}$ is equivalently rewritten as the operator commutator

$$[\mathcal{L}_{d/dt}, \mathcal{L}_{u_t}] = 0. \tag{456}$$

Consider now an arbitrary integral invariant of the MHD superfluid, governed by the Euler system (438):

$$I = \int_{D_t} \rho_t(x_t) \gamma_t(m_t; \rho_t, \sigma_t, B_t) d^3 x_t, \tag{457}$$

generated by some specific density functional $\gamma_t : \mathcal{G}^* \to C^\infty(M \times \mathbb{R}; \mathbb{R})$ and held over the domain $D_t = \eta_t(D)$ for any domain $D \subset M$, corresponding to the diffeomorphism subgroup $\eta_t \in \mathrm{Diff}(M), t \in \mathbb{R}$, generated by flow (454). Taking into account that there holds the following density relationship

$$\mathcal{L}_{d/dt}(\rho_t(x_t) d^3 x_t) = 0 \tag{458}$$

for any $t \in \mathbb{R}$, one easily derives from (457) and (458) that also

$$\mathcal{L}_{d/dt} \gamma_t(m_t; \rho_t, \sigma_t, B_t) = 0 \tag{459}$$

for any $t \in \mathbb{R}$. Thus, based on the commutation relationship (456) one can formulate the following important lemma.

**Lemma 1.** *Let vector fields (454) and (455) commute to each other and a density functional $\gamma_0 : \mathcal{G}^* \times \mathbb{R} \to C^\infty(M \times \mathbb{R}; \mathbb{R})$ satisfies for all $t \in \mathbb{R}$ the condition*

$$\mathcal{L}_{d/dt} \gamma_0(m_t; \rho_t, \sigma_t, B_t) = 0, \tag{460}$$

*then the following expressions*

$$I_{n,k} = \int_{D_t} \rho_t (\mathcal{L}_{u_t}^n \gamma_0(m_t; \rho_t, \sigma_t, B_t)^k d^3 x_t \tag{461}$$

*over the domain $D_t = \eta_t(D)$, generated by corresponding to the flow (454) diffeomorphism subgroup $\eta_t \in \mathrm{Diff}(M), t \in \mathbb{R}$, and arbitrary domain $D \subset M$, are for all integers $n \in \mathbb{Z}_+, k \in \mathbb{Z}$, the MHD invariants of the superfluid flow (438).*

**Proof.** A proof easily follows from the commutation condition (456) and the superfluid density relationship (458). $\square$

As examples, let us take following [81,194], the vector field $u_t := \rho_t^{-1} B_t \in \Gamma(T(M))$, commuting to the vector field $v_t \in \Gamma(T(M))$, and $\gamma_0 = i_{u_t}\langle A_t | dx_t \rangle = \langle A_t | \rho_t^{-1} B_t \rangle \in C^\infty(M \times \mathbb{R}; \mathbb{R})$, where the magnetic vector potential $A_t \in C^\infty(M; \mathbb{R}), t \in \mathbb{R}$, satisfies the

classical Maxwell relationships: the magnetic field $B_t = \nabla \times A_t$ and the electric field $E_t = -\partial A_t/\partial t = -v_t \times B_t$, owing to the net electric field superconductivity (437) condition $\tilde{E}_t = E_t + v_t \times B_t = 0$. Really, the commutativity condition (456) means that

$$\mathcal{L}_{d/dt}(\rho_t^{-1}B_t) - \langle \rho_t^{-1}B_t|\nabla > v_t = 0, \tag{462}$$

which is satisfied, owing to the second and fourth equations of the Euler MHD system (438), as well as to the invariance

$$\mathcal{L}_{d/dt}\gamma_0 = \mathcal{L}_{d/dt}i_{u_t}\langle A_t\,|dx_t\rangle = [\mathcal{L}_{d/dt}, i_{u_t}]\langle A_t\,|dx_t\rangle +$$
$$+i_{u_t}\mathcal{L}_{d/dt}\langle A_t\,|dx_t\rangle = i_{[d/dt,u_t]}\langle A_t\,|dx_t\rangle + i_{u_t}\mathcal{L}_{d/dt}\langle A_t\,|dx_t\rangle = 0, \tag{463}$$

which holds owing to the algebraic relationship

$$[\mathcal{L}_{d/dt}, i_{u_t}] = i_{[\partial/\partial t + v_t\nabla_t, u_t]}, \tag{464}$$

commutativity of vector fields $u_t$ and $v_t \in \Gamma(M)$ and the integral relationship

$$\begin{aligned}
\tfrac{d}{dt}\int_{\partial S_t}\langle A_t\,|dx_t\rangle &= \int_{\partial S_t}\mathcal{L}_{d/dt}\langle A_t\,|dx_t\rangle = \\
&= \int_{\partial S_t}[\langle \mathcal{L}_{d/dt}A_t\,|dx_t\rangle + \langle A_t\,|\mathcal{L}_{d/dt}dx_t\rangle] = \\
&= \int_{\partial S_t}[\langle \mathcal{L}_{d/dt}A_t\,|dx_t\rangle + \langle A_t\,|dv_t\rangle] = \\
&= \int_{\partial S_t}[\langle v_t \times B + \langle v_t|\nabla\rangle A_t\,|dx_t\rangle + \langle A_t\,|dv_t\rangle] = \\
&= \int_{\partial S_t}[\langle v_t \times (\nabla \times A) + \langle v_t|\nabla\rangle A_t\,|dx_t\rangle + \langle A_t\,|dv_t\rangle] = \\
&= \int_{\partial S_t}[\langle dA_t|v_t\rangle + \langle A_t\,|dv_t\rangle] = \int_{\partial S_t}[d\langle A_t\,|v_t\rangle] = 0,
\end{aligned} \tag{465}$$

equivalent to the condition $\mathcal{L}_{d/dt}\langle A_t\,|dx_t\rangle = 0$ for all $t \in \mathbb{R}$. The same statement we obtain from the slightly simpler reasoning:

$$\begin{aligned}
\tfrac{d}{dt}\int_{\partial S_t}\langle A_t\,|dx_t\rangle &= \tfrac{d}{dt}\int_{S_t}\langle \nabla \times A_t|dS_t^2\rangle = \\
&= \tfrac{d}{dt}\int_{S_t}\langle B_t|dS_t^2\rangle := -\int_{\partial S_t}\langle \tilde{E}_t|dx_t\rangle = 0,
\end{aligned} \tag{466}$$

following from the net electric field $\tilde{E}_t = 0$ superconductivity condition (437) along the boundary $\partial S_t$, where $S_t := \eta_t(S_0) \subset M$ is the surface, generated by the diffeomorphism subgroup $\eta_t \in \mathrm{Diff}(M), t \in \mathbb{R}$, and an arbitrarily chosen surface $S_0 = S_t|_{t=0} \subset M$. The latter is, evidently, equivalent to the equality $\mathcal{L}_{d/dt}\langle A_t\,|dx_t\rangle = 0$ modulo the gauge transformation $A_t \to A_t + \nabla\xi_t$, where $\mathcal{L}_{d/dt}\xi_t + \langle A_t|v_t\rangle = 0$ for some function $\xi_t \in C^\infty(M;\mathbb{R})$ and all $t \in \mathbb{R}$. Thus, one can formulate [81,194] the following proposition.

**Proposition 18.** *The functionals*

$$I_{n,k}^{(B)} = \int_{D_t} \rho_t \left( \mathcal{L}_{\rho_t^{-1}B_t}^n \langle A|\rho_t^{-1}B_t\rangle \right)^k d^3x_t \tag{467}$$

*over the domain $D_t = \eta_t(D)$, generated by corresponding to the flow (454) diffeomorphism subgroup $\eta_t \in \mathrm{Diff}(M), t \in \mathbb{R}$, and arbitrary domain $D \subset M$, are for all integers $n \in \mathbb{Z}_+, k \in \mathbb{Z}$, the MHD invariants of the superfluid flow (438). In particular, the following relationships $\{H, I_{n,k}^{(B)}\} = 0$ hold for all $n \in \mathbb{Z}_+$.*

It is natural here to mention [194,196] that the specific entropy functional $\gamma_0 = \sigma_t : M \to C^\infty(M \times \mathbb{R};\mathbb{R})$ satisfies the sufficient condition $\mathcal{L}_{d/dt}\sigma_t = 0, t \in \mathbb{R}$, a priori generates for the superfluid flow (438) the infinite hierarchy

$$I_{n,k}^{(\sigma)} = \int_{D_t} \rho_t \left( \mathcal{L}_{\rho_t^{-1}B_t}^n \sigma_t(x_t) \right)^k d^3x_t, \tag{468}$$

$n \in \mathbb{Z}_+, k \in \mathbb{Z}$, of the MHD invariants over the domain $D_t = \eta_t(D)$, generated by the corresponding to the flow (454) diffeomorphism subgroup $\eta_t \in \text{Diff}(M), t \in \mathbb{R}$, and arbitrary domain $D \subset M$.

To construct other MHD invariants, depending on the superfluid velocity $v_t \in \Gamma(T(M))$, $t \in \mathbb{R}$, let us consider, following [81], two differential one-forms $\langle \alpha_t | dx_t \rangle, \langle \beta_t | dx_t \rangle \in \Lambda^1(M)$, $x_t := \eta_t(X)$, $X \in M$, satisfying for all $t \in \mathbb{R}$ the following identity:

$$\mathcal{L}_{d/dt}\langle \alpha_t | dx_t \rangle = dh_t + \mathcal{L}_{u_t}\langle \beta_t | dx_t \rangle, \tag{469}$$

for some function $h_t \in \Lambda^0(M)$, where the vector field

$$dx_t/d\tau = u_t(x_t) \tag{470}$$

is uniform with respect to the evolution parameter $\tau \in \mathbb{R}$ and satisfies the following constraints:

$$[\mathcal{L}_{d/dt}, \mathcal{L}_{u_t}] = 0, \quad \langle \nabla | \rho_t u_t \rangle = 0 \tag{471}$$

and $u_t \parallel \partial M$ at almost all points $x_t \in \partial M$ for all evolution parameters $t, \tau \in \mathbb{R}$. Then one can formulate the following general proposition.

**Proposition 19.** *The following integral expressions*

$$I_0^{(\alpha,\beta)} = \int_M \rho_t \langle \alpha_t | u_t \rangle d^3 x_t, I_1^{(\alpha,\beta)} = \int_M \rho_t [\langle \alpha_t | v_t \rangle + h_t] d^3 x_t,$$

$$I_2^{(\alpha,\beta)} = \int_M \rho_t \langle \mathcal{L}_{d/dt}\alpha_t | u_t \rangle d^3 x_t \tag{472}$$

*over the whole domain $M \subset \mathbb{R}^3$ are for all integers $n \in \mathbb{Z}_+, k \in \mathbb{Z}$, the global MHD invariants.*

**Proof.** Consider, for example, a proof that $I_0^{(\alpha,\beta)} : \mathcal{G} \to \mathbb{R}$ is an invariant: taking into account that $\mathcal{L}_{d/dt}(\rho_t d^3 x_t) = 0$, one obtains the expression:

$$\begin{aligned} dI_0^{(\alpha,\beta)}/dt &= \int_M \rho_t \mathcal{L}_{d/dt}\langle \alpha_t | u_t \rangle d^3 x_t = \\ &= \int_M \rho_t i_{u_t}(dh_t + \mathcal{L}_{u_t}\langle \beta_t | dx_t \rangle) d^3 x_t = \\ &= \int_M \rho_t (i_{u_t} dh_t + i_{u_t}(i_{u_t}d + di_{u_t})\langle \beta_t | dx_t \rangle) d^3 x_t = \\ &= \int_M \rho_t i_{u_t} d(h_t + \langle \beta_t | u_t \rangle) d^3 x_t = \\ &= \int_M \langle \nabla | \tilde{h}_t \rho_t u_t \rangle d^3 x_t = \int_{\partial M} \langle \tilde{h}_t \rho_t u_t | dS_t^2 \rangle = 0 \end{aligned} \tag{473}$$

for all $t \in \mathbb{R}$, where we put, by definition, $\tilde{h}_t := (h_t + \langle \beta_t | u_t \rangle)$, denoted $dS_t^2$ the surface measure on the boundary $\partial M$, used the Cartan representation $\mathcal{L}_{u_t} = i_{u_t}d + di_{u_t}$ and the natural boundary tangency condition $\rho_t u_t \parallel \partial M$, thus proving the proposition. Exactly similar calculations ensue for the next two invariant $I_k^{(\alpha,\beta)} : \mathcal{G} \to \mathbb{R}, k = \overline{1,2}$, on which we will not stop here. $\square$

As a simple example, one can put $\alpha_t^{(0)} := \flat(v_t) \simeq v_t, \beta_t := B_t$, the vector field $u_t = \rho_t^{-1} B_t : M \to T(M), t \in \mathbb{R}$, and show by easy calculations, using the variational equality (439) that

$$\mathcal{L}_{d/dt}\langle v_t | dx_t \rangle = d(|v_t|^2/2 - h_t - |B_t|^2/\rho_t) + \mathcal{L}_{u_t}\langle B_t | dx_t \rangle + T_t d\sigma_t, \tag{474}$$

where, we have denoted the specific enthalpy [55,198,199] function $h_t := e_t + p_t \rho_t^{-1}$. As a consequence of equality (474), under the spatial temperature constancy $\nabla T_t = 0$ condition for all $t \in \mathbb{R}$, one obtains the following MHD superfluid invariant:

$$I_0^{(v,B)} := \int_M \langle v_t | B_t \rangle d^3 x_t = \int_M \langle m_t | \rho_t^{-1} B_t \rangle, \tag{475}$$

where $m_t \simeq \langle m_t(x_t)|dx_t \rangle \otimes d^3 x_t \in diff^*(M)$ and $\rho_t^{-1} B_t \simeq \langle \rho_t^{-1}(x) B_t | \partial/\partial x \rangle \in T(M)$, coinciding with the MHD invariant, presented before in [81,194]. If the above temperature condition does not hold, the equality (474) reduces to the differential relationship

$$\partial \langle v_t | B_t \rangle / \partial t + \langle \nabla | [v_t \langle v_t | B_t \rangle + B_t(h_t - |v_t|^2/2)] + \rho_t T_t \langle \rho_t^{-1} B_t | \nabla \sigma_t \rangle, \tag{476}$$

satisfied for all $x_t \in M$ and $t \in \mathbb{R}$.

*Remark.* It is worth remarking here that the following baroclinic relationship

$$\nabla \rho_t^{-1} \times \nabla p_t = -\nabla T_t \times \nabla \sigma_t \tag{477}$$

holds for all $x_t \in M$ and $t \in \mathbb{R}$.

Similarly, we also easily obtain the following invariant

$$I_1^{(v,B)} = \int_M \rho_t [|m_t|^2 / \left( 2\rho_t^2 \right) + w_t^{(0)}(\rho_t, \sigma_t) + |B_t|^2 / (2\rho_t)] d^3 x_t = H, \tag{478}$$

coinciding exactly with the Hamiltonian function for the flow (438) on the phase space $\mathcal{G}^*$. The third invariant is, eventually, closely related to the vorticity vector $\xi_t := \nabla \times v_t : M \to \mathbb{E}^3, t \in \mathbb{R}$, and needs a more detailed analysis.

It is instructive now to analyze the existence of integral invariants for the pure hydrodynamic case when the magnetic field $B_t = 0, t \in \mathbb{R}$, following the approach, devised before in [81]. In particular, owing to the relationship (477), there holds the following integral expression for the vorticity $\xi_t := \nabla \times v_t, t \in \mathbb{R}$ :

$$\mathcal{L}_{d/dt} \xi_t - \langle \xi_t | \nabla \rangle v_t = \nabla T_t \times \nabla \sigma_t \tag{479}$$

and define the vector field

$$u_t := \rho_t^{-1} \xi_t \exp f_t(x_t) \tag{480}$$

for some scalar smooth mapping $f_t : M \to \mathbb{R}$, which we will choose from the assumed commutation condition

$$[\mathcal{L}_{d/dt}, \mathcal{L}_{u_t}] = 0. \tag{481}$$

The latter gives rise to the equality $\quad \xi_t \mathcal{L}_{d/dt} f_t(x_t) = -\nabla T_t \times \nabla \sigma_t$ at any $x_t := \eta_t(X) \in M$, $X \in M$, or

$$\dot{f}_t (\nabla \times v_t) + \nabla T_t \times \nabla \sigma_t = 0, \tag{482}$$

where we took into account that $\mathcal{L}_{d/dt} f_t(x_t) = df_t(x_t)/dt := \dot{f}_t(x_t), \ x_t \in M$, with respect to the temporal parameter $t \in \mathbb{R}$. From (482), one obtains that the mapping $f_t : M \to \mathbb{R}$ should satisfy the following constraints:

$$\nabla \dot{f}_t = k_t v_t, \quad \dot{f}_t v_t = \rho_t^{-1} \nabla p(t) + \nabla \omega_t \tag{483}$$

for some scalar smooth functions $k_t$ and $\omega_t : M \to \mathbb{R}, t \in \mathbb{R}$. It is easy to check that the system (483) is compatible, i.e., the quasi-stationary thermodynamic relationship $p_t^{(0)} = \rho_t^2 \partial w_0(\rho_t, \sigma_t)/\partial \rho_t$ jointly with the Euler Equation (393) make it possible to determine these unknown scalar smooth functions $k_t$ and $\omega_t : M \to \mathbb{R}$ for all $t \in \mathbb{R}$.

Consider now, following [81], a slightly modified expression (474) at the magnetic field $B_t = 0$:

$$\mathcal{L}_{d/dt} \langle v_t \exp f_t | dx_t \rangle = \exp f_t d(\omega_t + |v_t|^2/2) \tag{484}$$

and calculate the related integral expression:

$$\begin{aligned} \tfrac{d}{dt} \int_M \rho_t (i_{u_t} \langle v_t | dx_t \rangle) d^3 x_t &= \int_M \rho_t \mathcal{L}_{d/dt} (i_{u_t} \langle v_t | dx_t \rangle) d^3 x_t = \\ = \int_M \rho_t (i_{u_t} \mathcal{L}_{d/dt} \langle v_t | dx_t \rangle) d^3 x_t &= \int_M \rho_t (i_{u_t} d\tilde{h}) d^3 x_t = \\ = \int_M (i_{\rho_t u_t} d\tilde{h}) d^3 x_t = \int_M \langle \nabla \tilde{h}_t | \rho_t u_t \rangle d^3 x_t &= \int_M \langle \nabla \tilde{h}_t | \xi_t \exp f_t(x_t) \rangle d^3 x_t, \end{aligned} \tag{485}$$

where we put, by definition, the function $\tilde{h}_t := \omega_t + |v_t|^2/2$.

If now to put that the mapping $f_t : M \to \mathbb{R}$ satisfies for all $t \in \mathbb{R}$ the constraint $\langle \nabla f_t | \xi_t \rangle = 0$, the integral expression (485) reduces to

$$
\begin{aligned}
\tfrac{d}{dt} \int_M \rho_t \, (i_{u_t} \langle v_t | dx_t \rangle) d^3 x_t &= \int_M \langle \nabla | (\exp f_t(x_t) \tilde{h}_t \xi_t) \rangle d^3 x_t = \\
&= \int_{\partial M} \langle \exp f_t(x_t) \tilde{h}_t \xi_t | d^2 S_t \rangle = 0,
\end{aligned}
\tag{486}
$$

where the vorticity vector tangency $\xi_t || \partial M$ constraint is assumed. Thus, under conditions assumed above, the following vortex functional

$$
I = \int_M \langle v_t | \nabla \times v_t \rangle d^3 x_t
\tag{487}
$$

persists to be conserved for all $t \in \mathbb{R}$.

If the function $f_t : M \to \mathbb{R}$, being defined by relationships (483), satisfies for all $t \in \mathbb{R}$ the scalar constraint $\langle \nabla f_t | \xi_t \rangle = 0$, one easily derives the following differential relationship:

$$
\begin{aligned}
\mathcal{L}_{d/dt} \langle \nabla f_t | \xi_t \rangle &= k_t \langle v_t | \xi_t \rangle + \langle \nabla | f_t \nabla T_t \times \nabla \sigma_t \rangle = \\
&= < \nabla \dot{f}_t | \xi_t \rangle + \langle \nabla | f_t \nabla T_t \times \nabla \sigma_t \rangle = 0,
\end{aligned}
\tag{488}
$$

or, equivalently, in the integral form

$$
\begin{aligned}
\tfrac{d}{dt} \int_{D_t} \langle \nabla f_t | \xi_t \rangle \rho_t d^3 x_t &= \int_{D_t} \mathcal{L}_{d/dt} \langle \nabla f_t | \xi_t \rangle \rho_t d^3 x_t = \\
&= \int_{D_t} \big[ \langle \nabla \dot{f}_t | \xi_t \rangle + \langle \nabla | f_t \nabla T_t \times \nabla \sigma_t \rangle \big] \rho_t d^3 x_t = \\
&= \int_{D_t} \big[ \langle \nabla \dot{f}_t | \xi_t \rangle - \langle \nabla f_t | \nabla \times \rho_t^{-1} \nabla p_t^{(0)} \rangle \big] \rho_t d^3 x_t \\
&= \int_{D_t} \big[ \langle \nabla \dot{f}_t | \xi_t \rangle \rho_t - \rho_t \langle \nabla \rho_t^{-1} | \nabla \times p_t^{(0)} \nabla f_t \rangle \big] d^3 x_t = \\
&= \int_{D_t} \big[ \langle \nabla \dot{f}_t | \xi_t \rangle \rho_t + \langle \nabla \ln \rho_t | \nabla \times p_t^{(0)} \nabla f_t \rangle \big] d^3 x_t = \\
&= \int_{D_t} \langle \nabla \dot{f}_t | \xi_t \rangle \rho_t d^3 x_t,
\end{aligned}
\tag{489}
$$

where we took into account that for the isentropic fluid flow under regard there holds the tangency $\nabla \rho_t || \partial D_t$ condition for all $t \in \mathbb{R}$. If the right hand side of (489) proves to be zero, i.e., $\langle \nabla \dot{f}_t | \xi_t \rangle = 0, t \in \mathbb{R}$, this will mean that the constraint $\langle \nabla f_t | \xi_t \rangle = 0$ for all $t \in \mathbb{R}$, if $\langle \nabla f_t | \xi_t \rangle |_{t=0} = 0$ at $t = 0$, thus producing the vortex conservation functional (487).

## 11. A Modified Current Lie Algebra Symmetry on Torus, Its Lie-Algebraic Structure and Related Integrable Heavenly Type Dynamical Systems

### 11.1. Introductory Notes

The main object of our study is integrable multidimensional dispersionless differential equations, which possess modified Lax–Sato type representations, related with their hidden Hamiltonian structures. Equations of this type arise and are widely applied in mechanics, general relativity, differential geometry and the theory of integrable systems. Among the most mentioned are the Boyer–Finley equations, heavenly type Plebański equations, which are descriptive of a class of self-dual 4-manifolds, as well as the dispersionless Kadomtsev–Petviashvili (dKP) equation, also known as the Khokhlov–Zabolotskaya equation, which arises in non-linear acoustics and the theory of Einstein–Weyl structures. Their integrability have been investigated by a whole variety of modern techniques including symmetry analysis, differential-geometric and algebrogeometric methods, dispersionless $\bar{\partial}$-dressing, factorization techniques, Virasoro constraints, hydrodynamic reductions, etc. The first important examples of the related Hamiltonian structures were previously demonstrated in [202–206] and later were developed in [207–214], where many examples of dispersionless differential equations were analyzed in detail as flows on orbits of the coadjoint action of loop vector field algebras $\widetilde{\mathrm{diff}}(\mathbb{T}^n)$, generated by specially chosen seed elements $\tilde{l} \in \widetilde{\mathrm{diff}}(\mathbb{T}^n)^*$. In these works, it was observed that many integrable multidimensional dispersionless differential equations are generated by seed elements of a very special structure, namely for them

there exist such analytical functional elements $\tilde{\eta}, \tilde{\rho} \in \Omega^0(\mathbb{T}^n) \otimes \mathbb{C}$ that $\tilde{l} = \tilde{\eta} d\tilde{\rho}$. As the latter naturally generates the symplectic structure $\tilde{\omega}^{(2)} := \int_{\mathbb{T}^n} d\tilde{\eta} \wedge d\tilde{\rho} \in \Omega^2(\mathbb{T}^n) \otimes \mathbb{C}$ on the moduli space [215,216] of flat connections on $\mathbb{T}^n$, related to coadjoint actions of the corresponding Casimir functionals, the geometric nature of many integrable multidimensional dispersionless differential equations can be also studied using cohomological techniques, devised in [215,217] for the case of Riemannian surfaces. It is also worth mentioning that in [207–209] a deep connection of the related Hamiltonian flows on $\widetilde{\text{diff}}(\mathbb{T}^n)^*$ was revealed with the well known in classical mechanics Lagrange–d'Alembert principle.

In this section, developing the approach, devised in [202,203,218], we will describe a Lie algebraic structure and integrability properties of a generalized hierarchy of the Lax-Sato type compatible systems of Hamiltonian flows and related integrable multidimensional dispersionless differential equations. Such systems are called the heavenly type equations and were first introduced by Plebański in [219]. The heavenly type equations were analyzed in many articles (see, e.g., [203,218,220–227]) using several different approaches. In [131,207,209,228] the heavenly type equations were analyzed by using non-associative and non-commutative current algebras on the torus $\mathbb{T}^m, m \in \mathbb{N}$. We also mention that [229,230] B. Szablikowski and A. Sergyeyev developed some generalizations of the classical AKS-algebraic and related $R$-structures [11,17–19]. In [203,218] and recently in [207,231], these ideas were applied to a semi-direct Lie algebra $\tilde{\mathcal{G}} := \widetilde{\text{diff}}(\mathbb{T}^n) \ltimes \widetilde{\text{diff}}(\mathbb{T}^n)^*$ of the loop Lie algebra $\widetilde{\text{diff}}(\mathbb{T}^n) := \widetilde{Vect}(\mathbb{T}^n)$ of vector fields on the torus $\mathbb{T}^n, n \in \mathbb{Z}_+$, and its dual space $\widetilde{\text{diff}}(\mathbb{T}^n)^*$. Several interesting and deep results about the orbits of the corresponding coadjoint actions on the space $\tilde{\mathcal{G}}^* \simeq \tilde{\mathcal{G}}$ and the classical Lie–Poisson type structures on them were presented. It is worth especially remarking here that the AKS-algebraic scheme is naturally embedded into the classical $R$-structure approach via the following construction.

Let a $(\tilde{\mathcal{G}}; [\cdot, \cdot])$ denote a Lie algebra over $\mathbb{C}$ and $\tilde{\mathcal{G}}^*$ be its natural adjoint space. Take some tensor element $r \in \tilde{\mathcal{G}} \otimes \tilde{\mathcal{G}} \simeq \text{Hom}(\tilde{\mathcal{G}}^*; \tilde{\mathcal{G}})$ and consider its splitting into symmetric and antisymmetric parts

$$r = k \oplus \sigma, \tag{490}$$

respectively, and assume that the symmetric tensor $k \in \tilde{\mathcal{G}} \otimes \tilde{\mathcal{G}}$ does not degenerate. That allows the definition on the Lie algebra $\tilde{\mathcal{G}}$ of a symmetric non-degenerate bi-linear product $(\cdot|\cdot) : \tilde{\mathcal{G}} \otimes \tilde{\mathcal{G}} \to \mathbb{C}$ via the expression

$$(a|b) := k^{-1}a(b) \tag{491}$$

for any $a, b \in \tilde{\mathcal{G}}$. The composed mapping $R := \sigma \circ k^{-1} : \tilde{\mathcal{G}} \to \tilde{\mathcal{G}}$, following the scheme

$$\tilde{\mathcal{G}} \xrightarrow{k^{-1}} \tilde{\mathcal{G}}^* \xrightarrow{\sigma} \tilde{\mathcal{G}}, \tag{492}$$

defines the following $R$-structure on the Lie algebra $\tilde{\mathcal{G}}$:

$$[a, b]_R := [Ra, b] + [a, Rb] \tag{493}$$

for all elements $a, b \in \tilde{\mathcal{G}}$. The following theorem, defining the related Poissonian structure [19,121,207,217,232,233] on the adjoint space $\tilde{\mathcal{G}}$ holds.

**Theorem 9.** *Let $\alpha, \beta \in \tilde{\mathcal{G}}^*$ be arbitrary and define the bracket*

$$\{\alpha, \beta\} := ad^*_{r\alpha}\beta - ad^*_{r\beta}\alpha. \tag{494}$$

*Then the bracket (494) is Poissonian if the R-structure on the Lie algebra $\tilde{\mathcal{G}}$ defines the Lie structure on $\tilde{\mathcal{G}}$, that is there holds the Yang–Baxter equation*

$$[Ra, Rb] - R[a, b]_R = -[a, b] \tag{495}$$

*for any $a, b \in \tilde{\mathcal{G}}$.*

**Remark 10.** *The above theorem makes it possible to consider the Hamiltonian flows on the coadjoint space $\tilde{\mathcal{G}}^*$ as those determined on the Lie algebra $\tilde{\mathcal{G}}$. The latter is exceptionally useful if for the scalar product (491) there exists such a trace-type $\mathrm{Tr}(\cdot)$ symmetric and ad-invariant functional (of Killing type) that*

$$\mathrm{Tr}(ab) := (a|b), \quad (a|[b,c]) = (([a,b]|,c) \tag{496}$$

*for any $a, b$ and $c \in \tilde{\mathcal{G}}$. Then any Hamiltonian flow of an element $a \in \tilde{\mathcal{G}}$ is representable in the standard Lax type form*

$$da/dt = [\mathrm{grad}(h), a], \tag{497}$$

*where $\mathrm{grad}(h) \in \tilde{\mathcal{G}}$ is generated by the corresponding smooth Hamiltonian function $h \in \mathrm{D}(\tilde{\mathcal{G}})$.*

Concerning the loop Lie algebra $\tilde{\mathcal{G}} := \widetilde{\mathrm{diff}}(\mathbb{T}^n)$ on the torus $\mathbb{T}^n$, it is well known that such a trace-type functional on $\tilde{\mathcal{G}}$ does not exist, thus we need to study the Hamiltonian flows on the adjoint loop space $\tilde{\mathcal{G}}^* \simeq \Omega^1(\mathbb{T}^n)$ of meromorphic differential forms on the torus $\mathbb{T}^n$ and obtain, as a result, integrable dispersionless differential equations as compatibility conditions for the related loop vector fields, generated by Casimir functionals on $\tilde{\mathcal{G}}^*$. This procedure is much more complicated for analysis than the standard one and employs more geometrical tools and considerations about the orbit space structure of the seed elements $\tilde{l} \in \tilde{\mathcal{G}}^*$, generating a hierarchy of integrable Hamiltonian flows. The latter, in part, is deeply related to its reduction properties, guaranteeing the existence of nontrivial Casimir invariants on its coadjoint orbits. By applying and extending these ideas to central extensions of Lie algebras, we construct new classes of commuting Hamiltonian flows on an extended adjoint space $\bar{\mathcal{G}} := \tilde{\mathcal{G}}^* \oplus \mathbb{C}$. These Hamiltonian flows are generated by seed elements $(\tilde{a} \ltimes \tilde{l}; \alpha) \in \bar{\mathcal{G}}^*$ and specially constructed Casimir invariants on the corresponding orbits of $\tilde{\mathcal{G}}^*$. In most cases, these seed elements appeared to be represented as specially factorized differential objects, whose real geometric nature is still much hidden and not clear. Moreover, we found that the corresponding compatibility condition of constructed Hamiltonian flows coincides exactly with the compatibility condition for a system of related three Lax–Sato type linear vector field equations. As examples, we found and described new multidimensional generalizations of the Mikhalev–Pavlov and Alonso–Shabat type integrable dispersionless equation, whose seed elements possess a special factorized structure, allowing to extend them to the multidimensional case of arbitrary dimension.

### 11.2. Differential-Geometric Setting: The Diffeomorphism Group $\mathrm{Diff}(\mathbb{T}^n)$ and Its Description

Consider the $n$-dimensional torus $\mathbb{T}^n$ and call points $X \in \mathbb{T}^n$ as the Lagrangian variables of a configuration $\eta \in \mathrm{Diff}(\mathbb{T}^n)$. The manifold $\mathbb{T}^n$, thought of as the target space of a configuration $\eta \in \mathrm{Diff}(\mathbb{T}^n)$, is called the spatial or Eulerian configuration, whose points, called spatial or Eulerian points, will be denoted by small letters $x \in \mathbb{T}^n$. Then any one-parametric configuration of $\mathrm{Diff}(\mathbb{T}^n)$ is a time $t \in \mathbb{R}$ dependent family [41,122,181,192,193] of diffeomorphisms written as

$$\mathbb{T}^n \ni x = \eta(X, t) := \eta_t(X) \in \mathbb{T}^n \tag{498}$$

for any initial configuration $X \in \mathbb{T}^n$ and some mappings $\eta_t \in \mathrm{Diff}(\mathbb{T}^n), t \in \mathbb{R}$.

Being interested in studying flows on the space of Lagrangian configurations $\eta \in \mathrm{Diff}(\mathbb{T}^n)$ with respect to the temporal variable $t \in \mathbb{R}$, generated by group diffeomorphisms $\eta_t \in \mathrm{Diff}(\mathbb{T}^n), t \in \mathbb{R}$, let us proceed to describing the structure of tangent $T_{\eta_t}(\mathrm{Diff}(\mathbb{T}^n))$ and cotangent $T_{\eta_t}^*(\mathrm{Diff}(\mathbb{T}^n))$ spaces to the diffeomorphism group $\mathrm{Diff}(\mathbb{T}^n)$ at the points $\eta_t \in \mathrm{Diff}(\mathbb{T}^n)$ for any $t \in \mathbb{R}$. Determine first the tangent space $T_{\eta_t}(\mathrm{Diff}(\mathbb{T}^n))$ to the diffeomorphism group manifold $\mathrm{Diff}(\mathbb{T}^n)$ at point $\eta \in \mathrm{Diff}(\mathbb{T}^n)$ for which we will make use of the construction, devised before in [122,181,194]. Namely, let $\eta \in \mathrm{Diff}(\mathbb{T}^n)$ be a Lagrangian configuration and try to determine the tangent space $T_\eta(\mathrm{Diff}(\mathbb{T}^n))$ at $\eta \in \mathrm{Diff}(\mathbb{T}^n)$ as

the collection of vectors $\xi_\eta := d\eta_\tau/d\tau|_{\tau=0}$, where $\mathbb{R} \ni \sim \to \eta_\tau \in \text{Diff}(\mathbb{T}^n)$, $\eta_\tau|_{\tau=0} = \eta$, is a smooth curve on $\text{Diff}(\mathbb{T}^n)$, and for arbitrary reference point $X \in \mathbb{T}^n$ there holds $\xi_\eta(X) = d\eta_\tau(X)/d\tau|_{\tau=0}$. The latter equivalently means that the vectors $\xi_\eta(X) \in T_{\eta(X)}(\mathbb{T}^n)$, $X \in \mathbb{T}^n$, represent a vector field $\xi : \mathbb{T}^n \to T(\mathbb{T}^n)$ on the manifold $\mathbb{T}^n$ for any $\eta \in \text{Diff}(\mathbb{T}^n)$. Thus, the tangent space $T_\eta(\text{Diff}(\mathbb{T}^n))$ coincides with the set of vector fields on $\mathbb{T}^n$:

$$T_\eta(\text{Diff}(\mathbb{T}^n)) \simeq \{\xi_\eta \in \Gamma(T(\mathbb{T}^n)) : \xi_\eta(X) \in T_{\xi(X)}(\mathbb{T}^n)\} \tag{499}$$

and similarly, the cotangent space $T_\eta^*(\text{Diff}(\mathbb{T}^n))$ consists of all one-form densities on $\mathbb{T}^n$ over $\eta \in \text{Diff}(\mathbb{T}^n)$:

$$T_\eta^*(\text{Diff}(\mathbb{T}^n)) = \{\alpha_\eta \in \Omega^1(\mathbb{T}^n) \otimes \Omega^3(\mathbb{T}^n) : \alpha_\eta(X) \in T_{\eta(X)}^*(\mathbb{T}^n) \otimes |\Omega^3(\mathbb{T}^n)|\} \tag{500}$$

subject to the canonical non-degenerate pairing $(\cdot|\cdot)_c$ on $T_\eta^*(\text{Diff}(\mathbb{T}^n)) \times T_\eta(\text{Diff}(\mathbb{T}^n))$ : if $\alpha_\eta \in T_\eta^*(\text{Diff}(\mathbb{T}^n))$, $\xi_\eta \in T_\eta(\text{Diff}(\mathbb{T}^n))$, where $\alpha_\eta|_X = \langle\alpha_\eta(X)|dx\rangle \otimes d^3X$, $\xi_\eta|_X = \langle\xi_\eta(X)|\partial/\partial x\rangle$, then

$$(\alpha_\eta|\xi_\eta)_c := \int_{\mathbb{T}^n} \langle\alpha_\eta(X)|\xi_\eta(X)\rangle d^3X. \tag{501}$$

The construction above makes it possible to identify the cotangent bundle $T_\eta^*(\text{Diff}(\mathbb{T}^n))$ at the fixed Lagrangian configuration $\eta \in \text{Diff}(\mathbb{T}^n)$ to the tangent space $T_\eta(\text{Diff}(\mathbb{T}^n))$, as the tangent space $T(\mathbb{T}^n)$ is endowed with the natural internal tangent bundle metric $\langle\cdot|\cdot\rangle$ at any point $\eta(X) \in \mathbb{T}^n$, identifying $T(\mathbb{T}^n)$ with $T^*(\mathbb{T}^n)$ via the related metric isomorphism $\sharp : T^*(\mathbb{T}^n) \to T(\mathbb{T}^n)$. The latter can be also naturally lifted to $T_\eta^*(\text{Diff}(\mathbb{T}^n))$ at $\eta \in \text{Diff}(\mathbb{T}^n)$, namely: for any elements $\alpha_\eta, \beta_\eta \in T_\eta^*(\text{Diff}(\mathbb{T}^n))$, $\alpha_\eta|_X = \langle\alpha_\eta(X)|dx\rangle \otimes d^3X$ and $\beta_\eta|_X = \langle\beta_\eta(X)|dx\rangle \otimes d^3X \in T_\eta^*(\text{Diff}(\mathbb{T}^n))$ we can define the metric

$$(\alpha_\eta|\beta_\eta) := \int_{\mathbb{T}^n} \langle\alpha_\eta^\sharp(X)|\beta_\eta^\sharp(X)\rangle d^3X, \tag{502}$$

where, by definition, $\alpha_\eta^\sharp(X) := \sharp\langle\alpha_\eta(X)|dx\rangle)$, $\beta_\eta^\sharp(X) := \sharp\langle\beta_\eta(X)|dx\rangle \in T_{\eta(X)}(\mathbb{T}^n)$ for any $X \in \mathbb{T}^n$. Based on the construction above, one can proceed to constructing smooth invariant functionals on the cotangent bundle $T^*(\text{Diff}(\mathbb{T}^n))$ subject to the corresponding coadjoint actions of the diffeomorphism group $\text{Diff}(\mathbb{T}^n)$. Moreover, as the cotangent bundle $T^*(\text{Diff}(\mathbb{T}^n))$ is *a priori* endowed with the canonical symplectic structure, equivalent [11,18,19,26,28,41,122,181,195] to the corresponding Poisson bracket on the space of smooth functionals on $T^*(\text{Diff}(\mathbb{T}^n))$, one can study both the related Hamiltonian flows on it and their adjoint symmetries and complete integrability.

Consider now the cotangent bundle $T^*(\text{Diff}(\mathbb{T}^n))$ as a smooth manifold endowed with the canonical symplectic structure [26,122] on it, equivalent to the corresponding canonical Poisson bracket on the space of smooth functionals on it. Taking into account that the cotangent space $T_\eta^*(\text{Diff}(\mathbb{T}^n))$ at $\eta \in \text{Diff}(\mathbb{T}^n)$, shifted by the right $R_{\eta^{-1}}$- action to the space $T_{Id}^*(\text{Diff}(\mathbb{T}^n))$, $Id \in \text{Diff}(\mathbb{T}^n)$, becomes diffeomorphic to the adjoint space $\text{diff}^*(\mathbb{T}^n)$ to the Lie algebra $\text{diff}(\mathbb{T}^n) \simeq \Gamma(T(\mathbb{T}^n))$ of vector fields on $\mathbb{T}^n$, as there was stated [31–33,41] still by S. Lie in 1887 this canonical Poisson bracket on $T_\eta^*(\text{Diff}(\mathbb{T}^n))$ transforms [26,31,32,41,181,195] into the classical Lie-Poisson bracket on the adjoint space $\mathcal{G}^*$. Moreover, the orbits of the diffeomorphism group $\text{Diff}(\mathbb{T}^n)$ on $T^*(\text{Diff}(\mathbb{T}^n))$ respectively transform into the coadjoint orbits on the adjoint space $\mathcal{G}^*$, generated by suitable elements of the Lie algebra $\mathcal{G}$. To construct in detail this Lie–Poisson bracket, we formulate the following preliminary simple lemma.

**Lemma 2.** *The Lie algebra* $\text{diff}(\mathbb{T}^n) \simeq \Gamma(T(\mathbb{T}^n))$ *is determined by the following Lie commutator relationships:*

$$[a_1, a_2] = \langle a_1|\nabla\rangle a_2 - \langle a_2|\nabla\rangle a_1 \tag{503}$$

*for any vector fields* $a_1, a_2 \in \Gamma(T(\mathbb{T}^n))$ *on the manifold* $\mathbb{T}^n$.

**Proof.** Proof of the commutation relationships (503) easily follows from the group multiplication

$$(\varphi_{1,t} \circ \varphi_{2,t})(X) = \varphi_{2,t}(\varphi_{1,t}(X)) \tag{504}$$

for any local group diffeomorphisms $\varphi_{1,t}, \varphi_{2,t} \in \text{Diff}(\mathbb{T}^n), t \in \mathbb{R}$, and $X \in \mathbb{T}^n$ under condition that $a_j(X) := d\varphi_{j,t}(X)/dt|_{t=0}$ and $\varphi_{j,t}|_{t=0} = Id \in \text{Diff}(\mathbb{T}^n), j = \overline{1,2}$. $\square$

To calculate the Poisson bracket on the cotangent space $T_\eta^*(\text{Diff}(\mathbb{T}^n))$ at any $\eta \in \text{Diff}(\mathbb{T}^n)$, let us consider the cotangent space $T_\eta^*(\text{Diff}(\mathbb{T}^n)) \simeq \text{diff}^*(\mathbb{T}^n)$, the adjoint space to the tangent space $T_\eta(\text{Diff}(\mathbb{T}^n))$ of left invariant vector fields on $\text{Diff}(\mathbb{T}^n)$ at any $\eta \in \text{Diff}(\mathbb{T}^n)$, and take the canonical symplectic structure on $T_\eta^*(\text{Diff}(\mathbb{T}^n))$ in the form $\omega^{(2)}(\mu, \eta) := \delta\alpha(\mu, \eta)$, where the canonical Liouville form $\alpha(\mu, \eta) := (\mu|\delta\eta)_c \in \Omega^1_{(\mu,\eta)}(T_\eta^*(\text{Diff}(\mathbb{T}^n)))$ at a point $(\mu, \eta) \in T_\eta^*(\text{Diff}(\mathbb{T}^n))$ is defined *a priori* on the tangent space $T_\eta(\text{Diff}(\mathbb{T}^n)) \simeq \Gamma(T(M))$ of right-invariant vector fields on the torus manifold $\mathbb{T}^n$. Having calculated the corresponding Poisson bracket of smooth functions $(\mu|a)_c, (\mu|b)_c \in C^\infty(T_\eta^*(\text{Diff}(\mathbb{T}^n)); \mathbb{R})$ on $T_\eta^*(\text{Diff}(\mathbb{T}^n)) \simeq \text{diff}^*(\mathbb{T}^n), \eta \in \text{Diff}(\mathbb{T}^n)$, one can formulate the following proposition.

**Proposition 20.** *The Lie–Poisson bracket on the coadjoint space $T_\eta^*(\text{Diff}(\mathbb{T}^n)) \simeq \text{diff}^*(\mathbb{T}^n), \eta \in M$, is equal to the expression*

$$\{f,g\}(\mu) = (\mu|[\delta g(\mu)/\delta\mu, \delta f(\mu)/\delta\mu])_c \tag{505}$$

*for any smooth functionals $f, g \in C^\infty(\mathcal{G}^*; \mathbb{R})$.*

**Proof.** By definition [26,122] of the Poisson bracket of smooth functions $(\mu|a)_c, (\mu|b)_c \in C^\infty(T_\eta^*(\text{Diff}(\mathbb{T}^n)); \mathbb{R})$ on the symplectic space $T_\eta^*(\text{Diff}(\mathbb{T}^n))$, it is easy to calculate that

$$\begin{aligned}\{\mu(a), \mu(b)\} &:= \delta\alpha(X_a, X_b) = \\ &= X_a(\alpha|X_b)_c - X_b(\alpha|X_a)_c - (\alpha|[X_a, X_b])_c,\end{aligned} \tag{506}$$

where $X_a := \delta(\mu|a)_c/\delta\mu = a \in \text{diff}(\mathbb{T}^n), X_b := \delta(\mu|b)_c/\delta\mu = b \in \text{diff}(\mathbb{T}^n)$. Since the expressions $X_a(\alpha|X_b)_c = 0$ and $X_b(\alpha|X_a)_c = 0$ owing the right-invariance of the vector fields $X_a, X_b \in T_\eta(\text{Diff}(\mathbb{T}^n))$, the Poisson bracket (506) transforms into

$$\begin{aligned}\{(\mu|a)_c, (\mu|b)_c\} &= -(\alpha|[X_a, X_b])_c = \\ &= (\mu|[b,a])_c = (\mu|[\delta(\mu|b)_c/\delta\mu, \delta(\mu|a)_c/\delta\mu])_c\end{aligned} \tag{507}$$

for all $(\mu, \eta) \in T_\eta^*(\text{Diff}(\mathbb{T}^n)) \simeq \text{diff}^*(\mathbb{T}^n), \eta \in \text{Diff}(\mathbb{T}^n)$ and any $a, b \in \text{diff}(\mathbb{T}^n)$. The Poisson bracket (506) is easily generalized to

$$\{f,g\}(\mu) = (\mu|[\delta g(\mu)/\delta\mu, \delta f(\mu)/\delta\mu])_c \tag{508}$$

for any smooth functionals $f, g \in C^\infty(\mathcal{G}^*; \mathbb{R})$, finishing the proof. $\square$

Based on the Lie–Poisson bracket (505), one can naturally construct Hamiltonian flows on the adjoint space $\text{diff}^*(\mathbb{T}^n)$ via the expressions

$$\partial l/\partial t = -ad^*_{\text{grad } h(l)} l \tag{509}$$

for any element $l \in \text{diff}^*(\mathbb{T}^n), t \in \mathbb{R}$, where, by definition, $\frac{d}{d\varepsilon}h(l + \varepsilon m)|_{\varepsilon=0} := (m|\text{grad } h(l))_c$, for some smooth Hamiltonian function $h \in C^\infty(\text{diff}^*(\mathbb{T}^n); \mathbb{R})$. If the system possesses enough additional invariants except the Hamiltonian function, one can expect its simplification often reducing to its complete integrability. Below, we proceed to developing an effective enough analytical scheme, before suggested in [207,209,234] for suitably constructed holomorphic loop diffeomorphism groups on tori, allowing to

generate infinite hierarchies of such completely integrable Hamiltonian systems on related functional phase spaces.

*11.3. A Modified Current Lie Algebra and Related Symmetry Analysis on Functional  Manifolds*

Consider a smooth manifold $M \subset \mathbb{R}^n, n \in \mathbb{N}$, endowed with the generalized quantum current group [26,181,216] group $G$ as the semidirect product $\mathrm{Diff}(M) \ltimes (\Lambda^0(M) \times \Lambda^1(M))$ of the diffeomorphism group  $\mathrm{Diff}(M)$ with the Abelian groups $\Omega^0(M)$  and  $\Omega^1(M)$, defined by the natural $\mathrm{Diff}(M)$—group action  $\mathrm{Diff}(M) \times G \to G$:

$$(\eta \circ \varphi)(X) := \varphi(\eta(X)), (\eta \circ r)(X) := r(\eta(X)),$$
$$\eta \circ \langle b(X)|dX\rangle := \eta^* \langle b(X)|dX\rangle \tag{510}$$

for $\eta \in \mathrm{Diff}(M), X \in M$, and any $(\varphi; r, b) \in \mathrm{Diff}(M) \times (\Omega^0(M) \times \Omega^0(M)$. The semidirect product group $G$ is endowed with the following internal right group multiplication subject to the Eulerian variable $x := \eta(X) \in M$:

$$(\varphi_1; r_1, \langle b_1|dx\rangle) \circ (\varphi_2; r_2, \langle b_2|dx\rangle) =$$
$$= (\varphi_2 \cdot \varphi_1; r_1 + r_2 \cdot \varphi_1, \langle b_1|dx\rangle + \langle b_2|dx\rangle \circ \varphi_1) \tag{511}$$

at a fixed point $\eta \in \mathrm{Diff}(M)$ and arbitrary elements $\varphi_1, \varphi_2 \in \mathrm{Diff}(M), r_1, r_2 \in \Omega^0(M)$ and $b_1 \simeq \langle b_1|dx\rangle, b_2 \simeq \langle b_2|dx\rangle \in \Omega^1(M)$.

Let $\mathcal{G} \simeq T_{Id}(G) = diff(M) \ltimes (\Omega^0(M) \times \Omega^1(M)), Id \in G$, denote the Lie algebra of the current group $G$, where we took into account that $T(\Omega^0(M)) \simeq \Omega^0(M), T(\Omega^1(M)) \simeq \Omega^1(M)$, and proceed to studying its coadjoint action on the adjoint space $\mathcal{G}^*$. Using (511), one can easily write down that

$$[(a_1; r_1, b_1), (a_2; r_2, b_2)] = (\mathcal{L}_{a_1} a_2; \mathcal{L}_{a_2} r_1 - \mathcal{L}_{a_1} r_2, \mathcal{L}_{a_2}\langle b_1|dx\rangle - \mathcal{L}_{a_1}\langle b_2|dx\rangle), \tag{512}$$

where $\mathcal{L}_a$ denotes the standard [122,123,181] Lie derivative with respect to a vector field $a \in diff(M)$. From (512) one easily ensues the following current Lie algebra $\mathcal{G}$ commutation relationships:

$$[(a_1; r_1, b_1), (a_2; r_2, b_2)] = (\langle\left(\langle a_1|\tfrac{\partial}{\partial x}\rangle a_2 - \langle a_2|\tfrac{\partial}{\partial x}\rangle a_1\right)|\tfrac{\partial}{\partial x}\rangle; \langle a_2|\tfrac{\partial}{\partial x}r_1\rangle - \tag{513}$$

$$- \langle a_1|\tfrac{\partial}{\partial x}r_2\rangle, \langle\langle a_2|\tfrac{\partial}{\partial x}\rangle b_1|dx\rangle - \langle\langle a_1|\tfrac{\partial}{\partial x}\rangle b_2|dx\rangle + \langle b_1|\langle dx|\tfrac{\partial}{\partial x}\rangle a_2\rangle - \langle b_2|\langle dx|\tfrac{\partial}{\partial x}\rangle a_1\rangle),$$

for any elements $a_1, a_2 \in diff(M) \simeq T(M), r_1, r_2 \in \Omega^0(M)$ and $b_1, b_2 \in \Omega^1(M)$, where we have also denoted the gradient vector $\tfrac{\partial}{\partial x} := \left(\tfrac{\partial}{\partial x_1}, \tfrac{\partial}{\partial x_2}, ..., \tfrac{\partial}{\partial x_n}\right)^{\mathsf{T}}$ at $x \in M$. The adjoint space $\mathcal{G}^*$ to the semidirect product Lie algebra $\mathcal{G} = diff(M) \ltimes (\Omega^0(M) \oplus \Omega^1(M))$ can be written symbolically as $\mathcal{G}^* = (\Omega^1(M) \otimes \Omega^n(M)) \times (*\Omega^0(M) \oplus *\Omega^1(M)) = diff^*(M) \times (\Omega^n(M) \oplus \Omega^{n-1}(M))$, where $* : \Omega(M) \to \Omega(M)$ denotes the corresponding Hodge isomorphism with respect to the natural scalar product

$$(\alpha^{(k)}|\gamma^{(s)}) := \delta_{sk} \int_M (\alpha^{(k)} \wedge *\gamma^{(s)}) \tag{514}$$

for any forms $\alpha^{(k)} \in \Omega^k(M)$ and $\gamma^{(s)} \in \Omega^s(M)$, $k, s = \overline{1, n}$. Then, taking into account that the adjoint space $\mathcal{G}^*$ is endowed [27,28,41,177,194,200] with the canonical Lie–Poisson bracket

$$
\begin{aligned}
\{f, h\}(l) := (l | [\nabla f(l), \nabla h(l)]) = &\int_M \left( \langle \mu | \langle \frac{\delta f}{\delta \mu} | \frac{\partial}{\partial x} \rangle \frac{\delta h}{\delta \mu} - \langle \frac{\delta h}{\delta \mu} | \frac{\partial}{\partial x} \rangle \frac{\delta f}{\delta \mu} \rangle \right) d^n x + \\
&+ \int_M \rho \left( \langle \frac{\delta f}{\delta \mu} | \frac{\partial}{\partial x} \frac{\delta h}{\delta \rho} \rangle - \langle \frac{\delta h}{\delta \mu} | \frac{\partial}{\partial x} \frac{\delta f}{\delta \rho} \rangle \right) d^n x + \\
&+ \int_M \left( \langle \beta | \langle \frac{\delta f}{\delta \mu} | \frac{\partial}{\partial x} \rangle \frac{\delta h}{\delta \beta} - \langle \frac{\delta h}{\delta \mu} | \frac{\partial}{\partial x} \rangle \frac{\delta f}{\delta \beta} \rangle + \\
&+ \langle \frac{\delta f}{\delta \beta}, \langle \beta | \frac{\partial}{\partial x} \rangle \frac{\delta h}{\delta \mu} - \langle \frac{\delta h}{\delta \beta}, \langle \beta | \frac{\partial}{\partial x} \rangle \frac{\delta f}{\delta \mu} \rangle \right) d^n x
\end{aligned}
\tag{515}
$$

for any smooth functionals $f, g \in \mathcal{D}(\mathcal{G}^*)$ on the $\mathcal{G}^*$, where we have denoted by $l := (\langle \mu | dx \rangle \otimes d^n x; \ \rho d^n x, * \langle \beta | dx \rangle \otimes d^n x) \in \mathcal{G}^*$ and by $\nabla(\circ)(l) := \left( \langle \frac{\delta(\circ)}{\delta \mu} | \frac{\partial}{\partial x} \rangle; \frac{\delta(\circ)}{\delta \rho}, \langle \frac{\delta(\circ)}{\delta \rho} | dx \rangle \right)$ the corresponding functional gradient.

**Remark 11.** *We remark here that the bracket (515) naturally derives, as it was demonstrated in [29,31,32,41], from the canonical symplectic structure on the cotangent phase space $T^*(G)$.*

Based on the Lie–Poisson bracket, one can construct the Hamiltonian system

$$
\frac{\partial}{\partial t}(\mu, \rho, \beta)^\intercal = \{H, (\mu, \rho, \beta)^\intercal\},
\tag{516}
$$

where $t \in \mathbb{R}$ is the related evolution parameter and $H \in \mathcal{D}(\mathcal{G}^*)$ is some suitably constructed Hamiltonian function. For the evolution flow (516) to be integrable, it should possess [11,122,181,235] enough commuting to each of the other invariant functionals $H_j \in \mathcal{D}(\mathcal{G}^*), j \in \mathbb{N}$, which is in most cases a very complicated problem. Thereby, taking this into account, we will proceed the following way: we will construct a set *a priori* commuting to each of the other invariants $h_j \in \mathcal{D}(\tilde{\mathcal{G}}^*), j \in \mathbb{N}$, defined on the coadjoint space $\tilde{\mathcal{G}}^*$ to a suitably generalized Lie algebra $\tilde{\mathcal{G}}$.

Namely, let us consider a group $\tilde{G} := \tilde{G}_+ \times \tilde{G}_-$, where $\tilde{G}_\pm := \widetilde{\text{Diff}}_\pm(M) \ltimes (\Omega^0_\pm(M) \times \Omega^1_\pm(M))$ are subgroups of the smooth loop mappings $\{\mathbb{C} \supset \mathbb{S}^1 \to G\}$, holomorphically extended, respectively, on the interior $\mathbb{D}^1_+ \subset \mathbb{C}$ and on the exterior $\mathbb{D}^1_- \subset \mathbb{C}$ domains of the unit centrally located disk $\mathbb{D}^1 \subset \mathbb{C}^1$ and such that for any $\tilde{g}(\lambda) \in \tilde{G}_-, \lambda \in \mathbb{D}^1_-, \tilde{g}(\infty) = Id \in G$. The corresponding Lie subalgebras $\tilde{\mathcal{G}}_\pm \simeq \widetilde{\text{diff}}_\pm(M) \ltimes (\Omega^0_\pm(M) \times \Omega^0_\pm(M))$ of the loop current subgroups $\tilde{G}_\pm$ consist, in general, of vector fields on $\mathbb{S}^1 \times \mathbb{T}^n$, holomorphically extended, respectively, on regions $\mathbb{D}^1_\pm \subset \mathbb{C}^1$, where for any $\tilde{p}(\lambda) \in \tilde{\mathcal{G}}_-$ the value $\tilde{p}(\infty) = 0$. The loop current Lie algebra splitting $\tilde{\mathcal{G}} = \tilde{\mathcal{G}}_+ \oplus \tilde{\mathcal{G}}_-$, where

$$
\tilde{\mathcal{G}}_+ = \bigcup_{m \in \mathbb{Z}_+} \left\{ \sum_{j=0}^m \lambda^j \langle a_{-j}(x) | \frac{\partial}{\partial x} \rangle \otimes d^n x; \sum_{j=0}^m \lambda^j \rho_{-j}(x), \sum_{j=0}^m \lambda^j \langle b_{-j}(x) | dx \rangle \right\},
\tag{517}
$$

$$
\tilde{\mathcal{G}}_- = \left\{ \sum_{j \in \mathbb{N}} \lambda^{-j} \langle a_j(x) | \frac{\partial}{\partial x} \rangle \otimes d^n x; \sum_{j \in \mathbb{N}} \lambda^{-j} \rho_j(x), \sum_{j \in \mathbb{N}} \lambda^{-j} \langle b_j(x) | dx \rangle \right\},
$$

can be naturally identified with a dense subspace of the dual space $\tilde{\mathcal{G}}^*$ through the pairing

$$
(\tilde{l} | \tilde{a}) := \underset{\lambda \in \mathbb{C}}{res} (l(x; \lambda) | p(x; \lambda))_{H^0}
\tag{518}
$$

with respect to the scalar product

$$(l(x;\lambda)|p(x;\lambda))_{H^0} := \int_M d^n x[\langle \mu(x;\lambda)|a(x;\lambda)\rangle + \rho(x;\lambda)r(x;\lambda) + \langle \beta(x;\lambda)|b(x;\lambda)\rangle]. \quad (519)$$

on the usual Hilbert space $H^0 := L_2(M; \mathbb{C}^{n+1} \times \mathbb{C}^1 \times \mathbb{C}^{n+1})$ for any elements $\tilde{l} := (\tilde{\mu}; \tilde{\rho}, \tilde{\beta}) \in \tilde{\mathcal{G}}^*$ and $\tilde{p} := (\tilde{a}; \tilde{r}, \tilde{b}) \in \tilde{\mathcal{G}}$, naturally represented in their component wise canonical form as

$$\tilde{p} := (\tilde{a}; \tilde{r}, \tilde{b}) = \left( \left\langle a(x;\lambda)| \frac{\partial}{\partial x} \right\rangle; r(x;\lambda), \langle b(x;\lambda)|dx\rangle \right), \tilde{l} := (\tilde{\mu}; \tilde{\rho}, \tilde{\beta}) = \quad (520)$$

$$= (\langle \mu(x;\lambda)|dx\rangle \otimes d^n x; \rho(x;\lambda)d^3 x, *\langle \beta(x;\lambda)|dx\rangle \otimes d^n x),$$

where for any $x := (x;\lambda) \in \mathbb{C} \times M$ we have denoted, for brevity, the gradient operator $\frac{\partial}{\partial x} := (\frac{\partial}{\partial \lambda}; \frac{\partial}{\partial x}) = \left( \frac{\partial}{\partial \lambda}; \frac{\partial}{\partial x_1}, \frac{\partial}{\partial x_2}, ..., \frac{\partial}{\partial x_n} \right)^\top$ in the Euclidean space $(\mathbb{E}^n; \langle \cdot, \cdot \rangle)$ and $\tilde{a} := \left\langle a(x;\lambda)| \frac{\partial}{\partial x} \right\rangle := a^{(0)}(x;\lambda)\frac{\partial}{\partial \lambda} + \left\langle a(x;\lambda)| \frac{\partial}{\partial x} \right\rangle, \tilde{b} := \langle b(x;\lambda)|dx\rangle := b^{(0)}(x;\lambda)d\lambda + \langle b(x;\lambda)|dx\rangle$, $\tilde{\mu} := \langle \mu(x;\lambda)|dx\rangle := \mu^{(0)}(x;\lambda)d\lambda + \langle \mu(x;\lambda)|dx\rangle$. The corresponding Lie commutator $[\tilde{p}_1, \tilde{p}_2] \in \tilde{\mathcal{G}}$ of any vectors $\tilde{p}_1 = (\tilde{a}_1; \tilde{r}_1, \tilde{b}_1), \tilde{p}_2 = (\tilde{a}_2; \tilde{r}_2, \tilde{b}_2) \in \tilde{\mathcal{G}}$ is calculated the standard way, using (513), and equals

$$[(\tilde{a}_1; \tilde{r}_1, \tilde{b}_1), (\tilde{a}_2; \tilde{r}_2, \tilde{b}_2)] = \left( \left\langle ((\langle a_1|\frac{\partial}{\partial x}\rangle a_2 - \langle a_2|\frac{\partial}{\partial x}\rangle a_1)| \frac{\partial}{\partial x} \right\rangle;$$

$$\langle a_2|\frac{\partial}{\partial x} r_1 \rangle - \langle a_1|\frac{\partial}{\partial x} r_2 \rangle, \langle a_2|\frac{\partial}{\partial x}\rangle\langle b_1|dx\rangle - \quad (521)$$

$$- \langle a_1|\frac{\partial}{\partial x}\rangle\langle b_2|dx\rangle + \langle b_1|\langle dx|\frac{\partial}{\partial x}\rangle a_2\rangle - \langle b_2|\langle dx|\frac{\partial}{\partial x}\rangle a_1\rangle \right).$$

The expression (521) makes it possible to construct the related Lie–Poisson bracket on the adjoint space $\tilde{\mathcal{G}}^*$, modifying that of (515):

$$\{f, h\} := res_\lambda \int_M \langle \mu | \langle \frac{\delta f}{\delta \mu}| \frac{\partial}{\partial x}\rangle \frac{\delta h}{\delta \mu} - \langle \frac{\delta h}{\delta \mu}| \frac{\partial}{\partial x}\rangle \frac{\delta f}{\delta \mu} \rangle d^n x +$$

$$+ res_\lambda \int_M \rho \left( \langle \frac{\delta f}{\delta \mu}| \frac{\partial}{\partial x} \frac{\delta h}{\delta \rho} \rangle - \langle \frac{\delta h}{\delta \mu}| \frac{\partial}{\partial x} \frac{\delta f}{\delta \rho} \rangle \right) d^n x +$$

$$+ res_\lambda \int_M \left( \langle \beta| \langle \frac{\delta f}{\delta \mu}| \frac{\partial}{\partial x}\rangle \frac{\delta h}{\delta \beta} - \left\langle \frac{\delta h}{\delta \mu}| \frac{\partial}{\partial x} \right\rangle \frac{\delta f}{\delta \beta} \rangle + \quad (522)$$

$$+ \frac{\delta f}{\delta \beta}| \langle \beta| \frac{\partial}{\partial x}\rangle \frac{\delta h}{\delta \mu} \rangle - \langle \frac{\delta h}{\delta \beta}| \langle \beta| \frac{\partial}{\partial x}\rangle \frac{\delta f}{\delta \mu} \rangle \right) d^n x$$

for any smooth functionals $f, h \in \mathcal{D}(\tilde{\mathcal{G}}^*)$.

The Lie–Poisson bracket (522) is strongly degenerate and possesses a lot of Casimir invariants $h_j \in \mathcal{D}(\tilde{\mathcal{G}}^*), j \in \mathbb{Z}_+$, satisfying the condition

$$\{f, h_j\} = 0 \quad (523)$$

for all smooth functionals $f \in \mathcal{D}(\tilde{\mathcal{G}}^*)$ and $j \in \mathbb{Z}_+$. As the Lie algebra $\tilde{\mathcal{G}}$ acts on its adjoint space $\tilde{\mathcal{G}}^*$ for any $\tilde{p} = (\tilde{a}; \tilde{r}, \tilde{b}) \in \tilde{\mathcal{G}}$ and $\tilde{l} = (\tilde{\mu}; \tilde{\rho}, \tilde{\beta}) \in \tilde{\mathcal{G}}^*$ as $ad^* : \tilde{\mathcal{G}} \times \tilde{\mathcal{G}}^* \to \tilde{\mathcal{G}}^*$, where

$$ad^*_{\tilde{p}} \tilde{l} = \left( -\langle \frac{\partial}{\partial x} \circ |a\rangle\langle \mu|dx\rangle \otimes d^n x - \langle \mu|\langle dx|\frac{\partial}{\partial x}\rangle a\rangle \otimes d^n x + \quad (524) \right.$$

$$+ \rho\langle dx| \frac{\partial r}{\partial x}\rangle \otimes d^n x + \langle \beta|\langle dx|\frac{\partial}{\partial x} x\rangle \otimes d^n x - \langle \frac{\partial}{\partial x} \circ |\beta\rangle\langle x|dx\rangle \otimes d^n x;$$

$$\langle \frac{\partial}{\partial x}|\rho a\rangle \otimes d^n x, *\langle\langle \frac{\partial}{\partial x} \circ |x\rangle\beta|dx\rangle \otimes d^n x - *\langle\langle \beta|\frac{\partial}{\partial x}\rangle a|dx\rangle \otimes d^n x \bigg),$$

the latter condition (523) is easily rewritten as

$$ad^*_{\nabla h(\tilde{l})}\,\tilde{l} = 0, \tag{525}$$

where $\nabla h(\tilde{l}) := \left( \langle \frac{\delta h}{\delta \mu}|\frac{\partial}{\partial x}\rangle; \frac{\delta h}{\delta \rho}, \langle \frac{\delta h}{\delta \beta}|dx\rangle \right)^{\mathsf{T}} \in \tilde{\mathcal{G}}$, being equivalent, owing to (524), to the following three differential-functional relationships:

$$\langle \frac{\partial}{\partial x} \circ |\frac{\delta h}{\delta \mu}\rangle \mu + \langle \mu| \circ \frac{\partial}{\partial x}\frac{\delta h}{\delta \mu}\rangle - \langle \beta| \circ \frac{\partial}{\partial x}\frac{\delta h}{\delta \beta}\rangle + \langle \frac{\partial}{\partial x} \circ |\beta\rangle \frac{\delta h}{\delta \beta} -$$

$$-\rho \frac{\partial}{\partial x}\frac{\delta h}{\delta \rho} = 0, \quad \langle \frac{\partial}{\partial x}|\rho \frac{\delta h}{\delta \mu}\rangle = 0, \quad \langle \frac{\partial}{\partial x} \circ |\frac{\delta h}{\delta \mu}\rangle \beta - \langle \beta| \frac{\partial}{\partial x}\rangle \frac{\delta h}{\delta \mu} = 0 \tag{526}$$

for any $(\tilde{\mu}; \tilde{\rho}, \beta) \in \tilde{\mathcal{G}}^*$. Recall now that the constructed above loop Lie algebra $\tilde{\mathcal{G}} = \tilde{\mathcal{G}}_+ \oplus \tilde{\mathcal{G}}_-$, as the direct sum of its subalgebras, possesses the additional Lie commutator

$$[\tilde{p}_1, \tilde{p}_2]_R := [R\tilde{p}_1, \tilde{p}_2] + [\tilde{p}_1, R\tilde{p}_2] = [\tilde{p}_{1,+}, \tilde{p}_{2,+}] - [\tilde{p}_{1,-}, \tilde{p}_{2,-}] \tag{527}$$

for any $\tilde{p}_1, \tilde{p}_2 \in \tilde{\mathcal{G}}$, where, by definition, the linear homomorphism $R := (P_+ - P_-)/2$, projectors $P_\pm : \tilde{\mathcal{G}} \to \tilde{\mathcal{G}}_\pm$, and $\tilde{p}_{j,\pm} := P_\pm \tilde{p}_j \in \tilde{\mathcal{G}}_\pm, j = \overline{1,2}$. Based on the second Lie commutator (527) we can construct, in the same way as above, the second Lie–Poisson bracket on the adjoint space $\tilde{\mathcal{G}}^*$ as

$$\{f, h\}_R := res_\lambda \int_M \langle \mu| \langle R\frac{\delta f}{\delta \mu}|\frac{\partial}{\partial x}\rangle \frac{\delta h}{\delta \mu} - \langle R\frac{\delta h}{\delta \mu}|\frac{\partial}{\partial x}\rangle \frac{\delta f}{\delta \mu}\rangle d^n x +$$

$$+ res_\lambda \int_M \langle \mu| \langle \frac{\delta f}{\delta \mu}|\frac{\partial}{\partial x}\rangle R\frac{\delta h}{\delta \mu} - \langle \frac{\delta h}{\delta \mu}|\frac{\partial}{\partial x}\rangle R\frac{\delta f}{\delta \mu}\rangle d^n x +$$

$$+ res_\lambda \int_M \rho \left( \langle R\frac{\delta f}{\delta \mu}|\frac{\partial}{\partial x}\frac{\delta h}{\delta \rho}\rangle - \langle R\frac{\delta h}{\delta \mu}|\frac{\partial}{\partial x}\frac{\delta f}{\delta \rho}\rangle \right) d^n x +$$

$$+ res_\lambda \int_M \rho \left( \langle \frac{\delta f}{\delta \mu}|\frac{\partial}{\partial x}R\frac{\delta h}{\delta \rho}\rangle - \langle \frac{\delta h}{\delta \mu}|\frac{\partial}{\partial x}R\frac{\delta f}{\delta \rho}\rangle \right) d^n x +$$

$$+ res_\lambda \int_M \left( \langle \beta| \langle R\frac{\delta f}{\delta \mu}|\frac{\partial}{\partial x}\rangle \frac{\delta h}{\delta \beta} - \left\langle R\frac{\delta h}{\delta \mu}|\frac{\partial}{\partial x}\right\rangle \frac{\delta f}{\delta \beta}\rangle + \right.$$

$$+ \langle \beta| \langle \frac{\delta f}{\delta \mu}|\frac{\partial}{\partial x}\rangle R\frac{\delta h}{\delta \beta} - \left\langle \frac{\delta h}{\delta \mu}|\frac{\partial}{\partial x}\right\rangle R\frac{\delta f}{\delta \beta}\rangle +$$

$$+ \langle \frac{\delta f}{\delta \beta}| \langle \beta|\frac{\partial}{\partial x}\rangle R\frac{\delta h}{\delta \mu}\rangle - \langle \frac{\delta h}{\delta \beta}| \langle \beta|\frac{\partial}{\partial x}\rangle R\frac{\delta f}{\delta \mu}\rangle +$$

$$\left. + \langle R\frac{\delta f}{\delta \beta}| \langle \beta|\frac{\partial}{\partial x}\rangle \frac{\delta h}{\delta \mu}\rangle - \langle R\frac{\delta h}{\delta \beta}| \langle \beta|\frac{\partial}{\partial x}\rangle \frac{\delta f}{\delta \mu}\rangle \right) d^n x \tag{528}$$

### 11.4. A New Modified Spatially Four-Dimensional Mikhalev–Pavlov Heavenly Type Integrable System

Let a seed element $\tilde{a} \ltimes \tilde{l} \in \tilde{\mathcal{G}}^*$ be chosen as

$$\tilde{a} \ltimes \tilde{l} = ((u_x + v_x \lambda - \lambda^2)\partial/\partial x \ltimes (w_x + \zeta_x \lambda)dx, \tag{529}$$

where $u, v, w, \zeta \in C^2(\mathbb{R}^2 \times (\mathbb{S}^1 \times \mathbb{T}^1); \mathbb{R})$. The asymptotic splits for the components of the gradient of the corresponding Casimir functional $h \in I(\tilde{\mathcal{G}}^*)$, as $|\lambda| \to \infty$ have the following forms:

$$\nabla h_{\tilde{l}} \sim 1 - v_x \lambda^{-1} - u_x \lambda^{-2} - v_z \lambda^{-3} - (u_z + v_x v_z - 2(\partial_x^{-1} v_{xx} v_z))\lambda^{-4} +$$

$$+ v_y \lambda^{-5} - (-u_y - v_x v_y + 2(\partial_x^{-1} v_{xx} v_y))\lambda^{-6} + \dots,$$

$$\nabla h_{\tilde{a}} \sim \zeta_x \lambda^{-1} + w_x \lambda^{-2} + \zeta_z \lambda^{-3} + (w_z - \zeta_x v_z + 2v_x \zeta_z - (\partial_x^{-1} v_x \zeta_x)_z)\lambda^{-4} -$$

$$- \zeta_y \lambda^{-5} + (-w_y + \zeta_x v_y - 2v_x \zeta_y + (\partial_x^{-1} v_x \zeta_x)_y)\lambda^{-6} + \dots.$$

In the case when

$$\nabla h^{(y)}_{\tilde{l},+} := \lambda^4 - v_x\lambda^3 - u_x\lambda^2 - v_z\lambda - (u_z + v_xv_z - 2(\partial_x^{-1}v_{xx}v_z)),$$

$$\nabla h^{(y)}_{\tilde{a},+} := \zeta_x\lambda^3 + w_x\lambda^2 + \zeta_z\lambda + (w_z - \zeta_xv_z + 2v_x\zeta_z - (\partial_x^{-1}v_x\zeta_x)_z),$$

and

$$\nabla h^{(t)}_{\tilde{l},+} = \lambda^6 - v_x\lambda^5 - u_x\lambda^4 - v_z\lambda^3 - (u_z + v_xv_z - 2(\partial_x^{-1}v_{xx}v_z))\lambda^2 + \qquad (530)$$
$$+ v_y\lambda - (-u_y - v_xv_y + 2(\partial_x^{-1}v_{xx}v_y)),$$

$$\nabla h^{(t)}_{\tilde{a},+} = \zeta_x\lambda^5 + w_x\lambda^4 + \zeta_z\lambda^3 + (w_z - \zeta_xv_z + 2v_x\zeta_z - (\partial_x^{-1}v_x\zeta_x)_z)\lambda^2 -$$
$$- \zeta_y\lambda + (-w_y + \zeta_xv_y - 2v_x\zeta_y + (\partial_x^{-1}v_x\zeta_x)_y),$$

the compatibility condition of the Hamiltonian vector flows leads to the system of new integrable evolution equations:

$$u_{zt} + u_{yy} = -u_yu_{xz} + u_zu_{xy} - v_yv_{xy} + v_zv_{xt} - u_zv_yv_{xx} + u_yv_zv_{xx} - \qquad (531)$$
$$- v_x^2v_zv_{xy} + v_x^2v_yv_{xz} - 2e_xu_{xy} - 2s_xu_{xz} + 2e_{xt} - 2s_{xy} + 2e_xv_yv_{xx} + 2s_xv_zv_{xx},$$
$$v_{zt} + v_{yy} = -u_yv_{xz} + u_zv_{xy} - v_yu_{xz} + v_zu_{xy} - 2e_xv_{xy} - 2s_xv_{xz} - 2v_xv_yv_{xz} + 2v_xv_zv_{xy},$$
$$- u_{xy} - u_{zz} = u_xu_{xz} - u_zu_{xx} - u_{xx}v_xv_z + u_xv_{xz}v_x - u_xv_{xx}v_z + (v_xv_z)_z + 2u_{xx}e_x - 2e_{xz},$$
$$- v_{xy} - v_{zz} = u_{xz}v_x - u_zv_{xx} - u_{xx}v_z + u_xv_{xz} - 2v_{xx}v_xv_z + v_x^2v_{xz} + 2v_{xx}e_x,$$
$$- u_{xt} + u_{yz} = -u_xu_{xy} + u_yu_{xx} + u_{xx}v_xv_y - u_xv_{xy}v_x + u_xv_{xx}v_y - (v_xv_y)_z + 2u_{xx}s_x - 2s_{xz},$$
$$- v_{xt} + v_{yz} = -u_{xy}v_x + u_yv_{xx} + u_{xx}v_y - u_xv_{xy} + 2v_{xx}v_xv_y - v_x^2v_{xy} + 2v_{xx}s_x,$$

where

$$e_{xx} = v_{xx}v_z, \quad s_{xx} = -v_{xx}v_y.$$

Under the constraint $v = 0$, one obtains a new spatially four-dimensional system

$$u_{zt} + u_{yy} = -u_yu_{xz} + u_zu_{xy}, \qquad (532)$$
$$- u_{xy} - u_{zz} = u_xu_{xz} - u_zu_{xx},$$
$$- u_{xt} + u_{yz} = -u_xu_{xy} + u_yu_{xx},$$

which reduces to the Mikhalev–Pavlov [204,208,223] integrable heavenly type equation, if to put $z = x \in \mathbb{R}$.

Here, we can observe that the seed element (529) can be presented in the following special compact form:

$$\tilde{a} \ltimes \tilde{l} := \frac{d\tilde{\eta}}{dx}\partial/\partial x \ltimes d\tilde{\rho}, \; \tilde{\eta} = u + v\lambda - \lambda^2 x, \; \tilde{\rho} = w + \zeta\lambda, \qquad (533)$$

deeply connected with the geometry of the related moduli space of flat connections, related to the coadjoint actions of the corresponding Casimir functionals. Its possible generalization to multidimensional Mikhalev–Pavlov type equations can be done by the seed element

$$\tilde{a} \ltimes \tilde{l} := \langle \nabla\tilde{\eta}|\nabla\rangle \ltimes d\tilde{\rho} \qquad (534)$$

for some elements $\tilde{\eta}, \tilde{\rho} \in \Omega^0(\mathbb{T}^n) \otimes \mathbb{C}, n \in \mathbb{N}$. An analysis of the case (534) and corresponding systems of multidimensional Mikhalev–Pavlov type equations is planned to be done in a separate study.

*11.5. A Modified Martinez Alonso-Shabat Heavenly Type Integrable System*

　　If the seed element $\tilde{a} \ltimes \tilde{l} \in \tilde{\mathcal{G}}^*$ is chosen as

$$\tilde{a} \ltimes \tilde{l} = (((u_{x_1} + cu_{x_2}) + \lambda)\partial/\partial x_1 + ((v_{x_1} + cv_{x_2}) + c\lambda)\partial/\partial x_2) \ltimes$$
$$\ltimes ((w_{x_1} + cw_{x_2})dx_1 + (\zeta_{x_1} + c\zeta_{x_2})dx_2), \tag{535}$$

where $u, v, w, \zeta \in C^2(\mathbb{R}^2 \times \mathbb{S}^1 \times \mathbb{T}^2; \mathbb{R})$, $c \in \mathbb{R} \setminus \{0\}$, one has the following asymptotic splits for the components of the gradients of the corresponding Casimir functionals $h^{(1)}$, $h^{(2)} \in \mathrm{I}(\tilde{\mathcal{G}}^*)$ as $|\lambda| \to \infty$:

$$\nabla h_{\tilde{l}}^{(1)} \sim \begin{pmatrix} 1 + (u_{x_1} + cu_{x_2})\lambda^{-1} - u_z\lambda^{-2} + \dots \\ c + (v_{x_1} + cv_{x_2})\lambda^{-1} - v_z\lambda^{-2} + \dots \end{pmatrix},$$

$$\nabla h_{\tilde{a}}^{(1)} \sim \begin{pmatrix} (w_{x_1} + cw_{x_2})\lambda^{-1} - w_z\lambda^{-2} + \dots \\ (\zeta_{x_1} + c\zeta_{x_2})\lambda^{-1} - \zeta_z\lambda^{-2} + \dots \end{pmatrix},$$

and

$$\nabla h_{\tilde{l}}^{(2)} \simeq \begin{pmatrix} 1 + (u_{x_1} - cu_{x_2})\lambda^{-1} + \chi\lambda^{-2} + \dots \\ -c + (v_{x_1} - cv_{x_2})\lambda^{-1} + \omega\lambda^{-2} + \dots \end{pmatrix},$$

$$\nabla h_{\tilde{a}}^{(2)} \simeq \begin{pmatrix} (w_{x_1} - cw_{x_2})\lambda^{-1} + \varrho\lambda^{-2} + \dots \\ (\zeta_{x_1} - c\zeta_{x_2})\lambda^{-1} + \chi\lambda^{-2} + \dots \end{pmatrix},$$

where

$$\chi_{x_1} + c\chi_{x_2} = -(u_{zx_1} - cu_{zx_2}) + 2c(u_{x_1}u_{x_1x_2} - u_{x_2}u_{x_1x_1} + v_{x_1}u_{x_2x_2} - v_{x_2}u_{x_1x_2}), \tag{536}$$
$$\omega_{x_1} + c\omega_{x_2} = -(v_{zx_1} - cv_{zx_2}) + 2c(u_{x_1}v_{x_1x_2} - u_{x_2}v_{x_1x_1} + v_{x_1}v_{x_2x_2} - v_{x_2}v_{x_1x_2}),$$

and

$$\rho_{x_1} + c\rho_{x_2} = -(w_{zx_1} - cw_{zx_2}) + 2c(u_{x_1}w_{x_1x_2} - u_{x_2}w_{x_1x_1} + 2w_{x_2}u_{x_1x_1} - $$
$$- 2w_{x_1}u_{x_1x_2} + v_{x_1}w_{x_2x_2} - v_{x_2}w_{x_1x_2} + w_{x_2}v_{x_1x_2} - w_{x_2}v_{x_2x_2} + \zeta_{x_2}v_{x_1x_1} - \zeta_{x_1}v_{x_1x_2}),$$
$$\chi_{x_1} + c\chi_{x_2} = -(\zeta_{zx_1} - c\zeta_{zx_2}) + 2c(v\zeta_{x_2x_2} - v_{x_2}\zeta_{x_1x_2} + 2\zeta_{x_2}v_{x_1x_2} - $$
$$- 2\zeta_{x_1}v_{x_2x_2} + u_{x_1}\zeta_{x_1x_2} - u_{x_2}\zeta_{x_1x_1} + \zeta_{x_2}u_{x_1x_1} - \zeta_{x_1}u_{x_1x_2} + w_{x_2}u_{x_1x_2} - w_{x_1}u_{x_2x_2}).$$

In the case when the reduced Casimir gradients are equal to the expressions

$$\nabla h_{\tilde{l},+}^{(y)} = \begin{pmatrix} \lambda^2 + (u_{x_1} + cu_{x_2})\lambda - u_z \\ c\lambda^2 + (v_{x_1} + cv_{x_2})\lambda - v_z \end{pmatrix}, \nabla h_{\tilde{a},+}^{(y)} = \begin{pmatrix} (w_{x_1} + cw_{x_2})\lambda - w_z \\ (\zeta_{x_1} + c\zeta_{x_2})\lambda - \zeta_z \end{pmatrix},$$

and

$$\nabla h_{\tilde{l},+}^{(t)} = \begin{pmatrix} \lambda^2 + (u_{x_1} - cu_{x_2})\lambda + \chi \\ -c\lambda^2 + (v_{x_1} - cv_{x_2})\lambda + \omega \end{pmatrix}, \nabla h_{\tilde{a},+}^{(t)} = \begin{pmatrix} (w_{x_1} - cw_{x_2})\lambda + \rho \\ (\zeta_{x_1} - c\zeta_{x_2})\lambda + \chi \end{pmatrix},$$

the Lax–Sato compatibility condition of the Hamiltonian vector flows leads to the system of evolution equations:

$$
\begin{aligned}
&u_{zt} + \chi_y = -u_{zx_1}\chi - u_{zx_2}\omega + u_{zx_1} + v_{zx_2}, \\
&v_{zt} + \omega_y = -v_{zx_1}\chi - v_{zx_2}\omega + u_z\omega_{x_1} + v_z\omega_{x_2}, \\
&u_{yx_1} + cu_{yx_2} = -(u_{x_1} + cu_{x_2})u_{zx_1} - (v_{x_1} + cv_{x_2})u_{zx_2} + (u_{x_1x_1} + cu_{x_1x_2})u_z + \\
&\quad + (u_{x_1x_2} + cu_{x_2x_2})v_z - u_{zz}, \\
&v_{yx_1} + cv_{yx_2} = -(u_{x_1} + cu_{x_2})v_{zx_1} - (v_{x_1} + cv_{x_2})v_{zx_2} + (v_{x_1x_1} + cv_{x_1x_2})u_z + \\
&\quad + (v_{x_1x_2} + cv_{x_2x_2})v_z - v_{zz}, \\
&u_{tx_1} + cu_{tx_2} = (u_{x_1} + cu_{x_2})\chi_{x_1} + (v_{x_1} + cv_{x_2})\chi_{x_2} - (u_{x_1x_1} + cu_{x_1x_2})\chi - \\
&\quad - (u_{x_1x_2} + cu_{x_2x_2})\omega + \chi_z, \\
&v_{tx_1} + cv_{tx_2} = (u_{x_1} + cu_{x_2})\omega_{x_1} + (v_{x_1} + cv_{x_2})\omega_{x_2} - (v_{x_1x_1} + cv_{x_1x_2})\chi - \\
&\quad - (v_{x_1x_2} + cv_{x_2x_2})\omega + \omega_z,
\end{aligned}
\tag{537}
$$

generalizing the Martinez Alonso–Shabat heavenly type integrable system. Thus, the following proposition holds.

**Proposition 21.** *The constructed system of heavenly type Equations (536) and (537) has the Lax–Sato vector field representation with the "spectral" parameter $\lambda \in \mathbb{C}$, which is related to the element $\tilde{a} \ltimes \tilde{l} \in \tilde{\mathcal{G}}^*$ in the form (535).*

The system of Equations (536) and (537) admits the reduction when $u = v$. In this case, under $c = 1$ one obtains such a system:

$$
\begin{aligned}
&u_{zt} + \chi_y = -(u_{zx_1} + u_{zx_2})\chi + u_z(\chi_{x_1} + \chi_{x_2}), \\
&\chi_{x_1} + \chi_{x_2} = -(u_{zx_1} - u_{zx_2}) - 2(u_{x_1}u_{x_2})_{x_1} - 2(u_{x_1}u_{x_2})_{x_2}.
\end{aligned}
\tag{538}
$$

The additional constraint $u_z = u_{x_1} + u_{x_2}$ transforms the system (538) into the following interesting integro-differential equation:

$$
\begin{aligned}
(u_{\tilde{t}x_1} + u_{\tilde{t}x_2}) &- (u_{\tilde{y}x_1} - u_{\tilde{y}x_2}) = u_{x_1x_2}(u_{x_1} - u_{x_2}) - u_{x_1x_1}u_{x_2} + u_{x_2x_2}u_{x_1} - \\
&- u_{x_1x_2}(u_{x_1}^2 - u_{x_2}^2) - u_{x_1x_1}u_{x_2}(u_{x_1} + u_{x_2}) + u_{x_2x_2}u_{x_1}(u_{x_1} + u_{x_2}) - \\
&- 2(\mathcal{P}(u_{x_1}u_{x_2})_{\tilde{y}}) + (u_{x_1x_1} + 2u_{x_1x_2} + u_{x_2x_2})(\mathcal{P}u_{x_1}u_{x_2}), \\
&\mathcal{P} = (\partial/\partial x_1 + \partial/\partial x_2)^{-1}(\partial/\partial x_1 - \partial/\partial x_2),
\end{aligned}
$$

where $\tilde{t} = 2t$ and $\tilde{y} = 2y$. Thus, the Equation (538) is integrable and can be considered as some multi-dimensional generalization of the Martinez Alonso–Shabat system [236].

### 11.6. A Modified Current Loop Algebra and Multidimensional Heavenly Type Integrable Equations: The Generalized Lie-Algebraic Structures

A further generalization of the multi-dimensional case related to the loop group $\widetilde{\mathrm{Diff}}(\mathbb{T}^n)$ on the torus $\mathbb{T}^n$, $n \in \mathbb{Z}_+$ can be developed [207–209] by the following approach. Since the Lie algebra $\widetilde{\mathrm{diff}}(\mathbb{T}^n)$ consists of the loop group elements, analytically continued from the circle $\mathbb{S}^1 := \partial\mathbb{D}^1$, being the boundary of the disk $\mathbb{D}^1 \subset \mathbb{C}$, by means of the complex "spectral" variable $\lambda \in \mathbb{C}$ both into the interior $\mathbb{D}^1_+ \subset \mathbb{C}$ and the exterior $\mathbb{D}^1_- \subset \mathbb{C}$ parts of the disk $\mathbb{D}^1 \subset \mathbb{C}$, one can take into account its analytical invariance to the circle diffeomorphism group $\mathrm{Diff}(\mathbb{S}^1)$. The latter gives rise to the naturally extended holomorphic Lie algebra $\widetilde{\mathrm{diff}}(\mathbb{T}^n \times \mathbb{C}) = \widetilde{\mathrm{diff}}(\mathbb{T}^n \times \mathbb{D}^1_+) \oplus \widetilde{\mathrm{diff}}(\mathbb{T}^n \times \mathbb{D}^1_-)$ on the torus $\mathbb{T}^n \times \mathbb{C}$, whose elements are representable as $\bar{a}(x; \lambda) := \left\langle a(x; \lambda), \frac{\partial}{\partial x} \right\rangle = \sum_{j=1}^{n} a_j(x; \lambda)\frac{\partial}{\partial x_j} + a_0(x; \lambda)\frac{\partial}{\partial \lambda}$ for some holomorphic in $\lambda \in \mathbb{D}^1_\pm$ vectors $a(x; \lambda) \in \mathbb{E} \times \mathbb{E}^n$ for all $x \in \mathbb{T}^n$, and where we

denoted by $\frac{\partial}{\partial \mathsf{x}} := (\frac{\partial}{\partial \lambda}, \frac{\partial}{\partial x_1}, \frac{\partial}{\partial x_2}, ..., \frac{\partial}{\partial x_n})^{\mathsf{T}}$ the generalized Euclidean vector gradient with respect to the vector variable $\mathsf{x} := (\lambda, x) \in \mathbb{C} \times \mathbb{T}^n$.

Let us construct a modified current loop Lie algebra $\bar{\mathcal{G}}$ as the semi-direct sum $\bar{\mathcal{G}} :=$ diff$(\mathbb{T}^n \times \mathbb{C}) \ltimes$ diff$(\mathbb{T}^n \times \mathbb{C})^*$ of the Lie algebra diff$(\mathbb{T}^n \times \mathbb{C})$ and its adjoint space diff$(\mathbb{T}^n \times \mathbb{C})^*$, taking into account their natural pairing

$$(\bar{l}|\bar{a}) := \operatorname*{res}_{\lambda \in \mathbb{C}} (l(\mathsf{x})|a(\mathsf{x}))_{H^0} \tag{539}$$

for any $\bar{l} \in$ diff$(\mathbb{T}^n \times \mathbb{C})^*$ and $\bar{a} \in$ diff$(\mathbb{T}^n \times \mathbb{C})$. The corresponding Lie commutator on the loop Lie algebra $\bar{\mathcal{G}}$ is given for any $\bar{a}_1 \ltimes \bar{l}_1, \bar{a}_2 \ltimes \bar{l}_2 \in \bar{\mathcal{G}}$ by

$$[\bar{a}_1 \ltimes \bar{l}_1, \bar{a}_2 \ltimes \bar{l}_2] := [\bar{a}_1, a_2] \ltimes ad_{a_1}^* \bar{l}_2 - ad_{a_2}^* \bar{l}_1. \tag{540}$$

The Lie algebra $\bar{\mathcal{G}}$ also splits into the direct sum of two subalgebras:

$$\bar{\mathcal{G}} = \bar{\mathcal{G}}_+ \oplus \bar{\mathcal{G}}_-, \tag{541}$$

allowing the introduction of the classical $R$-structure:

$$[\bar{a}_1 \ltimes \bar{l}_1, \bar{a}_2 \ltimes \bar{l}_2]_R := [R(\bar{a}_1 \ltimes \bar{l}_1), \bar{a}_2 \ltimes \bar{l}_2] + [\bar{a}_1 \ltimes \bar{l}_1, R(\bar{a}_2 \ltimes \bar{l}_2)] \tag{542}$$

for any $\bar{a}_1 \ltimes \bar{l}_1, \bar{a}_2 \ltimes \bar{l}_2 \in \bar{\mathcal{G}}$, where, by definition,

$$R := (P_+ - P_-)/2, \tag{543}$$

and

$$P_{\pm}\bar{\mathcal{G}} := \bar{\mathcal{G}}_{\pm} \subset \bar{\mathcal{G}}. \tag{544}$$

The space $\bar{\mathcal{G}}^*$ (adjoint to the Lie algebra $\bar{\mathcal{G}}$) can be identified with the space $\bar{\mathcal{G}}$ by using the symmetric and non-degenerate form

$$(\bar{a} \ltimes \bar{l}|\bar{r} \ltimes \bar{m}) := \operatorname*{res}_{\lambda \in \mathbb{C}} (\bar{a} \ltimes \bar{l}|\bar{r} \ltimes \bar{m})_{H^0}, \tag{545}$$

where, by definition,

$$(\bar{a} \ltimes \bar{l}|\bar{r} \ltimes \bar{m})_{H^0} = (\bar{m}|\bar{a})_{H^0} + (\bar{l}|\bar{r})_{H^0} \tag{546}$$

for any pair of elements $\bar{a} \ltimes \bar{l}, \bar{r} \ltimes \bar{m} \in \bar{\mathcal{G}}$.

**Remark 12.** *The above constructed Lie algebra $\bar{\mathcal{G}}$, being metrized by means of the symmetric, nondegenerate bilinear form (545), is owing to the construction described in the introduction, to uniquely represent the coadjoint orbits on $\bar{\mathcal{G}}^* \simeq \bar{\mathcal{G}}$ in the standard Lax type form on $\bar{\mathcal{G}}$, that will be used further.*

Owing to the convolution (546), the Lie algebra $\bar{\mathcal{G}}$ becomes metrized. For arbitrary smooth functions $f, g \in D(\bar{\mathcal{G}}^*)$ one can naturally determine two Lie–Poisson brackets

$$\{f, g\} := (\bar{a} \ltimes \bar{l}|[\nabla f(\bar{l}, \bar{a}), \nabla g(\bar{l}, \bar{a})]) \tag{547}$$

and

$$\{f, g\}_R := (\bar{a} \ltimes \bar{l}|[\nabla f(\bar{l}, \bar{a}), \nabla g(\bar{l}, \bar{a})]_R), \tag{548}$$

where at any seed element $\bar{a} \ltimes \bar{l} \in \bar{\mathcal{G}}^* \simeq \bar{\mathcal{G}}$ the gradient element $\nabla f(\bar{l}, \bar{a}) := \nabla f_{\bar{l}} \ltimes \nabla f_{\bar{a}} \simeq \langle \nabla f(l, a)|(\partial/\partial \mathsf{x}, d\mathsf{x})^{\mathsf{T}} \rangle \in \bar{\mathcal{G}}$ and $\nabla f_{\bar{l}} = \langle \nabla f_l|\partial/\partial \mathsf{x} \rangle$, $\nabla f_{\bar{a}} = \langle \nabla f_a|d\mathsf{x} \rangle$, and, similarly, the gradient element $\nabla g(\bar{l}, \bar{a}) := \nabla g_{\bar{l}} \ltimes \nabla g_{\bar{a}} \simeq \langle \nabla g(l, a)|(\partial/\partial \mathsf{x}, d\mathsf{x})^{\mathsf{T}} \rangle \in \bar{\mathcal{G}}^*$ and $\nabla g_{\bar{l}} = \langle \nabla g_l|\partial/\partial \mathsf{x} \rangle$, $\nabla g_{\bar{a}} = \langle \nabla g_a|d\mathsf{x} \rangle$ are calculated with respect to the metric (546).

Let us now assume that a smooth function $h \in \mathrm{I}(\bar{\mathcal{G}}^*)$ is a Casimir invariant, that is

$$ad^*_{\nabla h(\bar{l},\bar{a})}(\bar{a} \ltimes \bar{l}) = 0 \tag{549}$$

for a chosen seed element $\bar{a} \ltimes \bar{l} \in \bar{\mathcal{G}}^* \simeq \bar{\mathcal{G}}$. Since for an element $\bar{a} \ltimes \bar{l} \in \bar{\mathcal{G}}^* \simeq \bar{\mathcal{G}}$ and an arbitrary $f \in \mathrm{D}(\bar{\mathcal{G}}^*)$ the adjoint mapping is

$$ad^*_{\nabla f(\bar{l},\bar{a})}(\bar{a} \ltimes \bar{l}) = ([\nabla h_{\bar{l}}, \tilde{a}] \ltimes (ad^*_{\nabla h_{\bar{l}}}\tilde{l} + ad^*_{\tilde{a}}\nabla h_{\bar{a}})), \tag{550}$$

the condition (549) can be rewritten as

$$[\nabla h_{\bar{l}}, \tilde{a}] = 0, \quad ad^*_{\nabla h_{\bar{l}}}\tilde{l} + ad^*_{\tilde{a}}\nabla h_{\bar{a}} = 0, \tag{551}$$

and one can easily obtain that the Casimir functional $h \in \mathrm{I}(\bar{\mathcal{G}}^*)$ satisfies the system of determining equations

$$\langle \nabla h_l | \partial/\partial x \rangle a - \langle a | \partial/\partial x \rangle \nabla h_l = 0,$$

$$\langle \partial/\partial x | \circ \nabla h_l \rangle l + \langle l | (\partial/\partial x \nabla h_l) \rangle + \tag{552}$$

$$+ \langle \partial/\partial x | \circ a \rangle \nabla h_a + \langle a | (\partial/\partial x \nabla h_a) \rangle = 0.$$

For the Casimir functional $h \in \mathrm{D}(\bar{\mathcal{G}}^*)$ the Equation (552) should be be solved analytically. In the case when an element $\bar{l} \ltimes \bar{a} \in \bar{\mathcal{G}}^*$ is singular as $|\lambda| \to \infty$, one can consider the general asymptotic expansion

$$\nabla h^{(p)}(l,a) \sim \lambda^p \sum_{j \in \mathbb{Z}_+} (\nabla h^{(p)}_{l,j}; \nabla h^{(p)}_{a,j})\lambda^{-j} \tag{553}$$

for some suitably chosen $p \in \mathbb{Z}_+$, which is substituted into the Equation (552). The latter is then solved recurrently giving rise to a set of gradient expressions for the Casimir functionals $h^{(p)} \in \mathrm{D}(\bar{\mathcal{G}}^*)$ at the specially found integers $p \in \mathbb{Z}_+$.

Assume now that $h^{(y)}, h^{(t)} \in \mathrm{I}(\bar{\mathcal{G}}^*)$ are such Casimir functionals for which the Hamiltonian vector field generators

$$\nabla h^{(y)}(\bar{l},\bar{a})_+ := (\nabla h^{(p_y)}(\bar{l},\bar{a}))_+, \quad \nabla h^{(t)}(\bar{l},\bar{a})_+ := (\nabla h^{(p_t)}(\bar{l},\bar{a}))_+, \tag{554}$$

are, respectively, defined at some specially found integers $p_y, p_t \in \mathbb{Z}_+$. These invariants generate owing to the Lie–Poisson bracket (548) the following commuting to each other Hamiltonian flows:

$$\frac{\partial}{\partial y}(\bar{a} \ltimes \bar{l}) = -ad^*_{\nabla h^{(y)}(\bar{l},\bar{a})_+}(\bar{a} \ltimes \bar{l}), \tag{555}$$

$$\frac{\partial}{\partial t}(\bar{a} \ltimes \bar{l}) = -ad^*_{\nabla h^{(t)}(\bar{l},\bar{a})_+}(\bar{a} \ltimes \bar{l}),$$

on an element $\bar{a} \ltimes \bar{l} \in \bar{\mathcal{G}}^* \simeq \bar{\mathcal{G}}$ with respect to the corresponding evolution parameters $t, y \in \mathbb{R}$. Owing to the construction, the flows (554) can be rewritten equivalently as

$$\partial l/\partial t = -\left\langle \frac{\partial}{\partial x} | \circ \nabla h^{(p_t)}_l \right\rangle l - \left\langle l | (\frac{\partial}{\partial x}\nabla h^{(p_t)}_l) \right\rangle - \left\langle \frac{\partial}{\partial x} | \circ a \right\rangle \nabla h^{(p_t)}_a - \left\langle a | (\frac{\partial}{\partial x}\nabla h^{(p_t)}_a) \right\rangle, \tag{556}$$

$$\partial l/\partial y = -\left\langle \frac{\partial}{\partial x} | \circ \nabla h^{(p_y)}_l \right\rangle l - \left\langle l | (\frac{\partial}{\partial x}\nabla h^{(p_y)}_l) \right\rangle - \left\langle \frac{\partial}{\partial x} | \circ a \right\rangle \nabla h^{(p_y)}_a - \left\langle a | (\frac{\partial}{\partial x}\nabla h^{(p_y)}_a) \right\rangle,$$

$$\partial a/\partial t = -\left\langle \nabla h^{(p_t)}_l | \frac{\partial}{\partial x} \right\rangle a + \left\langle a | \frac{\partial}{\partial x} \right\rangle \nabla h^{(p_t)}_l, \quad \partial a/\partial y = -\left\langle \nabla h^{(p_y)}_l | \frac{\partial}{\partial x} \right\rangle a + \left\langle a | \frac{\partial}{\partial x} \right\rangle \nabla h^{(p_y)}_l,$$

where $y, t \in \mathbb{R}$ are the corresponding evolution parameters. Since the invariants $h^{(y)}, h^{(t)} \in I(\bar{\mathcal{G}}^*)$ are commuting to each other with respect to the Lie–Poisson bracket (548), the flows (556) are commuting too, meaning equivalently that the corresponding Hamiltonian vector field generators

$$\nabla h_+^{(t)} := \left\langle \nabla h_l^{(p_t)}(l)_+ | \frac{\partial}{\partial \mathrm{x}} \right\rangle, \quad \nabla h_+^{(y)} := \left\langle \nabla h_l^{(p_y)}(l)_+ | \frac{\partial}{\partial \mathrm{x}} \right\rangle \tag{557}$$

satisfy the Lax type compatibility condition

$$\frac{\partial}{\partial y} \nabla h_+^{(t)} - \frac{\partial}{\partial t} \nabla h_+^{(y)} = [\nabla h_+^{(t)}, \nabla h_+^{(y)}] \tag{558}$$

for all $y, t \in \mathbb{R}$. On the other hand, the condition (558) is equivalent to the compatibility condition of two linear equations

$$\left( \frac{\partial}{\partial t} + \nabla h_+^{(t)} \right) \psi = 0, \quad \langle a | \frac{\partial}{\partial \mathrm{x}} \rangle \psi = 0, \quad \left( \frac{\partial}{\partial y} + \nabla h_+^{(y)} \right) \psi = 0 \tag{559}$$

for a function $\psi \in C^2(\mathbb{R}^2 \times \mathbb{T}^n \times \mathbb{C}; \mathbb{C})$, all $y, t \in \mathbb{R}$ and any $\mathrm{x} \in \mathbb{T}^n \times \mathbb{C}$. The results obtained above can be formulated as the following proposition.

**Proposition 22.** *Let a seed element $\bar{a} \ltimes \bar{l} \in \bar{\mathcal{G}}^*$ and $h^{(y)}, h^{(t)} \in I(\bar{\mathcal{G}}^*)$ are some Casimir functionals subject to the metric $(\cdot | \cdot)$ on the holomorphic current loop algebra $\bar{\mathcal{G}}$ and the natural coadjoint action on the co-algebra $\bar{\mathcal{G}}^* \simeq \bar{\mathcal{G}}$. Then the following dynamical systems*

$$\frac{\partial}{\partial y}(\bar{a} \ltimes \bar{l}) = -ad^*_{\nabla h^{(y)}(\bar{l}, \bar{a})_+}(\bar{a} \ltimes \bar{l}), \quad \frac{\partial}{\partial t}(\bar{a} \ltimes \bar{l}) = -ad^*_{\nabla h^{(t)}(\bar{l}, \bar{a})_+}(\bar{a} \ltimes \bar{l}) \tag{560}$$

*are commuting to each other Hamiltonian flows for evolution parameters $y, t \in \mathbb{R}$. Moreover, the compatibility condition of these flows is equivalent to the vector field representation*

$$(\partial/\partial t + \nabla h_+^{(t)})\psi = 0, \quad \langle a | \partial/\partial \mathrm{x} \rangle \psi = 0, \quad (\partial/\partial y + \nabla h_+^{(y)})\psi = 0, \tag{561}$$

*where $\psi \in C^2(\mathbb{R}^2 \times \mathbb{T}^n \times \mathbb{C}; \mathbb{C})$ and the vector fields $\nabla h_+^{(t)}, \nabla h_+^{(y)} \in \mathrm{diff}(\mathbb{T}^n \times \mathbb{C})$ are given by the expressions (557).*

**Remark 13.** *As it was mentioned above, the expansion (553) is effective if a chosen seed element $\bar{a} \ltimes \bar{l} \in \bar{\mathcal{G}}^*$ is singular as $|\lambda| \to \infty$. In the case when it is singular as $|\lambda| \to 0$, the expression (553) should be respectively replaced by the expansion*

$$\nabla h^{(p)}(\bar{l}, \bar{a}) \sim \lambda^{-p} \sum_{j \in \mathbb{Z}_+} \nabla h_j^{(p)}(\bar{l}, \bar{a}) \lambda^j \tag{562}$$

*for suitably chosen integers $p \in \mathbb{Z}_+$, and the reduced Casimir function gradients are then given by the Hamiltonian vector field generators*

$$\nabla h^{(y)}(\bar{l}, \bar{a})_- = \lambda(\lambda^{-p_y-1} \nabla h^{(p_y)}(\bar{l}, \bar{a}))_-, \quad \nabla h^{(t)}(\bar{l}, \bar{a})_- = \lambda(\lambda^{-p_t-1} \nabla h^{(p_t)}(\bar{l}, \bar{a}))_- \tag{563}$$

*for suitably chosen positive integers $p_y, p_t \in \mathbb{Z}_+$ and the corresponding Hamiltonian flows are, respectively, written as*

$$\frac{\partial}{\partial t}(\bar{a} \ltimes \bar{l}) = ad^*_{\nabla h^{(t)}(\bar{l}, \bar{a})_-}(\bar{a} \ltimes \bar{l}), \quad \frac{\partial}{\partial y}(\bar{a} \ltimes \bar{l}) = ad^*_{\nabla h^{(y)}(\bar{l}, \bar{a})_-}(\bar{a} \ltimes \bar{l}) \tag{564}$$

*for evolution parameters $y, t \in \mathbb{R}$.*

As it was demonstrated above, the presented construction of Hamiltonian flows on the adjoint space $\bar{\mathcal{G}}^*$ can be generalized proceeding to the point product $\bar{\mathfrak{G}} := \bar{\mathcal{G}}^{\mathbb{S}^1} = \prod_{z \in \mathbb{S}^1} \bar{\mathcal{G}}$ of the holomorphic current Lie algebra $\bar{\mathcal{G}}$, endowed with the central extension, generated by a two-cocycle $\omega_2 : \bar{\mathfrak{G}} \times \bar{\mathfrak{G}} \to \mathbb{C}$, where

$$\omega_2(\bar{a}_1 \ltimes \bar{l}_1, \bar{a}_2 \ltimes \bar{l}_2) := \int_{\mathbb{S}^1} [(\bar{l}_1, \partial \bar{a}_2 / \partial z)_1 - (\bar{l}_2, \partial \bar{a}_1 / \partial z)_1] dz \tag{565}$$

for any pair of elements $\bar{a}_1 \ltimes \bar{l}_1, \bar{a}_2 \ltimes \bar{l}_2 \in \mathfrak{G}$. The resulting $R$-deformed Lie–Poisson bracket for any smooth functionals $h, f \in D(\widehat{\mathfrak{G}}^*)$ on the adjoint space $\widehat{\mathfrak{G}}^*$ to the centrally extended loop Lie algebra $\widehat{\mathfrak{G}} := \bar{\mathfrak{G}} \oplus \mathbb{C}$ becomes equal to

$$\{h, f\}_R := (\bar{a} \ltimes \bar{l} | [\nabla h(\bar{l}, \bar{a}), \nabla f(\bar{l}, \bar{a})]_R) + \tag{566}$$
$$+ \; \omega_2(R \nabla h(\bar{l}, \bar{a}), \nabla f(\bar{l}, \bar{a})) + \omega_2(\nabla h(\bar{l}, \bar{a}), R \nabla f(\bar{l}, \bar{a})).$$

The corresponding Casimir functionals $h^{(p)} \in I(\widehat{\mathfrak{G}}^*)$ for specially chosen $p \in \mathbb{Z}_+$, are defined with respect to the standard Lie–Poisson bracket as

$$\{h^{(p)}, f\} := (\bar{a} \ltimes \bar{l} | [\nabla h^{(p)}(\bar{l}, \bar{a}), \nabla f(\bar{l}, \bar{a})]) + \omega_2(\nabla h^{(p)}(\bar{l}, \bar{a}), \nabla f(\bar{l}, \bar{a})) = 0 \tag{567}$$

for all smooth functionals $f \in D(\widehat{\mathfrak{G}}^*)$. Based on the equality one easily finds that the gradients $\nabla h^{(p)} \in \widehat{\mathfrak{G}}$ of the Casimir functionals $h^{(p)} \in I(\widehat{\mathfrak{G}}^*)$, $p \in \mathbb{Z}_+$, satisfy the following equations:

$$[\nabla h_{\bar{l}}, \bar{a}] - \frac{\partial}{\partial z} \nabla h_{\bar{l}} = 0, \quad ad^*_{\nabla h_{\bar{l}}} \bar{l} + ad^*_{\bar{a}} \nabla h_{\bar{a}} - \frac{\partial}{\partial z} \nabla h_{\bar{a}} = 0 \tag{568}$$

for a chosen element $\bar{a} \ltimes \bar{l} \in \widehat{\mathfrak{G}}^*$. Making use of the suitable Casimir functionals $h^{(y)}, h^{(t)} \in I(\widehat{\mathfrak{G}}^*)$, one can construct, making use of (566), the following commuting Hamiltonian flows on the adjoint space $\widehat{\mathfrak{G}}^*$:

$$\frac{\partial}{\partial y}(\bar{a} \ltimes \bar{l}) = \{h^{(y)}, \bar{a} \ltimes \bar{l}\}_R, \quad \frac{\partial}{\partial t}(\bar{a} \ltimes \bar{l}) = \{h^{(t)}, \bar{a} \ltimes \bar{l}\}_R, \tag{569}$$

which are equivalent to the evolution equations

$$\frac{\partial}{\partial y}\bar{a} = -[\nabla h^{(y)}_{\bar{l},+}, \bar{a}] + \frac{\partial}{\partial z} \nabla h^{(y)}_{\bar{l},+}, \quad \frac{\partial}{\partial t}\bar{a} = -[\nabla h^{(t)}_{\bar{l},+}, \bar{a}] + \frac{\partial}{\partial z} \nabla h^{(t)}_{\bar{l},+} \tag{570}$$

and

$$\frac{\partial}{\partial y}\bar{l} = -ad^*_{\nabla h^{(y)}_{\bar{l},+}} \bar{l} - ad^*_{\bar{a}}(\nabla h^{(y)}_{\bar{a},+}) + \frac{\partial}{\partial z} \nabla h^{(y)}_{\bar{a},+}, \tag{571}$$

$$\frac{\partial}{\partial t}\bar{l} = -ad^*_{\nabla h^{(t)}_{l,+}} \bar{l} - ad^*_{\bar{a}}(\nabla h^{(t)}_{\bar{a},+}) + \frac{\partial}{\partial z} \nabla h^{(t)}_{\bar{a},+}.$$

The results obtained above are summarized as

**Proposition 23.** *The Hamiltonian flows (569) on the adjoint space $\widehat{\mathfrak{G}}^*$ generate the separately commuting evolution flows (570) and (571), giving rise to the following unique Lax type compatibility condition:*

$$[\nabla h^{(y)}_{l,+}, \nabla h^{(t)}_{l,+}] - \frac{\partial}{\partial t} \nabla h^{(y)}_{\bar{l},+} + \frac{\partial}{\partial y} \nabla h^{(t)}_{\bar{l},+} = 0, \tag{572}$$

*being equivalent to some system of nonlinear heavenly type equations in partial derivatives. More-over, the system of evolution flows (570) and (571) can be considered as the compatibility condition for the following set of linear vector equations*

$$\frac{\partial \psi}{\partial y} + \nabla h_{\tilde{l},+}^{(y)} \psi = 0, \quad \frac{\partial \psi}{\partial z} + \langle a | \partial / \partial x \rangle \psi = 0, \quad \frac{\partial \psi}{\partial t} + \nabla h_{\tilde{l},+}^{(t)} \psi = 0 \tag{573}$$

*for all $(y, t, z; x) \in (\mathbb{R}^2 \times \mathbb{S}^1) \times \mathbb{T}^n \times \mathbb{C}$ and a function $\psi \in C^2((\mathbb{R}^2 \times \mathbb{S}^1) \times \mathbb{T}^n \times \mathbb{C}; \mathbb{C})$.*

**Remark 14.** *The Lie-algebraic scheme of constructing heavenly type integrable equations on respectively chosen smooth functional manifolds, applied above for the modified current loop Lie algebra $\bar{\mathfrak{G}} := \widetilde{\mathrm{diff}}(\mathbb{T}^n \times \mathbb{C}) \ltimes \widetilde{\mathrm{diff}}(\mathbb{T}^n \times \mathbb{C})^*$ as the semi-direct sum of the Lie algebra $\widetilde{\mathrm{diff}}(\mathbb{T}^n \times \mathbb{C})$ and its dual space $\widetilde{\mathrm{diff}}(\mathbb{T}^n \times \mathbb{C})^*$, can be naturally reformulated within a respectively generalized Lagrange–d'Alembert mechanical principle, as was done in the work [214], and which will be analyzed in a separate work under preparation.*

*11.7. A New Modified Spatially Four-Dimensional Mikhalev-Pavlov type Heavenly Equation*

Let a seed element $\tilde{a} \ltimes \tilde{l} \in \widehat{\mathfrak{G}}^*$ be chosen as

$$\tilde{a} \ltimes \tilde{l} = ((u_x - \lambda)\partial / \partial x + v_x \partial / \partial \lambda) \ltimes (w_x dx + \eta_x d\lambda), \tag{574}$$

where $u, v, w, \eta \in C^2(\mathbb{R}^2 \times (\mathbb{S}^1 \times \mathbb{C}); \mathbb{R})$. The asymptotic expressions for the components of the gradients (562) of the corresponding Casimir functionals $h^{(p)} \in I(\widehat{\mathfrak{G}}^*)$, $p \in \mathbb{Z}_+$, as $|\lambda| \to \infty$ have the following forms:

$$\nabla h_{\tilde{l}} \sim \lambda^p \begin{pmatrix} 1 - u_x \lambda^{-1} + (-u_z + (p-1)v)\lambda^{-2} + (u_y + (p-2)(u_x v + \chi_x))\lambda^{-3} + \dots \\ -v_x \lambda^{-1} - v_z \lambda^{-2} + (v_y - (p-2)v_x v)\lambda^{-3} + \dots \end{pmatrix},$$

$$\nabla h_{\tilde{a}} \sim \lambda^p \begin{pmatrix} w_x \lambda^{-1} + w_z \lambda^{-2} + (-w_y + (p-2)(wv)_x)\lambda^{-3} + \dots \\ \eta_x \lambda^{-1} + (\eta_z + (p-1)w)\lambda^{-2} + (-\eta_y + (p-2)\omega_x)\lambda^{-3} + \dots \end{pmatrix},$$

$p \in \mathbb{Z}_+$, where

$$\chi_{xx} = v_z + u_x v_x, \quad \omega_{xx} = w_z - u_x w_x - v_x \eta_x + v \eta_x.$$

In the case when

$$\nabla h_{\tilde{l},+}^{(y)} := \begin{pmatrix} \lambda^2 - u_x \lambda + (-u_z + v) \\ -v_x \lambda - v_z \end{pmatrix},$$

$$\nabla h_{\tilde{a},+}^{(y)} := \begin{pmatrix} w_x \lambda + w_z \\ \eta_x \lambda + (\eta_z + w) \end{pmatrix},$$

and

$$\nabla h_{\tilde{l},+}^{(t)} = \begin{pmatrix} \lambda^3 - u_x \lambda^2 + (-u_z + 2v)\lambda + (u_y + u_x v + \chi_x) \\ -v_x \lambda^2 - v_z \lambda + (v_y - v_x v) \end{pmatrix},$$

$$\nabla h_{\tilde{a},+}^{(t)} = \begin{pmatrix} w_x \lambda^2 + w_z \lambda + (-w_y + (wv)_x) \\ \eta_x \lambda^2 + (\eta_z + 2w)\lambda + (-\eta_y + \omega_x) \end{pmatrix},$$

the compatibility condition of the Hamiltonian vector flows (569) leads to the system of evolution equations:

$$u_{zt} + u_{yy} = -u_y u_{zx} + u_z u_{xy} - u_{xy} v - u_{zz} v - \chi_x u_{xz},$$  (575)

$$v_{zt} + v_{yy} = v v_x^2 - v_z^2 - v v_{xy} - v v_{zz} - u_y v_{xz} + u_z v_{xy} - u_z v_x^2 - \chi_x v_{xz},$$

$$- u_{xy} - u_{zz} = u_x u_{xz} - u_z u_{xx} + u_{xx} v,$$

$$- v_{xy} - v_{zz} = v_x^2 + v_{xx} v + u_x v_{xz} - u_z v_{xx},$$

$$- u_{xt} + u_{yz} = -u_x u_{xy} + u_y u_{xx} + u_{xz} v + u_{xx} \chi_x,$$

$$- v_{xt} + v_{yz} = -u_x v_{xy} + u_y v_{xx} + u_x v_x^2 + v_{xx} v + 2 v_x v_z.$$

Under the constraint $v = 0$ one obtains the modified Michalev–Pavlov type integrable system (532).

Here, we can also observe that the seed element (574) can also be presented in the compact form:

$$\tilde{a} \ltimes \tilde{l} := \left( \frac{\partial \tilde{\eta}_1}{\partial x} \frac{\partial}{\partial x} + \frac{\partial \tilde{\eta}_0}{\partial \lambda} \frac{\partial}{\partial \lambda} \right) \ltimes d\tilde{\rho},$$  (576)

$$\tilde{\eta}_0 = \lambda v_x, \tilde{\eta}_1 = u - \lambda x, \quad \tilde{\rho} = w + \eta_x \lambda,$$

being closely connected with the geometry of the related moduli space of flat connections, related to the coadjoint actions of the corresponding Casimir functionals. Its suitable generalization to multidimensional Mikhalev–Pavlov type equations can be chosen as

$$\tilde{a} \ltimes \tilde{l} := \left( \langle \nabla_x \tilde{\eta} | \nabla_x \rangle + \nabla_\lambda \tilde{\eta}_0 \nabla_\lambda \right) \ltimes d\tilde{\rho}$$  (577)

for some elements $\tilde{\eta}, \tilde{\eta}_0, \tilde{\rho} \in \Omega^0(\mathbb{T}^n) \otimes \mathbb{C}, n \in \mathbb{N}$. The analysis of corresponding systems of integrable multidimensional Mikhalev–Pavlov type equations is planned to be presented in a separate study.

## 12. Conclusions

A wide variety of multidimensional completely integrable evolution flows on smooth functional manifolds have been constructed. Our approach was based on a generalized Lie-algebraic Adler–Kostant–Symes scheme, applied to the modified holomorphic current loop algebra $\mathfrak{G} := \widetilde{\mathrm{diff}}(\mathbb{T}^n \times \mathbb{C}) \ltimes \widetilde{\mathrm{diff}}(\mathbb{T}^n \times \mathbb{C})^*$, the semi-direct sum of the loop Lie algebra $\widetilde{\mathrm{diff}}(\mathbb{T}^n \times \mathbb{C}) := \widetilde{Vect}(\mathbb{T}^n \times \mathbb{C})$ of vector fields on the $\mathbb{T}^n \times \mathbb{C}, n \in \mathbb{Z}_+$, and its adjoint space $\widetilde{\mathrm{diff}}(\mathbb{T}^n \times \mathbb{C})^*$. Its relation to the classical $R$-structure on the loop Lie algebra $\widetilde{Vect}(\mathbb{T}^n \times \mathbb{C})$ is also discussed. The structure of the corresponding seed elements is analyzed, its multidimensional generalizations are presented. We also demonstrated that the obtained Hamiltonian flows are equivalent to the compatibility conditions for the suitably related Lax–Sato type linear vector field equations. We also mentioned a very interesting Lagrange–d'Alembert type mechanical interpretation, naturally related to the devised Lax–Sato vector field equations and their compatibility conditions. As interesting examples, we constructed new modified spatially four-dimensional Mikhalev–Pavlov and Alonso–Shabat type completely integrable equations, appearing in the study of some differential geometric structures on Riemannian spaces with symmetries.

**Funding:** This research was funded by the Department of Computer Science and Telecommunication of the Cracov University of Technology for a local research grant F-2/370/2018/DS.

**Acknowledgments:** I would like to convey my warm thanks to Gerald A. Goldin for many discussions of the work and instrumental help in editing a manuscript during the XXVIII International Workshop on "Geometry in Physics", held on 30 June–7 July 2019 in Białowieża, Poland. My special appreciation belongs to Stefan Duplij for friendly encouragement to write this article and to Joel

Lebowitz for the invitation to take part in the 121-st Statistical Mechanics Conference, held 12–14 May 2019 at the Rutgers University, New Brunswick, NJ, USA. I cordially appreciate Joel Lebowitz, Denis Blackmore and Nikolai N. Bogolubov for instructive discussions, useful comments and remarks on the work during the Conference. My warm acknowledgements also belong to my close collaborators Alex A. Balinsky, Radoslaw Kycia, Yarema A. Prykarpatsky, Valeriy H. Samoilenko for the support during my work on manuscript.

**Conflicts of Interest:** The author declares no conflict of interest.

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
