# Peer review of "Quantum Current Algebra in Action: Linearization, Integrability of Classical and Factorization of Quantum Nonlinear Dynamical Systems"

_universe, doi:10.3390/universe8050288_

Round 1
Reviewer 1 Report
This is a very extensive review about the quantum current algebra as a universal algebraic structure of integrable classical and quantum dynamical systems. With my limited expertise, I cannot assess critically the huge amount of mathematical material from functional analysis and other areas as presented in this work. Apart from minors corrections concerning spelling: idiabatic -> adiabaric, Nether -> Noether, I see objections for publication in MDPI.
Author Response
I thank Refereee for reading a manuscript and mentioning remarks and pointing out diverse drawbacks entered the exposition. All these point became with attaention checked and corrected. The English as well as style aspect were also improved. Sincerely thanks.
Reviewer 2 Report
The review reads ok but too complicated. It woukd be bettet if the author can write mote consice
Author Response
I cordially thank for reading the stuff of the review and comment. In a revised
version I presented a wide enough introduction with explanations of main points related with technical details. In fact, presentation is not looking too complicated, if to recall classical quantum-mechanical backgrounds commonly with group theory elements. Moreover, I have added a classical example of the coherent vector scheme owing to Bargmann, which is based on the Hilbert space of many-dimensional holomorphic functions.
Reviewer 3 Report
This manuscript is about the universal algebraic structure of quantum and classical integrable nonlinear dynamical systems. The universal algebraic and geometric properties of the non-relativistic quantum current algebra symmetry and their representations are reviewed. The applications to describing geometrical and analytical properties of quantum and classical integrable Hamiltonian systems are suggested.
The manuscript gives complete picture on quantum current algebra symmetry in theoretical and mathematical physics.
Nevertheless, the title is much long. It would be better to omit the words “of theoretical and mathematical physics” in the title of the manuscript.
The section 1 Introduction looks like “Notations and Definitions or Methods and Model Initial Conditions”. It would be better to write in brief form another Introduction about the place of the author’s approach among other approaches and to demonstrate goals and application to the problems of theoretical physics.
In this case the section 2 can be under the title “Notations and Definitions or Methods and Model Initial Conditions” with the text given now in section 1.
Latex error in text after formula (21).
I recommend to publish this very useful and high level manuscript after the minor revisions.
Author Response
I am so thankful to Referee for a very important remarks and style correction suggestions, as well as reordering some sections and chapters, which were made owing to the Referee adbices. Owing to Refgeree suggestion, we have shortened the review title to the following one: "Quantum current algebra in action: linearization, integrability of classical and factorization of quantum nonlinear dynamical systems"
which may serve more adequately relevant to accepting the topic by specialists. The English presentation was also checked and corrected by an English native professional, that strongly improved many points of presentation. I would like also to remark that a section devoted to coherent vector representations was slightly extended by a classical example of the Bargmann-Segal coherent vector constructioin. The latter serves as a scheme for introducing coherent vector representation on other Fock type space objects. Thanks so much.
Reviewer 4 Report
The review article is devoted to some mathematical tool that is current algebra symmetry and it representations subject to applications to describing geometrical and analytical properties of non-relativistic quantum and classical integrable Hamiltonian systems of theoretical and mathematical physics. It is interesting review but not easy to read, as it links various fields of mathematical physics and contains a strong mathematical formalism. So, is seems absolutely necessary to add at the beginning of the article an introductory section, where the author defines and describes the notion of current algebra symmetry and explains why and how it connects (links) such different fields like particular class of quantum systems, soliton systems and integrable dispersionless systems. That is, in which sense some important mathematical formalism with a common clamp holds so many different systems together.
Author Response
I am very thankful to Referee for reading the manuscript and suggesting so nreasonable and instrumental suggestions concerning both rewriting the introduction and improiving other aspects of presentations. I have took into account all these remarks and suggestions, adding a fairly detailed introduction and making some logical improvements to the other sections, taking great care to ensure the logical consistency of the presentation.
I so obliged to Referee for this instrumental help, much imroving a present exposiotion.